

# Supersymmetric grey galaxies, dual dressed black holes and the superconformal index

Sunjin Choi[1*], Diksha Jain[2†], Seok Kim[3‡], Vineeth Krishna[4,5○], Goojin Kwon[3§], Eunwoo Lee[5¶], Shiraz Minwalla[5‖] and Chintan Patel[5♣]

**1** Kavli Institute for the Physics and Mathematics of the Universe (WPI),
The University of Tokyo Institutes for Advanced Study,
The University of Tokyo, Kashiwa, Chiba 277-8583, Japan
**2** Department of Applied Mathematics and Theoretical Physics,
University of Cambridge, Cambridge CB3 0WA, UK
**3** Department of Physics and Astronomy & Center for Theoretical Physics,
Seoul National University, Seoul 08826, Korea
**4** Leinweber Center for Theoretical Physics, University of Michigan,
Ann Arbor, MI 48109, United States
**5** Department of Theoretical Physics, Tata Institute of Fundamental Research,
Homi Bhabha Rd, Mumbai 400005, India

★ sunjin.choi@ipmu.jp , † diksha.2012jain@gmail.com , ‡ seokkimseok@gmail.com ,
○ vineethbannu@gmail.com , § gk404@snu.ac.kr , ¶ eunwoo.lee@tifr.res.in ,
‖ minwalla@theory.tifr.res.in , ♣ chintan.patel@tifr.res.in

## Abstract

Motivated by the recent construction of grey galaxy and dual dressed black hole solutions in $AdS_5 \times S^5$, we present two conjectures relating to the large $N$ entropy of supersymmetric states in $\mathcal{N} = 4$ Yang-Mills theory. Our first conjecture asserts the existence of a large number of supersymmetric states which can be thought of as a non interacting mix of supersymmetric black holes and supersymmetric "gravitons". It predicts a microcanonical phase diagram of supersymmetric states with eleven distinct phases, and makes a sharp prediction for the supersymmetric entropy (as a function of 5 charges) in each of these phases. The microcanonical version of the superconformal index involves a sum over states - with alternating signs - over a line in 5 parameter charge space. Our second (and more tentative) conjecture asserts that this sum is dominated by the point on the line that has the largest supersymmetric entropy. This conjecture predicts a large $N$ formula for the superconformal index as a function of indicial charges, and predicts a microcanonical indicial phase diagram with nine distinct phases. It predicts agreement between the superconformal index and black hole entropy in one phase (so over one range of charges), but disagreement in other phases (and so at other values of charges). We compare our predictions against numerically evaluated superconformal index at $N \leq 10$, and find qualitative agreement.

# 1   Introduction

## 1.1   Supersymmetric black holes and the superconformal index

Twenty years ago, Gutowski and Reall discovered a one parameter set of $(1/16)^{th}$ BPS supersymmetric black hole solutions in IIB supergravity on $AdS_5 \times S^5$ [1]. Intriguingly, regular Gutowski-Reall (GR) black holes (and their generalizations) exist only on a particular codimension one surface in the space of charges[1] – a phenomenon that we will call $R$−charge concentration. This fact may appear to suggest the following "concentration" conjecture

- $\mathcal{N} = 4$ Yang-Mills theory (on $S^3$, at strong coupling) hosts order $e^{N^2}$ supersymmetric states only on the black hole sheet: the codimension one surface in the 5 dimensional space parameterized by the three $SO(6)$ charges $Q_1, Q_2, Q_3$ and two angular momenta $J_1, J_2$ on which nonsingular supersymmetric black holes exist.

From the field theory side, the complete enumeration of supersymmetric states is a difficult task.[2] Partial information on the spectrum of supersymmetric states can be obtained more easily from the superconformal index [11], defined by

$$\mathcal{I}_W = \operatorname{Tr} \exp \left( -\sum_{i=1}^{5} \nu_i Z_i \right), \quad \text{where} \tag{1}$$

$$\nu_1 + \nu_2 + \nu_3 - \nu_4 - \nu_5 = 2n\pi i \qquad (n \text{ is an odd integer}). \tag{2}$$

The trace (1) counts states annihilated by the supercharge with charges listed in the fifth line of Table 1. In (1), the charges $\{Z_i\}$ are defined by

$$(Z_1, Z_2, Z_3, Z_4, Z_5) = (Q_1, Q_2, Q_3, J_1, J_2), \tag{3}$$

and we sometimes use the alternate notation

$$(\nu_1, \nu_2, \nu_3, \nu_4, \nu_5) = (\mu_1, \mu_2, \mu_3, \omega_1, \omega_2), \tag{4}$$

---

[1]Gutowski and Reall specialized their study to black holes with equal $SO(6)$ charges and equal angular momenta. In subsequent works [2–6] this restriction was relaxed. The basic phenomenon described above continues. One finds a four (rather than five) parameter set of regular black hole solutions. Smooth solutions only exist on a specific codimension one surface in the 5 dimensional space parameterized by $Q_1, Q_2, Q_3, J_1, J_2$. See [7–10] for suggested field theory explanations of this surface.

[2]Though one that has seen progress in recent years [11–22]; see below for more detail.

for chemical potentials.[3]

Consider two states with charges[4]

$$(Z_1, Z_2, Z_3, Z_4, Z_5), \quad \text{and} \quad \left(Z_1 + \frac{n}{2}, Z_2 + \frac{n}{2}, Z_3 + \frac{n}{2}, Z_4 - \frac{n}{2}, Z_5 - \frac{n}{2}\right), \quad n \in \mathbf{Z}. \quad (5)$$

It follows from (2) that the contribution of these two states to the index differs only by the factor $(-1)^n$. In other words, the index does not record information of the number of states at definite values of the charges, but instead computes the sum of the number of states (with oscillating minus signs) along lines in charge space. In equations,[5]

$$\mathcal{I}_W = \sum_{Z'} n_I(Z')(-1)^{2(Z_1' + Z_2' + Z_3' - Z_4' - Z_5')} e^{-\nu_i Z_i'}, \quad (6)$$

where $n_I(Z_i')$ in (6) is given in terms of $n(Z_i)$, the number of supersymmetric states (in the interacting theory) with charges $Z_i$, by[6]

$$n_I(Z_i') = (-1)^{2(Z_1' + Z_2' + Z_3' - Z_4' - Z_5')} \sum_m (-1)^m\, n\left(Z_1' + \frac{m}{2}, Z_2' + \frac{m}{2}, Z_3' + \frac{m}{2}, Z_4' - \frac{m}{2}, Z_5' - \frac{m}{2}\right). \quad (7)$$

The summation in (6) is taken over the set of index lines, which, in turn, may be thought of as the space of distinct equivalence classes in $\{Z_i\}$, where two charge vectors are equivalent if they differ by a multiple of the vector $t_I$ in charge space,[7] where

$$t_I = \frac{1}{2}(1, 1, 1, -1, -1). \quad (8)$$

Let us now return to supersymmetric black holes. Recall that the black hole sheet is a codimension one surface in 5 dimensional charge space $\{Z_i\}$ (see §2 for a detailed geometrical exploration of this embedding). Every point on the black hole sheet has definite values of charges $\{Z_i\}$ and so is intersected by a unique index line. In fact no index line ever intersects the black hole sheet more than once (see Appendix §C.9 for a proof), so the map from black holes to index lines is one to one.[8] Given that this map exists, it is natural to wonder about the relationship between the indicial entropy $\ln|n_I(Z')|$ of a given index line, and the Bekenstein-Hawking entropy of the associated supersymmetric black hole. Recent works [7–9, 23] (improving on early attempts in [11]) have demonstrated that the indicial entropy equals the Bekenstein-Hawking entropy of the associated black hole, over a range of the charge equivalence classes $Z_i'$.[9]

The match described above is remarkable. However one might wish to better understand why it works. Why does the index, which counts states along an entire line, end up evaluating

---

[3]Equivalently, the index is given by

$$\text{Tr}\left((-1)^F e^{-\sum_{i=1}^3 \mu_i Q_i - \sum_{m=1}^2 \omega_m J_m}\right),$$

subject to the constraint $\mu_1 + \mu_2 + \mu_3 - \omega_1 - \omega_2 = 0$. An inspection of Table 1 will convince the reader of the equivalence of these two definitions.

[4]Charges in our convention are quantized in half integer units.

[5]The factor $(-1)^{2(Z_1' + Z_2' + Z_3' - Z_4' - Z_5')}$ in (6) ensures that the quantity $n_I(Z')$ is "'class valued", i.e. it depends only on the equivalence class (line) under study, and not on the choice of representative $Z_i'$. (6) can be reworded as saying that the index is the sum over the various $n_I(Z_i')$ defined in (7), weighted by $e^{-\nu_i Z_i}$ evaluated on any bosonic state on each line.

[6]By shifting the dummy variable in the sum, we see that the RHS of (7) is left unchanged if we shift $\{Z_i\}$ by a vector proportional to (8). The overall factor on the RHS has been chosen in a manner that ensures that Bosons contribute +1 and Fermions contribute −1 to each of the terms in a line.

[7]See around 2.18 of [11] for a representation theory explanation of this chain.

[8]However the map is not "onto". There are index lines that nowhere intersect the black hole sheet.

[9]See §C.3 for details and references.

the entropy of the intersection black hole, which lives at a single point on this line? The concentration conjecture presented above suggests the following "concentration explanation" of this point

- The Index agrees with the entropy of its intersecting black hole because the number of supersymmetric states is small everywhere along the index line except in the neighbourhood of the point at which it intersects that black hole sheet, and so receives its dominant contribution from the immediate neighbourhood of this intersection point.

The experimentally observed agreement between the indicial and black hole entropies might thus seem to provide some support for the concentration conjecture. Moreover, the recent paper [24] has demonstrated that an analogue of the concentration conjecture does indeed hold in simple toy models like supersymmetric SYK theory.[10]

## 1.2 A conjecture for the large $N$ supersymmetric cohomology at generic charges

The "concentration explanation" feels satisfyingly economical. However, the results of recent field theoretic enumerations of supersymmetric states [11–22] have not found any clear evidence for the concentration conjecture, at least at the (usually small) values of $N$ and charges at which these studies have been performed. In this paper we propose a modification of the concentration conjecture. Our proposal is motivated by the recent construction [25–27] of new $AdS_5 \times S^5$ solutions: Grey galaxies [25, 27], revolving black holes (RBHs) [25] and dual dressed black holes (DDBHs) [26]. We refer the reader to the papers [25–27] for the details of these solutions. Here we only recall the following aspects of these (in general non supersymmetric) solutions.

- These new solutions can approximately be thought of as a non interacting mix[11] of (vacuum) black holes and "gravitons".[12] The "gravitons" carry angular momentum in grey galaxy solutions, but carry $SO(6)$ charge in DDBH solutions.

- Grey galaxies/ DDBHs appear in distinct families [26, 27], labeled by the rank of the $SO(4)$ / $SO(6)$ angular momentum carried by their gravitons. The rank also equals twice the number of $\Omega_i$ that are parametrically close to unity (in the case of grey galaxies or RBHs) and twice the number of $\Delta_i$ that are close to unity (for DDBHs).[13] Grey galaxies and RBHs are either of rank 2 or 4, while DDBHs are of rank 2, 4 or 6.[14]

- At leading order in large $N$, the entropy of these solutions equals the entropy of their vacuum black hole component. In particular, grey galaxies and RBHs with the same

---

[10]Indeed, we have borrowed the term "R-charge concentration" from the paper [24]. We emphasize that the authors of [24] were aware of the possible subtleties in R-charge concentration conjecture applied to $\mathcal{N} = 4$ Yang Mills theory (see section 5.2 of [24]).

[11]In quantum mechanical terms, as a tensor product.

[12]Through this paper we use the term "gravitons" to refer to large angular momentum gravitons (in the case of grey galaxies), one, two or three large dual giant gravitons (in DDBH solutions) and descendant derivatives (i.e. derivatives that act on the collection of primaries that make up the supersymmetric black hole and so set it revolving [25]) -in the case of RBHs.

[13]$\Omega_i$ and $\Delta_i$ are the usual thermodynamic chemical potentials; see §2.5 for precise definitions.

[14]There are 2 distinct rank 2 grey galaxy phases, depending on which two plane the graviton angular momentum lies in, but only one rank 4 grey galaxy. The central black hole in a rank 4 grey galaxy carries charges $J_1 = J_2$. Similarly there are 3 distinct rank 2 DDBH phases (depending on which of the 3 two planes the $SO(6)$ charges lie in), $\binom{3}{2}$=3 distinct rank 4 DDBHs and one rank 6 DDBH. While the central black hole in rank 6 DDBHs carries equal values of all three $SO(6)$ charges, it carries equal values of two of these three charges in rank 4 DDBHs.

central black holes (and other charges) have identical entropies at leading order in the large $N$ limit, and so lead to identical leading order predictions.[15]

- These new solutions (rather than vacuum black holes) dominate the phase diagram of large $N$ $\mathcal{N} = 4$ Yang-Mills in a band of energies around the BPS plane.

- In the BPS limit, these solutions appear to reduce to a non interacting mix of supersymmetric vacuum black hole[16] states and supersymmetric "gravitons". There are several reasons to believe that at least some 'supersymmetric black hole plus supersymmetric "graviton" states are exactly supersymmetric. Revolving black holes [25] are the bulk duals of supersymmetric descendants of SUSY black hole states, and so are exactly supersymmetric. Turning to grey galaxies, the authors of [17, 20, 22] have constructed large classes of fortuitous cohomologies by taking the product of high angular momentum gravitons with (what appears to be) core black holes, giving some direct field theory evidence for the existence of at least some supersymmetric grey galaxy states.[17] The existence of supersymmetric DDBH solutions is suggested by the fact that special dual giants around Gutowski-Reall black are supersymmetric in the probe limit, and also by the construction of similar cohomologies on the field theory side [22]. See §6 for more on this point.

- While the SUSY black holes exist only on the black hole sheet[18] (and so obey the concentration conjecture), supersymmetric gravitons carry arbitrary charges $\{Z_i\}$, subject only to the restriction[19] $Z_i \geq 0$ $\forall i$.

The discussion above suggests that $\mathcal{N} = 4$ Yang-Mills hosts several different supersymmetric configurations with entropy of order $N^2$ at generic values of charges. In fact we have configurations of this sort for every decomposition of the charges $\{Z_i\}$ into

$$Z_i = Z_i^{BH} + Z_i^{gas}, \tag{9}$$

with $Z_i^{BH}$ on the black hole sheet, and $Z_i^{gas} \geq 0$, $\forall i$. At leading order, the entropy of the collection of configurations with the charge decomposition (9) equals the entropy of the vacuum SUSY black hole with charges $Z_i^{BH}$.

We thus propose the

- **Dressed concentration conjecture**: At leading order in the large $N$ limit, the supersymmetric entropy of $\mathcal{N} = 4$ Yang-Mills at any given values of the charges $\{Z_i\}$, is given by

$$\max_{Z_i'} S_{BH}(Z_i') \qquad (Z_i' \leq Z_i \quad \forall i = 1 \ldots 5), \tag{10}$$

where the charges $Z_i'$ lie on the supersymmetric black hole sheet (43).

---

[15] Consequently, every "grey galaxy" phase listed in this paper has a corresponding RBH phase with identical leading order thermodynamics. While grey galaxies have higher entropy than RBHs at first subleading order in $1/N$, the argument (see below) that grey galaxies are exactly supersymmetric seems less watertight than the corresponding argument for RBHs. For this reason, the conservative reader may choose to read every mention of a "grey galaxy" as reference to a "grey galaxy/ RBH phase" in the rest of this paper.

[16] All supersymmetric black holes turn to automatically obey $\Omega_i = \Delta_j = 1$ ($i = 1, 2$, and $j = 1 \ldots 3$), and so, to that extent, automatically "qualify" as black holes at the centres of grey galaxies and DDBHs. See §2.5 for details.

[17] The analysis of [17, 20, 22] was performed using the nonlinear but tree level supersymmetry operator, and so computes supersymmetric states of the Beisert one loop Hamiltonian [28]. There is, however, some evidence that the number of supersymmetic states- which jumps discontinuously from the zero loop to the one loop Hamiltonian - does not change further at higher loop orders (see [13, 15]). It is thus possible that the weak coupling results of [17, 20, 22] (at the level of counting) apply all the way to strong coupling.

[18] We use the word " "black hole sheet" for the codimension one surface in the 5d charge space on which regular black holes lie.

[19] See §2.10 for an explanation of these inequalities.

The dressed concentration conjecture leads to a fully quantitative prediction for the super-symmetric entropy of large $N$ $\mathcal{N} = 4$ Yang-Mills theory, as a function of charges. In §3 we implement the maximization in (10) to produce a (relatively) explicit formula for the number of supersymmetric states for arbitrary values of the five charges $Z_i$. In different ranges of the charges $\{Z_i\}$, the nature of the maximum entropy solution changes qualitatively. Consequently, the maximization in (10) produces a rich phase diagram (in the microcanonical ensemble) with several sharp phase transitions between three distinct grey galaxy (or RBH) phases and 7 distinct DDBH phases.[20] These phase transitions are all continuous - roughly speaking of second order - in the microcanonical sense.[21] In §3.3 (see Fig. 7 for a summary) we present an algorithm that determines an explicit expression for the supersymmetric entropy $S_{BPS}$ in each of these phases.

We end this subsection with a disclaimer. In our presentation of the dressed concentration conjecture (and our use of it in §3 §4) we have effectively assumed that $AdS_5 \times S^5$ hosts no single centred supersymmetric black holes other than the known four parameter generalizations of Gutowski-Reall black holes [6].[22] We are unaware of any clear indication that additional single centred solutions exist. If new families of such supersymmetric black holes happen to be found in the future, however, it is conceivable that (10) would continue to apply, but with $Z_i^{BH}$ now allowed to range over the full set of single centred supersymmetric black holes (not just the currently known ones). If this possibility pans out, the detailed results of §3 and §4 would then likely require modification. In the rest of this paper we simply proceed assuming that the known black holes are the only ones that exist (see some brief related remarks in §6).

## 1.3 A conjecture for the superconformal index

Our proposal for $S_{BPS}(Z_i)$ (described in subsection §1.2, i.e. the previous subsection) is clearly in tension with the concentration explanation for the agreement between $\ln |n_I(Z_i)|$ and the entropy of the associated black hole (reviewed in §1.1). We now propose an alternate explanation for that striking agreement - but also predict deviation from this agreement over appropriate ranges of $Z_i$.

The alternating signs in (7) makes the microcanonical index difficult to analyze. It is simpler to first study a related but simpler quantity $\tilde{n}_I$ (obtained by dropping all the signs in (7))[23]

$$\tilde{n}_I(Z_i) = \sum_m n\left(Z_1 + \frac{m}{2}, Z_2 + \frac{m}{2}, Z_3 + \frac{m}{2}, Z_4 - \frac{m}{2}, Z_5 - \frac{m}{2}\right). \tag{11}$$

As we have explained above, our dressed concentration conjecture above leads to a formula for a coarse-grained version of $n(Z_i)$ that takes the structural form $n(Z_i) = e^{N^2 S_{BPS}\left(\frac{Z_i}{N^2}\right)}$.[24] Inserting this into (11) we find

$$
\begin{aligned}
\tilde{n}_I(Z_i) &\approx 2N^2 \int dx\, e^{N^2 S_{BPS}\left(\frac{Q_1+x}{N^2}, \frac{Q_2+x}{N^2}, \frac{Q_3+x}{N^2}, \frac{J_1-x}{N^2}, \frac{J_2-x}{N^2}\right)} \\
&\approx 2N^2 e^{N^2 S_{BPS}\left(\frac{Q_1+x_m}{N^2}, \frac{Q_2+x_m}{N^2}, \frac{Q_3+x_m}{N^2}, \frac{J_1-x_m}{N^2}, \frac{J_2-x_m}{N^2}\right)},
\end{aligned}
\tag{12}
$$

where $x_m$ is that value of $x$ on which $S_{BPS}$ is maximized.

---

[20]The DDBH and grey galaxy phases are separated from each other by the codimension one SUSY black hole sheet.

[21]More precisely, the configuration with maximum entropy remains continuous across phase boundaries.

[22]While multi centred configurations of these known black holes presumably exist, they are entropically sub-dominant to grey galaxies and DDBHs for the reasons spelt out in section 6.3 of [25].

[23]$\tilde{n}_I(Z_i)$ is the coefficient $e^{-\nu_i Z_i'}$ in the supersymmetric partition function (26), upon setting $\nu_1 + \nu_2 + \nu_3 - \nu_4 - \nu_5 = 0$.

[24]The fact that $n(Z_i)$ takes this structural form follows from $N$ scaling.

It is clear from (11) and (7) that $n_I$ and $\tilde{n}_I$ are closely related quantities. The second - and more tentative - conjecture of this paper asserts that the oscillations in (7) do not invalidate naive saddle point maximization in (12), so that $n_I$ is also given by the RHS of (12), at leading order in the large $N$ limit. More precisely we propose

- **Unobstructed saddle conjecture:** At leading order in the large $N$ limit, $|n_I(Z_i)|$ equals the maximum of $n(Z_i)$ along the corresponding index line.

Purely mathematically, it is easy to cook up examples of sequences $\{n(m)\}$ for which $\sum_m (-1)^m n(m)$ well approximates $\sum_m n(m)$. However it is equally easy to cook up examples in which $|\sum_m (-1)^m n(m)| \ll |\sum_m n(m)|$. The dressed concentration conjecture asserts that the numbers $n(m)$ that actually appear on the RHS of (7) and (11) are of the first rather than the second sort. In Appendix A we present (indicative rather than conclusive) arguments for why we suspect this is indeed the case.

Even though the *a priori* evidence in favour of the unobstructed saddle conjecture is not strong, in §4 we proceed by initially putting all doubt aside. We assume the validity of this conjecture and proceed to spell out its rather dramatic consequences for the superconformal index. in §5 we then proceed to compare these predictions with independent field theoretic computations, and find (as yet qualitative) agreement. We view this agreement as stronger *a posteriori* evidence[25] for the unobstructed saddle conjecture. In the rest of this introduction we elaborate on these points.

As already mentioned above, the unobstructed saddle conjecture, together with the results from §3 (which employ the dressed concentration conjecture to determine $n(Z_i)$), makes definite predictions for the indicial entropy $\frac{\ln|n_I|}{N^2}$; one is simply instructed to maximize the quantity $N^2 S_{BPS}\left(\frac{Q_1+x}{N^2}, \frac{Q_2+x}{N^2}, \frac{Q_3+x}{N^2}, \frac{J_1-x}{N^2}, \frac{J_2-x}{N^2}\right)$ over $x$. In §4, we study this maximization problem as a function of the indicial charges $\{Z_i\}$(modulo shifts given in (11)). We demonstrate that $x_m$ (the value of $x$ that maximizes the entropy) is a non analytic function of $Z_i$ on codimension one surfaces in the space of indicial charges. These special surfaces are phase boundaries for the microcanonical index. In one particular phase[26] (the so called black hole phase), the indicial entropy equals the entropy of the black hole at the intersection of the index line and the black hole sheet. In the other 8 phases the index is dominated by a DDBH or a grey galaxy solution. The algorithm of §4.3 yields definite formulae for the indicial entropy as a function of charges in each of these phases. As a consequence, the analysis of §4 predicts a rich microcanonical phase diagram as a function of indicial charges, describing sharp phase transitions between 9 distinct phases, each of which has its own formula for the indicial entropy. As in the previous section, the relevant phase transitions are never of first order across phase boundaries in a microcanonical sense.[27] See Figs 8, 12, 13 and 14 for a visualization of the microcanonical indicial phase diagram.

Through the main text of paper we focus entirely - and directly - on the microcanonical index; i.e the computation of the indicial entropy as a function of indicial charges. The Legendre transformation of our microcanonical results to the grand canonical ensemble is a very interesting - but a somewhat confusing exercise. In Appendix I we outline the procedure that implements this Legendre transformation. In particular we demonstrate that grey galaxy (or DDBH) phases that involve a condensation of angular momenta $J_i$ (or charges $Q_j$) correspond

---

[25]While the detailed quantitative predictions for the superconformal index (described below) rely on the dressed concentration conjecture holding in a precise manner, it seems possible to us that the qualitative results reviewed below- namely the existence of an indicial phase diagram with 9 phases, and the structure of the indicial phase diagram - would survive a mild modification of the unobstructed saddle conjecture, if future work suggests the need for such a modification.

[26]This is the dominant phase at charges at which the $Q_i \in \{Z_i\}$ are not too asymmetrical (i.e. not too different from each other) and also at which the $J_i \in \{Z_i\}$ are not too asymmetrical.

[27]As above, we mean that the configuration that maximizes the index is continuous across phase boundaries.

to Grand Canonical phases with the corresponding $\omega_i$ (or $\mu_j$) simply equal to zero (the real and imaginary parts of these chemical potentials both vanish).[28] The explicit determination of the grand canonical partition function as a function of indicial chemical potentials appears to be an algebraically involved exercise, one that we leave to future work.

## 1.4  Comparison with the explicit evaluation of the index at $N \leq 10$

As we have explained above, the two conjectures presented in this paper have both been motivated by gravitational analysis (of new black hole solutions) on the bulk side of the duality. It is, of course, of great interest to test the predictions of this paper against explicit field theoretic computations of the supersymmetric cohomology and the superconformal index.

As we have explained in §1.1, the $\mathcal{N} = 4$ cohomology is not yet very well understood. However, there is a clear and well established integral formula for the index in $\mathcal{N} = 4$ supersymmetric Yang-Mills theory [11]. The evaluation of this matrix integral, in the large $N$ limit, has proved to be surprisingly intricate. Most of the (extremely impressive) computations performed to date, proceed by determining one "saddle point" to the problem, but leave open the problem of establishing that the saddle point is dominant. In order to test our predictions against the integral formula for the index - in a way that is free of assumptions - we have evaluated the superconformal index in a "brute force" manner (on a computer) at values of $N \leq 10$. Ignoring the fact that 10 is not a strikingly large number, we have then (boldly or rashly, according to taste) proceeded to compare these results with our predictions presented earlier in this paper at a special simple cut of charges. In Fig 1 we display our "brute force" results at $N = 10$, together with the large $N$ predictions obtained using the method of §4, for the indicial entropy as a function of one indicial charge with all others held fixed.[29] The match between the data and our predictions appears rather good to us, at least at the qualitative level. We view Fig. 1 as substantial - but not, as yet, foolproof - evidence for the correctness of our conjectures.

The plot of Fig. 1 displays surprisingly good qualitative agreement between "theory" and "numerics", despite the fact that we are working only at $N = 10$. We explain in §5.4 that this unexpectedly good agreement is a consequence of the following:

- On the cut of charges studied in this subsection, the deviation between the grey galaxy prediction and the naive black hole prediction happens to be large at numerically accessible values of charges.

- Where the naive black hole and grey galaxy predictions differ maximally, it turns out that the contributions of supersymmetric gravitons to the index is negligible. Consequently, our numerically evaluated index sees the black hole entropy "unpolluted" by the entropy of gravitons.

These two favourable features do not persist in other cuts in charge space that we have investigated. In §5.3 we once again compare "theory" to "numerics", this time on the charge cut $q_1 = q_2 = q$, $q_3 = q'$, $j_1 = j_2 = j$. In this case, it turns out that DDBH prediction does not deviate substantially from the naive black hole entropy at accessible values of charges. Moreover the (indicial) entropy of gravitons turns out to be rather substantial where the deviation is maximum (see Fig. 17, and §5.4). Consequently, the comparison between "theory" and "experiment" is ineffective in testing the predictions of our paper in this case.[30]

---

[28]See [29] for a study of the thermodynamics of vacuum BPS black holes in the canonical ensemble.

[29]We focus on the charges $q_1 = q_2 = q_3 = q$ (see (15) for definitions of these symbols). In this case the Indicial entropy depends on two indicial charges, $\alpha = q + j_L$ and $j_R$. In Fig 1, we plot $S_{BPS}(\alpha, j_R)$ versus $j_R$ at fixed $\alpha$.

[30]It may be possible to present a more serious check between theory and experiment by theoretically filtering out the contribution of gravitons near the tail (or, of course, by going to higher values of $N$). We leave this to future work.

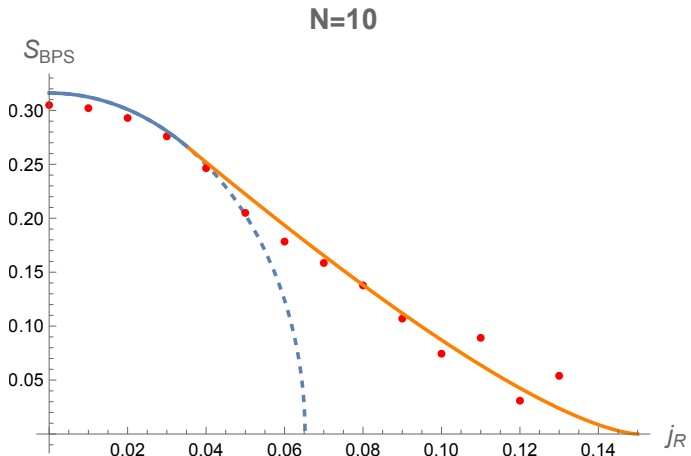

Figure 1: The red dots represent the numerical values of the indicial entropy, expressed as $\frac{1}{N^2}\log(|n_I(Z_i)|)$, for $N = 10$, $2(Z_1 + Z_2 + Z_3) + 3(Z_4 + Z_5) = 90$, and $j_R = \frac{Z_4 - Z_5}{N^2}$ (see §5 for the detailed setup). The blue solid and dashed lines depict the black hole entropy, determined at the intersection of the black hole sheet and an index line. The orange line depicts the indicial entropy for indicial charges where grey galaxy solution dominates. The solid line, combining the blue and orange segments, represents the indicial entropy $S_{BPS}$ as computed using the unobstructed saddle conjecture.

## 1.5 Structure of this paper

The rest of this paper is organized as follows. In §2 we explore constraints on the charges that can be accessed by multiparticling supersymmetric letters in $\mathcal{N} = 4$ Yang-Mills theory, and present a detailed study of the embedding of the supersymmetric black hole sheet into this 5 dimensional charge space. In §3 we explore the implications of the dressed concentration conjecture, and in particular present a quantitative prediction for the number of supersymmetric states as a function of charges. In §4 we explore the implications of the unobstructed saddle conjecture, and in particular present a quantitative prediction for the indicial entropy as a function of indicial charges. In §5, we study two, three parameter cuts of 5 dimensional charge space, namely $Q_1 = Q_2 = Q_3 = Q$ (with arbitrary angular momenta) and $J_1 = J_2$ and $Q_1 = Q_2$ and compare our predictions for indicial entropy with the explicit computer based evaluation of the superconformal index at $N \leq 10$. In §6 we conclude with a discussion of our results. Technical results that support the analysis in the main text are enclosed in several Appendices.

## 2 Supersymmetric states and the black hole sheet

$\mathcal{N} = 4$ Yang-Mills enjoys invariance under the superconformal algebra $PSU(2,2|4)$. This algebra has 32 supercharges (16 $\mathcal{Q}$s and 16 $\mathcal{Q}^\dagger$s or $\mathcal{S}$s). Consider $\mathcal{N} = 4$ Yang-Mills on $S^3$. We say that a state $|\psi\rangle$ in this theory is supersymmetric if $|\psi\rangle$ is annihilated by both $\mathcal{Q}$ and $\mathcal{Q}^\dagger$ for at least one choice of $\mathcal{Q}$.

All states in $\mathcal{N} = 4$ Yang-Mills on $S^3$ transform in (infinite dimensional, module type) representations of $PSU(2,2|4)$. These representations have all been classified [30]. It was demonstrated in [11] that complete information of the spectrum of states annihilated by any

particular left-moving[31] $\mathcal{Q}$, plus corresponding information for any particular right-moving supersymmetry, together completely determine the full spectrum of supersymmetric states.[32] Moreover, the CP symmetry maps states annihilated by a left-moving supercharge to states annihilated by its right-moving partner, so once we have full information of states annihilated by a left-moving supercharge, the right-moving information comes for free.

In this paper, we follow the conventions of previous papers (starting with [11]) to study the spectrum of states annihilated by the left-moving supercharge $\mathcal{Q}$ with charges given in fifth row of table 1.

## 2.1 Anti commutation relations

The anticommutation relations between the supercharge with charges mentioned in 5th row of table 1 and its (radial quantization) Hermitian conjugate $\mathcal{S} = \mathcal{Q}^\dagger$ is given by

$$2\{\mathcal{Q}, \mathcal{S}\} = E - (J_1 + J_2 + Q_1 + Q_2 + Q_3) \equiv \Delta . \tag{13}$$

It follows from (13) that states are annihilated by both $\mathcal{Q}$ and $\mathcal{S}$, if and only if[33] they obey the BPS bound[34]

$$E = J_1 + J_2 + Q_1 + Q_2 + Q_3 . \tag{14}$$

## 2.2 Supersymmetric letters

Supersymmetric states of $\mathcal{N} = 4$ Yang-Mills theory are made out of supersymmetric "letters", i.e. adjoint valued fields that obey the condition (14). All such letters are listed in Table 1,[35] together with their charges. As explained around (3), we will often use the symbols $Z_i$ $i = 1 \ldots 5$ to label the five commuting conserved charges of supersymmetric states. We also use the symbols $\zeta_i$ defined by

$$\zeta_i = \frac{Z_i}{N^2} = (q_1, q_2, q_3, j_1, j_2) , \tag{15}$$

for "intensive" charges, i.e. for charges in units of $N^2$.

## 2.3 Allowed values of charges for supersymmetric states

In this subsection we ask the following question: what range of charges do multi-particle states (made up of the letters listed in Table 1) access?[36]

---

[31]Supersymmetries transform in the $(2, 1) + (1, 2)$ representation of $SO(4) = SU(2)_L \times SU(2)_R$. We refer to supersymmetries that are singlets under $SU(2)_R$ as left-moving, and supersymmetries that are singlets under $SU(2)_L$ as right-moving.

[32]Including SUSY states annihilated by supercharges other than the special ones discussed above. The argument proceeds using representation theory. The subgroup of $PSU(2, 2|4)$ that commutes with any particular left-moving $\mathcal{Q}$ is $PSU(2, 1|3)$. The spectrum of $\mathcal{Q}$ supersymmetric states can be organized - in a unique way - into multiplets of this commuting superalgebra. This organization gives us a list of representations of $PSU(2, 1|3)$. Now the map from (left-moving) short representations of $PSU(2, 2|4)$ to representations of $PSU(2, 1|3)$ is one to one. Consequently the list of representations of the commuting superalgebra (referred to above) can be uplifted to a complete list of (left-moving) short representations of $PSU(2, 2|4)$. A similar argument holds on the right-moving side.

[33]The "only if" part of this statement uses the fact that $\mathcal{N} = 4$ Yang-Mills theory is unitary. As a consequence, only state that has zero norm is the zero state.

[34]By abuse of terminology, in the rest of this paper, we refer to a state as supersymmetric if and only if it obeys the BPS bound (14), i.e. if and only if it is annihilated by $\mathcal{Q}^{+++}_{-\frac{1}{2},0}$ and its complex conjugate, regardless of whether or not it is annihilated by some other supercharge.

[35]More precisely, the letters are those obtained by acting arbitrary numbers of derivatives (sixth line of Table 1) on any of the fields listed in the first four lines of that table. In the 5th line of the table we also enumerate the charges of our special supercharge - the one that annihilates all these letters.

[36]In asking this question, we ignore the Gauss Law. We also ignore the Fermi Exclusion principle in §2.3.1 and §2.3.2 below, but take this aspect into account in §C.2.1 and §C.2.2.

Table 1: The BPS letters in $\mathcal{N} = 4$ Yang-Mills and their charges. Here $E$ is the energy, $J_{L/R}$ or $J_{1/2}$ label the angular momenta and are related by $J_{L/R} = \frac{J_1 \pm J_2}{2}$. $Q_i$ and $R_i$ are the R-charges and their relationship is given in Appendix §B.1.

| Letters | $E$ | $(J_L, J_R)$ | $(J_1, J_2)$ | $(Q_1, Q_2, Q_3)$ | $(R_1, R_2, R_3)$ |
|---|---|---|---|---|---|
| $X, Y, Z$ | 1 | $(0,0)$ | $(0,0)$ | $(1,0,0), (0,1,0), (0,0,1)$ | $(0,1,0), (1,-1,1), (1,0,-1)$ |
| $\bar{\psi}_{0,\pm}$ | $\frac{3}{2}$ | $(0, \pm\frac{1}{2})$ | $(\pm\frac{1}{2}, \mp\frac{1}{2})$ | $(\frac{1}{2}, \frac{1}{2}, \frac{1}{2})$ | $(1,0,0)$ |
| $\psi_{+,0}$ | $\frac{3}{2}$ | $(\frac{1}{2}, 0)$ | $(\frac{1}{2}, \frac{1}{2})$ | $(\frac{1}{2}, -\frac{1}{2}, \frac{1}{2}), (-\frac{1}{2}, \frac{1}{2}, \frac{1}{2}), (\frac{1}{2}, \frac{1}{2}, -\frac{1}{2})$ | $(1,-1,0), (0,1,-1), (0,0,1)$ |
| $F_{++}$ | 2 | $(1,0)$ | $(1,1)$ | $(0,0,0)$ | $(0,0,0)$ |
| $\mathcal{Q}^{+++}_{-\frac{1}{2},0}$ | $\frac{1}{2}$ | $(-\frac{1}{2}, 0)$ | $(-\frac{1}{2}, -\frac{1}{2})$ | $(\frac{1}{2}, \frac{1}{2}, \frac{1}{2})$ | $(1,0,0)$ |
| $D_{+\pm}$ | 1 | $(\frac{1}{2}, \pm\frac{1}{2})$ | $(1,0),(0,1)$ | $(0,0,0)$ | $(0,0,0)$ |

### 2.3.1 The bosonic cone

If we view the derivatives as independent letters, we have five bosonic letters ($X, Y, Z$, $D_{++}$ and $D_{+-}$). Let us denote the (rather simple) charge vectors $(Z_1, Z_2, Z_3, Z_4, Z_5)$ of these letters by

$$b_1 = (1,0,0,0,0), \qquad b_2 = (0,1,0,0,0), \qquad b_3 = (0,0,1,0,0),$$
$$b_4 = (0,0,0,1,0), \qquad b_5 = (0,0,0,0,1). \tag{16}$$

States built out of these letters can clearly access those (and only those) charges that obey[37]

$$\zeta_i \geq 0 \qquad (i = 1 \dots 5). \tag{17}$$

In preparation for the next subsubsection, we now outline a more formal argument for the (obvious) result (17). The bosonic cone is given by

$$z = \sum_{i=1}^{5} \lambda_i b_i \qquad (\lambda_i \geq 0 \; \forall i). \tag{18}$$

The boundary in charge space of this region is given by charges that saturate the inequalities in (18), i.e. by vectors of the form (18) with one (or more) $\lambda_i = 0$. This boundary is built out of the 5 codimension one walls, $\lambda_i = 0$, $i = 1 \dots 5$. The accessible region consists of the "inside" of each of these 5 regions,[38] in other words of the region (17).

### 2.3.2 The Bose-Fermi cone

Let us now also include fermions. In this subsection we ignore the Fermi exclusion principle (for completeness, this is accounted for in §C.2.1 and §C.2.2, but we will never have occasion to use this more complete analysis).

We see from Table 1 that the charges that multi particle bosons and fermions can together access take the form

$$z = \sum_{i=1}^{10} \lambda_i b_i \qquad (\lambda_i \geq 0 \; \forall i). \tag{19}$$

---

[37]The normalized charges $\zeta_i$ are quantized in units of $1/N^2$ and so are effectively continuous in the large $N$ limit of interest to this paper.

[38]The "inside" of surface $\zeta_i = 0$ defined by the normal vector that has a positive dot product with $\hat{e}_i$, the unit vector in the $i^{th}$ charge direction.

$b_i$ were defined in the previous subsubsection for $i = 1 \ldots 5$, and for $i = 6 \ldots 10$, $v_i$ are given by the charges of the 5 fermionic letters (three $\psi_{0,+}$, $\bar{\psi}_{0,+}$ and $\bar{\psi}_{0,-}$), i.e. by

$$b_6 = \left(-\frac{1}{2}, \frac{1}{2}, \frac{1}{2}, \frac{1}{2}, \frac{1}{2}\right), \qquad b_7 = \left(\frac{1}{2}, -\frac{1}{2}, \frac{1}{2}, \frac{1}{2}, \frac{1}{2}\right), \qquad b_8 = \left(\frac{1}{2}, \frac{1}{2}, -\frac{1}{2}, \frac{1}{2}, \frac{1}{2}\right),$$
$$b_9 = \left(\frac{1}{2}, \frac{1}{2}, \frac{1}{2}, -\frac{1}{2}, \frac{1}{2}\right), \qquad b_{10} = \left(\frac{1}{2}, \frac{1}{2}, \frac{1}{2}, \frac{1}{2}, -\frac{1}{2}\right). \tag{20}$$

As in the previous subsection, the boundary of the region of allowed charges is made up of a collection of codimension one planes. The four dimensional tangent space of each of these planes is spanned by (at least) 4 of the 10 vectors $v_i$. The outermost of these planes are those whose outward pointing normals obey

$$n.b_i \leq 0, \quad \forall i = 1 \ldots 10 \tag{21}$$

(this condition asserts that there does not exist an allowed motion that takes us beyond these boundaries, so that the boundaries really are the outermost ones). It is not difficult to convince oneself that the legal region is bounded by the 10 planes[39]

$$\zeta_i + \zeta_j = 0, \quad \text{legal region } \zeta_i + \zeta_j \geq 0. \tag{22}$$

The equation (22) has an interesting physical interpretation (see Appendix C.1). Each of the 10 boundary surfaces (22) represents the condition for states in the theory to be annihilated by an additional supercharge (other than $\mathcal{Q}$). Consequently the 10 planes (22) - the boundaries of the legal region - are surfaces on which states are $(1/8)^{th}$ (rather than $(1/16)^{th}$) BPS.

### 2.4 The charge lattice $\Lambda$

In our discussion of the bosonic cone and the Bose-Fermi cone we have, so far, ignored on point, namely that the charges of supersymmetric states are quantized. The charge of every supersymmetric state is a (positive) integer linear combination of the eleven charge vectors listed in (16) and (20). Integer linear combinations of these eleven vectors define a lattice, which we call the charge lattice and denote by $\Lambda$. All supersymmetric states have charges that are represented by a vector in that part of the charge lattice that is contained within the Bose Fermi cone.

In §1.1 we defined indicial charges as charge vectors modulo shifts by the vector $t_I$ (see (8)). Correspondingly, we define the lattice of indicial charge vectors $\Lambda_I$ as the charge lattice $\Lambda$ modulo shifts by the vector $t_I$. In other words, two distinct vectors in the charge lattice $\Lambda$ represent the same vector in the indicial charge lattice $\Lambda_I$, if they differ by an integer multiple of $t_I$.

### 2.5 The supersymmetric partition function

The density of (not necessarily supersymmetric) states in $\mathcal{N} = 4$ Yang-Mills theory is encoded in the partition function defined by

$$Z_{\text{gen}} = \text{Tr} e^{-\beta(E - \Omega_1 J_1 - \Omega_2 J_2 - \Delta_1 Q_1 - \Delta_2 Q_2 - \Delta_3 Q_3)}. \tag{23}$$

---

[39]It is easily verified by inspection that $\zeta_i + \zeta_j \geq 0$ for each of the 10 letters $b_i$. So all vectors (18) clearly obey (22). Furthermore, this condition is the most stringent possible condition is established by demonstrating that each of the boundary planes is spanned by a subset of the vectors $b_i$, and that the dot product of the remaining $n.b_j > 0$ for the remaining $b_j$. Consider, for instance, the case $(i, j) = (4, 5)$. The normal to this surface is $n = (0, 0, 0, 1, 1)$. The dot product of this normal with $b_1$, $b_2$, $b_3$, $b_9$ and $b_{10}$ all vanish, so these 5 vectors are all tangent to this codimension one surface. Moreover $n.b_i = 1$ for $i = 4, .., 8$ so the relevant dot products are all positive.

A convenient way (see [31]) to focus on the BPS states (14) is to set

$$
\begin{aligned}
\Omega_i &= 1 - \frac{\omega_i}{\beta} \qquad (i = 1 \ldots 2), \\
\Delta_j &= 1 - \frac{\mu_j}{\beta} \qquad (j = 1 \ldots 3).
\end{aligned}
\tag{24}
$$

Plugging (24) into (23) we obtain

$$
Z_{\text{gen}} = \text{Tr} e^{-\beta(E - J_1 - J_2 - Q_1 - Q_2 - Q_3)} \times e^{-\omega_1 J_1 - \omega_2 J_2 - \mu_1 Q_1 - \mu_2 Q_2 - \mu_3 Q_3} \,.
\tag{25}
$$

Recall that non supersymmetric state has an energy that is strictly larger than the BPS bound. It follows that in the zero temperature limit, $\beta \to \infty$, $e^{-\beta(E - J_1 - J_2 - Q_1 - Q_2 - Q_3)}$ evaluates to unity on supersymmetric states, but to zero on all other states. It follows, in other words, that the $\beta \to \infty$ limit of this quantity is effectively a projector onto the space of BPS states. Consequently, $Z_{\text{gen}}$ (under the condition (24), and in the limit $\beta \to \infty$) reduces to

$$
Z_{BPS} = \text{Tr}_{\text{BPS}} \, e^{-\omega_1 J_1 - \omega_2 J_2 - \mu_1 Q_1 - \mu_2 Q_2 - \mu_3 Q_3} \,,
\tag{26}
$$

where the trace in (26) runs only over the states that obey (14). We refer to the finite quantities $\omega_i$ and $\mu_i$ as the renormalized chemical potentials.[40] The chemical potentials that will appear through the rest of the paper will always be the renormalized potentials; the thermodynamic chemical potentials in the LHS of (24) never appear in our analysis. For this reason we will sometimes drop the adjective "renormalized" in later discussions. Note that the chemical potentials that appear (4) are all of the renormalized variety. Using the abbreviated notation of (4), the SUSY partition function takes the form

$$
Z_{BPS} = \text{Tr} e^{-\sum_j \nu_j Z_j} \,.
\tag{29}
$$

## 2.6 The dual charge lattice $\Lambda^*$

Recall (see §2.4) that supersymmetric states carry quantized charges, which may be thought of as vectors in the charge lattice $\Lambda$. It follows that the shift of chemical potentials

$$
\nu_j \to \nu_j + (2\pi i) s_j \,,
\tag{30}
$$

leave the partition function invariant, provided the shift vector $s$ is chosen to ensure

$$
s.Z \in \mathbb{Z}, \quad \forall \, Z \text{ in } \Lambda \,.
\tag{31}
$$

It follows immediately that the imaginary parts of renormalized chemical potentials are effectively periodic variables, with periods that lie in the dual charge lattice $\Lambda^*$.

It is not difficult to characterize $\Lambda^*$ in detail. The condition (31) is met if all $s_j$ ($j = 1 \ldots 5$) are integers, and if either 0, 2 or 4 of the $s_j$ are odd. An (overcomplete) basis of such shifts is given by $(1, 1, 0, 0, 0)$, $(1, -1, 0, 0, 0)$ plus permutations.[41]

---

[40]It is sometimes convenient to work with the redefined angular velocities

$$
\omega_L = \omega_1 + \omega_2 \,, \qquad \omega_R = \omega_1 - \omega_2 \,,
\tag{27}
$$

in terms of which (26) becomes

$$
Z_{BPS} = \text{Tr}_{\text{BPS}} \, e^{-\omega_L J_L - \omega_R J_R - \mu_1 Q_1 - \mu_1 Q_2 - \mu_1 Q_3} \,.
\tag{28}
$$

As usual, $J_L = \frac{J_1 + J_2}{2}$ and $J_R = \frac{J_1 - J_2}{2}$ are the $J^z$ Cartan's of $SU(2)_L$ and $SU(2)_R$ respectively.

[41]Note that every $\nu$ that belongs to the dual charge lattice obeys the condition $\nu.t_I \in \mathbb{Z}$. It follows that if the chemical potential $\nu_i$ obeys (2), then $\nu_i$ shifted by a vector in the dual lattice also obeys (2) - though possibly with a shifted value of $n$.

When studying the superconformal index, we are interested in chemical potentials that obey the condition (2), i.e. the equation

$$\nu.t_I = \pi i \, . \tag{32}$$

The difference, $\delta \nu$, between any two solutions of (32) obeys

$$\delta \nu.t_I = 0 \, . \tag{33}$$

We define the dual indicial lattice $\Lambda_I^*$ to be the restriction of $\Lambda^*$ to dual vectors (chemical potentials) that obey (33). The space of inequivalent indicial chemical potentials is given by solutions of (32) modulo shifts by vectors in $\Lambda_I^*$.

## 2.7 Allowed values of renormalized chemical potentials

The supersymmetric partition function converges if and only if the dot product $\mathrm{Re}(\nu).\zeta$ is positive for every charge vector $\zeta$ that lies within the bosonic cone.[42] In equations, the partition function converges if and only if

$$\mathrm{Re}(\nu_i) > 0 \, , \quad i = 1 \ldots 5 \, . \tag{34}$$

In the domain (34), partition function (29) is an analytic function of its arguments $\nu_i$. While we have not carefully considered this question, we suspect that an analytic continuation of the supersymmetric partition function, beyond this domain (34), is not possible.[43]

Note that the partition function is well defined at chemical potentials that obey (34), even if $\mathrm{Re}(\nu).\zeta$ is negative for some $\zeta$ that lie in the Bose-Fermi (as opposed to the bosonic) cone. Intuitively, we expect the partition function at such values of $\nu_i$ to involve Fermi seas of those fermionic letters whose charge vectors have a negative dot product with $\mathrm{Re}(\nu)$, and so have a large charge to entropy ratio.[44]

## 2.8 Coarse grained supersymmetric entropy

The supersymmetric partition function is given by

$$Z_{BPS} = \sum_i n(Z_i) e^{-\nu_i Z_i} \, , \tag{35}$$

where $n(Z_i)$ denotes the number of supersymmetric states at charges $Z_i$. It follows from large $N$ scaling that

$$\langle n(Z_i) \rangle = e^{N^2 S_{BPS}(\zeta_i)} \, , \tag{36}$$

where $\zeta_i = \frac{Z_i}{N^2}$ (see (15)) and the symbol $\langle \rangle$, in (36) denotes smearing over a smooth envelop function, with a width large compared to one but small compared to $N^2$. On physical grounds we expect $S_{BPS}(\zeta_i)$ to be a smooth function (that varies on scale of order unity) of its arguments

---

[42]If this condition were violated, the contribution of at least one bosonic letter to the partition function would be Boltzmann enhanced rather than suppressed, resulting in a divergence.

[43]In much the same way that the modular function $\eta(\tau)$ cannot be analytically continued to negative values of $\mathrm{Im}(\tau)$.

[44]Note that a Fermi Sea like

$$\prod_{i,j=1}^{N} \bar{\psi}_{ij}$$

is gauge invariant: see [12] for related discussions.

$\zeta_i$ (at least when all chemical potentials are real). We thus expect the summation in (35) to be well approximated by the integral

$$Z_{BPS} = N^{10} \int d\zeta_i \exp\left[N^2\left(S_{BPS}(\zeta_i) - \nu_i\zeta_i\right)\right],\tag{37}$$

which, in turn, can be evaluated in the saddle point approximation. When all chemical potentials are real, we expect the saddle point to lie on the contour of integration, and so, to find the usual thermodynamical relationship (Legendre transform) between $S_{BPS}$ and $\ln Z_{BPS}$.

## 2.9 The superconformal index

As we have explained in the introduction, the superconformal index is simply the BPS partition function

$$\mathcal{I}_W(\nu_i) = \sum_{Z_i} n(Z_i)e^{-\nu_i Z_i},\tag{38}$$

evaluated on chemical potentials that obey the constraint (2). As explained in the introduction, the index is equivalently given by (6) with $n_I(Z')$ listed in (7). Finally, the index is also given by the expression (25) at any value of $\beta$ (when the renormalized chemical potentials obey (2)) - despite appearances, the trace in (25) is independent of $\beta$, see [11]).

As the superconformal index is a specialization of the supersymmetric partition function, it also enjoys invariance under the shifts (31). Recall, however, that the condition (2) involves an arbitrary odd integer on the RHS. The most general shifts (31) do not leave this integer invariant, but, instead, change it by an even number. We can thus (if we like) use these shits to set the number $n$ on the RHS of (2) to unity (or any other convenient value). If we adopt this choice of "fundamental domain", the index is invariant under only those shifts (30) that obey (31), together with the condition

$$s_1 + s_2 + s_3 - s_4 - s_5 = 0.\tag{39}$$

As any continuous evolution of the spectrum leaves the superconformal index unchanged [11], it is independent of coupling, and so can be evaluated in free Yang -Mills theory. The authors [11] used this fact to obtain a simple expression for the superconformal index in terms of an integral over a single $N \times N$ unitary matrix (the holonomy matrix). The last few years have seen impressive progress in the evaluation of this matrix integral in the large $N$ limit (see Appendix C.3 for a listing of some of these results).

In the large $N$ limit, one can move between the canonical index (38) and the microcanonical index (the focus of the current paper) by performing the appropriate Legendre transform: see Appendix I for a brief discussion.

## 2.10 The supersymmetric gas

In addition to supersymmetric black holes (see later in this section), the bulk hosts multi graviton supersymmetric states. The spectrum of $(1/16)^{th}$ BPS gravitons in $AdS_5 \times S^5$ was enumerated in §6.2 of [11]; we present a rederivation - and retabulation (see Table 2 - of this result (in the notation used in this paper) in Appendix B.

The boundary duals of multi gravitons are products of single trace operators built out of fermionic as well as bosonic letters. These single trace operators are either bosonic or fermionic (depending on whether they are build out of an even or odd number of fermionic letters). It turns out (see the last column of lines 1, 4, 5, 6, 9 of Table 2) that all bosonic single traces have charges that lie within the bosonic cone.[45] The charges of some of the fermionic traces -

---

[45]A state lies within the bosonic cone if and only if the contribution of this state to the partition function, $e^{-\alpha_i\mu_i-\zeta_l\omega_L-\zeta_R\omega_R}$, is such that $\alpha_i \geq 0\ \forall i$ and $\zeta_L \geq |\zeta_R|$.

those listed in lines 7 and 8 (also the equation of motion listed in line 10) of Table 2 - also lie within the bosonic cone. On the other hand, the charges of the fermionic traces listed in lines 2 and 3 of Table 2 do not all lie completely within the bosonic cone.

It follows from the discussion above that the supersymmetric gas has states with charges that lie outside the bosonic cone. We will now explain that the fractional "violation" of the bosonic cone condition (in gas states) goes to zero at high energies (or large charges). This fact follows from the observation that while the number of bosonic letters with each "bosonic cone violating" trace is unbounded, the number of fermionic letters in these traces is never larger than 1 (see lines 2 and 3 of Table 2). We demonstrate in Appendix §C.4 that - as a consequence - the ratio of fermionic to bosonic letters (in potentially bosoinc cone violating multi-graviton configurations) is $\lesssim \frac{1}{E^{\frac{1}{4}}}$ where $E$ is the energy of the state in question. In the large $E$ limit, the charges of all multi graviton configurations (asymptotically) lie within the bosonic cone. This estimate is parametrically good in $\frac{1}{N}$ at values of $E \sim N^2$, of interest to this paper. Moreover, this statement is of the if and only if variety. In the limit of large charges, one can find a multi graviton configuration that well approximates any charge vector in the bosonic cone (with a fractional error that goes to zero as a power of the inverse charge, and so as an inverse power of $N$, at the charges of interest to this paper).

## 2.11 Supersymmetric Euclidean black holes

The superconformal index may be evaluated by performing the appropriate boundary Euclidean path integral.[46] At large $N$, the usual AdS/CFT rules map this computation to the action of the bulk Euclidean solution that obeys the corresponding boundary conditions. One class of solutions that fits the bill may be found [7] (see also [32]) by analytically continuing the usual 6 parameter set of Lorentzian black hole solutions to Euclidean space at arbitrary complex values of the chemical potentials that obey (2). This construction yields candidate bulk saddle points for arbitrary complex values of the indicial chemical potentials.[47] These saddles are regular in Euclidean space[48] even though their analytic continuations to Lorentzian space are generically pathological. They have regular killing spinors and so are supersymmetric. Though the bulk solutions depend on the parameter $\beta$ in (25), their action is independent of $\beta$, in agreement with field theoretic expectations.[49]

At leading order in the large $N$ limit, the on-shell action of these black holes is given by [23][50]

$$\frac{N^2 \mu_1 \mu_2 \mu_3}{2\omega_1 \omega_2}. \tag{40}$$

---

[46]Recall that in the superconformal index, the trace (25) is evaluated at values of chemical potentials that obey (2).

[47]There are, actually, an infinite number of such saddle points at any fixed values of chemical potentials. Recall that the superconformal index is left invariant by the shifts (39). However the action of these shifts on Euclidean black holes produces new Euclidean black holes, with (in general) different values of the Euclidean action. The bulk path integral (at any given value of boundary holonomies) thus receives contributions from an infinite number of distinct saddles (which, in general, have distinct Euclidean actions). The final answer in the large $N$ limit is, presumably, dominated by the saddle with the smallest action. The situation here is conceptually similar to the bulk computation of the torus partition function in the $AdS_3/CFT_2$ correspondence, which receives contributions from various "filling ins" of the bulk torus. See [33] for relevant discussion.

[48]See §6 for some comments on the interplay between these solutions and the conditions of [34].

[49]The solutions at finite values of $\beta$ are non extremal (in the sense that they do not have an $AdS_2$ near horizon factor) even though they are supersymmetric.

[50]Note that this answer does not enjoy invariance under the shifts (39). This is a reflection of the fact that the corresponding black hole solutions are not left invariant by (39). The invariance of the index over (39) is restored once we sum over all relevant saddle points (Euclidean black holes). In the large $N$ limit only the dominant saddle contributes, obscuring this point.

When these solutions are legal (see §6 for some discussion of the legality), and are also dominant (40) is the bulk prediction for the superconformal index as a function of chemical potentials. As mentioned above, several different methods have recently been used to reproduce (40) as a "saddle point" contribution to the matrix model that evaluates the index (see §C.3 for references).

The Legendre transform of (40) (following the method of Appendix I) yields

- The indicial entropy as a function of the four real indicial charges associated with these saddle points.

- The five complex chemical potentials as a function of the four real indicial charges.

For the chemical potentials one finds the following results [8, 23]:

$$
\begin{aligned}
\mu_I &= \frac{2\pi i z_I}{1 + z_1 + z_2 + z_3 + z_4}, && I = 1, 2, 3, \\
\omega_1 &= -\frac{2\pi i z_4}{1 + z_1 + z_2 + z_3 + z_4}, && \omega_2 = -\frac{2\pi i}{1 + z_1 + z_2 + z_3 + z_4},
\end{aligned}
\tag{41}
$$

where $\mu_1 + \mu_2 + \mu_3 - \omega_1 - \omega_2 = 2\pi i$ and the $z_I$'s are auxiliary parameters that conveniently express chemical potentials and are determined in terms of the on-shell black hole charges:

$$
z_I = -\frac{S + 2\pi i J_2}{S - 2\pi i Q_I}, \qquad z_4 = \frac{S + 2\pi i J_2}{S + 2\pi i J_1}.
\tag{42}
$$

Note that the fact that the indicial charges are real, forces these chemical potentials to be complex, as emphasized above. For the entropy one finds (45). (45) is also the entropy ($S$) that appears in the formulae (42).

## 2.12 Supersymmetric Lorentzian black holes

The rest of his section is devoted to the study of the space of regular Lorentzian supersymemtric black holes.

As we have mentioned in the introduction, IIB theory on $AdS_5 \times S^5$ admits supersymmetric solutions on a four dimensional submanifold of the five dimensional charge space. This four dimensional submanifold is given by

$$
\frac{j_1 j_2}{2} + q_1 q_2 q_3 = \left(q_1 + q_2 + q_3 + \frac{1}{2}\right)\left(q_1 q_2 + q_1 q_3 + q_2 q_3 - \frac{1}{2}(j_1 + j_2)\right),
\tag{43}
$$

together with the requirement that the black hole sheet lies within the Bose-Fermi cone,[51] and also obey the positivity conditions imposed by the following equations

$$
\begin{aligned}
q_1 + q_2 + q_3 + \frac{1}{2} &> 0, \\
\frac{j_1 j_2}{2} + q_1 q_2 q_3 &> 0, \\
q_1 q_2 + q_1 q_3 + q_2 q_3 - \frac{1}{2}(j_1 + j_2) &> 0
\end{aligned}
\tag{44}
$$

(the first of these conditions is actually automatic from the requirement that the manifold lies within the Bose-Fermi cone). On this manifold, the entropy of these black holes as a function of charge is given by [35]

$$
S \equiv N^2 S_{BH}(\zeta_i) = 2\pi N^2 \sqrt{q_1 q_2 + q_2 q_3 + q_3 q_1 - (j_1 + j_2)/2}.
\tag{45}
$$

---

[51]The space of solutions to the equation (43) has multiple connected components. The requirement that solutions lie within the Bose-Fermi cone plus positivity chooses out a single physically connected component.

The formula (45) may be thought of as the gravitational prediction for the number of super-symmetric states at least on this special codimension one manifold in the space of charges. Later in this paper, we will also make use of this formula to propose a formula for the super-symmetric entropy as a function of charges, even away from the black hole sheet.

One can compute the renormalized chemical potential of these supersymmetric black holes by evaluating the temperature and chemical potential of general non-extremal black holes, and taking the limit to the SUSY manifold using (24). The final results for these chemical potentials turn out to be the real parts of the quantities $\mu_I$ [31]. In particular, these chemical potentials always obey the constraint (2), and so seem automatically well suited for computing the index.

We end this subsection with a brief aside. In §2.7 we have explained that there is a range of chemical potentials $\{\nu_i\}$ over which all bosonic SUSY letters are stable, but some fermionic letters are "unstable" to the formation of a Fermi sea. In Appendix C.10 we demonstrate that the chemical potentials on the black hole sheet never lie in this range.

## 2.13 Visualizing the black hole sheet as a cone

Through this paper we refer to the space of charges of legal Lorentzian supersymmetric black holes - i.e. solutions to (43) and (44) - as the "black hole sheet". In the next few subsections, we study the geometry of the black hole sheet - and its embedding into the Bose-Fermi cone - in some detail. In this preliminary subsection we present a rough analysis that will be justified and sophisticated in subsequent subsections.

The black hole sheet is, topologically, a $B_3$ (a solid three dimensional ball) fibered over the half line $R^+$. In this subsection we will explain roughly how the surfaces $j_1 = j_2$ and $q_i = q_j$ slice the base $B_3$ of this cone.

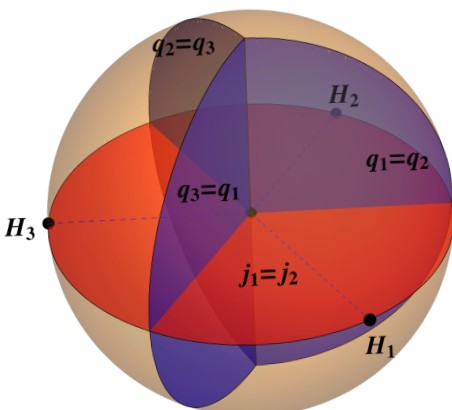

Figure 2: The black hole sheet has the topology of a $\mathbb{B}^3$ fibered over a half line $\mathbb{R}^+$. In the above figure we split the $\mathbb{B}^3$ into various regions. The (blue) planes of two equal charges split the $\mathbb{B}^3$ into three regions. The black holes in the bulk of these three regions form the core black holes of the three rank 2 DDBH phases. The black holes on the three blue surfaces similarly give rise to the three rank 4 DDBH phases and the black holes on the line where the planes meet give rise to the rank 6 DDBH phase. Similarly, the $j_1 = j_2$ (red) plane splits the Ball $\mathbb{B}^3$ into two regions. The bulk of these two regions will give rise to the two rank 2 grey galaxy phases. The black holes on the Red surface will give rise to the rank 4 grey galaxy phase.

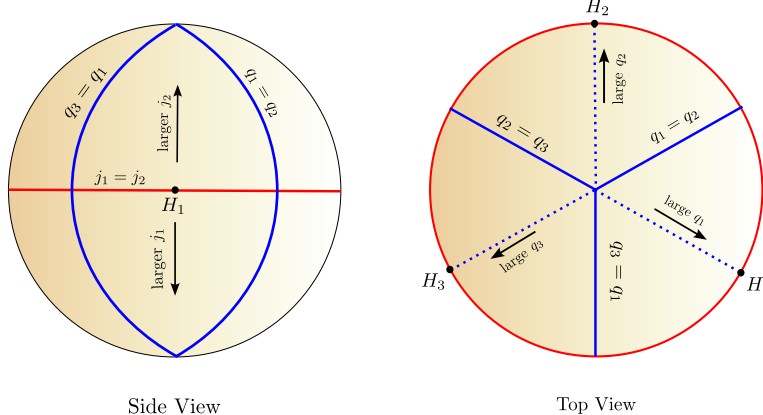

Figure 3: The side view and top view of the surface of the apple of Fig. 2. The red equator is the boundary of the red disk in Fig 2, while the blue curves are the "longitudes" at the boundary of the blue disks in Fig. 2.

For visual purposes, it is useful to think of the $B_3$ fiber as an apple, sitting on a table, with its core vertical (see Fig 2 and 3). The plane $j_1 = j_2$ - the red disk in Fig 2 - slices through the apple in a horizontal manner. The $j_1 = j_2$ submanifold of the black hole sheet is thus given by the red disk in Fig 2 fibered over $R^+$. All black holes at the centre of rank 4 grey galaxies live on this submanifold (see §3 for more about this). The boundary of this submanifold (the red curves in Fig. 3) is $S^1$ fibred over $R^+$.[52]

As shown in Fig 2, the same apple is sliced into three equal wedges by the blue half disks located along the planes $q_1 = q_2$, $q_2 = q_3$ and $q_3 = q_1$. Black holes that live at the core of rank 4 DDBHs lie along these three blue half disks. These three blue cuts meet at the core of the "apple" drawn in Fig 2. Black holes in the three different wedges (shown in Fig 2) carry distinct relative orderings of the charges $q_1, q_2, q_3$. As explained in the second of Fig. 3), $q_i$ is the largest of the three charges in the wedge that contains the point $H_i$

Black holes along vertical blue slices in Fig. 2 have (for instance) $q_1 = q_2 > q_3$ and lie at the centre of the rank 4 DDBHs.[53]

The core of the apple - the vertical axis in Fig 2 - is the codimension two submanifold of the black hole sheet that represents black holes with $q_1 = q_2 = q_3$. The core and the $j_1 = j_2$ surface meet a single point (the centre of the apple in Fig 2). The fibration of these points over $R^+$ yields the line of the simplest - so called Gutowski-Reall - supersymmetric black holes.

The boundary of the black hole sheet has the topology of a fibration of $S^2$ (the surface of the apple in Fig 2) over $R^+$.

The Bose-Fermi and bosonic cone both have the topology of a solid cone in 5 dimensions. These cones can (topologically) be viewed as fibrations of[54] $B_4$ over the half line $R^+$, with the $B_4$ shrinking to zero at the origin of $R^+$. In order to better understand the relationship between this black hole "sheet" and the bosonic and Bose-Fermi cones, in Appendix C.8 and C.6 we demonstrate that

---

[52]This surface is presented in equations in (C.6).

[53]The surface $q_1 = q_2$ extends beyond the core of the apple to its boundary at the other side. This extension is denoted by the dashed blue lines in Fig 3. On the solid blue region $q_1 = q_2 > q_3$. In the dotted blue region, on the other hand, $q_1 = q_2 < q_3$.

[54]$B_4$ is the solid ball in 4 dimensions, i.e. a filled in $S^3$.

- Apart from the three special half BPS points $H_i$, every point on the boundary of the black hole sheet has a negative value for exactly one of the five charges $Z_i$, and so (in particular) lies outside the bosonic cone.[55] Consequently, the black hole sheet slices through the bosonic cone, dividing it into two parts.[56]

- The black hole sheet lies within the Bose-Fermi cone, merely touching it on three two dimensional submanifolds (see the next subsection). Consequently, it is possible to go around the black hole sheet while staying entirely within the Bose-Fermi cone.

## 2.14 The boundary of the black hole sheet and $(1/8)^{th}$ BPS planes

The boundary of the black hole sheet is given by those points that saturate the two inequalities listed in the second and third of (44). Substituting $j_L = \frac{j_1 + j_2}{2}$ and $j_R = \frac{j_1 - j_2}{2}$, we find that this boundary surface is given by the solutions of

$$
\begin{aligned}
j_L &= q_1 q_2 + q_2 q_3 + q_3 q_1 \,, \\
j_R &= \pm \sqrt{(q_1 q_2 + q_2 q_3 + q_3 q_1)^2 + 2 q_1 q_2 q_3} \,.
\end{aligned}
\tag{46}
$$

The space described in (46) clearly consists of two sheets, glued together at the surface $j_R = 0$, i.e. $j_1 = j_2$ (the red equator in Fig. 2). From (45), it is easy to check that the entropy of black holes that lie on the boundary of the black hole sheet vanishes.

In §C.5 and §C.6 we study the boundary surface (46) (and its filling to give the full black hole sheet) in some detail. We investigate the positioning of the black hole sheet w.r.t. the boundaries of the Bose-Fermi cone, i.e. w.r.t the ten $(1/8)^{th}$ BPS planes (see the end of §2.3.2). We demonstrate that

- The black hole sheet touches the four planes $q_i + q_j = 0$ and $j_1 + j_2 = 0$ only along the three half BPS lines with charges $(q_1, 0, 0, 0, 0)$, $(0, q_2, 0, 0, 0)$ and $(0, 0, q_3, 0, 0)$ (for arbitrary positive values of $q_1$, $q_2$ and $q_3$ respectively). The black hole sheet develops a cuspy non analyticity at these touching points. These half BPS lines reduce to points on the "apple" base of the black hole sheet depicted in Fig 2. These points are named $H_1$, $H_2$ and $H_3$ in the next subsection.

- In contrast, the black hole sheet touches the six planes $q_i + j_m = 0$ along two-dimensional surfaces. On (for example) the plane $q_1 + j_1 = 0$, the equation of this "touching" surface is

$$
q_1 = 0 \,, \qquad j_1 = 0 \,, \qquad j_2 = 2 q_2 q_3
\tag{47}
$$

(analogous equations apply to all the other 5 planes). On the base of the black hole sheet, depicted in Fig 2, these surfaces reduce to curves. We study these curves in more detail in the next subsection.

## 2.15 $(1/8)^{th}$ BPS curves on the surface of the black hole apple

As we have mentioned above, while the "apple" of Fig 2 is entirely enclosed within the Bose-Fermi cone, its boundary surface touches the boundary of the Bose-Fermi on certain exceptional (two dimensional) submanifolds. As "touching submanifolds" will turn out to play an

---

[55] When $q_1$ is negative and all the other charges are positive, $\mu_2$ and $\mu_3$ are negative, while all the other chemical potentials are positive. When $j_1$ is negative and all the other charges are positive, $\omega_2$ is negative, while all the other chemical potentials are positive. (See Appendix C.11 for proof.)

[56] The relationship between the black hole sheet and the bosonic and Bose-Fermi cones may be visualized in the toy model in which we reduce the dimensionality of the bosonic cone to 3 and the dimension of the black hole sheet to 2. In this toy model, the bosonic and Bose-Fermi cones have the topology of a solid ice cream cone, and the black hole sheet has the topology of a semi infinite triangular cardboard sheet (that happens to be curved).

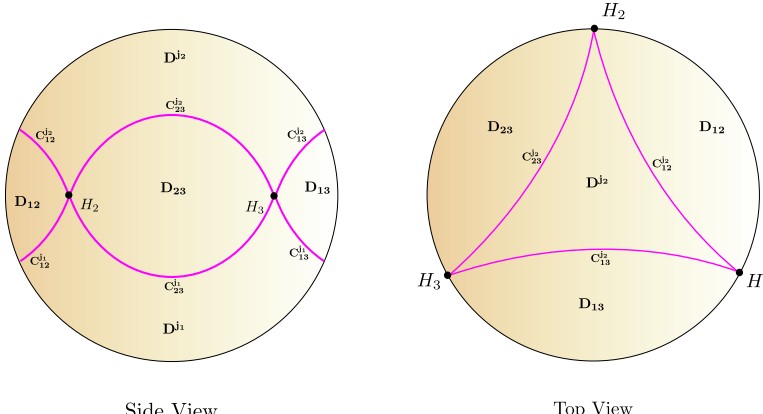

Side View          Top View

Figure 4: In this figure we depict a side and a top view of the 1/2 BPS cusps marked by $H_1, H_2, H_3$ and the $1/8^{th}$ BPS boundaries of the black hole sheet marked as $C^{j_i}_{kl}$. Here $i$ represents the nonzero angular momentum on a particular $1/8^{th}$ BPS curve and $k, l$ represent the non-zero charges on that curve. These curves divide the boundary of the black hole sheet into five separate regions marked as $D^{j_{1,2}}$ and $D_{12}, D_{23}, D_{13}$.

important role in our study of the superconformal index in §4, we study them in some detail in this subsection.

We have already seen that the boundary of the apple in Fig 2 touches the boundary of the Bose-Fermi cone at the three half BPS "cusps" marked $H_1$, $H_2$ and $H_3$ in Figs 2 and 3.[57] In Appendix C.6.3 we demonstrate that each pair of the half BPS cusps $H_i$ are also connected by a pair of $(1/8)^{th}$ BPS curves on the surface of the apple (see Fig 4). The curve that lies on the plane $q_1 + j_1 = 0$ connects the cusps $H_2$ and $H_3$ along the top surface of the apple (see Fig 4). We name this $(1/8)^{th}$ BPS curve $C^{j_2}_{23}$ (see the figure). In a similar manner, the curve that lies on the plane $q_1 + j_2 = 0$ connects $H_2$ and $H_3$ via the bottom surface of the apple, on the curve $C^{j_1}_{23}$. As depicted in Fig. 4, the two curves $C^{j_2}_{23}$ and $C^{j_1}_{23}$ together form a closed curve on the apple (with the topology of a circle). We name this closed curve $C_{23}$. The (segment type) curves $C^{j_2}_{31}$, $C^{j_1}_{31}$ $C^{j_2}_{12}$ and $C^{j_1}_{12}$, and the closed curves $C_{31}$ and $C_{12}$ are all defined in an entirely analogous manner.

As depicted in Fig 4, the closed curves $C_{12}$ and $C_{23}$ touch at the single point $H_2$ (similar statements hold with $2 \leftrightarrow 1$ and $2 \leftrightarrow 3$). Together, the three closed curves $C_{12}$, $C_{23}$ and $C_{31}$ divide the boundary of the black hole sheet into five disjoint regions. We will find it useful, below, to have names for these 5 regions. We denote the regions bounded by $C_{ij}$ and by $D_{ij}$, the upper region (i.e. the region bounded by $C^{j_2}_{12}$, $C^{j_2}_{23}$ and $C^{j_2}_{31}$ as $D^{j_2}$[58]). Similarly, we denote the lower region - i.e. the region bounded by $C^{j_1}_{12}$, $C^{j_1}_{23}$ and $C^{j_1}_{31}$ by $D^{j_1}$. All these 5 regions are displayed in Fig 4.[59]

## 2.16  Surfaces of vanishing renormalized chemical potentials

In addition to charges, the black hole sheet is characterized by its 5 renormalized chemical potentials. These chemical potentials represent directional derivatives of the entropy along

---

[57]The cusp $H_1$, on the $q_2 = q_3$ plane represents the 1/2 BPS configuration with $q_2 = q_3 = j_1 = j_2 = 0$ but $q_1 \neq 0$. The same statement, with $2 \leftrightarrow 1$, defines the cusp $H_2$, and with $3 \leftrightarrow 1$ defines the cusp $H_3$.

[58]While $C$ stands for curve, $D$ stands for Disk.

[59]Very roughly, one can think of $D^{j_i}$ as the boundary region around which $j_i$ is the largest charge. The charge $q_i$ is largest in the direction of $H_i$. $q_i$ and $q_j$ are the largest charges two $SO(6)$ charges in the region $D_{ij}$. Near $H_i$, $q_i$ is larger than $q_j$, and vice verca.

the black hole sheet, and will play an important in our analysis of the indicial phase diagram below. The five distinguished codimension one surfaces on the black hole sheet - along which the (real parts of the) chemical potentials $\nu_i$ $i = 1 \ldots 5$ respectively vanish - will be of particular importance. We study these surfaces in this subsection.

These five surfaces are described by the following equations (refer to Appendices C.12 and C.13 for the detailed derivation):[60]

$$
\begin{aligned}
\mu_1 = 0 \quad &\implies \quad \frac{j_1 + j_2}{2} = \frac{(q_2 + q_3)(q_1 + 2q_1^2 - 2q_2q_3)}{1 + 2q_1}, &&\text{and } 1 \leftrightarrow 2, 3, \\
\omega_1 = 0 \quad &\implies \quad j_1 = -2(1 + q_1 + q_2 + q_3)j_2 + 2(q_1q_2 + q_2q_3 + q_3q_1), &&\text{and } 1 \leftrightarrow 2.
\end{aligned}
\tag{49}
$$

As indicated above, the formula for surfaces with $\mu_{2,3} = 0$ can be obtained by replacing $q_1$ with $q_{2,3}$ in the first line. The formula for surfaces with $\omega_2 = 0$ can be obtained by replacing $j_1$ with $j_2$ in the second line. We will, once again, find it useful to have names for these surfaces. We will use the symbol $S^{\nu_i = 0}$ to denote the surface (on the black hole sheet) along which $\nu_i = 0$. As we will see below, the surfaces $S^{\nu_i = 0}$ are not closed, but have boundaries (that themselves lie on the boundary of the black hole sheet).

It is easy to check from (49) that $\mu_2$, $\mu_3$ and $\omega_2$ all vanish along the curve $C_{23}^{j_2}$.[61] Similar statements hold for the touching other planes of the form $j_i + q_m = 0$ and the black hole sheet (for instance, $\mu_1$, $\mu_3$ and $\omega_1$ all vanish along $C_{13}^{j_1}$). Moreover, it is possible to check that the curves $C_{mn}$ are the only locations at which any of the surfaces $\nu_i = 0$ touch the boundary of the black hole sheet (see §C.11).

The boundary[62] of the surface $S^{\omega_i = 0}$ ($i = 1, 2$) coincides with the boundary of the disk $D^{j_i}$ (see previous subsection for terminology).[63] The boundary of the surface $S^{\mu_1 = 0}$, is the union of $C_{12}$ and $C_{13}$ (see the previous subsection for terminology). The sheets $S^{\nu_i = 0}$ divide the black hole into the "unstable" part (where at least one of the $\nu_i < 0$) and "stable"[64] part where $\nu_i \geq 0$ $\forall i = 1 \ldots 5$.

In the rest of this section we describe the shape of these surfaces - and their dissection of the black hole sheet - in more detail.

### 2.16.1 $S^{\omega_i = 0}$

As we have explained above, the boundary of the sheet $S^{\omega_i = 0}$ coincides with the boundary of the disk $D^{j_i}$. In fact, the sheet $S^{\omega_i = 0}$ turns out to be homologous to the disk $D^{j_i}$. We can think of $S^{\omega_i = 0}$ as the boundary of what is left over after we take an "ice cream scoop" out of $D^{j_i}$ region of the black hole apple. The part of the black hole sheet that lies between $S^{\omega_i}$ and $D^{j_i}$

---

[60]More precisely, we have

$$
\begin{aligned}
\mu_1 < 0 \quad &\iff \quad \frac{j_1 + j_2}{2} < \frac{(q_2 + q_3)(q_1 + 2q_1^2 - 2q_2q_3)}{1 + 2q_1}, &&\text{plus } 1 \leftrightarrow 2, 3, \\
\omega_1 < 0 \quad &\iff \quad j_1 < -2(1 + q_1 + q_2 + q_3)j_2 + 2(q_1q_2 + q_2q_3 + q_3q_1), &&\text{and } 1 \leftrightarrow 2.
\end{aligned}
\tag{48}
$$

[61]Recall this is where the black hole sheet touches the plane $q_1 + j_1 = 0$, at $j_1 = q_1 = 0$ and on the curve $j_2 = 2q_2q_3$.

[62]The surfaces $\nu_i = 0$ are three dimensional submanifolds of the black hole sheet. The boundary of these surfaces is thus two dimensional, and its projection to the "apple" base is one dimensional. These boundaries occur on the boundary of the black hole sheet.

[63]In other words, this boundary is given by the union of $C_{12}^{j_i}$, $C_{23}^{j_i}$ and $C_{31}^{j_i}$.

[64]We use the word "stable" to describe this part of the black hole sheet to remind the reader that the grand-canonical ensemble would be stable (convergent) at the values of black hole chemical potentials encountered at stable points on the black hole sheet (see (34) ), but would be unstable (divergent) at the values of the chemical potentials encountered at other points on the black hole sheet. We emphasize that all supersymmetric black holes represent the configurations of largest entropy (at least among known solutions) at the given values of charges, and so everywhere appear to represent stable configurations in the microcanonical ensemble.

has $\omega_i < 0$; the rest of the black hole sheet has $\omega_i > 0$. $\omega_1$ and $\omega_2$ are never simultaneously negative (see Appendix C.12 for a proof). As a consequence, the $\omega_1$ and $\omega_2$ scoops are non overlapping, and the surfaces $S^{\omega_1=0}$ and $S^{\omega_2=0}$ do not intersect.[65]

In Appendix C.13 we also prove that $\omega_i$ (where $i$ is either 1 or 2) and $\mu_j$ (where $j = 1, 2, 3$) are also never simultaneously negative on the black hole sheet. As a consequence, the surfaces $S^{\omega_1=0}$ and $S^{\omega_2=0}$ do not intersect each other.[66]

In the rest of this section we discuss the shape of the surface $S^{\mu_i=0}$. It turns out that this surface extends into the black hole "apple" in a manner that is different at small and large charges (i.e. depending on whether the "apple" is located near to - or far away from- the apex of the black hole sheet cone). We study these two cases in turn.

### 2.16.2 $S^{\mu_i=0}$: Small charges

At low values of black hole charges (i.e. for apples that are not too far from the tip of the black hole sheet),[67] the sheet $S^{\mu_1=0}$ can be thought of as the boundary of the part of the apple that is left over after we take two ice cream scoops out of the black hole apple. The first scoop is taken out of the part of the apple bounded by $D_{12}$, so its boundary is homologous to $D_{12}$. The second scoop is taken out of the part of the apple bounded by $D_{13}$, so its boundary is homologous to $D_{13}$. These two scoops do not intersect each other.[68] A similar discussion applies with $1 \leftrightarrow 2$ and $1 \leftrightarrow 3$.

Notice that the given disk $D_{12}$ is associated with two separate scoops; one associated with the surface $S^{\mu_1=0}$, and the second associated with the surface $S^{\mu_2=0}$. Though these two scoops are homologous, they are not identical. The $S^{\mu_1=0}$ scoop is deeper near $H_1$ and shallower near $H_2$,[69] while the $S^{\mu_2=0}$ scoop is deeper near $H_2$ and shallower near $H_1$. Of course similar remarks apply to the surface $D_{23}$ and $D_{31}$ with $1 \leftrightarrow 3$ and $2 \leftrightarrow 3$.

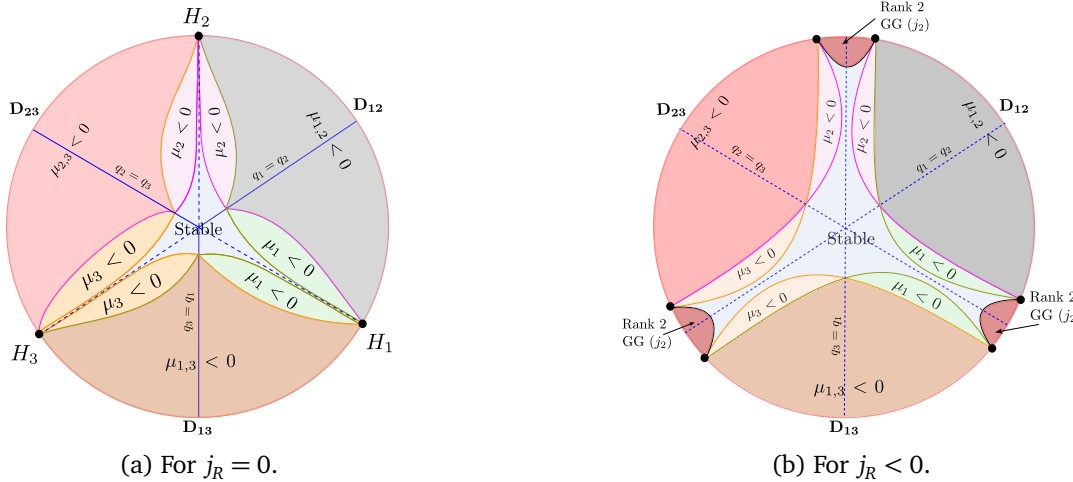

(a) For $j_R = 0$.

(b) For $j_R < 0$.

Figure 5: Cross section of the $\mathbb{B}^3$ (apple) at various values of $j_R$ and $\frac{q_1+q_2+q_3}{3} + j_L < \frac{1}{6}$. The blue region represents stable black holes, i.e. black holes with $v_i > 0$ for all $i$. Every other region has at least one $\mu_i < 0$, as depicted in the diagram. The green, purple and orange curves are, respectively, intersections of the $S^{\mu_1=0}$, $S^{\mu_2=0}$ and $S^{\mu_3=0}$ with constant $j_R$ cross-sections.

---

[65] The boundary of these sheets touch at the points $H_i$.

[66] Of course these sheets touch at boundaries. For instance, $C_{12}^{j_2}$ lies on the boundary of all of $S^{\omega_2=0}$, $S^{\mu_2=0}$ and $S^{\mu_3=0}$ (and similar statements hold with permutation of indices).

[67] See §4 for quantification of this statement.

[68] Except for the fact that they share a common boundary point, namely $H_1$.

[69] See the previous subsection for notation.

In Fig 5a we present a sketch of the horizontal equatorial plane - the surface $j_1 = j_2$ - of a small charge "black hole apple" (see Fig 2). The blue region in Fig 5a is the stable part of the black hole apple. All other regions in Fig 5a correspond to black holes that are "unstable", because at least one $\mu_i$ ($i = 1 \ldots 3$) is less than zero (see the diagram for details). Note that the stable blue region extends all the way to the points $H_1$, $H_2$ and $H_3$. The intersection of the sheets $S^{\mu_1=0}$, $S^{\mu_2=0}$ and $S^{\mu_3=0}$ with our horizontal cut, are, respectively, depicted by the green, pink and orange curves in Fig. 5a. In Fig. 5b we present a second horizontal cut of the black hole apple, this time "above" the equitorial plane (i.e. at a value of $j_2 > j_1$). Note that this cross section cuts through the surface $S^{\omega_2=0}$, and so includes regions of the black hole sheet with $\omega_2 < 0$.

### 2.16.3 $S^{\mu_i=0}$: Large charges

At large values of charges (when the black hole "apple" is located far from the vertex of the black hole cone), the sheet $S^{\mu_1=0}$ has a different nature; it turns out to be topologically a tube (or a hollow cylinder), whose two circular boundaries ends respectively lie on the curves $C_{12}$ and $C_{13}$. This tube can be thought of as forming because the two ice cream scoops (of the previous section) now overlap. The tube is "folded" so that its boundaries touch along a particular line. See Appendix C.7 for a much more detailed description of the geometry of this tube.

We have, so far, focused on the tube bounded by $S^{\mu_i=0}$ at $i = 1$. Of course the black hole sheet hosts two additional such tubes corresponding to $i = 2$ and $i = 3$. The (upper half of) stable region of the black hole apple consists of a set of three arches - with (point) bases at the three points $H_1$, $H_2$ and $H_3$. These arches extend towards the centre of the apple as they rise, until they eventually meet up and merge into a central column near the central axis of the apple. The lower half of the stable part of the black hole apple is the mirror reflection of the upper half.

As an aid to visualization, we have sketched three separate horizontal cross sections of the black hole apple in this case. The first of these is the equatorial cross section - i.e. the horizontal cross section at $j_1 = j_2$. This takes the form depicted in Fig. 6a. Once again the blue region in both figures refers to stable black holes; the remaining regions are all unstable in one way or another. Note that the stable blue region does not, in this case, extend all the way to the points $H_i$.

In Figs. 6b and 6c we plot successively higher horizontal cross sections of the black hole apple. The cross sections of Fig. 6 cut through the three pillars of the arch that makes up the stable part of the black hole sheet along the three peripheral blue blobs in the figure.[70] It also cuts through the surface $S^{\omega_2=0}$. The cross section of Fig. 6c is taken at a height that is large enough so that the three arches have all merged,[71] so the blue (stable black hole region) in this cross section is now connected.

The curve $C_{23}^{j_2}$ represents the end point of three different surfaces, namely the surface $S^{\omega_2=0}$, the surface $S^{\mu_2=0}$ and the surface $S^{\mu_3=0}$. See §C.6.4 for a quantitative description of these three sheets in the neighbourhood of the curves $C_{23}^{j_2}$.

---

[70]Each of these blobs is the analogue of the white region in the second of Fig. 19).

[71]Locally, the stable parts of the black hole sheet take the form of the white region in the third of Fig. 19.

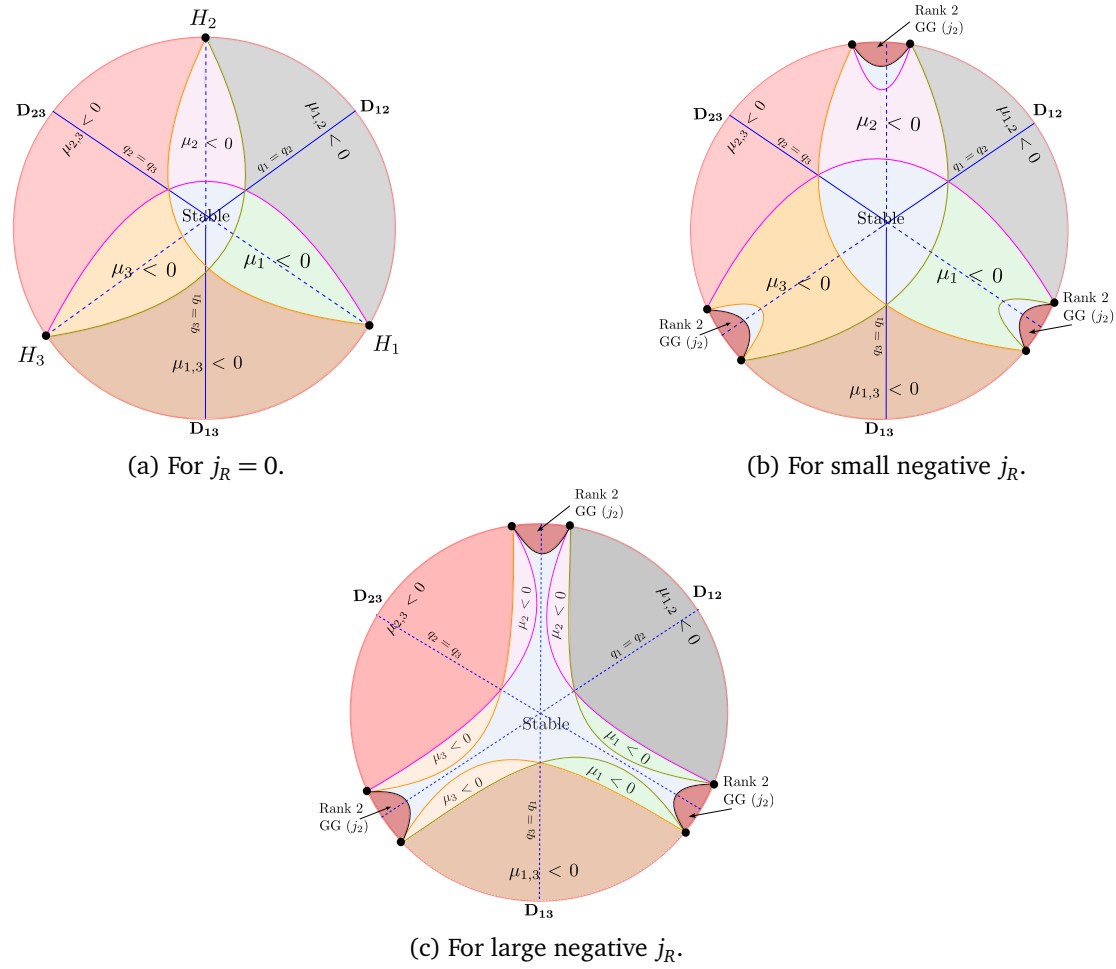

(a) For $j_R = 0$.

(b) For small negative $j_R$.

(c) For large negative $j_R$.

Figure 6: Cross sections of the $\mathbb{B}^3$ (apple) at various values of $j_R$ and $\frac{q_1+q_2+q_3}{3}+j_L > \frac{1}{6}$. The blue region represents stable black holes, i.e. black holes with $\nu_i > 0$ for all $i$. Every other region has at least one $\mu_i < 0$, as depicted in the diagram. The green, purple and orange curves are, respectively, intersections of the $S^{\mu_1=0}$, $S^{\mu_2=0}$ and $S^{\mu_3=0}$ with constant $j_R$ cross-sections.

# 3 Supersymmetric entropy from the dressed concentration conjecture

In the introduction we have presented our dressed concentration conjecture, which - roughly speaking - postulates that dressing supersymmetric black holes with a supersymmetric "gas" yields new legal supersymmetric configurations. The dressed concentration conjecture leads to a definite prediction for the large $N$ spectrum of supersymmetric states in $\mathcal{N} = 4$ Yang-Mills theory. In this section, we spell this prediction out in detail.

## 3.1 A conjecture for the supersymmetric entropy as a function of charges

It follows from the dressed concentration conjecture (10),[72] that the large $N$ supersymmetric entropy of $\mathcal{N} = 4$ Yang Mills (at least at large $\lambda$) is given, as a function of charges, $N^2 S_{BPS}(\zeta_i)$ by

---

[72]The inequality in (10) arises from the fact (see §2.10 ) that the charges of the supersymmetric gas always lie within the bosonic cone (§2.3.1) when all charges $Z_i$ are large compared to unity, and so, in particular, when charges of order $N^2$.

- First constructing a bosonic cone with vertex at each point on the black hole sheet. The interior of the cone centred about any particular SUSY black hole represents the set of all charges carried by grey galaxy or DDBHs solutions (or a combination of the two) that have the given black hole at their centre.

- Identifying the collection of all black holes whose cone includes the charge $\{\zeta_i\}$ of interest.

- Maximizing over the entropy $N^2 S_{BH}(\zeta_i)$ of these black holes.

We implement this procedure in the rest of this section.

## 3.2 The extensive entropy region

In §2.3.1, §2.3.2, §C.2.1 and §C.2.2, we have already carved out four interesting subregions of charge space. The discussion above prompts us to define a fifth "Extensive Entropy Region (EER)" as follows:

- The Extensive Entropy Region (EER) (of charge space) is the union of the interior of the bosonic cones with a vertex at each point on the black hole sheet (43).

Our first conjecture above implies that $\mathcal{N} = 4$ Yang-Mills theory has a supersymmetric entropy of order $N^2$ (i.e. an entropy that takes the form $N^2 f(\zeta_i)$) in (and only in) the EER.[73] As the black hole sheet (43) passes through the origin in charge space, it follows that all of the bosonic cone (see §2.3.1) lies within the EER. As the black hole sheet is not contained entirely in the bosonic cone, it follows that the Extensive Entropy Region extends outside the bosonic cone.[74]

Note that charges outside the EER also lie outside the bosonic cone, and so do not host pure gas states (see §2.10). It is possible that supersymmetric states simply do not exist at charges that lie (well) outside the EER.[75]

Every point on the black hole sheet is the seed for one dominant DDBH phase, and one dominant grey galaxy phase. Correspondingly the EER has two separate parts; the grey galaxy part of the EER and the DDBH component of the EER.

## 3.3 An algorithm to compute supersymmetric phase and entropy

As we have explained in §3.1, the dressed concentration conjecture, turns the problem of evaluating the large $N$ cohomology into a problem of constrained maximization. In Appendix D, we explain one approach to the solution of this problem. Our solution is given by starting with any charge in the EER, identifying a core black hole that lies on the black hole sheet for that charge, and then flowing on the black hole manifold, along a vector field defined by chemical potentials. The flow in question is similar to gradient flow; it is designed to ensure that the black hole entropy always increases along the flow. The flow is constrained by hard walls (a wall is reached when the flow reaches a point on the black hole manifold such that the charge under study lies on the boundary of the bosonic cone from that point). On hitting a wall, the flow continues parallel to the wall along the "gradient flow" vector projected orthogonal to the wall. Points at which the flow vector vanishes represent (as yet local) maxima of entropy,

---

[73]The entropy vanishes at the boundary of EER since these points lie on the bosonic cone originating from the zero entropy black holes.

[74]The Extensive Entropy region does, of course, lie entirely within the Bose-Fermi cone.

[75]We thank A. Zaffaroni for discussion on this point.

and so potential phases of our system. In Appendix D, we demonstrate that these phases are always one of[76]

- A rank 6 DDBH in which the core black hole carries $q_1 = q_2 = q_3$.

- One of 3 possible rank 4 DDBH solution, in which the core black hole carries either $q_1 = q_2$ or $q_2 = q_3$ or $q_3 = q_1$.

- A rank 4 grey galaxy in which the core black hole carries $j_1 = j_2$.

- Any one of the 3 possible rank 2 DDBH or the 2 possible rank 2 grey galaxy solutions.

We also explain that the map from charges to phases is one to one: no given collection of charge $\{\zeta_i\}$ belongs to more than one phase.[77] In the rest of this subsection, we present an algorithm that may be used to determine which phase any given charge $\zeta_i$ lies in, and also to determine the supersymmetric entropy associated with this $\zeta_i$.

We assume that $\zeta_i$ lies inside the Bose-Fermi cone (if this is not the case then the charge in question belongs to no phase, and the supersymmetric entropy vanishes). We also assume, without losing generality, that $q_1 \leq q_2 \leq q_3$ and $j_1 \leq j_2$.

### 3.3.1 Step 1

First, determine which category the set of charges belongs to by evaluating the following relation:

$$\mathcal{P}(q_i, j_i) = \frac{j_1 j_2}{2} + q_1 q_2 q_3 - \left( q_1 + q_2 + q_3 + \frac{1}{2} \right) \left( q_1 q_2 + q_1 q_3 + q_2 q_3 - \frac{1}{2}(j_1 + j_2) \right). \quad (50)$$

There are three possibilities: $\mathcal{P}(q_i, j_i) > 0$, $\mathcal{P}(q_i, j_i) < 0$ or it vanishes.

### 3.3.2 Step 2

Next, analyze one of the three cases based on the result from Step 1.

- **Case 1: $\mathcal{P}(q_i, j_i) > 0$**
  In this case, the solutions are either rank 4 grey galaxy, rank 2 grey galaxy, or the point is outside of the EER (See Appendix D.9 for the proof). With given $q_1, q_2, q_3$,

  1. if there exists a $j$ such that $0 < j < j_1, 0 < j < j_2$ and it satisfies $\mathcal{P}(q_i, j) = 0$, it belongs to the rank 4 grey galaxy, and the entropy of the corresponding black hole computes $S_{BPS}(\zeta_i)$.

  2. If the above condition is not met, but there exists a $j$, $j_1 < j < j_2$ such that $(q_1, q_2, q_3, j_1, j)$ satisfies $\mathcal{P}(q_i, j_1, j) = 0$, it belongs to the rank 2 grey galaxy with the gas carrying $j_2$, and the entropy of the corresponding black hole computes $S_{BPS}(\zeta_i)$.

---

[76]In the discussion above we use the term DDBH and grey galaxy to denote a solution in which a central black hole (with charges as described above) is surrounded by (dual giant) charge or (graviton) angular momentum, respectively. As always, the rank of solutions denote the rank of the "graviton" $SO(6)$ charge or $SO(4)$ angular momentum. We emphasize that, in contrast with the situation in the next section, §4, the central black holes can carry arbitrary charges, and -in particular - are not restricted by the requirement that the relevant chemical potentials vanish. All chemical potentials are allowed to take arbitrary values: positive, zero or negative. Consequently, the black holes that lie at the centre of the DDBHs and grey galaxies of this section can be either "stable" or "unstable" in the terminology of §2.16. Irrespective of which kind of black holes they host, these solutions are always entropy maximizing in the microcanonical ensemble (at least within the space of known solutions), and so are always stable in this ensemble.

[77]Consequently the microcanonical phase diagram exhibits no first order phase transitions.

3. If neither of the conditions above are satisfied, the point is outside the EER, and the supersymmetric entropy at the corresponding charge is sub-extensive i.e. is not of order $N^2$.

- **Case 2:** $\mathcal{P}(q_i, j_i) < 0$
  In this case, the solution is either a rank 6 DDBH, a rank 4 DDBH, a rank 2 DDBH, or the point is outside of the EER (See Appendix D.9 for the proof). With given $j_1$ and $j_2$,

  1. if there exists $0 < q < q_1, 0 < q < q_2, 0 < q < q_3$ such that $\mathcal{P}(q, j_1, j_2) = 0$, it belongs to the rank 6 DDBH and the entropy of the corresponding black hole computes $S_{BPS}(\zeta_i)$.

  2. If the above is not met, but there exists $q_1 < q < q_{2,3}$ such that $\mathcal{P}(q_1, q, q, j_1, j_2) = 0$, it belongs to the rank 4 DDBH and the entropy of the corresponding black hole computes $S_{BPS}(\zeta_i)$.

  3. If the above is not met, but there exists $q_2 < q < q_3$ such that $\mathcal{P}(q_1, q_2, q, j_1, j_2) = 0$, it belongs to the rank 2 DDBH and the entropy of the corresponding black hole computes $S_{BPS}(\zeta_i)$.

  4. If neither of the above conditions is satisfied, the charges lie outside of the EER, and the entropy at the corresponding charge is sub extensive.

- **Case 3:** $\mathcal{P}(q_i, j_i) = 0$
  If $\frac{j_1 j_2}{2} + q_1 q_2 q_3 > 0$, the microcanonical phase is described by a BPS black hole. Else, outside of the EER.

The algorithm presented in this subsection is summarized in Fig. 7.

Note that the phase transition from a grey galaxy phase to a DDBH phase always occurs through a pure black hole configuration. In other words, the black hole sheet divides EER into two regions: one with DDBH phases and the other with grey galaxy phases.

## 3.4 The boundary of the EER

We have argued above that the microcanonical phase diagram of Supersymmetric Yang-Mills theory fills the EER with an intricate patchwork of one rank 6, three rank 4 and three rank 2 DDBH phases together with two rank 2 grey galaxy phases. Each phase is dominated by a solution of Einstein equations that has a central vacuum black hole dressed with "gravitons". The boundary of the EER is made up of those solutions whose central black holes lie at the boundary of the black hole sheet.

As we see from Fig. 2, the $j_1 = j_2$ slice of the black hole meets the boundary of the black hole sheet in a codimension one submanifold (this submanifold is depicted by red curves in Fig. 3). Rank 4 grey galaxies that emanate from this boundary (C.6) surface make up one component of the boundary of the grey galaxy part of the EER. The other two components of this part of the boundary consist of rank 2 grey galaxies that are seeded, respectively, by black hole on the upper and lower "curved surfaces" of the apple (the regions above and below the red line in the first of Fig. 3).

Let us now turn to the boundary of the DDBH component of the EER. The $q_1 = q_2 = q_3$ "core" meets the surface of the apple at two points (see Fig 2: one of these points is the central meeting point of the blue lines in the second of Fig. 3). Accounting for the fibration over $R^+$, these two points form two continuous curves,[78] the boundary of the fibration of the core

---

[78]See fig. 21. One of the two curves described here is the red curve in that figure. The other point is a similar curve with $j_1$ and $j_2$ exchanged.

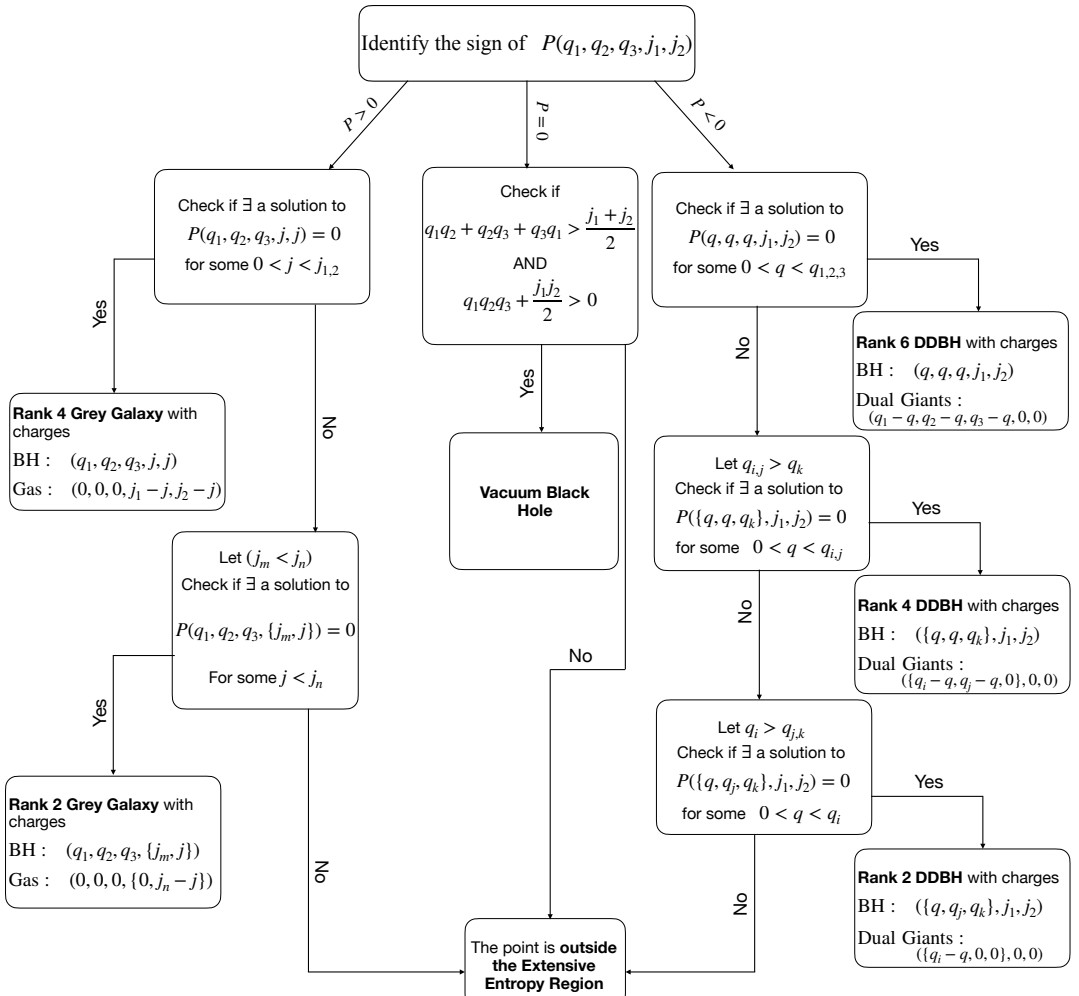

Figure 7: A flowchart of the algorithm to identify the dominant supersymmetric phase of the $\mathcal{N} = 4$ SYM in the microcanonical ensemble at charges $(q_1, q_2, q_3, j_1, j_2)$. The non-linear charge constraint $\mathcal{P}(q_i, j_m)$ is defined in (50). The subscripts take the values $i, j, k = 1, 2, 3$ and $m, n = 1, 2$. The charges written in curly braces ({}) should be arranged based on the subscripts in the following order: $q_1, q_2, q_3$ for charges and $j_1, j_2$ for angular momentum.

"interval" over $R^+$. These boundary points yield a rank 6 DDBH component of the DDBH part of the boundary of the EER.

The surfaces $q_i = q_j$ meet the boundary of the apple on the "longitudes" that lie the boundaries of the half disks in Fig. 2 (these are be blue curves in Fig. 3). The corresponding black holes lie at the centre of the three distinct rank 4 components of the boundary of the DDBH part of the EER.

Finally, each of the three curved surfaces (of the three apple wedges, the regions between any of the two blue half disks in Fig. 2), form the core of the last 3 rank 2 DDBH components of the boundary of the EER.

Each component of the boundary of the EER described above is of unit codimension in five dimensional charge space. This comes about as follows. The rank 2, rank 4 and rank 6 components of the boundary of the black hole sheet are, respectively, of codimension zero, 1

and 2 on the boundary of the black hole sheet. As the phases seeded by these black holes, respectively have one, two or three additional parameters (the independent charges of the "hair" in these phases), the number of parameters in each of these components is one more than the number of parameters on the boundary of the black hole sheet, and so equals 4 (which is codimension 1 in five dimensional charge space) in every case.

## 4 The microcanonical index from the unobstructed saddle conjecture

The unobstructed saddle conjecture, together with results of §3, yield a prediction for the microcanonical version of the superconformal index. We elaborate on this prediction in this section.

### 4.1 Indices with extensive entropy

As we have explained in the introduction, index lines are parameterized by charges $\zeta_i$, modulo shifts along the vector (8). The space of indices is constrained by the $(1/8)^{th}$ BPS inequalities[79]

$$\zeta_i + \zeta_m > 0, \quad \forall i = 1 \ldots 3, \quad \forall m = 4, 5. \tag{51}$$

Indices vanish if they fail to obey any of (51). In this subsection, we demonstrate that the condition (51) is also sufficient to guarantee that the corresponding index (somewhere) passes through the EER, and so has extensive indicial entropy.

First recall that an index line is parameterized by the five charges $\zeta_i$ modulo shifts

$$(\zeta_1 + x, \zeta_2 + x, \zeta_3 + x, \zeta_4 - x, \zeta_5 - x). \tag{52}$$

The microcanonical data of an index line supplies us with differences between - and so the relative ordering of - the three $\zeta_i$ with $i = 1, 2, 3$. Similarly it supplies us with differences between - and so ordering of the two $\zeta_m$ with $m = 4, 5$.[80] Let us suppose that $\zeta_1 \le \zeta_2 \le \zeta_3$, and $\zeta_4 \le \zeta_5$. In this case, the index line intersects the boundary of the Bose-Fermi cone on the surfaces $\zeta_1 + \zeta_2 = 0$ and $\zeta_4 + \zeta_5 = 0$, respectively, at the points

$$\left( -\frac{\zeta_2 - \zeta_1}{2}, \frac{\zeta_2 - \zeta_1}{2}, \frac{2\zeta_3 - \zeta_1 - \zeta_2}{2}, \frac{2\zeta_4 + \zeta_1 + \zeta_2}{2}, \frac{2\zeta_5 + \zeta_1 + \zeta_2}{2} \right), \quad \text{and}$$
$$\left( \frac{2\zeta_1 + \zeta_4 + \zeta_5}{2}, \frac{2\zeta_2 + \zeta_4 + \zeta_5}{2}, \frac{2\zeta_3 + \zeta_4 + \zeta_5}{2} - \frac{\zeta_5 - \zeta_4}{2}, \frac{\zeta_5 - \zeta_4}{2} \right). \tag{53}$$

The segment of the Index line between these two points lies within the Bose-Fermi cone; all other points on this line lie outside it. Notice that the $x$ value (see (52)) of the second point in (53) differs from the $x$ value of the first point (53) by $\Delta x_{max}(\zeta) = \frac{\zeta_1 + \zeta_2 + \zeta_4 + \zeta_5}{2} \ge 0$.[81]

Now consider a point on the same index line, whose $x$ value is larger than that of the first point (see (53)) by $\Delta x$ given by $\delta x = \frac{\zeta_2 - \zeta_1}{2} \le x_{max}(\zeta)$.[82] The corresponding point on the index line has charges

$$(0, \zeta_2 - \zeta_1, \zeta_3 - \zeta_1, \zeta_4 + \zeta_1, \zeta_5 + \zeta_1). \tag{54}$$

---

[79]These are 6 of the 10 inequalities (22). The other four inequalities (namely $q_i + q_j \ge 0$ and $j_1 + j_2 \ge 0$) are on charges that are not constant along the index line. As we will see below, the role of these other four inequalities is to demarcate an interval of nontriviality along each index line (not to constrain the space of indices).

[80]More generally, the 10 quantities, $\zeta_i - \zeta_j$, $\zeta_m - \zeta_n$ and $\zeta_i + \zeta_m$ ($i, j = 1 \ldots 3$, $m, n = 4, 5$) are all constant along index lines. Linear combinations of these 10 quantities create a four dimensional space, the space of Index lines.

[81]The inequality follows from the conditions (51)).

[82]The last inequality follows because $x_{max} - x = \frac{\zeta_4 + \zeta_5 + 2\zeta_1}{2} > 0$ (using (51)).

As $\zeta_2 \geq \zeta_1$ and $\zeta_3 \geq \zeta_1$, (and using (51)), it follows that all four nonzero entries in the charges (54) are positive. Consequently, the charges (54) lie in the bosonic cone - and also in the EER.[83] We thus see that the conjectures of this paper, in particular, predict that every microcanonical Index line that obeys (51) has an Indicial Entropy of order $N^2$, and that this is true whether or not the corresponding index line intersects the black hole sheet.[84]

## 4.2 Dominant phases along an indicial line

Consider any index line labeled by $\{\zeta_i\}$ modulo the shifts (52). As the parameter $x$ varies from its smallest to its largest values (53), the points of the index line transit between regions outside the EER and regions that traverse through one of the 3 grey galaxy or 7 DDBH phases. Some index lines also intersect the black hole sheet at one particular value of $x$.

Recall that every grey galaxy or DDBH phase consists of a central black hole surrounded by charged or rotating matter, and that the leading order entropy of the grey galaxy/ DDBH simply equals that of its central black hole. For the purposes of tracking the entropy of solutions, therefore, we are interested only in the central black hole content of the grey galaxy / DDBH we happen to be in. For this reason we define

- The shadow of a grey galaxy / DDBH solution to be the point on the black hole sheet that describes its central black hole.

In Appendix E, we study how the shadow black hole - and its entropy - varies as $x$ is varied along a given index line (see (52)). In particular, in that Appendix, we demonstrate that

- When the index line intersects the black hole sheet at a "stable" point (i.e. a point at which all $\nu_i \geq 0$), then the entropy along the index line is always maximized at this intersection.

- If the index line intersects the black hole sheet at a point where one or more chemical potentials are negative, or if the index line fails to intersect the black hole sheet, then there is exactly one point on the shadow of the index line at which the chemical potential (relevant to the phase that the index line is then passing through) vanishes. The entropy along the index line is maximized at this point.[85]

- Since no point on the black hole sheet has $\omega_1 = \omega_2 = 0$ or $\mu_1 = \mu_2 = \mu_3 = 0$ (see §2.16) it follows that the maximum of the entropy along the index line never occurs in a rank 4 grey galaxy or rank 6 DDBH phase. The 9 possible indicial phases consist of the black hole phase[86] three rank 2 DDBH phases, three rank 4 DDBH phases and two rank 2 grey galaxy phases.

As we have explained above, the microcanonical index vanishes unless $q_i + j_m > 0$, for $i = 1 \ldots 3$ and $m = 1 \ldots 2$. When one of these inequalities is saturated, the corresponding index line intersects the EER at only a single point. When one of the inequalities is "just obeyed" (e.g. if $q_1 + j_1 = \epsilon$ where $\epsilon$ is small), the portion of the index line that passes through

---

[83]This follows, as the bosonic cone originating from a very small black hole coincides with the full bosonic cone itself.

[84]This illustrates a key consequence of our conjecture: the indicial entropy is not always given by the entropy of the intersecting black hole; see below.

[85]So the maximum entropy is attained if (for instance) the index line has a segment in a rank 4 DDBH phase with $q_1 = q_2$ and the shadow black hole passes through a point with $\mu_1 = \mu_2 = 0$. Or if the index line has a segment in a rank 2 grey galaxy phase (with $j_1 > j_2$) and the shadow black hole passes through a point with $\omega_1 = 0$. etc.

[86]Note that the microcanonical indicial phase diagram has a vacuum black hole phase. In the microcanonical phase diagram presented in §3, in contrast, configurations with a pure vacuum black hole represent a boundary between DDBH and grey galaxy phases.

the EER is also small (it turns out to be of order $\sqrt{\epsilon}$ in the example above). As an illustration of the ideas presented in this section, in Appendix F, we present a detailed analysis of such index lines. Working at leading order in $\epsilon$, we track the passage of the index line through the EER for every value of the other three indicial charges $q_2 + j_1$, $q_3 + j_1$ and $j_1 - j_2$, and in particular determine the dominant phase as a function of indicial charges. The explicit results of Appendix F are a quantitative illustration of the points in this subsection and the next.

## 4.3 Algorithm to determine the index as a function of indicial charges

We are now in a position to formulate an algorithm to determine the phase a given collection of indicial charges lie in, and also the indicial entropy as a function of indicial charges $\{\zeta_i\}$ modulo shifts,

$$(\zeta_1 + x, \zeta_2 + x, \zeta_3 + x, \zeta_4 - x, \zeta_5 - x). \tag{55}$$

### 4.3.1 Checking the (1/8) BPS conditions

First check whether (55) obeys all the inequalities (51). The charges $\{\zeta_i\}$ lie outside the indicial cone and the index vanishes if even one of (51) fails.

If, on the other hand, all of (51) are obeyed we proceed to the next subsubsection.

### 4.3.2 The stable black hole phase

We insert the charges

$$(\zeta_1 + x, \zeta_2 + x, \zeta_3 + x, \zeta_4 - x, \zeta_5 - x), \tag{56}$$

into (43) and solve for $x$ as a function of the indicial charges $\{\zeta_i\}$. A solution is legal only if $q_i$ in (56) obey the inequalities (44).[87] If such a solution exists, we plug it into (41) to determine the chemical potentials $\nu_i$ ($\mu_i$ and $\omega_i$). If (the real part of) all these chemical potentials is positive, then our index lies in the stable black hole phase, and the indicial entropy is obtained by plugging the solution of (56) into (45).

On the other hand, if a legal solution to (45) (with $\text{Re}(\nu_i) \geq 0$ $\forall i$) does not exist, then the index is not in the stable black hole phase, and we proceed to the next subsubsection.

### 4.3.3 Grey galaxy phases

If $\zeta_4 - \zeta_5$ vanishes (i.e. if $j_1 = j_2$), the index is not in a grey galaxy phase and we proceed to the next subsubsection.

If $\zeta_4 - \zeta_5 = j_1 - j_2 \neq 0$, we assume $j_1 - j_2 > 0$ (the analysis of $j_2 > j_1$ proceeds identically). The index is in a grey galaxy phase if and only if there exists a $\omega_1 = 0$ central black hole with charges

$$(\zeta_1 + x, \zeta_2 + x, \zeta_3 + x, \zeta_4 - x - y, \zeta_5 - x), \quad y > 0, \tag{57}$$

for some any real value of $x$[88] and any positive value of $y$. Here $y$ is the $j_1$ charge of the grey galaxy gas. In order to determine whether these conditions are met, we plug (57) into (43) and the second of (49), and solve for $x, y$ as functions of $\{\zeta_i\}$. If such a solution exists with $y > 0$, and if it obeys all of the inequalities (44) then the index is in the $j_1$ rank 2 grey galaxy phase. Its indicial entropy is given by plugging the charges (57) into (45).

On the other hand, if no legal solution to these equations (with the properties outlined above) exists, the index is not in a grey galaxy phase, and we proceed to the next subsubsection.

---

[87]We have proved in Appendix C.9 that such a solution is unique, if it exists.

[88]Of course all the charges for such an $x$ should lie inside the Bose-Fermi cone.

### 4.3.4 Rank 4 DDBH phases

The analysis of this case proceeds differently depending on how many $q_i$s are equal to each other.

- *Three Charges equal* If $q_1 = q_2 = q_3$, then the index is always in either the stable black hole or grey galaxy phase. One never encounters this case at this stage of our algorithm.

- *Two charges equal* Let us suppose $q_1 = q_2 = q$ (the discussion of $1 \leftrightarrow 3$ and $2 \leftrightarrow 3$ is identical). If $q_3 > q$, the index is not in a rank 4 DDBH phase, and we proceed to the next subsection. If $q > q_3$ (and we have reached this far in the algorithm) then the index *must* be in a rank 4 DDBH phase. The black hole in this phase carries charges

$$(q + x - y, q + x - y, q_3 + x, j_1 - x, j_2 - x), \quad y > 0 \tag{58}$$

  (here $y$ is the charge of dual giants in the DDBH in both the 1 and 2 directions). This black hole must have $\mu_1 = \mu_2 = 0$. We thus plug the charges (58) into (43) and the first of (49)[89] and solve for $x$ and $y$ in terms of $q, q_3, j_1, j_2$. If we have reached this far in the algorithm, there exists exactly one solution with $y > 0$ - and the property that the charges (58) obey (44) - to these equations. The indicial entropy is then obtained by plugging the charges (58) (corresponding to this legal solution) into (45).

- *All charges unequal* Let us suppose $q_1 > q_2 > q_3$ (the other possibilities are dealt with in an identical manner). We search for a rank 4 DDBH solution whose dual carries charges in the 1 and 2 directions. The black hole in such a solution must carry charges

$$(q_2 + x - y, q_2 + x - y, q_3 + x, j_1 - x, j_2 - x), \quad y > 0 \tag{59}$$

  (in this case the charge of duals in the 1 direction is $(q_1 - q_2) + y$ and in the 2 direction is $y$. This black hole must obey both (43) and the first of (49) (for $i = 1, 2$: these are the same equation). We plug (58) into these two equations, and solve for $x$ and $y$ in terms of $q, q_3, j_1, j_2$. If a solution with $y > 0$ exists, and if the black hole charges (59) obey (44) then our index is in the rank 4 DDBH phase. The indicial entropy is then obtained by plugging the charges (59) into (45). If an acceptable solution does not exist, we proceed to the next subsubsection.

### 4.3.5 Rank 2 DDBH phases

If we have reached this far in the algorithm, the largest of the three charges $(q_1, q_2, q_3)$ must be unique. We suppose this charge is $q_1$. It does not matter if two charges ($q_2$ and $q_3$) are equal or not.

We search for a DDBH solution whose dual carries $q_1$ charge. The black hole in this solution carries charges

$$(q_1 + x - y, q_2 + x, q_3 + x, j_1 - x, j_2 - x), \quad y > 0 \tag{60}$$

(here $y$ is the $q_1$ charge carried by the dual giant in the solution). We plug (60) into the first of (49) for $\mu_1 = 0$ and (43) and search for a solution with $y > 0$ (and on which the black hole charges obey (44)). If we have reached this far in the algorithm, such a solution must exist and be unique. The index is in the rank 2 DDBH phase. The indicial entropy is obtained by plugging the charges (60) into (45).

---

[89]The first of (49) must be satisfied for $i = 1$ or $i = 2$: these two are the same equation.

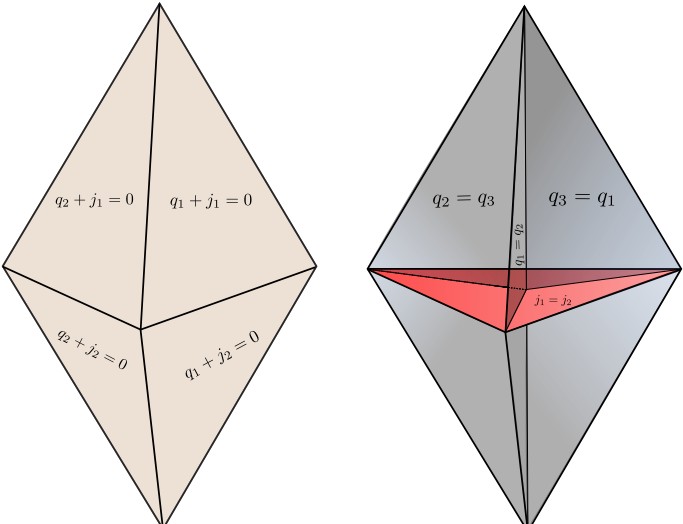

Figure 8: Base of the indicial cone - topologically a $\mathbb{B}^3$ but best visualized as a six faced diamond obtained by gluing two tetrahedra along the horizontal face. The left figure shows the external faces of the diamond. The right side figure shows the intersection of various equal charge planes inside the bulk of the diamond. The two angular momentum are equal on the horizontal red colored plane. Three pairs of charges are equal on the three vertical grey colored planes.

## 4.4 The indicial cone

In this subsection, we present a global visualization of the indicial phase diagram in the microcanonical ensemble. As the space of indices is closely related to the space of black holes, the space of indices depicted in this section, will turn out to be rather similar to the black hole apples of Fig 2, enhanced with the details of the sheets $S^{\nu_i=0}$ on this apple (see §2.16).

The space of indicial charges - subject to the inequalities (51) - is a cone, i.e. a fibration of a base over $R^+$. We refer to this as the indicial cone. The base of the cone - topologically a unit $B_3$ - is most usefully visualized as a solid six faced diamond, where each face of the diamond corresponds to a surface on which one of the six inequalities in (51) is saturated. This diamond is produced by gluing two (four sided) tetrahedra together along their common base (which we imagine as horizontal, see Fig. 8). The diamond that results from this gluing has six external faces. The three faces above the horizontal represent the surfaces $j_1 + q_1 = 0$, $j_1 + q_2 = 0$ and $j_1 + q_3 = 0$, respectively (see Fig 8). Each pair of these faces has a common edge, along which the corresponding $q_i$ values are equal,[90] see Fig 8. The equality $j_1 = j_2$ is obeyed along all horizontal edges. The part of the diamond below the horizontal - obtained by reflecting the top half about horizontal plane - is the analogue of the top half but with $j_1 \leftrightarrow j_2$. The "lower" face glued to $q_i + j_1 = 0$ represents $q_i + j_2 = 0$.

In the bulk of the diamond - as on its boundary - the horizontal bases of the two tetrahedra obey $j_1 - j_2 = 0$. As depicted in Fig. 8, the three vertical planes that symmetrically cut through the tetrahedron along one pair of slanting edges - are the planes $q_1 - q_2 = 0$, $q_2 - q_3 = 0$ and $q_3 - q_1 = 0$.

Let us imagine coloring those points in the indicial diamond, whose index lines intersect the black hole sheet. On the other hand, we leave points in the indicial diamond uncolored if the corresponding index line does not intersect the black hole sheet. We explain immediately below that such a color assignment divides the diamond described above into six distinct re-

---

[90]For example, $q_1 = q_2$ on the edge that is common to the faces $q_1 + j_1 = 0$ and $q_2 + j_1 = 0$.

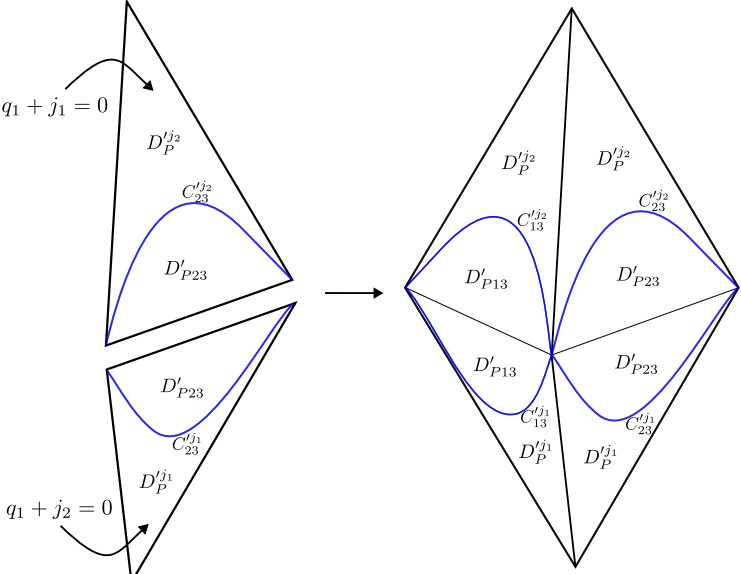

Figure 9: On the left figure we have two 1/8 BPS planes $q_1 + j_{1,2} = 0$ which touch at $j_1 = j_2 = 0$. The blue curves represent the points where indicial charges match the black hole charges on this boundary. On the right, we have depicted the intersection of four such boundaries: $q_2 + j_i = 0$ and $q_1 + j_i = 0$. These surfaces meet at points where both $q_1$ and $q_2$ are zero.

gions: a central, simply connected colored region which touches the boundary of the diamond along codimension one curves, together with five disconnected uncolored regions. In order to understand how this comes about, it is useful to first study the boundary of the indicial diamond.

### 4.4.1 The black hole sheet and the boundary of the indicial cone

Consider one of the six boundary faces (of the diamond), let us say $q_1 + j_1 = 0$. In charge (rather than indicial charge) space, the corresponding plane was studied in Appendix C.6.3 and reviewed in §3. Recall that the black hole sheet touches this four dimensional plane along the two dimensional surface $q_1 = j_1 = 0$, $j_2 = 2q_2 q_3$.

Let us now return to the space of indicial charges. Consider the three dimensional subspace of index lines that obey $q_1 + j_1 = 0$.[91] As index lines are one dimensional, it follows that only a codimension $4 - (2 + 1) = 1$ collection of these lines (i.e. those with $q_1 + j_1 = 0$) touches the boundary of the black hole sheet. Generic indicial lines with $q_1 + j_1 = 0$ completely miss the black hole sheet. The $q_1 + j_1$ index lines that touch the black hole sheet, pass through a point that obeys $q_1 = 0$, $j_1 = 0$, $j_2 = 2q_2 q_3$. It follows that the indicial charges of this special line obey the equation

$$j_2 - j_1 = 2(q_2 + j_1)(q_3 + j_1). \tag{61}$$

(61) describes a "tent" in the boundary indicial charge space parameterized by positive values of the "$x, y$ and $z$ coordinates", respectively identified with $(q_2 + j_1)$, $(q_3 + j_1)$ and $(j_2 - j_1)$.[92] This tent is attached to the ground (defined by $z \equiv j_2 - j_1 = 0$) along the positive

---

[91]Recall that an index line runs parallel to the charge space plane $q_1 + j_1 = 0$. This explains why this plane is a four dimensional surface in charges space, but a three dimensional surface in indicial space.

[92]The inequalities $(q_2 + j_1) \geq 0$ and $(q_3 + j_1) \geq 0$ are immediate from (51). The inequality $j_2 - j_1 \geq 0$ also follows from (51) on using $q_1 + j_1 = 0$. These conditions ensure that all inequalities in (51) are met, as follows from the fact that $q_2 + j_2 = (q_2 + j_1) + (j_2 - j_1)$ and $q_3 + j_2 = (q_3 + j_1) + (j_2 - j_1)$.

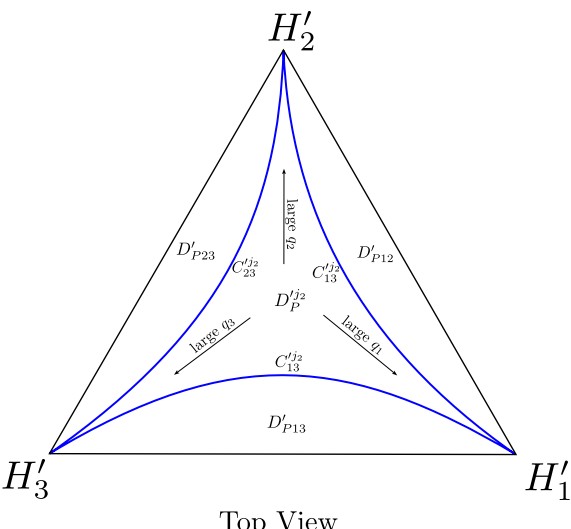

Figure 10: Top view of the boundary of the indicial diamond (the base of the indicial cone) shown in Figure 9.

$x$ and $y$ axes. It rises to its highest along the line $x = y$, (i.e. $q_2 + j_1 = q_3 + j_1$). There $j_2 - j_1$ rises parabolically as a function of $q_2 + j_1 = q_3 + j_1$.[93]

As we have seen above, the $j_1 + q_1 = 0$ segment of the boundary discussed in the previous two paragraphs may be thought of as a cone, whose base is a triangle (one of the six triangles that make up the face of the diamond). The "tent" described above intersects this triangle on the parabola depicted in Fig. 9. As this curve represents those index lines that intersect the black hole sheet on the curve $C_{23}^{j_2}$ (see Fig. 4), we name this curve $C_{23}^{'j_2}$ (see Fig. 9).

As depicted in Fig. 9, the $j_1 + q_1 = 0$ triangle is glued, along its horizontal edge, to a second $j_2 + q_1 = 0$ triangle. The black hole sheet touches this triangle along a second parabola (the reflection of the first about the horizontal plane). Together these two intersection curves make up a closed loop with the topology of an $S^1$ (see Fig. 9). As this curve consists of those index lines that intersect the black hole sheet in the curve $S_{12}$, we name this curve $S'_{12}$.

An identical discussion applies to the pair of triangles corresponding to $q_2 + j_1 = 0$ and $q_2 + j_2 = 0$, as well as the pair of triangles corresponding to $q_3 + j_1 = 0$ and $q_3 + j_2 = 0$. Sewing these three pairs of triangles into the diamond we see that the "black hole sheet" (colored part of the diamond) divides the boundary of the base of the indicial cone into 5 disconnected regions: the insides of the three circles described above, the region above the three circles and the regions below the three circles. See Fig. 10 for a top view of these separating curves on the diamond.

It is useful to have names for these disconnected regions of the boundary. Following the nomenclature of our discussion of the black hole sheet, we call the region above/below all three circles, $D_P^{'j_2}$ or $D_P^{'j_1}$ respectively. We call the region inside the circle located on the two planes $q_1 + j_1 = 0$ and $q_1 + j_2 = 0$ $D'_{P23}$ (similarly with $1 \leftrightarrow 2$ and $1 \leftrightarrow 3$), see Fig. 9 and 10. The subscripts $P$ in this notation emphasize that the regions in question lie on the boundary of the *prisms* that make up the base of the indicial diamond (below we will define similar regions on the boundary of the black hole or blue part of the boundary diamond).

---

[93]We will see below that this tent represents a dividing line on the boundary of the space of indices, between index lines that pass through the EER in a grey galaxy phase, and index lines that pass through the EER in the DDBH phase.

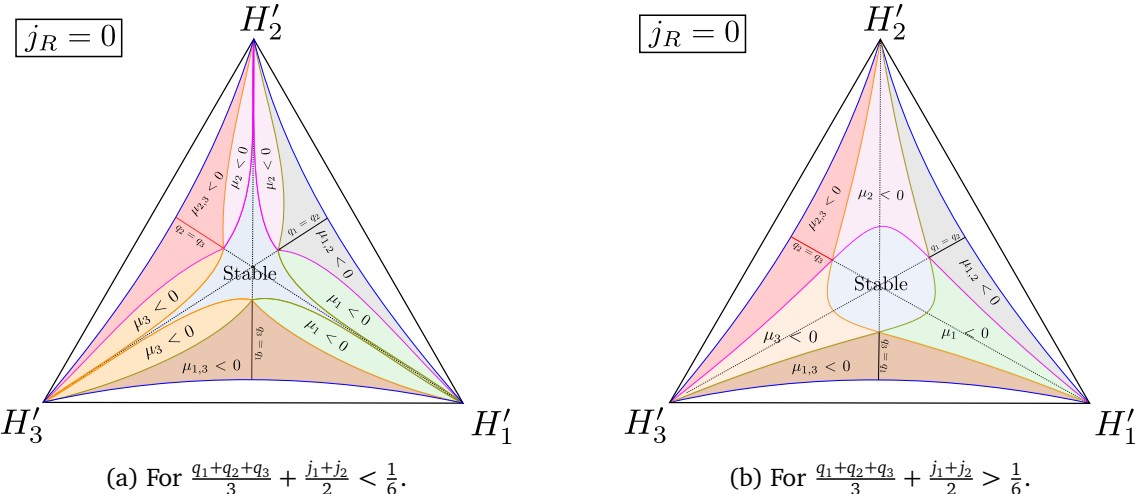

(a) For $\frac{q_1+q_2+q_3}{3} + \frac{j_1+j_2}{2} < \frac{1}{6}$.  (b) For $\frac{q_1+q_2+q_3}{3} + \frac{j_1+j_2}{2} > \frac{1}{6}$.

Figure 11: Black hole sheet inside the index diamond. The colored part of the above figures denotes the region of indicial charges that intersects the black hole sheet and the uncolored region is the space of indicial charges that do not intersect the black hole sheet.

### 4.4.2 The black hole sheet in the bulk of the diamond

The base of the indicial cone forms a filled out solid diamond. The colored (black hole) region within this diamond is a simply connected region of codimension zero. It can be visualized as an amoeba that extends out three pseudopodia that just touch the boundary surface along the three "circles" described above.

As the black hole sheet is itself simply connected, it divides the bulk of the diamond into 6 regions - the colored black hole region itself, and five disconnected uncolored regions (see fig. 11). The boundary of the colored region within the diamond also has 5 regions. We denote these by $D^{'j_2}$, $D^{'j_1}$, $D'_{12}$, $D'_{23}$ and $D'_{31}$ respectively. The notation here is the obvious one; $D^{'j_1}$ consists of those indices that touch the black hole sheet on a boundary point that lies in $D^{j_1}$. The sheet $D^{'j_1}$ has the same boundary as - and is homologous to - the sheet $D_P^{'j_1}$. Identical remarks hold for the remaining 4 sheets.[94]

### 4.4.3 Surfaces of vanishing chemical potential within the diamond

Because there is a one to one map from the black hole sheet to the space of indices, the colored (black hole) region of the indicial diamond is topologically identical to the black hole sheet. And the colored region of the diamonds in Fig 8 are topologically identical to the black hole apple in Fig 2. As was the case for the black hole diamond, the indicial diamond hosts 5 physically important surfaces, $S^{'\nu_i=0}$. These surfaces, respectively, distinguish the space of index lines that intersect the black hole sheet along the curves $S^{\nu_i=0}$.

The sheets $S^{'\nu_i=0}$ extend into the colored part of the indicial diamond, in a manner that is identical to the discussion of §2.16. As in that section, the nature of the extension of these sheets into the indicial diamond depends on whether our indicial charges are small or large (i.e. whether our indicial diamond lies near to - or far from - the apex of the indicial cone).

At small charges (when $\frac{q_1+q_2+q_3}{3} + \frac{j_1+j_2}{2} < \frac{1}{6}$, see Appendix H for a derivation of this value) the extension of the $S^{'\nu_i=0}$ into the colored part of the indicial diamond, is topologically iden-

---

[94]The surface $D^{'j_2}$ "false ceiling" for the diamond. The surface $D^{'j_1}$ is its mirror reflection along the horizontal $j_1 = j_2$ surface. The surfaces $D'_{12}$, $D'_{23}$ and $D'_{31}$ are disk like protrusions (from the boundary circles) into the black hole amoeba.

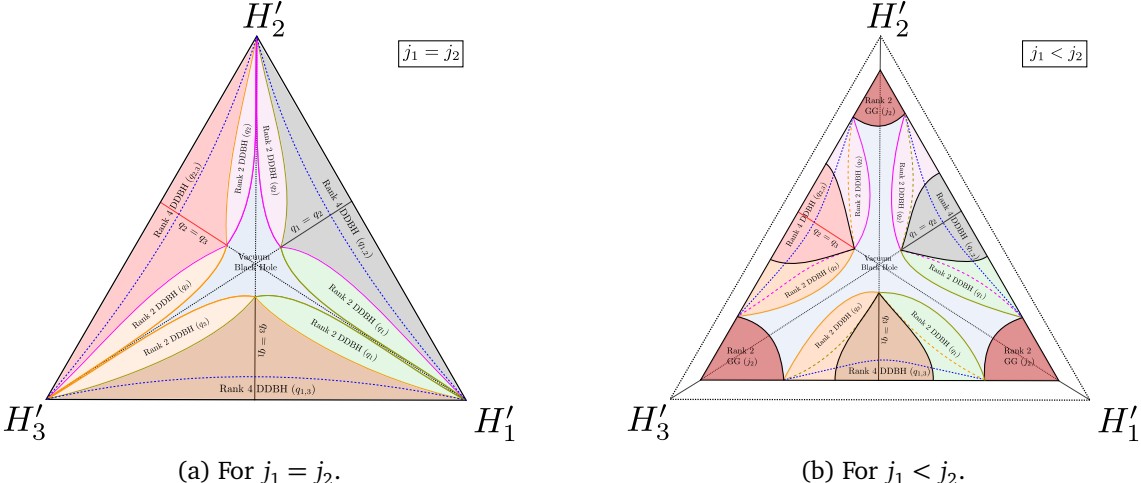

(a) For $j_1 = j_2$.      (b) For $j_1 < j_2$.

Figure 12: Horizontal cross sections of the indicial cone for $\frac{q_1+q_2+q_3}{3} + \frac{j_1+j_2}{2} < \frac{1}{6}$. One of the three renormalized chemical potentials $(\mu_1, \mu_2, \mu_3)$ vanishes on each of the three (green, pink, orange respectively) colored curves (including the dashed component of the curves). For $j_1 < j_2$, there is a fourth region (maroon colored) where $\omega_2 < 0$. In the blue region, the microcanonical index is in the vacuum black hole phase. In the colored regions, the index is dominated by either rank 2 the grey galaxy phase, rank 2 DDBH or rank 4 DDBH.

tical to the "small charge" extension of the sheets $S^{\nu_i=0}$ into the black hole apple described in §2.16.2. It follows that two horizontal cross sections of the colored part of the indicial diamond, in this case, take the forms depicted in Fig. 11a (compare Fig. 5a). When the indicial diamond is far from the apex of the indicial cone), the extension of the $S'^{\nu_i=0}$ into the colored part of the indicial diamond, is topologically identical to the "large charge" extension of the sheets $S^{\nu_i=0}$ into the black hole apple described in §2.16.3. It follows that two horizontal cross sections of the colored part of the indicial diamond, in this case, take the forms depicted in Fig. 11b (compare Fig. 6a).

In both cases, the index lines that lie in the stable (light blue) region of Fig. 12)will turn out to lie in the black hole phase. All other index lines will lie in either a rank 2 grey galaxy phase, a rank 2 or rank 4 DDBH phase. All points in the diamond that lie above the surface $S'^{j_2=0}$ lie in a $j_2$ grey galaxy phase. Points lie below the surface $S'^{j_1=0}$ lie in a $j_1$ grey galaxy phase. The remaining regions are divided between rank 2 and rank 4 DDBH phases, in the manner described in the next subsection.

## 4.5 Visualization of the microcanonical indicial phase diagram

Since every index line passes through some part of the EER, however, the prediction of this paper is that the indicial entropy is of order $N^2$ everywhere within the indicial cone, even in the uncoloured regions of Fig. 13 (as explained above, this region consists of index lines that fail to intersect the black hole sheet). In more detail our analysis predicts that the diamond at the base of this cone is broken up into regions that lie in several distinct indicial phases. The precise phase diagram differs, depending on whether we are working at small or large values of indicial charges.

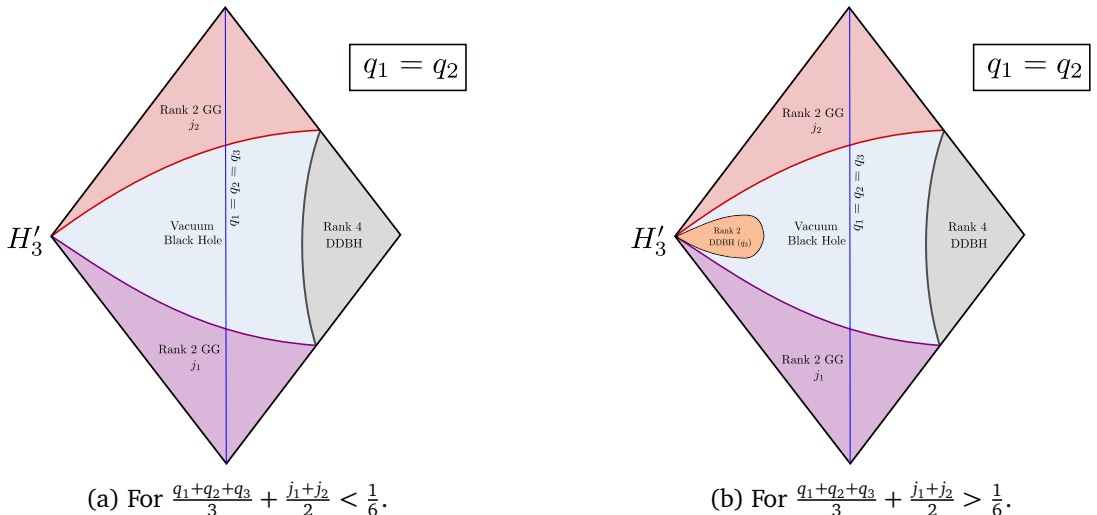

(a) For $\frac{q_1+q_2+q_3}{3} + \frac{j_1+j_2}{2} < \frac{1}{6}$.

(b) For $\frac{q_1+q_2+q_3}{3} + \frac{j_1+j_2}{2} > \frac{1}{6}$.

Figure 13: Vertical cross-section of the base of the indicial cone along the $q_1 = q_2$ plane. The bulk of the indicial cone is split into four or five regions corresponding to the four phases as mentioned in the figure. The renormalized chemical potentials $(\omega_2, \omega_1, \mu_{1,2})$ vanish on the boundaries between various regions. The blue line indicates points with $q_1 = q_2 = q_3$.

### 4.5.1 Small charge, i.e. $\frac{q_1+q_2+q_3}{3} + \frac{j_1+j_2}{2} < \frac{1}{6}$

As we have explained above, the indicial phase diagram is a fibration of the indical diamond over $R^+$. In this subsection we describe how the diamond "fibres" of this phase diagram look at small values of indicial charges (not too far from the tip of the cone).

In Fig. 12 we display two horizontal cross sections of the low charge indicial phase diagram diamond. Fig. 12a depicts the "equatorial" cross section $j_1 = j_2$. Notice that this diagram simply "extends" the diagram of Fig 11 to the boundary of the diamond.

Fig12b depicts a second "higher" horizontal cross section of the same diamond, i.e. a cross section at a value $j_2 > j_1$. Notice the similarity of the shape of the stable light blue region with the corresponding region in Fig 5b

In order to help the reader better understand the structure of the phase diagram, in Fig 13a we also present vertical, $q_1 = q_2$ slice of the same diamond.

### 4.5.2 Large charge, i.e. $\frac{q_1+q_2+q_3}{3} + \frac{j_1+j_2}{2} < \frac{1}{6}$

Focussing now on diamonds of large charge, we once again display two horizontal cross sections of our the diamond in our phase diagram in Fig 14. Fig. 14a depicts the "equatorial" cross section $j_1 = j_2$. As above, this diagram simply extends the diagram Fig 11 to the boundary of the diamond. Fig 14b depicts a second horizontal cross section of the same diamond, this time at a value $j_2 > j_1$ (compare with Fig 6ab). Finall, Fig. 14c depicts a third horizontal cross section of the same diamond, at a still larger value $j_2 - j_1$ - a value that is high enough that the three arches have met up (compare with Fig 6ac).

Finally, in the second of Fig 13 we also present vertical, $q_1 = q_2$ slice of the same diamond (that makes up the base of the microcanonical indicial phase diagram).

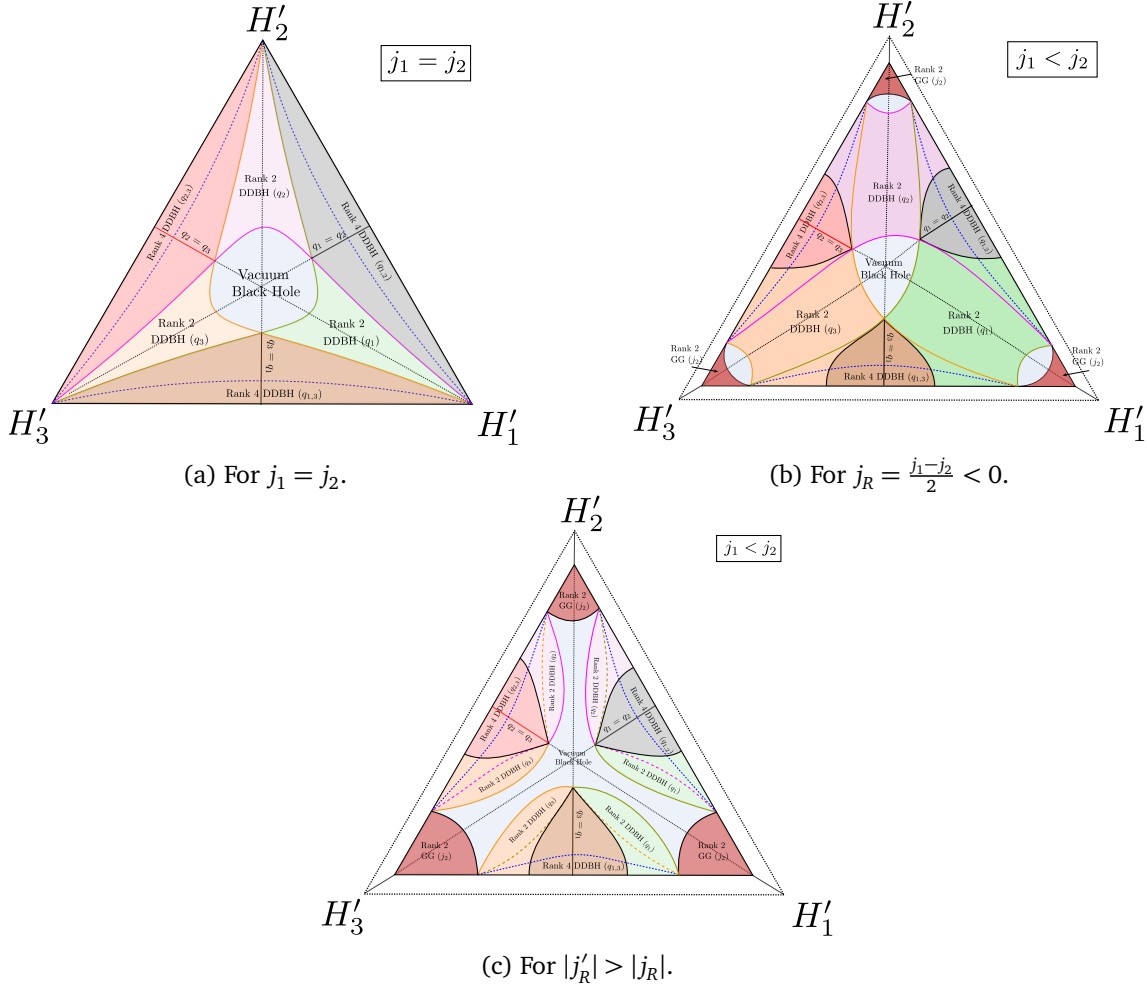

(a) For $j_1 = j_2$.

(b) For $j_R = \frac{j_1 - j_2}{2} < 0$.

(c) For $|j_R'| > |j_R|$.

Figure 14: Horizontal cross sections of the indicial cone for $\frac{q_1 + q_2 + q_3}{3} + \frac{j_1 + j_2}{2} > \frac{1}{6}$ at various values of $j_R = \frac{j_1 - j_2}{2}$. One of the three renormalized chemical potentials $(\mu_1, \mu_2, \mu_3)$ vanishes on each of the three (green,pink,orange respectively) colored curves (including the dashed component of the curves). For $j_1 < j_2$, there is a fourth region (maroon colored) where $\omega_2 < 0$. In the blue region, the microcanonical index is in the vacuum black hole phase. In the colored regions, the index is dominated by either rank 2 the grey galaxy phase, rank 2 DDBH or rank 4 DDBH.

## 5 Comparison against numerics for special charge configurations

### 5.1 Summary of the comparison and its results

In this section, we numerically compute the microcanonical index defined in (1) at the largest finite values of $N$ that we are able to tackle, and compare the results of this computation with our prediction for the index (from §4) in two special classes of charge configurations that we now describe.

#### 5.1.1 Three equal charges

Our first special case is obtained by specializing our charges to $q_1 = q_2 = q_3 = q$.[95] In Appendix G, we present a detailed study of both the supersymmetric entropy (as a function of the three

---

[95]And so setting the indicial charge $q_1 - q_2$ and $q_2 - q_3$ both to zero.

charges $q$, $j_L$ and $|j_R|$) as well as the superconformal index (as a function of the two indicial charges $q + j_L$ and $|j_R|$) (see §G.7 for a detailed summary of final results).

In the case of the index, our special configurations lie on the vertical axis of the blue line drawn in the indicial diamonds sketched in Fig 13.[96] We traverse the blue line (from bottom to top) by letting $j_R$ range between its largest to its smallest allowed values, at fixed $q + j_L$. As is clear from Fig 13, the microcanonical indicial phase diagram with these charges has three phases; the black hole phase, and the two rank 2 grey galaxy phases. While the black hole phase dominates at small values of $|j_R|$, the grey galaxy phases dominate at larger values of $|j_R|$. Consequently, the indicial entropy undergoes two phase transitions as $j_R$ is varied at fixed $q + j_L$ and the qualitative nature of the behaviour of the index as a function of these two indicial charges is similar at small and large values of indicial charges, Figs 13(a) and 13(b).

In Appendix G.7 we present a detailed computation of the location of this phase transition, and the value indicial entropy as a function of $|j_R|$ and $q + j_L$, In §5.2 below we compare our prediction (summarized in Appendix G.7) for this indicial entropy as we traverse the blue line in 13 (at a particular value of $q + j_L$) to data obtained from numerics, and find a good match, at least at the qualitative level. At the end of this section we explain that gravitons do not substntially "contaminate" the black hole index on this cut of charges. This partly explains why the match between black hole predictions and the numerical data is so good in this case.

### 5.1.2 Equal angular momenta and two equal charges

The second special case we study involves specializing to the charges $q_1 = q_2 = q$ and $j_1 = j_2 = j$ (we use the notation $q'$ for $q_3$). In Appendix H, we present a detailed study of both the supersymmetric entropy (as a function of the three charges $q$, $q'$ and $j$) as well as the superconformal index (as a function of the two indicial charges $q - q'$ and $\frac{2q+q'}{3} + j$).

The special charges described here correspond to the red lines in the equatorial cross section of the indicial diamond depicted in Fig. 15.[97] One moves from one end to the other of the red lines in Fig. 15 by varying $q - q'$ at fixed $\frac{2q+q'}{3} + j$. As is clear from the Figures above, we explore either two or three phases in this process, depending on whether the indicial charges are small Fig 15(a)) or large 15(b)).

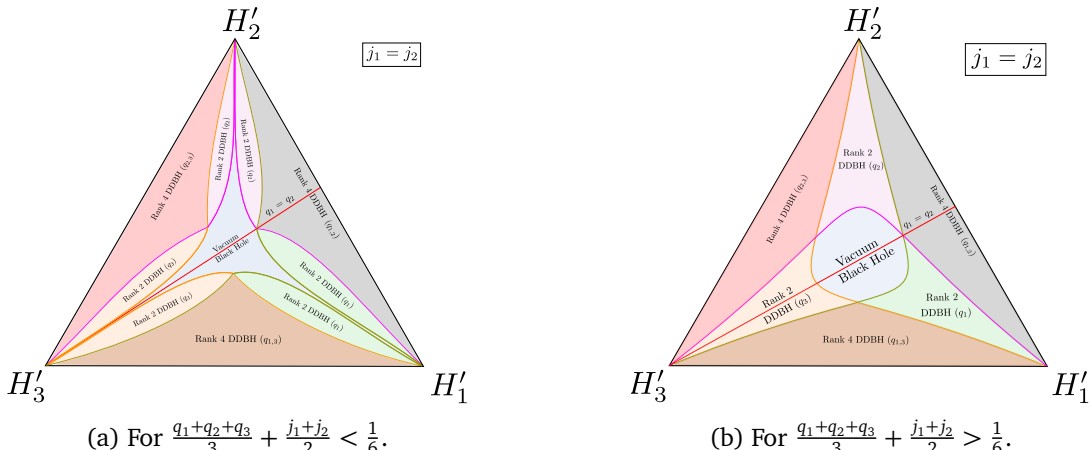

(a) For $\frac{q_1+q_2+q_3}{3} + \frac{j_1+j_2}{2} < \frac{1}{6}$.      (b) For $\frac{q_1+q_2+q_3}{3} + \frac{j_1+j_2}{2} > \frac{1}{6}$.

Figure 15: Horizontal cross sections of the indicial diamond depicting the set of indicial charges (Red line) considered in the two equal charges and equal angular momentum case.

---

[96]Figs 13 are simply the vertical cross sections of the indicial diamond 13 redrawn together with the blue line.

[97]Fig. 15 redraws Fig. 12a (when $\frac{2q+q'}{3} + j < \frac{1}{6}$) and the same line in Fig. 14a when (when $\frac{2q+q'}{3} + j < \frac{1}{6}$), but with the red line of interest inserted.

At small indicial charges Fig 15(a), the rank 4 DDBH phase dominates when $q - q'$ is large (top right region of the red line). As this quantity is reduced, we make a phase transition into the black hole phase, and stay there all the way down to the half BPS point $H_3'$.

At large values of indicial charges, 15(b), the rank 4 DDBH phase once again dominates when $q - q'$ is large and positive. On lowering this quantity we make a phase transition into the black hole phase. On further lowering (to a region where $q - q'$ is negative enough) we make a second phase transition into a rank 2 DDBH phase, and then stay there all the way down to the half BPS point $H_3'$.

In the Appendix we have, once again, obtained detailed predictions for the locations of these phase transitions, and the value of the indicial entropy in each of the three phases. In §5.3, below we compare our prediction for this indicial entropy to data obtained from numerics. Unfortunately, in this case our prediction for the indicial entropy of the black hole and DDBH phases do not differ substantially from each other at values of the charge $\frac{2q+q'}{3} + j$ that we were able to numerically handle. Moreover the data at these relatively small values of $N$ is significantly "contaminated" by the contribution of pure gravitons.

In summary, data that we were able to obtain (at relatively small values of the charge $\frac{2q+q'}{3} + j$) turns out, in this case, to be too noisy (and too contaminated by gravitons) to either confirm or contradict our predictions. We hope that better numerics - or a better choice of charge cut (in future work) will improve this situation.

## 5.2 Numerics for black holes with three equal charges and different angular momenta

In this subsection we study the grand canonical index relevant to the special family of charges $q_1 = q_2 = q_3 = q$ (see Appendix G for a detailed study). In other words we study

$$\mathcal{I}_W = \text{Tr}\left((-1)^F e^{-\mu \frac{Q_1+Q_2+Q_3}{3} - \mu J_L - \omega_R J_R}\right). \tag{62}$$

Note that in (62) (as in Appendix G) we have set $3\mu_1 = 3\mu_2 = 3\mu_3 = \mu = \omega_L$.

In this section we use a computer to evaluate the "Taylor Expansion" of $\mathcal{I}_W$ defined by

$$\mathcal{I}_W(\mu, \omega_R) = \sum_{A, J_R} n(A, J_R) e^{-\mu A - \omega_R J_R}, \tag{63}$$

where the sum over $J_R$ runs over half integers, and the sum over $A = \frac{Q_1+Q_2+Q_3}{3} + J_L$ runs over integers divided by six. Comparing with the notation of Appendix G,[98]

$$A = N^2 \alpha, \qquad J_R = N^2 j_R. \tag{64}$$

We evaluate the integers $n(A, J_R)$ using the method presented in [36, 37]. Briefly, we first Taylor expand the integrand (in the formula for the index as an integral over holonomies) in a double expansion in $e^{-\frac{\mu}{6}}$ and $e^{-\frac{\omega_R}{2}}$, and then evaluate the integral over $U$, in each term (and so identify the singlets of $U(N)$) using Cauchy's theorem.

The indicial entropy is given by

$$N^2 S_{BPS}(\alpha, j_R) = \ln\left(|n(A, J_R)|\right), \tag{65}$$

where $\alpha$ (on the RHS of (65)) is defined[99] by (64). Consequently, every term in (63) gives us a computation of $S_{BPS}(\alpha, j_R)$ at some values of $\alpha$, $j_R$ and $N$.

---

[98]Therefore $\alpha = q + j_L$ where $q = \frac{q_1+q_2+q_3}{3}$.

[99]$\alpha$ is the variable referred to as $q + j_L$ in the introduction to this section.

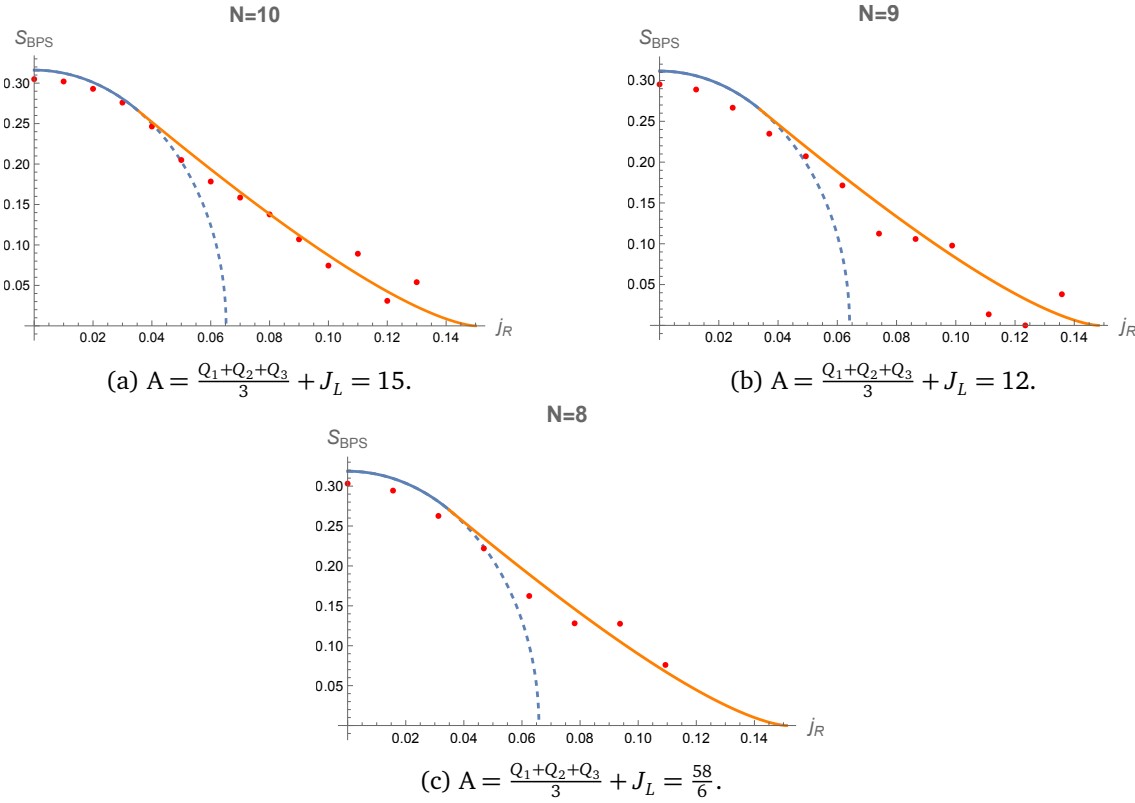

Figure 16: The figures above compare the numerically computed indicial entropy (red dots) with the predicted formula (blue and orange solid lines) for various values of $N$. The indicial entropy $S_{BPS}$ is calculated as $\frac{1}{N^2} \ln(|n(A, J_R)|)$, with $A = \frac{Q_1+Q_2+Q_3}{3} + J_L$ fixed, while varying $J_R = N^2 j_R$. The blue line (solid and dashed) represents the black hole entropy, determined at the intersection of the black hole sheet and an index line. The orange line depicts the entropy of grey galaxy solutions with $\omega_1 = 0$. Deviations between the black hole entropy and the indicial entropy arise when $j_R$ becomes significantly asymmetric.

In Fig 16 below, we fix $\alpha = 0.15$ and plot $S_{BPS}(\alpha, j_R)$ as a function of $j_R$[100] at (respectively) $N = 10, 9, 8$.[101,102] The graphs in these figures display both our large $N$ predictions for the indicial entropy as well as the values obtained by explicit computer based integration. In more detail, the solid blue curve (representing the black hole indicial phase) and the solid orange curves (representing the grey galaxy phase) in these figures, together constitute our prediction for the indicial entropy. The dotted blue curve represents the continuation of the black hole phase - the naive prediction for the superconformal index. The solid and dotted blue curves, together, represent the entropy of the intersection of the index line with the black hole sheet.

A glance at Fig 16 will convince the reader that the numerical results (red dots) are in reasonable agreement with the entropy of the "intersection black hole" at small values of $|j_R|$, but begin to deviate significantly from this intersection entropy at large $|j_R|$. This divergence begins to happen at approximately the value of $|j_R|$ at which the theoretical analysis of this

---

[100]In this case the indicial entropy is an even function of $j_R$, so we plot it only for positive values of $j_R$.

[101]At $N = 10$, the corresponding indicial charge (normalized so it always evaluates to an integer) equals $6(Q + J_L) = 90$. Similarly, for $N = 9, 6(Q + J_L) = 72$ and $N = 8, 6(Q + J_L) = 58$. These are all reasonably large numbers.

[102]Note that $0.15 < \frac{1}{6}$, so we are in the scenario sketched in Fig. 13(a), and so should expect an indicial phase diagram with only two phases; the rank 4 DDBH phase and the black hole phase.

paper predicts a phase transition from the black hole to the grey galaxy indicial phases. At larger values of $|j_R|$, the red dots are in reasonably good qualitative agreement with the entropy of the grey galaxy phase of the index.

In summary, while the computer generated data appears to be in rather good qualitative agreement with our predictions at all values of $j_R$, it deviates significantly from the naive prediction (entropy of intersection of the index line with the black hole sheet) when $|j_R|$ is larger than our predicted phase transition. While we find this agreement between data and our predictions very encouraging, we emphasize that it is (as yet) qualitative. We do not (yet) have enough data to perform a systematic fit of deviation of the data from the predicted value as a function of $1/N$. We hope that future work will improve this situation.

## 5.3 Numerics for black holes with two equal charges and equal angular momenta

In this subsection we study the grand canonical index relevant to the analysis of the special case studied in Appendix H, namely

$$\mathcal{I}_W = \text{Tr}\left((-1)^F e^{-\mu \frac{Q_1+Q_2+Q_3}{3} - \mu J_L - \frac{\mu_3-\mu_1}{3}(2Q_3-Q_1-Q_2)}\right). \tag{66}$$

Note that, as in Appendix H, we have set $\mu_1 = \mu_2$ and $\mu_1 + \mu_2 + \mu_3 = \mu = \omega_L$.

As in the previous subsection, we evaluate the "Taylor Expansion" of $\mathcal{I}_W$ defined by

$$\mathcal{I}_W(\mu, \mu_3 - \mu_1) = \sum_{A_1, 2Q_3-Q_1-Q_2} n(A_1, 2Q_3 - Q_1 - Q_2)e^{-\mu A_1 - \frac{\mu_3-\mu_1}{3}(2Q_3-Q_1-Q_2)}, \tag{67}$$

where the sum over $2Q_3 - Q_1 - Q_2$ runs over integers, and the sum over $A_1 = \frac{Q_1+Q_2+Q_3}{3} + J_L$ runs over integers divided by six. Comparing with the notation of Appendix H,

$$2Q_3 - Q_1 - Q_2 = 2N^2(q' - q), \qquad A_1 = N^2 \alpha_1. \tag{68}$$

The indicial entropy is given by

$$N^2 S_{BPS}(\alpha_1, 2(q' - q)) = \ln\left(|n(A_1, 2Q_3 - Q_1 - Q_2)|\right). \tag{69}$$

Consequently, every term in (67) gives us a computation of $S_{BPS}(\alpha_1, 2(q' - q))$ at some values of $\alpha_1, 2(q' - q)$ and $N$.[103]

As in the previous subsection, we fix $\alpha_1 = 0.15$ and plot the indicial entropy $S_{BPS}$ varying $2(q' - q)$), and compare it with the conjectured indicial entropy given in Appendix H.4 for $N = 10$. Figure 17 illustrates the comparison between the predicted entropy and numerical results. The red dots represent the numerical values of the indicial entropy $S_{BPS}$. The blue solid line (and blue dashed line) corresponds to the entropy of the black hole, where the black hole sheet intersects with an index line. The orange line represents the entropy of the (dominant) rank 4 DDBH solutions with $\mu_1 = \mu_2 = 0$, using the equations in H.4.[104] The union of these two solid lines is our prediction for the indicial entropy. Note that the solid and dashed lines in Figure 17 are essentially identical to those in the first plot of Figure 34, and so are difficult to numerically distinguish.

---

[103]In Appendix H $\alpha_1$ is defined as $\frac{q_1+q_2+q_3}{3} + j_L = \frac{2q+q'}{3} + j_L$.

[104]For $\alpha_1 < \frac{1}{3}$, the rank 2 DDBH phase never dominates the indicial entropy. Conversely, for $\alpha_1 > \frac{1}{3}$, the rank 2 DDBH phase dominates over the intersection entropy for large $q' - q$. See H.4.3 for further details.

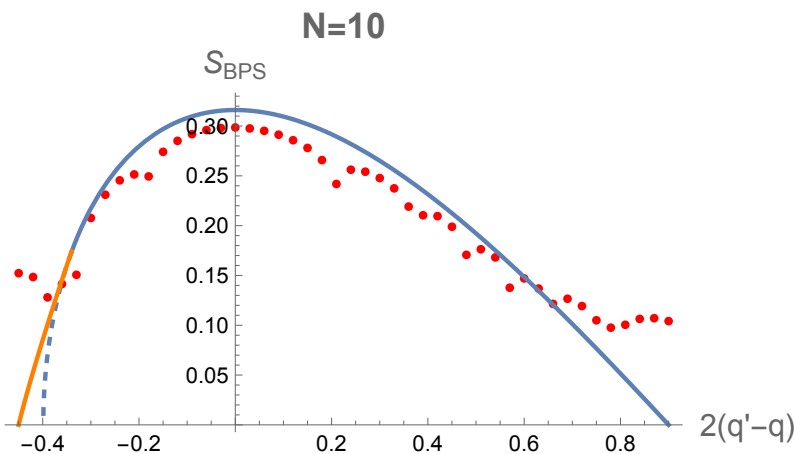

Figure 17: The figure compares the numerically computed indicial entropy (red dots) with the predicted formula (blue and orange solid lines) at $N = 10$. The indicial entropy $S_{BPS}$ is calculated as $\frac{1}{N^2}\ln(|n(A_1, 2Q_3 - Q_1 - Q_2)|)$, with $A_1 = \frac{Q_1 + Q_2 + Q_3}{3} + J_L = 15$ fixed, while varying $2Q_3 - Q_1 - Q_2 = 2N^2(q' - q)$. The blue line (solid and dashed) represents the black hole entropy, determined at the intersection of the black hole sheet and an index line. The orange line depicts the entropy of DDBH solutions with $\mu_1 = \mu_2 = 0$.

### 5.3.1 Explanation of oscillations

As an aside, we note that the curve described by the red dots in Fig. 17 displays an oscillatory behavior, with clearly visible "kinks". These kinks have a simple origin. On the $y$ axis of Fig. 17, we have plotted the logarithm of $|n_I(Z_i')|$. However, our numerical data yields $n_I(Z_i')$ with a sign. The kinks in the data mark points at which $n_I(Z_i')$ switch sign.

As explained in [36] (and reviewed briefly in footnote 177 at the end of Appendix §I.1) these oscillations in sign are a reflection of the fact that the Euclidean black hole indicial saddle point has an imaginary component to its entropy (in addition to the real part we have focused on through most of this paper). Once we account for the imaginary part of the entropy, saddle point analysis predicts that $n_I(Z_i')$ takes the form

$$n_I(Z_i') = e^{\mathrm{Re}(S_{BH})}\cos(\mathrm{Im}(S_{BH}) + \alpha), \tag{70}$$

where $S_{BH}$ is the complex entropy obtained by Legendre transforming the grand canonical index (40)[105] (in the manner described in detail in Appendix I)[106] and $\alpha$ is a function of $Z_i'/N^2 = \zeta_i'$. The oscillating cosines above explain the flips in sign of the function $n_I(Z_i')$.

In principle, the function $\alpha$ in (70) is determined by a one loop computation. As this computation has not been done, in practice we treat $\alpha$ as an unknown constant[107] fitting

---

[105]For e.g., in the special case of $\mu_1 = \mu_2 = \mu_3 = \mu$ and $\omega_1 = \omega_2 = \omega$, $S_{BH}$ is obtained by extremization of

$$\frac{N^2 \mu^3}{2\left(\frac{3\mu - 2\pi i}{2}\right)^2} + \mu(Q + J),$$

w.r.t. $\mu$. The entropy obtained after this extremization procedure is, in general, complex.

[106]Earlier in this paper, we have worked with the real part of the black hole entropy only. The entropy $S_{BH}$ presented in (45) is actually $Re(S_{BH})$ in the language of this subsubsection and in the language of Appendix I.

[107]In the large $N$ limit, $\alpha$ is effectively constant, as it is a function of $Z_i'/N^2$, (rather than $N^2$ times a function of $Z_i'/N^2$ as is the case with the imaginary part of the leading order black hole entropy). Treating $\alpha$ as effectively constant at finite $N$ - as in the numerics above- is a less justified approximation, but one that appears to work rather well (see below).

parameter. On the black hole saddle one can check that the black hole charge relation tells us that $\mathrm{Im}(S_{BH}) = 2\pi J$ (here $J_1 = J_2 = J$). Consequently, (70) simplifies to

$$n_I(Z'_i) = e^{\mathrm{Re}(S_{BH})} \cos(2\pi J + \alpha). \tag{71}$$

We have checked that, for a suitable choice of $\alpha \approx -\frac{\pi}{2}$, the sign oscillations our data for $n_I$ (i.e. of the computer generated values for these quantities), almost perfectly matches the sign oscillations in the formula (71) for $2(q - q')$ in the range $-0.4$ to $0.6$ - where the black hole phase should dominate. This yields a quantitative explanation of these oscillations along the lines of [36].

We could, in principle, attempt a similar analysis of the oscillations in Fig. 16. In Appendix I we have conjectured that the full complex entropy of a grey galaxy matches the complex entropy of its central black hole. We could thus attempt to match the imaginary part of the entropy of the central black holes in the grey galaxy phases in Figs 16 with the signs of the data in the orange (grey galaxy) phases of those diagrams. While it is possible to roughly perform such a match, unfortunately, the data of Fig. 16 exhibits too few oscillations for this match to be very convincing. The situation may improve in the future if one is able to obtain data at larger values of $N$.

## 5.4 The tail contribution of pure gravitons

As can be observed, the indicial entropy at the tails of Fig 17 does not fit either the "intersection black hole" prediction or the DDBH prediction of this paper particularly well at the two tails of the graph. In particular, the data fails to match both the naive (pure black hole) as well as the DDBH predictions that the entropy should vanish at the two extreme ends of the graph.

The discrepancy noted in the last paragraph arises due to finite $N$ effects, in particular due to the contribution of pure gravitons. In order to see this, it is useful to first focus on the rightmost red data point in Fig. 17. This point lies at $2(q' - q) = 0.9$. Now $q = \frac{q_1 + q_2}{2} \geq 0$ (the inequality follows from (22)) Consequently, all states that contribute to this index line have $q' \geq 0.45$. Using $q \geq 0$ and $j_L \geq 0$ (both inequalities follow from (22)) it follows that $\alpha_1 = \frac{q' + 2q}{3} + j_L \geq \frac{q'}{3} = 0.15$. But the whole of Fig. 17 is plotted at $\alpha_1 = \frac{q' + 2q}{3} + j_L = 0.15$. Consequently, the only states that contribute to the rightmost red point in Fig. 17 are those with $q = j_L = 0$. On the special charge configuration under study, this tells us that $q_1 = q_2 = j_1 = j_2 = 0$. It follows that all states that contribute to the rightmost red point in Fig. 17 are half-BPS operators made of scalar $Z$. Though the entropy of these half BPS operators is subleading in the large $N$ limit, it is nonzero. This subleading entropy is easily computed. The rightmost point has $q' = q_3 = 0.45$, and so $Q_3 = 0.45N^2$. At $N = 10$ the charges are given by $(Q_1, Q_2, Q_3, J_1, J_2) = (0, 0, 45, 0, 0)$. and the number of half BPS states at this charge is by the power of $q^{45}$ in $\prod_{n=1}^{10} \frac{1}{1-q^n}$ and equals 33401, leading to $S_{BPS} = 0.104163$, which precisely matches the numerical results. At least at the extreme right end of Fig 17 (and so, presumably, also for the points near this boundary), the red dots capture the entropy of gravitons modulo trace relations.

A similar analysis applies to the left end of Fig 17. The leftmost red dot in Fig 17 lies at $2(q' - q) = -0.45$, and so captures states with $q = q' + \frac{0.45}{2}$. Plugging this into the equation $\alpha_1 = \frac{q' + 2q}{3} + j_L \geq \frac{q'}{3} = 0.15$, we conclude that this point only gets contributions from those states that have $q' + j_L = 0$. However $q' + j_L = \left( \frac{q_3 + j_1}{2} + \frac{q_3 + j_2}{2} \right) \geq 0$ (the inequality follows from (22)). Consequently, the only states that contribute to this point are those with $q' + j_L = 0$ (since no states carry negative values of this charge, we cannot have cancellation between positive and negative values). The only letters in Table 1 that meet this condition are the scalars $X, Y$ and the third of $\psi_{+,0}$. States made out of these letters were counted in Eq 6.5

of [11]. The coefficient of the index at this charge is 411258, leading to $S_{BPS} = 0.152295$, which again precisely matches the numerical results, once again explaining the entropy of the leftmost red dot. Once again we conclude that the tail counts gravitons, which, while formally subleading to black holes at large $N$, are not terribly suppressed at $N = 10$.

Given this discussion, it is now natural to wonder why gravitons did not also contaminate the right tail of Fig. 16, allowing for such a close match between the grey galaxy prediction and the numerically computed entropy at at $N = 8, 9, 10$. The rightmost end of Fig. 16 lies at $j_R = 0.15$. Since all of Fig. 16 is plotted at $q + j_L = 0.15$, this point only receives contributions from states that obey $q + j_L = j_R$, which implies $q + j_1 = 0$. Once again, it follows from (22) that $q + j_1 \geq 0$, so once again this point only receives contributions from letters that carry $q + j_1 = 0$ (since no letter carries a negative $q + j_L$ charge, positive and negative charge contributions cannot cancel). The only letter in Table 1 that obeys this condition is $\bar{\psi}_{0+}$ and one type of derivative $D_{++}$. The only gravitons one can construct out of these letters are $D_{++}$ descendants of the $m = 0$ state on the third line of Table 2. These gravitons are rather trivial (they lie entirely in the $U(1)$ sector of $U(N)$). Moreover it has been shown that the index in this sector exactly vanishes [38]. We believe that the fact gravitons do not "contaminate" the rightmost tail of Fig 16 is the key reason for the rather remarkable fit between predictions and data in that Figure.

We were lucky in the analysis of $q_1 = q_2 = q_3 = 3$ in one additional respect. In that case grey galaxy solution predicts long tails beyond the unhairy black holes, even for relatively small values of $\frac{q_1+q_2+q_3}{3} + j_L$. These long tails makes it easier to distinguish the entropy of the grey galaxy from that of the unhairy black holes. Since computing the index is numerically simpler for smaller values of $\frac{q_1+q_2+q_3}{3} + j_L$, this comparison becomes more accessible. In contrast, the DDBH solution exhibits shorter tails for relatively small $\frac{q_1+q_2+q_3}{3} + j_L$ (see Fig. 34) and requires $\frac{q_1+q_2+q_3}{3} + j_L$ to be significantly larger to observe these tails. This makes the numerical computations more challenging.

## 6 Conclusions and discussion

In this paper, we have presented two conjectures relating to the spectrum of supersymmetric states in $\mathcal{N} = 4$ Yang-Mills theory. The first of these (the dressed concentration conjecture, see around (10)) leads to a definite prediction for the large $N$ entropy of supersymmetric states as a function of the five conserved charges $Z_i$ (see §3 and, in particular, §3.3). Our second conjecture (the unobstructed saddle conjecture, see around (7)), *together with the results of §3*, makes a definite prediction for the indicial entropy[108] as a function of the four indicial charges (§4, see esp. §4.3).

The dressed concentration conjecture essentially asserts the existence of supersymmetric grey galaxies or RBHs, and supersymmetric DDBHs (see the introduction for the expansion of these acronyms). As we have mentioned in the introduction, the evidence in favour of this conjecture seems rather strong to us. Supersymmetric RBHs - supersymmetric descendants of supersymmetric black holes - certainly exist, and mimic the thermodynamics of supersymmetric grey galaxies [25]. In the case of charge, supersymmetric DDBHs have been shown to exist in the probe approximation [26, 33], and there seems no reason to suspect that bulk back-reaction effects alter the situation. Moreover, recent results on the direct enumeration of supersymmetric cohomology - at $N = 2, 3$ and beyond - can be interpreted as displaying the contributions of both grey galaxies (or RBHs) and DDBHs [17, 20, 22]. Indeed the authors of [22] have constructed an infinite number of grey galaxy type states (i.e. elements of the supersymmetric cohomology that are products of "core black hole" type and graviton cohomologies) at all values of $N$, providing rather convincing evidence for the existence of at least

---

[108]Associated with the superconformal index.

some supersymmetric grey galaxy states.

Despite the (in our opinion persuasive) arguments reviewed above, it would certainly be useful to seek additional evidence in support of the dressed concentration conjecture. One way to proceed would be to construct exact bulk solutions for supersymmetric dressed dual black holes (DDBHs). In [26], DDBHs were constructed as an expansion in a power series (in an inverse power of the radial location of the dual giant graviton). In the supersymmetric limit, it may be possible to do better - to simply find these solutions exactly (see the discussion in the first few paragraphs of section 7 of [26]). Exact solutions of this nature would, in our opinion, put the existence of supersymmetric DDBH solutions beyond reasonable doubt.

Unlike DDBHs, grey galaxies represent ensembles - rather than particular configurations - over a large phase space associated with gravitons of large angular momentum propagating around the seed black hole. Supersymmetric grey galaxies exist should emerge from the quantization of a large gravitational phase space of supersymmetric solutions describing supersymmetric gravitons propagating around supersymmetric black holes. One way to make progress here would be to first search for solutions in any convenient (even if atypical) corner of phase space. One could, for instance, search for the classical solution corresponding to $\sim \frac{N^2}{\zeta}$ gravitons in a single coherent state, sharply localized around a mode that carries angular momentum $\zeta$ (where $\zeta$ is a large but fixed number). A linearized solution for such a mode (in a particular context) was presented in section 4 of [17]. An exact, nonlinear supersymmetric completion of this solution (and other similar solutions)[109] would, in our opinion, put the existence of supersymmetric grey galaxies beyond reasonable doubt.

A complementary approach (to the quest for evidence for the dressed concentration conjecture) would be to continue to push the field theoretic studies of $Q$ cohomology [11–22]. As we have explained above, the dressed concentration conjecture supplies a very definite prediction for the large $N$ cohomology of $Q$ as a function of charges. A verification of this result from the field theory side would constitute spectacular evidence for this conjecture.

In our opinion, the current evidence in favour of the unobstructed saddle conjecture is less overwhelming (than that for the dressed concentration conjecture). While the "numerical" check presented in §5 lends some support for this conjecture (the agreement between the data and our prediction in Fig 16 certainly appears striking to the eye), much more remains to be done. To start with it would be useful to push the "numerical" evaluation of the matrix integral to higher values of $N$.[110] Additional data over a range of values of $N$ would allow us to fit the deviation of the data from our predictions against $1/N$, ruling out the (anyway unlikely) possibility that the match in Fig 16 happens to be a numerical fluke.

Next, it would be useful to test the match between our predictions and numerical evaluations of the microcanonical index for different cuts of charges. Unfortunately, practicalities make this more difficult that one might initially suspect. The charge specialization of §5.2 (which led to the impressive agreement between our prediction and data, presented in Fig 16) was special for two reasons. First, our analytical predictions deviated significantly from the naive predictions of the black hole sheet even at manageable values of charges. Second, the contribution of gravitons to the index turned out to be negligible at values of charges at which this difference is significant (i.e. at the right tail of Fig. 16). These two facts together allowed the numerics of §5.2 to act as an effect check of our conjecture[111] at accessible values of $N$ and the charge. Other charge cuts will be useful for checking our conjecture, only if they have

---

[109]J. Santos has informed us that he is currently undertaking an investigation along these lines together with O. Dias and P. Mitra.

[110]Unfortunately, it seems difficult to go beyond $N = 10$ using the the method presented in this paper. Going to larger values of $N$ may require coming up with a new method for the computer based evaluation of the matrix integral.

[111]In contrast, the cut of charges studied in §5.3 satisfied neither of these properties. For this reason, the numerics of that section were ineffective in checking our conjecture.

these two properties. It would be interesting to search for such cuts.

It would eventually, of course, be most satisfying to verify the agreement between the superconformal index and the predictions of §4 in an analytical manner. Recall that while the matrix integral yields a formula for superconformal index in the grand canonical ensemble, in the main text of this paper have worked entirely in the microcanonical ensemble. In Appendix I, we have, however, presented a preliminary analysis of our new phases in the grand canonical ensemble.

Recall that we found in §4 that the microcanonical index lies in the "black hole phase" if, and only if, the index line in question intersects the black hole sheet at a "stable point", i.e. at a point at which[112]

$$\text{Re}(\nu_i) \geq 0, \quad \forall i. \tag{72}$$

Beyond this region, the analysis of §4 predicts that the index undergoes a phase transition. The resultant phases all include central black holes whose chemical potentials (approximately) saturate one of the inequalities listed above. In Appendix I we explain that the grand canonical version of the new phases presented in this paper - those that replace the "unstable" black holes with one or more $\nu_i$ negative - are phases in which one or more of the $\nu_i$ are simply zero. Infact a phase that has a central black hole with $Re(\nu_i) = 0$ turns out to be a grey galaxy (or DDBH) phase with the corresponding $\nu_i$ simply equal to zero (both the real and imaginary parts of the chemical potential vanish). It would be fascinating to reproduce this prediction directly from the unitary matrix integral or from Euclidean gravity.

These facts suggest that the grand canonical index is indeed given by the usual formula (40) at chemical potentials that obey (72), but suggests the existence of new saddle points (or substantially, loop corrected values for the action of old saddle points) at values of the chemical potential that approach parametrically near to[113] saturation values of (72).

From the viewpoint of the bulk, the existence of the "wall", (72), for chemical potentials, may plausibly turn out to be an application of Witten's adaptation [34] of the Kontsevich-Segal [39] criterion for the legality of saddle points.[114] Witten demonstrated in [34] that the Euclidean version of (ordinary, non supersymmetric) rotating black holes in $AdS$ space pass this criterion when their rotational chemical potentials obey $\Omega_i < 1, \quad \forall i$, but fail this criterion when any $\Omega_i$ exceeds unity. It seems plausible that a similar result applies to supersymmetric black holes, with the role of $\Omega_i < 1$ being played by the requirement (34). It would be interesting to further investigate this point.

We have explained that the conjectured existence of supersymmetric grey galaxies and DDBH solutions makes dramatic predictions (i.e. a phase transition) for the leading order superconformal index at asymmetric values of indicial charges. At indicial charges that are nearer to symmetrical, the prediction of §4 is less dramatic; we learn that the superconformal index is dominated by vacuum black holes at these charges, and so the existence of grey galaxy and DDBH solutions does not affect the predicted value of the index at leading order in the large $N$ limit. Even at these relatively tame indicial charges, however, it seems likely that

---

[112]Benini and Milan [9] (see Fig. 3 of that paper) have argued that the saddles with Index equal to (40) become subdominant in the grand canonical ensemble even before the bound (72) is reached. The relevant saddles may well, however, continue to dominate the microcanonical ensemble all the way to the edge of the region (72) (we are assuming this is the case). A similar situation arises in the study of usual (non supersymmetric) black holes in $AdS_5$. While Schwarzschild black holes dominate the canonical ensemble only above the Hawking Page transition, they dominate the microcanonical ensemble at all energies of order $N^2$. We thank O. Aharony for a discussion on this point.

[113]In the case of DDBHs, this happens at $\mu \sim \mathcal{O}(1/N^2)$. In the case of grey galaxies this happens at $\omega \sim \mathcal{O}(1/N)$ when the grey galaxy is of rank 2, and at $\omega \sim \mathcal{O}(1/N^{\frac{2}{3}})$ when the grey galaxy is of rank 4. See [25–27] for an explanation of these estimates.

[114]We thank R. Mouland and S. Murthy for suggesting this possibility, and for very interesting discussion and correspondence on this point.

the existence of a large number of supersymmetric states at every point on the indicial line can be deduced from a careful analysis of the one loop determinants around the dominant saddle point. In particular the factor of $N^{10}$ in the indicial version of (37) yields a "one loop" contribution proportional to $\ln N$ to the index (see [40] for a discussion of other sources of $\ln N$ corrections to the Euclidean determinant). It would be interesting to isolate this contribution in the Euclidean determinant around supersymmetric black holes (see section 3.4 of [33] and [41,42] for related work).

The current paper is based on the assumption that supersymmetric grey galaxy (or RBH) and DDBH solutions are the entropically dominant supersymmetric phases at generic values of charges. As we have mentioned in the introduction, we, of course, cannot rule out the possible existence of as-yet-unknown new bulk supersymmetric black hole solutions that carry even higher entropy than supersymmetric grey galaxies and DDBHs. While we are unaware of any results that suggest this;[115] a definite refutation of this possibility would require a convincing match of field theory results with the prediction of §3 and 4.

This paper has been dedicated to an analysis of supersymmetric states in $\mathcal{N} = 4$ Yang Mills theory. However key qualitative features in our analysis - namely the existence of (known) supersymmetric black holes on a codimension one surface in charge space, plus the existence of new grey galaxies and (analogues of) DDBH solutions - are shared by many other context, including the bulk dual of the ABJM theory and the 6d $(0,2)$ theory. It would be interesting to repeat the analysis of this paper in these other contexts.[116]

Finally, as the analysis of §4 makes use of the results of §3, its results might reasonably appear to be contingent on the correctness of the dressed concentration conjecture. As an outlandish thought experiment, one could, however, conceive of scenario in which the dressed concentration conjecture[117] is somehow violated by subtle $\frac{1}{N}$ effects (so the prediction of §3 hold only in the strict large $N$ limit but are violated at finite $N$) but the predictions of §4 nonetheless continue to hold anyway. This could come about because the states that remain supersymmetric (after accounting for these $1/N$ effects) are concentrated around the charges at which the infinite $N$ cohomology was largest along every given index line.[118] Something like this does indeed happen in some simple toy models like supersymmetric SYK theory [24], with the role of $1/N$ (in the scenario spelt out above) being played by the couplings "$C_{i_1 \ldots i_q}$" of [24] (see around Fig. 2 of that paper). While this scenario seems highly unlikely in the current situation, it is, perhaps (barely) conceivable. We leave further study of this point - and several others raised in this section - to future work.

# Acknowledgments

We would like to thank O. Aharony, F. Benini, C. Chang, Y. Chen, M. Cvetic, A. Gadde, G. Horowitz, L. Iliesiu, Z. Komargodski, V. Kumar, F. Larsen, G. Mandal, D. Marolf, J. Minahan, P. Mitra, R. Mouland J. Mukherjee, S. Murthy, L. Pando Zayas, K. Papadodimas, O. Parrikar, A. Rahaman, J. Santos, A. Sen, K. Sharma, S. Trivedi, S. Van Leuven, Z. Yang and A. Zaffaroni for very useful discussions. We would also like to thank O. Aharony, F. Benini, C. Chang, Y. Chen, A.Grassi, Z. Komargodski, J. Maldacena, J. Minahan, R. Mouland, S. Murthy, J. Santos, A. Sen, B. Sia, S. Van Leuven, Z. Yang and A. Zaffaroni for useful comments on the manuscript.

---

[115]In particular localized $10d$ black holes appear always to be entropically subdominant - compared to 5d black holes - near the BPS limit. Also the evidence presented in the recent paper [43] suggests that hairy supersymmetric solutions may not exist. See also [44] for a related discussion.

[116]We thank A. Zaffaroni for pointing this out.

[117]Which, after all was motivated by bulk analysis in the semi classical limit.

[118]In this scenario, neither the dressed concentration conjecture nor the concentration conjecture would hold. Instead, the full supersymmetric cohomology - along every given index line - would be located at the "index maximizing" charges identified in §4.

**Funding information**   The work of S.C. was supported by World Premier International Research Center Initiative (WPI), MEXT, Japan. The work of D.J., S.M., C.P. and V.K. was supported by the J C Bose Fellowship JCB/2019/000052, and the Infosys Endowment for the study of the Quantum Structure of Spacetime. The work of S.K. and E.L. was supported in part by the National Research Foundation of Korea (NRF) Grant 2021R1A2C2012350. V.K. was supported in part by the U.S. Department of Energy under grant DE-SC0007859. V.K. would like to thank the Leinweber Center for Theoretical Physics for support. The work of E.L. was supported by Basic Science Research Program through the National Research Foundation of Korea (NRF) funded by the Ministry of Education RS-2024-00405516. C.P was supported in part by grant NSF PHY-2309135 to the Kavli Institute for Theoretical Physics (KITP). D.J. would like to thank Simons Collaboration on Celestial Holography for their support. D.J, V.K, S.M, and C.P would all also like to acknowledge our debt to the people of India for their steady support for the study of the basic sciences.

# A   A discussion of the unobstructed saddle conjecture

The unobstructed saddle conjecture asserts that the summation (7) is well approximated by its largest term in the large $N$ limit.

## A.1   An example of a sequence for which the unobstructed saddle conjecture fails

Consider

$$(1-x)^M = \sum_{m=0}^{M} \frac{(-1)^m (x)^m M!}{m!(M-m)!} \,. \tag{A.1}$$

In the large $M$ limit (and focussing on terms for which $m$ and $M-m$ are also large[119]), Sterling's approximation can be use to simplify the summand on the RHS (A.1), yielding

$$(1-x)^M \approx \sum_{m=0}^{M} \frac{(-1)^m x^m}{\left(1-\frac{m}{M}\right)^{M-m} \left(\frac{m}{M}\right)^m} \,. \tag{A.2}$$

In the large $M$ limit, $y = \frac{m}{M}$ is an effectively continuous variable. The modulus of summand (on the RHS of (A.2)) can be rewritten in terms of $y$ as

$$\left( \frac{|x|^y}{(1-y)^{1-y} y^y} \right)^M \,. \tag{A.3}$$

Notice that the generic term in the summand is exponentially large or exponentially small in $M$. It is easily verified that the quantity in (A.3) is maximized at $y = \frac{|x|}{1+|x|}$. Setting $y = \frac{|x|}{1+|x|} + \delta y$ into (A.3) we find

$$\left( \frac{|x|^y}{(1-y)^{1-y} y^y} \right)^M \bigg|_{y \to \frac{|x|}{1+|x|} + \delta y} = (1+|x|)^M e^{-M \frac{(1+|x|)^2}{2|x|} (\delta y)^2} + \mathcal{O}((\delta y)^3) \,. \tag{A.4}$$

When $x$ is negative, we see that the largest term on the RHS of (A.2) ($\delta y = 0$) is an excellent approximation to the LHS of (A.2). This happy situation arises when all terms in the summand of (A.2) are positive. When $x$ is positive, on the other hand, the largest term (approximately $(1+|x|)^M$) vastly overestimates the true answer, namely $(1-|x|)^M$. Consequently, when $x$ is positive, the analogue of the unobstructed saddle conjecture fails badly for the sum (A.2). Morally speaking, in this case the "saddle point" for the summation lies "off the summation contour".

---

[119]Such terms give the dominant contribution at large $M$.

## A.2  An intuitive explanation for the failure above

At positive values of $x$, the sum on the RHS of (A.2) clearly fails to be well approximated by its largest term because of cancellations against neighboring terms. These cancellations fail to be effective when the modulus of the terms with $m$ and $m+1$ differ significantly. As changing $m$ to $m+1$ changes $y$ by $\frac{1}{M}$, it follows from (A.4) that the ratio of the $m^{th}$ to the $(m+1)^{th}$ term approximately equals $e^{\frac{(1+|x|)^2\delta y}{|x|}}$ (at small $\delta y$, where the approximation (A.3) is valid). Consequently, cancellations are effectively obstructed at $\delta y \sim \frac{\alpha|x|}{(1+|x|)^2}$ where $\alpha$ is a number of order unity. Since we are working in a Taylor expansion in $\delta y$ this estimate is valid only when $x$ is small, and so $\delta y \sim \alpha|x|$. Plugging this value of $\delta y$ into the RHS of (A.3), we find the value

$$\left(1 + |x|(1 - \frac{\alpha}{2})\right)^N . \tag{A.5}$$

The crude estimate of this subsection thus suggests that the RHS of (A.2) should - in the the large $N$ limit, be well approximated by an expression of the form (A.5) for some choice of the order one number $\alpha$. Of course we know independently that this is the case, with $\alpha = 4$.

## A.3  The impact of randomness

The summation over the integer $m$ in (7) has many similarities to the summation on the RHS of (A.2). In the large $M$ limit, the modulus of the summand in (A.2) is an expression of the form $e^{Mg(m/M)}$. In a similar manner, we expect the modulus of the summand in (7), on average and at leading order in the large $N$ limit, to take the form $e^{N^2S(m/N^2)}$ for a value of the function $S$ whose form is determined (on average) by the analysis of §3.

Were the summand in (7) to be (the restriction to integer values of) an analytic function with a smooth maximum (as was the case in (A.2)) then, we would once again expect the analysis of the sum in (7) to be very similar to the analysis of (A.2) in the previous two subsections. In this situation, as above, phase cancellations would ensure that terms around the maximum do not contribute significantly to the sum and the unobstructed saddle conjecture would almost certainly not hold.

However we do not expect the modulus of the summand in (7) to be a completely smooth function of $m/N^2$. The number of black hole states as a function of charge is expected to be determined by diagonalizing a "one loop" Hamiltonian that is expected to display features of chaos [45]. For this reason it is natural to expect the number fortuitous states of $\mathcal{N} = 4$ Yang Mills theory not to be a completely smooth function of charges, but to include a random element.[120] We thus expect that the modulus of the summand in (7) is better modelled, in the large $N$ limit, by an expression of the form

$$e^{N^2S(m/N^2)+r(m)}, \tag{A.6}$$

---

[120]In contrast, one should expect the indicial entropy to be a smooth function of indicial charges, as the index can be computed entirely within the free theory, and this computation lacks the chaotic element that could give rise to randomness. Indeed one can experimentally check that the microcanonical superconformal index *is not* equal (in the large $N$ limit) to the maximal Free Yang Mills entropy along an indicial line. In the special case $Q_1 = Q_2 = Q_3 = Q$ and $J_1 = J_2 = J$, this point was already investigated in [11]. In the limit of large charges, the entropy of supersymmetric states as a function of charges was computed in Eq. 5.9 of that paper. In the language of that paper, the entropy along a given microcanonical indicial line is maximum when the chemical potential along the indicial line vanishes, i.e. when the chemical potentials obey the "indicial" condition $\mu = \frac{\beta}{3}$, and so $\mu \approx 0$ in the large charge limit. As explained at the end of subsection 5.1 of [11], setting $\mu = 0$ does not reproduce the correct relation between angular momentum and charge for large charge supersymmetric black holes (the scalings are right but the order one coefficient is wrong) and also yields and entropy larger than the indicial entropy at that charge. One can force the black hole relation between charge and angular momentum by setting $\mu = \mu_c$ defined in 5.24 of [11]. Even if one chooses to work at this non-entropy-maximizing point one still obtains an entropy that is larger than indicial entropy. (See, however, [10], for another viewpoint on the data).

where $r(m)$ is an effectively random number. We proceed by modeling this randomness in a crude manner, by taking each $r(m)$ is an independent Gaussian random variable with mean zero and standard deviation $\sigma^2(m/N)$ (postponing a discussion of the magnitude of $\sigma$ to later in this subsection). In other words, we assume that the random number $r(m)$ obeys the effective statistics

$$\langle r(m)\rangle = 0\,, \qquad \langle r(m)r(n)\rangle = \sigma^2(m/N)\delta_{m,n}\,. \tag{A.7}$$

We will now explain that the random fluctuations described above effectively obstruct phase cancellations provided that the standard deviation of the "noise" $r$ is parametrically larger than $e^{-N^2}$. Let us suppose that the function $S(m/N^2)$ is maximized at $y \equiv m/N = y_{max}$. Consider an interval of size $\delta y$ centred $y_{max}$, chosen so that the average entropy function $e^{S(m/N^2)}$ is approximately constant within this range.[121] The contribution to (7) from $m$ in this range is given approximately by

$$e^{N^2 S(y_{max})}C\,, \qquad C = \sum_{m=[N^2(y_{max}-\frac{\delta y}{2})]}^{m=[N^2(y_{max}+\frac{\delta y}{2})]}(-1)^m e^{r(m)}\,, \tag{A.8}$$

where we also assume $y_{max}$ is chosen so that the number of integers in the summation range is even (all these assumptions are made to ensure we are dealing with the most hostile possible case, i.e. a case in which phase cancellation would be perfect in the absense of randomness).

Upon averaging over randomness (and using $\langle e^{r(m)}\rangle = e^{\frac{\sigma^2}{2}}$) we find

$$\langle C\rangle = e^{\frac{\sigma^2}{2}}\sum_m (-1)^m = 0\,. \tag{A.9}$$

The expectation value of $C^2$ is also easily computed; we find

$$\langle C^2\rangle = N^2(\delta y)e^{2\sigma^2}(1-e^{-\sigma^2})\,, \tag{A.10}$$

where $\sigma = \sigma(y_{max})$ (we have ignored the variation of $\sigma$ in the $y$ range under study).

The value of $C$ for a typical draw of the random ensemble is of order

$$\sqrt{\langle C^2\rangle} = N(\delta y)e^{N^2 S(y_{max})}e^{\sigma^2}\sqrt{1-e^{-\sigma^2}}\,. \tag{A.11}$$

The unobstructed saddle conjecture holds if $\sigma$ is of order unity. If $\sigma$ is a smaller number then (A.11) simplifies to $(\delta y)N\sigma e^{N^2 S(y_{max})}$. Clearly, the unobstructed saddle conjecture holds provided

$$\sigma \gg \mathcal{O}(e^{-N^2})\,. \tag{A.12}$$

If $\sigma$ takes this exponentially small value, on the other hand, then randomness is likely insufficient to obstruct phase cancellations.

While we do not have a particularly clear expectation for the size of $\sigma$, we feel that it may even be or order unity because

- This is the leading order at which quantum corrections are expected to modify the entropy.

- The difference between the average entropy of states at charge $m$ and the average entropy at charge $m+1$ is of order unity.

---

[121]In the simple example studied in the previous section, we would have $\delta y = \alpha x$ with $\alpha \ll 1$ when $x$ is small.

- Numerical studies of the index show oscillations of the entropy (of order unity) upon changing charges by order unity.[122]

In summary, the lesson of this Appendix is that any randomness in the summand in the entropy that is larger that of order $e^{-N^2}$ effectively obstructs phase cancellation, leading to the unobstructed saddle conjecture.

# B   The partition function over the supersymmetric gas

## B.1   Transforming between Cartan bases of $SU(4)$

In (3), and through much of this paper, we use the eigenvalues $Q_1$, $Q_2$ and $Q_3$ under the the rotations in the three orthogonal two planes in an embedding $R^6$, as a basis for $SO(6)$ eigenvalues. We will occasionally find it useful to use a second basis, $R_i$, $i = 1 \ldots 3$ for $SU(4)$. $R_i$ are defined to be the diagonal $SU(4)$ matrices with 1 in the $i^{th}$ diagonal entry, $-1$ in the $(i+1)^{th}$ diagonal entry, and zero everywhere else. For the highest weight, $R_i$ are the number of columns of height $i$ in the Young Tableaux. The translation between $Q_1, Q_2, Q_3$ and $R_1, R_2, R_3$ is given by

$$Q_1 = \frac{R_1 + 2R_2 + R_3}{2}, \qquad R_1 = Q_2 + Q_3,$$
$$Q_2 = \frac{R_1 + R_3}{2}, \qquad R_2 = Q_1 - Q_2, \tag{B.1}$$
$$Q_3 = \frac{R_1 - R_3}{2}, \qquad R_3 = Q_2 - Q_3.$$

Our special supercharge $\mathcal{Q}$ carries $R_1 = 1$, $R_2 = 0$ and $R_3 = 0$.

## B.2   The (anti) commuting subalgebra $PSU(2, 1|3)$

As mentioned in the main text, states annihilated by $\mathcal{Q}$ transform in representations of the part of the superconformal algebra that commutes (or anticommutes) with $\mathcal{Q}$ and its Hermitian conjugate, i.e. of the sub superalgebra $PSU(2, 1|3)$. The bosonic subalgebra of this superalgebra is $SU(3) \times SU(2, 1)$. The Cartan charges of $SU(2, 1)$ can be taken to be $E + J_1$ and $J_1 - J_2$.[123] The $SU(3)$ is the obvious subalgebra of $SU(4)$. The Cartans of this $SU(3)$ can be taken to be $R_2 = Q_1 - Q_2$ and $R_3 = Q_2 - Q_3$. The 9 anticommuting charges of $PSU(2, 1|3)$ all have $\Delta = 0$ and transform in the bifundamental under $SU(2, 1) \times SU(3)$. They carry charges

$$\begin{pmatrix} (\frac{1}{2}, -\frac{1}{2}, -\frac{1}{2}, \frac{1}{2}, \frac{1}{2}) & (-\frac{1}{2}, \frac{1}{2}, -\frac{1}{2}, \frac{1}{2}, \frac{1}{2}) & (-\frac{1}{2}, -\frac{1}{2}, \frac{1}{2}, \frac{1}{2}, \frac{1}{2}) \\ (-\frac{1}{2}, \frac{1}{2}, \frac{1}{2}, \frac{1}{2}, -\frac{1}{2}) & (\frac{1}{2}, -\frac{1}{2}, \frac{1}{2}, \frac{1}{2}, -\frac{1}{2}) & (\frac{1}{2}, \frac{1}{2}, -\frac{1}{2}, \frac{1}{2}, -\frac{1}{2}) \\ (-\frac{1}{2}, \frac{1}{2}, \frac{1}{2}, -\frac{1}{2}, \frac{1}{2}) & (\frac{1}{2}, -\frac{1}{2}, \frac{1}{2}, -\frac{1}{2}, \frac{1}{2}) & (\frac{1}{2}, \frac{1}{2}, -\frac{1}{2}, -\frac{1}{2}, \frac{1}{2}) \end{pmatrix}. \tag{B.2}$$

The first row in (B.2) represent the supersymmetries $\mathcal{Q}^i_{\frac{1}{2}}$, where $i = 2, 3, 4$ are fundamental $SU(4)$ labels and the subscript denotes the value of $J_L$. The supersymmetries in the second and third rows of (B.2) are the supercharges $(\bar{\mathcal{Q}}_i)_{\pm\frac{1}{2}}$, where the first subscript denotes anti-fundamental $SU(4)$ indices, and the second subscript denotes $J_R$ values.

## B.3   Partition function of the supersymmetric gas at low energy

The evaluation of the supersymmetric partition function over the supersymmetric chiral gas is a simple exercise [11]. In this appendix, we present a brief review of this computation.

---

[122]As we have mentioned in the previous footnote, it seems reasonable to us that these indicial oscillations are the regular analogues more chaotic fluctuations in the number of black hole state.

[123]Using the BPS relation, the first of these can be rewritten as $Q_1 + Q_2 + Q_3 + 2J_1 + J_2$.

We wish to compute the partition function

$$Z = \text{Tr}\left(x^{2E} z^{2J_L} y^{2J_R} v^{R_2} w^{R_3}\right)$$
$$= \text{Tr}\left((x^2 z)^{2L_L} y^{2J_R} x^{3R_1} (x^2 v)^{R_2} (xw)^{R_3}\right) \tag{B.3}$$
$$= \text{Tr}\left((x^2)^{2J_L + Q_1 + Q_2 + Q_3} v^{Q_1 - Q_2} w^{Q_2 - Q_3} y^{2J_R} z^{2J_L}\right),$$

where in the second line, we have used the BPS condition: $E = 2J_L + \frac{3R_1}{2} + R_2 + \frac{R_3}{2}$ (recall the charges $R_i$ are related to the charges $Q_i$ via (B.1)). The fugacities $x, y, z, v$ and $w$ are related to the renormalized chemical potentials defined in (24) via

$$x^2 v = e^{-\mu_1}, \qquad x^2 w/v = e^{-\mu_2}, \qquad x^2/w = e^{-\mu_3}, \qquad z = e^{\frac{\mu_1 + \mu_2 + \mu_3}{3} - \frac{\omega_L}{2}}, \qquad y = e^{-\frac{\omega_R}{2}}. \tag{B.4}$$

The partition function over the supersymmetric gas is given (using the formulae of Bose

Table 2: A list of the supersymmetric gas "primaries". For clarity in presentation, we set $\mu_1 = \mu_2 = \mu_3 = \mu$ (Note that the conventions differ from the notation used in (G.10).). Each line in Table 2 denotes a "conformal primary": one obtains supersymmetric states from these "primaries" by acting on them with all the supersymmetric derivatives. Each line in Table 2 denotes an irreducible representation of $SU(3) \times SU(2)_R$. We use the following notation for $SU(3)$ representations: $R'_1 = R_2$ denotes the denotes the number of columns of length unity, while $R'_2 = R_3$ denotes the number of columns of length two in the $SU(3)$ Yong Tableaux. $J_R$ representations also listed in the usual manner. $J_L$ and $R_1$ are charge rather than representation labels. The $J_L$ lists the $z$ component of the $J_L$ charge, while $R_1$ lists the values of the highest weight $SU(4)$ charge. The last line denotes a supersymmetric equation of motion (null state). Note that a supersymmetric equation of motion exists only in the $U(1)$ sector: there are no such equations of motion in descendants of $tr(X^m)$ for $m \geq 2$. The partition function in the last column is obtained after summing over all values of $m$ from $m = 1$ to $m = \infty$. For each row we get the answer by multiplying the summation of characters over chiral primaries (first row divided by denominator - this is Bose statistics - the $-1$ is because there is no operator with $m = 0$) with the character of the extra letter, and then subtracting away "contractions" (to implement the Clebsch-Gordon to the representations of interest) in the case that the extra letters are charged under $SU(3)$i.e. in rows containing $\psi$, i.e. in rows 2, 6 and 8. The form of the operators listed in Table 2 is highly schematic - its merely a suggestive way of listing the charges of the corresponding representations.

| Word | $J_L$ | $J_R$ | $R_1$ | $R_2$ | $R_3$ | Numerator($n_r$) |
|---|---|---|---|---|---|---|
| $\text{Tr}(X^m)$ | 0 | 0 | 0 | $m$ | 0 | $1 - (1 - e^{-\mu_1})(1 - e^{-\mu_2})(1 - e^{-\mu_3})$ |
| $\text{Tr}(\psi X^m)$ | $\frac{1}{2}$ | 0 | 0 | $m$ | 1 | $(e^{\mu_1} + e^{\mu_2} + e^{\mu_3} - 1) e^{\frac{1}{2}(-\mu_1 - \mu_2 - \mu_3 - \omega_L)}$ |
| $\text{Tr}(\bar{\psi} X^m)$ | 0 | $\frac{1}{2}$ | 1 | $m$ | 0 | $(e^{\omega_R} + 1) e^{\frac{1}{2}(-\mu_1 - \mu_2 - \mu_3 - \omega_R)}$ |
| $\text{Tr}(F X^m)$ | 1 | 0 | 0 | $m$ | 0 | $e^{-\omega_L}$ |
| $\text{Tr}(\bar{\psi}\bar{\psi} X^m)$ | 0 | 0 | 2 | $m$ | 0 | $e^{-\mu_1 - \mu_2 - \mu_3}$ |
| $\text{Tr}(\psi\bar{\psi} X^m)$ | $\frac{1}{2}$ | $\frac{1}{2}$ | 1 | $m$ | 1 | $(e^{\mu_1} + e^{\mu_2} + e^{\mu_3} - 1)(e^{\omega_R} + 1) e^{-\mu_1 - \mu_2 - \mu_3 - \frac{\omega_L}{2} - \frac{\omega_R}{2}}$ |
| $\text{Tr}(F\bar{\psi} X^m)$ | 1 | $\frac{1}{2}$ | 1 | $m$ | 0 | $(e^{\omega_R} + 1) e^{\frac{1}{2}(-\mu_1 - \mu_2 - \mu_3 - 2\omega_L - \omega_R)}$ |
| $\text{Tr}(\psi\bar{\psi}\bar{\psi} X^m)$ | $\frac{1}{2}$ | 0 | 2 | $m$ | 1 | $(e^{\mu_1} + e^{\mu_2} + e^{\mu_3} - 1) e^{\frac{1}{2}(-3(\mu_1 + \mu_2 + \mu_3) - \omega_L)}$ |
| $\text{Tr}(F\bar{\psi}\bar{\psi} X^m)$ | 1 | 0 | 2 | $m$ | 0 | $e^{-\mu_1 - \mu_2 - \mu_3 - \omega_L}$ |
| $\text{Tr}(D_{+\dot{\alpha}}\bar{\psi}^{\dot{\alpha}})$ | $\frac{1}{2}$ | 0 | 0 | 0 | 0 | $(e^{\mu_1} - 1)(e^{\mu_2} - 1)(e^{\mu_3} - 1) e^{\frac{1}{2}(-3(\mu_1 + \mu_2 + \mu_3) - \omega_L)}$ |

and Fermi statistics) by

$$Z_{mp} = \exp\left\{\sum_n \left(\frac{Z_{SP}^B(x^n, y^n, z^n, w^n, v^n)}{n} + (-1)^{n+1}\frac{Z_{SP}^F(x^n, y^n, z^n, w^n, v^n)}{n}\right)\right\}, \qquad \text{(B.5)}$$

where $Z_{SP}^B(x^n, y^n, z^n, w^n, v^n)$ is the bosonic part of the "word partition function" (and $Z_{SP}^F(x^n, y^n, z^n, w^n, v^n)$ is its fermionic counterpart.

Concretely,

$$Z_{SP}^{B/F} = \frac{n_r}{d_r}, \qquad \text{(B.6)}$$

where $n_r$ is the sum of the entries in the last column of Table 2 and $d_r$ is the denominator[124]

$$d_r = \left(1 - e^{-\mu_1}\right)\left(1 - e^{-\mu_2}\right)\left(1 - e^{-\mu_3}\right)\left(1 - e^{-\frac{\omega_L}{2} - \frac{\omega_R}{2}}\right)\left(1 - e^{\frac{\omega_R}{2} - \frac{\omega_L}{2}}\right). \qquad \text{(B.7)}$$

Using Table 2, we derive the explicit expressions for the single-particle (single-word) bosonic and fermionic partition functions.

$$Z_{SP}^B = \frac{n_B}{d_r}, \qquad Z_{SP}^F = \frac{n_F}{d_r}, \qquad \text{(B.8)}$$

$$n_B = e^{-\mu_1} + e^{-\mu_2} + e^{-\mu_3} - (e^{-\mu_1-\mu_2} + e^{-\mu_1-\mu_3} + e^{-\mu_2-\mu_3})(1 - e^{-\frac{\omega_L}{2} - \frac{\omega_R}{2}} - e^{-\frac{\omega_L}{2} + \frac{\omega_R}{2}})$$
$$\qquad + e^{-\mu_1-\mu_2-\mu_3}(2 - e^{-\frac{\omega_L}{2} + \frac{\omega_R}{2}} + e^{-\omega_L}) + e^{-\omega_L},$$

$$n_F = 2e^{\frac{1}{2}(-\mu_1-\mu_2-\mu_3-\omega_L)}\left(\cosh(\mu_1) + \cosh(\mu_2) + \cosh(\mu_3) + 2\cosh\left(\frac{\omega_L}{2}\right)\cosh\left(\frac{\omega_R}{2}\right) - 1\right).$$

# C   More about supersymmetric charge space and the black hole sheet

## C.1   The boundaries of the Bose-Fermi cone are all $(1/8)^{th}$ BPS

In this brief subsection, we explain why (22) represents the condition for $(1/8)^{th}$ supersymmetry.

Recall that the 16 $\mathcal{Q}s$ (i.e. supercharges with energy $\frac{1}{2}$ rather than $-\frac{1}{2}$) carry charges $(\alpha_1, \alpha_2, \alpha_3, \alpha_4, \alpha_5)$ with each $\alpha$ either $+\frac{1}{2}$ or $-\frac{1}{2}$, and subject to the constraint that the product of the $\alpha$'s is positive. The anticommutator of any of these supercharges with their complex conjugate takes the form (13), but with the RHS of (13) replaced by

$$E - (\alpha_1 Q_1 + \alpha_2 Q_2 + \alpha_3 Q_3 - \alpha_4 J_1 - \alpha_5 J_2). \qquad \text{(C.1)}$$

We see from (53) that our special supercharge has $\alpha_1 = \alpha_2 = \alpha_3 = \frac{1}{2}$, and $\alpha_4 = \alpha_5 = -\frac{1}{2}$. Now consider the supercharge with the $i^{th}$ and $j^{th}$ signs of $\alpha_i$ flipped. A state that is annihilated by both this supercharge and $\mathcal{Q}$ must obey both (13) as well as the analogous equation with the RHS replaced by (C.1), and so must have $\zeta_i + \zeta_j = 0$.

States that are annihilated by two of the 16 (positive energy) supersymmetries are often referred to as $(1/8)^{th}$ BPS. These states (and so the boundaries of the Bose-Fermi cone) are of three different qualitative types.

---

[124]$d_r$ is a result of summing over all insertions of symmetrized $X, Y, Z$ as well as overall supersymmetric derivatives.

1. When $(i, j) = (4, 5)$. In this case $J_1 + J_2 = 0$. States in this sector have $J_L = 0$, arbitrary values of $J_R$, and have $E = Q_1 + Q_2 + Q_3$. The corresponding operators are often called the $\frac{1}{8}$-BPS chiral ring sector.[125]

2. When $(i, j) = (1, 2)$ or $(1, 3)$ or $(2, 3)$. When, for example, $(i, j) = (1, 2)$, $Q_1 + Q_2 = 0$, states carry $E = Q_3 + J_1 + J_2$.[126]

3. The case $i \in (1, 2, 3)$, $j \in (4, 5)$. E.g. $(i, j) = (1, 4)$. In this case $Q_1 + J_1 = 0$. The BPS bound is $E = Q_2 + Q_3 + J_2$. The corresponding states lie in the, so called, $\frac{1}{8}$-BPS Macdonald sector.[127] Such states will turn out to form the boundary of the "Indicial cone" that we defined in the main text.

## C.2 Allowed charges for supersymmetric states after accounting for Pauli exclusion

### C.2.1 The fermionic polyhedron

The Fermi exclusion principle ensures that no particular fermionic letter can be occupied more than $N^2$ times (this degeneracy is permitted because each fermionic letter has $N^2$ gauge indices). Consequently, there are now states with the charges in the interior of Bose-Fermi cone that will be excluded due to Fermi exclusion principle.

In order to proceed, we first present a complete listing of all fermionic letters (including the action of an arbitrary number of supersymmetric derivatives). These are

$$
\begin{aligned}
v_6^{m,n} &= \left( -\frac{1}{2}, \frac{1}{2}, \frac{1}{2}, m + \frac{1}{2}, n + \frac{1}{2} \right) & (m, n \geq 0), \\
v_7^{m,n} &= \left( \frac{1}{2}, -\frac{1}{2}, \frac{1}{2}, m + \frac{1}{2}, n + \frac{1}{2} \right) & (m, n \geq 0), \\
v_8^{m,n} &= \left( \frac{1}{2}, \frac{1}{2}, -\frac{1}{2}, m + \frac{1}{2}, n + \frac{1}{2} \right) & (m, n \geq 0), \\
v_9^{n} &= \left( \frac{1}{2}, \frac{1}{2}, \frac{1}{2}, -\frac{1}{2}, n + \frac{1}{2} \right) & (n \geq 0), \\
v_{10}^{m} &= \left( \frac{1}{2}, \frac{1}{2}, \frac{1}{2}, m + \frac{1}{2}, -\frac{1}{2} \right) & (m \geq 0), \\
v_{11}^{m,n} &= \left( \frac{1}{2}, \frac{1}{2}, \frac{1}{2}, m + \frac{1}{2}, n + \frac{1}{2} \right) & (m, n \geq 0).
\end{aligned}
\tag{C.2}
$$

States built entirely out of fermionic letters, in a manner that obeys the Pauli principle, have charge vectors $z$ that take the form

$$
z = \sum_{i=6}^{11} \sum_{m,n \geq 0} \lambda_i^{m,n} v_i^{m,n} \qquad (0 \leq \lambda_i^{m,n} \geq 1 \; \forall (i, m, n)).
\tag{C.3}
$$

This condition defines an infinite polyhedron. This polyhedron is bounded by surfaces obtained by setting all but 4 of the $\lambda_i^{m,n}$ to either 0 or 1. The normal vector of each segment of the

---

[125] Accounting for both the Gauss Law and interactions, all states of this sort have been (conjecturally) enumerated (see section 6 of [11]). The entropy of such states is parametrically smaller than $N^2$ at values of $\zeta_i$ that are of order unity.

[126] After accounting for both the Gauss law and interactions, states of this form have been explicitly analyzed in [13]. Similar to the previous case, no BPS states with $N^2$ entropy scaling have been found at values of $\zeta_i$ of order unity.

[127] While states in this sector have not been fully enumerated (after accounting for the Gauss law and interactions) it is believed that the number of states with charges $\zeta_i$ of order unity is less than $N^2$ [46, 47].

outermost boundary of this polyhedron, once again, obeys the condition (21) for all vectors $i$ (once again the normal is always inward pointing: note that inward pointing means toward greater values if $\lambda_i = 0$, but towards smaller values if $\lambda_i = 1$).

### C.2.2 The allowed region

The fermionic polyhedron describes all charges one can obtain only from fermionic letters. One can now add Bosons to the mix by regarding each point on the polyhedron as the origin of a bosonic cone. The union of all points swept out by all these cones gives the allowed charge region.[128]

Explicitly characterizing the boundary of the Allowed Region at generic charges appears to be an involved task. It is, however, easy to give an explicit description of this boundary at small and large values of charges.

At small enough values of $\zeta_i$ (compared to unity), the Fermi principle is unimportant, and the full allowed region reduces exactly (i.e. without error) to the inside of the Bose-Fermi cone.

Conversely, at large charges all fermionic letters that matter have a large number of derivatives, and so large angular momentum to charge ratio. Thus the charges of these letters are linear combinations of $\zeta_4$ and $\zeta_5$. Consequently, at these large values of charges, the allowed region tends to the bosonic cone from outside. In the rest of this subsection, we make this last point quantitative.[129]

The reason that the allowed region always extends outside the bosonic cone is that fermionic letters are allowed to carry negative values of each of the $q_i$ and also of each of the $j_i$. Let first estimate how negative $q_1$ can become at any given value of $j_L = \frac{j_1 + j_2}{2}$. We clearly get the most negative possible value of $q_1$ by building Fermi Seas out of $v_6^{m,n}$ for the smallest possible values of $m + n$. Let us suppose we occupy all $v_6^{m,n}$ up to $m + n = p$ for some large value of $p$. This gives us $q_1 \sim -\frac{p^2}{4}$ but $j_L \sim \frac{p^3}{6}$. In other words we see that

$$|q_1|^{\frac{3}{2}} \leq \frac{4}{3} j_L, \quad \text{when } q_1 < 0. \tag{C.4}$$

In a similar manner, the way to make $j_1$ maximally negative is to occupy the letters $v_9^n$ for values of $n$ that are as small as possible. If we occupy these letters up to $n = p$ (for large $p$ we have $j_1 = -\frac{p}{2}$ but $j_L = \frac{p^2}{4}$. In other words

$$|j_1|^2 \leq j_L \approx \frac{j_2}{2}, \quad \text{when } j_1 < 0 \tag{C.5}$$

(this equation bounds how negative $j_1$ can become).

The bounds (C.4) and (C.5) apply to both the fermionic polyhedron and the Allowed Region. They demonstrate that both these structures tend (from the outside) to the bosonic cone at large values of the charges.

### C.3 Field theory evaluation of the superconformal index

We have already explained (see the end of section 2.9) that direct analysis of the field theory path integral allows one to evaluate the path integral that evaluates the $\mathcal{N} = 4$ in terms of a matrix integral over $N \times N$ unitary matrix. The last few years has seen dramatic progress in

---

[128]Alternate, we can imagine a fermionic polyhedron living at each point within the bosonic cone and take the union of all these polyhedra.

[129]The reader who finds herself uninterested in this exercise can safely skip to the next subsection.

the evaluation of this matrix integral in the large $N$ limit (as well as in the ultra high charge "Cardy limit"). This evaluation has been performed[130]

- By directly finding a saddle point of the matrix integral [48–53]

- By using the (so called) Bethe Ansatz method [9, 42, 54–58].

- Using the "giant graviton" expansion [59–61].

- At large charges in the so called Cardy limit [8, 58, 62–73]

- By an exact evaluation of the unitary integral (on a computer) at $N \le 10$ [36, 37].

## C.4 The charges of high energy graviton states Lie within the bosonic cone

Gas modes are in one to one correspondence with single trace operators built out of Yang-Mills letters, and so all lie within the Allowed Region (see §C.2.2). However the charges of gas modes are further constrained by the fact that (see §2.10)

- All bosonic gas modes have charges that lie within the bosonic cone.

- The only fermionic gas modes with charges outside the bosonic cone carry no more than a single fermionic (but arbitrarily many bosonic) letters (see lines 2 and 3 Table 2).

As single trace fermionic operators cannot be occupied more than once, this point imposes a lower bound on the fractional violation of the bosonic cone condition (in supersymmetric gas states) that is increasingly stringent at large charges.

In order to get a sense of how this works, consider states built entirely out of the traces listed in the second line of Table 2 (the analysis of states from the third line of the table is similar; the traces listed in all other lines carry charges within the bosonic cone and so are unoccupied in maximally violating configurations). The $m^{th}$ operator in line 2 of Table 2 has 1 fermionic letter but $m$ bosonic letters. These states transform in a representation of $SU(3)$ with $m + 1$ boxes in the first row, and 1 box in the second row. The number of states in this representation equals $(m+1)(m+3) \approx m^2$ (we assume that $m \gg 1$). The state in which all of these traces (for $n = 1, 2, \ldots m$) are occupied thus has $\approx \sum_{n=1}^{m} n \times n^2 \approx \frac{m^4}{4}$ bosonic letters but only $\sum_{n=1}^{m} n^2 \approx \frac{m^3}{3}$ fermionic letters. As the total number of letters is a rough estimate of the charge $Q$ of the state, we see that the ratio of the number of fermionic to bosonic letters scales (at most) like $\frac{m^3}{m^4} = \frac{1}{m} \sim 1/Q^{\frac{1}{4}}$, and so becomes very small for $Q \gg 1$. We conclude that the charges accessed by the supersymmetric gas all lie within a region that is increasingly well approximated by the bosonic cone at values of charges that are large compared to unity.[131]

We have, so far, performed our analysis ignoring interactions. From a field theory viewpoint, however, it is easy to argue that interactions do not lift any multi graviton states at energies less than $N$. The argument proceeds as follows. We first recall that the spectrum of single trace operators is half BPS, and so completely protected against renormalization [11]. In order to deal with product of single trace operators, we recall that there is a one to one correspondence between states annihilated by both $\mathcal{Q}$ and $\mathcal{Q}^\dagger$, and cohomology classes of the nilpotent operator $\mathcal{Q}$ [13]. If $O_1$ and $O_2$ belong to the cohomology of $Q$, the same is true of

---

[130]Apart from the leading order large $N$ limit computation flagged above, the Bethe Ansatz formalism has been used to determine the one loop determinant about this saddle point [42] (this computation has not yet been matched against a similar calculation in gravity).

[131]This discussion has some similarities to the analysis of the Allowed region at large charge (see §C.2.2), but there is one important difference. While the allowed region goes over to the Bose Pyramid at charges that are large in units of $N^2$ (large values of $\zeta_i$) the charges of the supersymmetric gas becomes effectively bosonic at charges that are large compared to unity.

$O_1 O_2$. Moreover $(O_1 + QA)O_2 = O_1 O_2 \pm Q(AO_2)$, so the product is well defined for cohomology classes.[132] Now it is possible for the product of two nontrivial cohomology elements to yield an element that is trivial in cohomology. This only happens, however, upon using trace relations. At energies below $N$ all traces are independent; there are no nontrivial trace relations, and so multi gravitons are exactly supersymmetric.

## C.5 More about the boundary of the black hole sheet

As explained in the main text, the boundary of the black hole sheet is described by the equation (46). In this subsection we study, in turn

- The gluing cut $j_R = 0$

- The boundary of the black hole manifold formed by gluing the two sheets along the gluing cut

- The bulk of the black hole manifold, formed by filling in the boundary

### C.5.1 The gluing cut $j_R = 0$ of the boundary

The gluing cut of the boundary of the black hole manifold (the red equator of Fig. 2) is given by the equation

$$(q_1 q_2 + q_2 q_3 + q_3 q_1)^2 + 2q_1 q_2 q_3 = 0 \,. \tag{C.6}$$

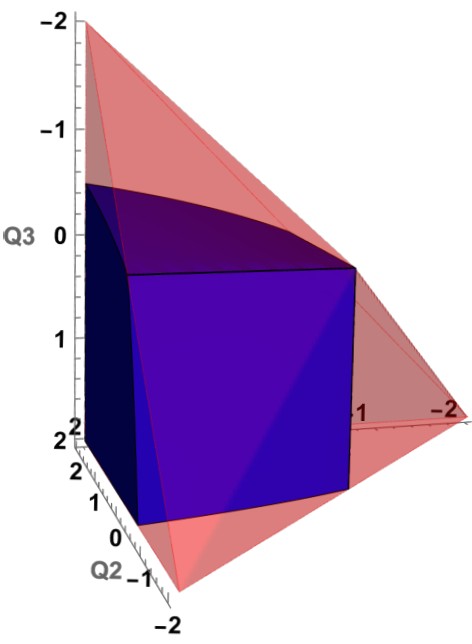

Figure 18: In the above figure, boundary of the black hole sheet is given by the solid blue region. Every point in the bulk of the blue region corresponds to two supersymmetric black hole solutions. These solutions degenerate on the boundary of the blue region. Therefore the topology of the blue region is that of an $S^2$ fibered over a half line. The red region represents the Bose-Fermi cone. The boundary of the black hole sheet intersects the Bose-Fermi cone on the three lines given in (C.7).

---

[132]The map to cohomology thus allows us to define a multiplication operation on the set of supersymmetric states.

In order to understand the nature of this surface, let us first note that 3 lines

$$q_1 = q_2 = 0, \qquad q_2 = q_3 = 0, \qquad q_3 = q_1 = 0, \tag{C.7}$$

clearly lie obey (C.6). Note that $j_1$ and $j_2$ are also both zero on each of these lines, so the lines describe half BPS configurations with a single nonzero $q_i$ greater than zero; each of these lines lies on the intersection of 6 of the 10 edge planes of the Bose-Fermi cone (see the next subsection for more on this).

Away from these lines, all solutions to (C.6) have one of the 3 $q_i$ negative.[133] If we linearize around the line $q_1 = q_2 = 0$ (at some positive value of $q_3$) we find the two surfaces

$$\frac{q_1}{q_2} = -1 - \frac{1}{q_3} \pm \frac{\sqrt{1+2q_3}}{q_3}.$$

The choice of $\pm$ is fixed by the requirement that we lie within the Bose-Fermi cone. After some processing we find that the relevant surfaces are

$$\begin{aligned}
q_3(q_1+q_2) &= -q_2\left(1+\sqrt{1+2q_3}\right) & (q_1 > 0, q_2 < 0), \\
q_3(q_1+q_2) &= -q_1\left(1+\sqrt{1+2q_3}\right) & (q_1 < 0, q_2 > 0).
\end{aligned} \tag{C.8}$$

When $q_3$ is small, the surfaces defined in (C.8) are approximately $q_2 = 0, q_1 > 0$ and $q_1 = 0, q_2 > 0$ (i.e. surfaces of the bosonic cone). When $q_3$ is large, on the other hand, these two surfaces both tend to $q_1 + q_2 = 0$ (a boundary of the Bose-Fermi cone). We see that the surface (C.6) is non analytic (kinky) in the neighbourhood of the three half BPS lines.

Of course the three lines described above meet at the origin. A little thought will convince the reader that the gluing cut of the boundary of the black hole sheet is thus a cone, whose base is a triangle. This cone is the union of the three boundaries of the surface depicted in Fig 18 (these are the visible boundary on the left side, plus the two boundaries -one below and one behind - that cannot be seen in the figure.

In terms of the schematic diagram Fig. 2, the triangle at the base of this cone is the red equator of the Fig 2 (which, though depicted as a circle in Fig 2, actually has 3 kinks, and so is really a triangle). The kinks on the red equator of 2- the vertices of this triangle occur exactly at the centre of the boundary regions of of the 120 degree red pie slices in Fig 2.

### C.5.2 The boundary of the black hole manifold

The boundary of the black hole manifold consists of two copies of the bulk of the shaded region in Fig. 18, glued together at the gluing cut (the boundary of the shaded region in Fig. 18).

Topologically the bulk of the shaded region in Fig. 18 is a cone whose base is a disk (formed by filling in the triangles of the previous subsubsection). Gluing these two disks together at their boundary gives an $S^2$. Consequently, the boundary of the black hole sheet is a cone with base $S^2$. This $S^2$ is the boundary of the apple in Fig 2. As we have seen above, the $S^2$ in question is not completely smooth: its surface has 3 kinks (we will see below that it has no other singularities).

### C.5.3 The black hole sheet itself

Clearly, the black hole sheet itself is a cone with base $B^3$, i.e. a fibration of $B^3$ over $R^+$. The $B^3$ is formed by filling in the $S^2$ of the previous subsubsection. This filling in happens in the angular momentum space (see the discussion around (C.12).

---

[133](C.6) cannot be obeyed unless $q_1 q_2 q_3$ is negative, and values with all three $q_i$ negative lie outside the Bose-Fermi cone.

## C.6 The black hole sheet in the neighbourhood of $(1/8)^{th}$ BPS surfaces

In this section we study precisely how the black hole sheet is located with respect to the boundaries of the Bose-Fermi cone. In particular we find all the locations at which this sheet touches the Bose-Fermi cone, and study the neigbhbourhood of these regions.

### C.6.1 The surface $q_i + q_j = 0$

We have seen above that the black hole sheet intersects the $(1/8)^{th}$ BPS plane $q_1 + q_2 = 0$ along the line $q_1 = q_2 = j_1 = j_2 = 0$ (configurations along this line are $\frac{1}{2}$ BPS). To better understand the structure of the black hole sheet in the neighbourhood of this line, we work at a fixed value $q_1 + q_2 = \epsilon$ (with small and a fixed $\epsilon$ and a fixed value of $q_3$ (not necessarily small)). From (C.8) it follows that that the boundary of the black hole sheet is given by the straight line connecting the points $(q_1, q_2)$ given by,

$$\frac{\epsilon}{2}\left(1 + \sqrt{1 + 2q_3}, 1 - \sqrt{1 + 2q_3}\right), \qquad \frac{\epsilon}{2}\left(1 - \sqrt{1 + 2q_3}, 1 + \sqrt{1 + 2q_3}\right). \tag{C.9}$$

The angular momenta of the black hole along this line both of order $\epsilon$, and are given, at leading order by

$$\begin{aligned}
j_1 &= q_3(q_1 + q_2) \pm \sqrt{q_3^2(q_1 + q_2)^2 + 2q_1 q_2 q_3}, \\
j_2 &= q_3(q_1 + q_2) \mp \sqrt{q_3^2(q_1 + q_2)^2 + 2q_1 q_2 q_3}.
\end{aligned} \tag{C.10}$$

Adding these two equations gives

$$j_1 + j_2 = 2q_3 \epsilon. \tag{C.11}$$

Consequently, three of the 5 charges (namely $q_3$, $q_1 + q_2$ and $j_1 + j_2$) are fixed in terms of $q_3$ and $\epsilon$. In order to see the shape traced out in a plane formed by the remaining charges, namely $q_R = \frac{q_1 - q_2}{2}$ and $j_R = \frac{j_1 - j_2}{2}$, we subtract the two equations (C.10) (and use $(q_1 + q_2) \to \epsilon$, $q_1 q_2 \to \frac{1}{4}(\epsilon^2 - 4q_R^2)$ ) to find

$$j_R^2 + (2q_3)q_R^2 = \epsilon^2\left(q_3^2 + \frac{q_3}{2}\right). \tag{C.12}$$

Clearly (C.12) describes a small ellipse in the $j_R$-$q_R$ space. Note that the radius of the ellipse is proportional to the distance, $\epsilon$, from the BPS manifold. Varying over $\epsilon$, thus, gives us a pointy cone with apex on the BPS sheet.

Let us summarize. The black hole sheet in the neighbourhood of the $(1/8)^{th}$ BPS sheet $q_1 + q_2 = 0$, has the structure $R \times C$. $R$ is a line parameterized by $q_3$, while $C$ is a solid ice cream cone. The apex of the cone lies on the $(1/8)^{th}$ BPS sheet along the line $q_1 = q_2 = j_1 = j_2 = 0$. The axis of the cone is $\epsilon = q_1 + q_2 = \frac{j_1 + j_2}{2q_3}$ (the equality between the last two quantities may be taken as a statement of the black hole sheet in the neighbourhood of the $(1/8)^{th}$ BPS black hole). The base of the cone is the inside of the ellipse (C.12) the $j_1 - j_2$ and $q_1 - q_2$ plane. The boundary of the black hole sheet is the surface of this cone.

### C.6.2 The surface $j_1 + j_2 = 0$

As we have explained above, the intersection of the black hole sheet with the plane $q_i + q_j = 0$ automatically has $j_1 + j_2 = 0$. We have shown, above, that small deformations of this line obey $j_1 + j_2 = (q_1 + q_2)q_3 = \epsilon q_3$. Consequently, no small deformation of this line obeys $j_1 + j_2 = 0$. This already suggests that the black hole sheet intersects the plane $j_1 + j_2 = 0$ only on the 3 half BPS lines described above. This result is easily established in generality. For finite $q_1, q_2, q_3$ with $j_1 + j_2 = 0$, there is no black hole solution. The $q$'s must be positive to satisfy the second

equation of (44). However, the RHS of (43) is always greater than the LHS, preventing it from satisfying the non-linear charge relation (43). Similarly, if two of the $q$'s are finite and one is near zero, a black hole solution cannot be found. Thus, we conclude that the black hole sheet intersects the plane $j_1 + j_2 = 0$ only along the three half-BPS lines described above.

### C.6.3 The surface $q_1 + j_1 = 0$

As $q_1 + j_1$ (and the 5 similar charges) are all constant along index lines, this surface will be of particular interest to the study of the index.

It is easy to see that the intersection of the black hole sheet and this $(1/8)^{th}$ surface BPS surface is given by the points with $q_1 = 0$ on the "second sheet" of the boundary (46).[134] Note that this intersection is two dimensional, and so is codimension one on the boundary of the black hole sheet (but is codimension 2 on the black hole sheet itself, as well as on the plane $q_1 + j_1 = 0$). We can use $q_2$ and $q_3$ as coordinates for this intersection region.[135] On this intersection, $j_2 = 2q_2 q_3$.

Let us now study the black hole sheet in the neighbourhood of this plane, i.e. at small $q_1$. Taylor expanding the second of (46) in $q_1$ and using $j_1 = j_L + j_R$, $j_2 = j_L - j_R$, we find that the boundary of the black hole sheet is given by

$$j_1 = -q_1 + \frac{1 + 2q_2 + 2q_3}{2q_2 q_3} q_1^2 + O(q_1^3),$$
$$j_2 = 2q_2 q_3 + q_1(1 + 2q_2 + 2q_3) + O(q_1^2). \tag{C.13}$$

At leading order $q_1 + j_1 = \frac{1 + 2q_2 + 2q_3}{2q_2 q_3} q_1^2$ while (again at leading order) $q_1 - j_1 = 2q_1$. In the neighbourhood of $q_1 = j_1 = 0$, consequently, the boundary of the black hole sheet is given by the parabola

$$(q_1 + j_1) = \frac{1 + 2q_2 + 2q_3}{8q_2 q_3}(q_1 - j_1)^2. \tag{C.14}$$

The black hole sheet itself is given by the region

$$\left( \frac{1 + 2q_2 + 2q_3}{8q_2 q_3} \right)(q_1 - j_1)^2 \le (q_1 + j_1). \tag{C.15}$$

At every fixed $q_2$ and $q_3$, the black hole sheet, consequently, is a parabolic cardboard sheet that just touches the surface BPS surface $q_1 + j_1 = 0$ at its apex. Unlike for the surface $q_1 + q_2 = 0$, the black hole sheet touches the surface $q_1 + j_1 = 0$ in a smooth manner.

### C.6.4 Meeting of three vanishing potential sheets

Let us consider a point on $C_{23}^{j_2}$ on which $q_2 > q_3$. In this case, we will demonstrate below that the three sheets meet on the $C_{23}^{j_2}$. Note, in particular, that the $\omega_2 = 0$ ice cream scoop does not intersect with the $\mu_2 = 0$ and $\mu_3 = 0$ ice cream scoops, so we have a region in which all chemical potentials are positive (though this region is rather small near the boundary of the black hole sheet). We now investigate these points in equations.

For black holes near the surface $q_1 + j_1 = 0$, the zero chemical potential surfaces are as follows: suppose black hole carries, $q_1 + j_1 = \frac{1 + 2q_2 + 2q_3}{8q_2 q_3} \epsilon^2$. We can see from (C.15) that such

---

[134]Plugging $j_1 = 0$ into (46) gives $j_L = -j_R = q_2 q_3$. Consequently, $j_1 = \frac{j_L + j_R}{2} = 0$. In a similar manner, points with $q_1 = 0$, on the first sheet of (46), have $q_1 = j_2 = 0$.

[135]Note that this intersection includes, in particular, both the (1/2) BPS lines with $q_2 \ne 0$ and the half BPS lines with $q_3 \ne 0$. These lines lie on the interface of the two sheets of (46).

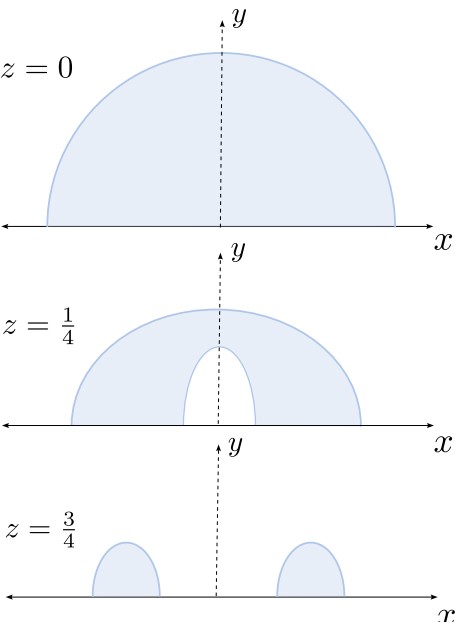

Figure 19: Horizontal cross sections of the toy model of the tube describing the $S^{\mu_1=0}$.

black holes carry $q_1$ from $-\epsilon/2$ to $\epsilon/2$. Then it is easy to see from (49), that for given $q_2, q_3$, the zero chemical potential black holes appear with the following charges:

$$
\begin{aligned}
\omega_2 = 0 &\implies q_1 = \frac{4\left(q_2{}^2 + q_2 q_3 + q_2 + q_3 + q_3{}^2\right) + 1}{8 q_2 q_3 (2 q_2 + 2 q_3 + 1)} \epsilon^2, \\
\mu_{2,3} = 0 &\implies q_1 = \frac{1 + 2 q_{2,3}}{4 q_{3,2}(1 + 2 q_2 + 2 q_3)} \epsilon^2.
\end{aligned}
\tag{C.16}
$$

Substituting $q_1$ from (C.16) in second of (C.13) and using the charge constraint to find $j_2$, we can easily see that the point where $\omega_2 = 0$ have $j_2 > 2 q_2 q_3$ and the points where $\mu_{2,3} = 0$ have $j_2 < 2 q_2 q_3$ as expected. In between (in a region of size $\epsilon^2$) all chemical potentials are positive. Also notice from (C.15) that for $q_1 + j_1 = O(\epsilon^2)$, the boundary of the black hole sheet has $(q_1, j_1)$ in the range $(-\epsilon/2, \epsilon/2)$.

## C.7 A toy model for the $S^{\mu_i=0}$ tube in the black hole sheet at large charges

In this brief subsection, we present a local "toy model" that captures qualitative aspects of the geometry of this tube. Consider the half space (of a Cartesian $R^3$) defined by the equation $y \geq 0$ ($x$, $y$ and $z$ are the usual flat coordinates on this space). Consider a unit circle, in the $xz$ plane (i.e. on the boundary of this half space), centred at $x = 1$, $y = z = 0$. This circle just touches the origin. We generate a tube by rotating this circle, counterclockwise around the $z$ axis, by the angle $\pi$. We then vertically squash the part of this tube that lies near the $yz$ plane by rescaling $z \to f(|x|)z$, where $f(a)$ is a monotonically increasing function in $[0, 1]$ with $f(0) = \frac{1}{2}$[136] and $f(2) = 1$. This rescaling reduces that maximum height of the intersection of the tube and the $yz$ plane to half. In Fig. 19 we present three horizontal (constant $z$) cross sectional cuts of the space described in this toy model. These cuts are taken at $z = 0$, $z = \frac{1}{4}$ (note that $\frac{1}{4} < \frac{1}{2}$), and at $z = \frac{3}{4}$ (note that $\frac{3}{4} > \frac{1}{2}$).

The interior of the tube is our toy model maps to the unstable region in the scooped out tube (bounded by $S^{\mu_1=0}$) in the black hole apple. The $xy$ plane in our toy model maps to the

---

[136]This number is chosen arbitrarily, it could have been replaced by any number between 0 and 1.

$j_1 = j_2$ surface in the apple of Fig 2. The two boundaries of the tube (on the $xz$ plane) in our toy model, maps to the closed curves $C_{12}$ and $C_{13}$. The $y$ axis in our toy model corresponds to the line $j_1 = j_2$, and $q_2 = q_3$ (in Fig 2).

## C.8 Positioning of the black hole sheet w.r.t the bosonic cone, Bose-Fermi cone and fermionic polyhedron

In §2.12 (see under (40)) we made several claims about the positioning of the black hole sheet w.r.t. the bosonic cone, the Bose-Fermi cone and the fermionic polyhedron. In this Appendix, we establish these claims.

- The black hole sheet extends beyond the bosonic cone can be seen, for instance, from a set of charges $\{\zeta_i\} = (-\frac{1}{4}, 2, 2, \frac{5}{2}, \frac{5}{2})$, which satisfies both (43) and (44). In particular, we can find values of $\{\zeta_i\}$ with one of the charges being negative and still satisfy both (43) and (44). Note that, on the black hole sheet, at most one of the $\zeta_i$ can be negative.

- The black hole sheet lies within the Bose-Fermi cone follows from the well known fact the black holes are more than $(1/16)^{th}$ BPS [1–6] together with the fact that the space of $(1/8)^{th}$ BPS black holes constitutes the boundary Bose-Fermi cone.

- The black hole sheet lies within the allowed region can be argued as follows:
  **For small charges:** Recall that the allowed region coincides with the Bose-Fermi cone at small charges. Hence the last point establishes that the black hole sheet lies in the Allowed region at small values of the charges.
  **For large charges:** We now demonstrate that when all charges are large (i.e. when $|\zeta_i| \gg 1$ for all $i \in (1, \ldots, 5)$), the supersymmetric black hole sheet always lies within the fermionic polyhedron. More precisely, we will demonstrate that charges that violate (C.4) necessarily violate the second of (44) (and so do not live on the black hole sheet). We will also demonstrate that charges that violate (C.5) necessarily violate one of the first or third of (44)

The first argument proceeds as follows. Suppose we are given charges that violate (C.4).

$$j_L \gtrsim \frac{4}{3}|q_1|^{3/2}. \tag{C.17}$$

It follows that

$$q_1 q_2 q_3 + \frac{j_1 j_2}{2} \leq q_1 q_2 q_3 + \frac{j_L^2}{2} \leq -|q_1| q_2 q_3 + \frac{8}{9}|q_1|^3 < 0, \tag{C.18}$$

where, in the second last step, we have used (C.17), and in the last step we have used $q_2 > |q_1|$ and $q_3 > |q_1|$ (these are simply the equations $\zeta_1 + \zeta_2 > 0$ and $\zeta_1 + \zeta_3 > 0$). Clearly (C.18) contradicts the second of (44).

The second argument proceeds as follows. Let us suppose we are at charges that violate (C.5), i.e. that obey

$$j_2 \gtrsim 2|j_1|^2, \tag{C.19}$$

where $j \equiv j_1$ The nonlinear charge relation (43) cannot be satisfied under the violation of the inequality. To see this, consider the following:

$$q_1 q_2 q_3 - j^3 \geq q_1 q_2 q_3 + \frac{j j_2}{2} = \left(q_1 + q_2 + q_3 + \frac{1}{2}\right)\left(q_1 q_2 + q_2 q_3 + q_3 q_1 - \frac{-j + j_2}{2}\right)$$

$$\geq \left(q_1 + q_2 + q_3 + \frac{1}{2}\right)(q_1 q_2 + q_2 q_3) \geq q_1 q_2 q_3, \tag{C.20}$$

which is a contradiction comparing the first and the last term. Note that we have used the inequalities $q_3 q_1 > j_2/2$ from $q_1 > j$ and $q_3 > j$.

## C.9 The index line never intersects the black hole sheet more than once

In this subsection, we show that the index line never intersects the black hole sheet more than once. First we assume that a set of charges $(q_1, q_2, q_3, j_1, j_2)$ satisfy the black hole charge relation:

$$\frac{j_1 j_2}{2} + q_1 q_2 q_3 = \left(q_1 + q_2 + q_3 + \frac{1}{2}\right)\left(q_1 q_2 + q_2 q_3 + q_3 q_1 - \frac{1}{2}(j_1 + j_2)\right). \tag{C.21}$$

Now, suppose the charges are shifted along the index line according to (5), such that

$$\left(q_1', q_2', q_3', j_1', j_2'\right) = (q_1 + a, q_2 + a, q_3 + a, j_1 - a, j_2 - a).$$

If the index line intersects the black hole sheet more than once, the shifted charges also satisfy the non-linear charge relation when they intersect the black hole sheet:

$$\frac{j_1' j_2'}{2} + q_1' q_2' q_3' = \left(q_1' + q_2' + q_3' + \frac{1}{2}\right)\left(q_1' q_2' + q_2' q_3' + q_3' q_1' - \frac{1}{2}(j_1' + j_2')\right). \tag{C.22}$$

Subtracting the original equation from the shifted one, we obtain (besides the trivial solution):

$$a = \frac{1}{4}\left(\pm 2\sqrt{\left(\frac{j_1 + j_2}{2} - q_1 q_2 - q_1 q_3 - q_2 q_3\right)} - 2q_1 - 2q_2 - 2q_3 - 1\right). \tag{C.23}$$

The term under the square root is always negative because it is proportional to the negative of the entropy squared. Hence, the shift $a$ becomes imaginary, which implies that it is not possible to have two distinct black hole configurations on the same index line.

## C.10 Black hole Fermi seas?

We demonstrate that fermionic condensation cannot occur on the black hole sheet. For fermionic condensation to take place, one of the $\nu_i$ values must exceed the sum of the remaining $\nu$ values:

$$\exists \nu_i, \quad \text{such that} \quad \text{Re}[\nu_i] > \sum_{j \neq i} \text{Re}[\nu_j]. \tag{C.24}$$

However, on the black hole sheet, we find:

$$\text{Re}[\nu_1 + \nu_2 + \nu_3] = \text{Re}[\nu_4 + \nu_5]. \tag{C.25}$$

It is straightforward to see that, when all the $\nu$'s are positive, none of the $\nu_i$ values can be greater than the sum of the others.

## C.11 Renormalized chemical potentials at the boundary of the black hole sheet

The boundary of the black hole sheet is inside the Bose-Fermi cone and defined as follows:

$$\frac{j_1 j_2}{2} + q_1 q_2 q_3 = 0,$$
$$q_1 q_2 + q_1 q_3 + q_2 q_3 - \frac{1}{2}(j_1 + j_2) = 0. \tag{C.26}$$

Solving them, the boundary of the black hole sheet can be parametrized by $q_1, q_2, q_3$:

$$j_L = q_1 q_2 + q_1 q_3 + q_2 q_3,$$
$$|j_R| = \sqrt{(q_1 q_2 + q_1 q_3 + q_2 q_3)^2 + 2q_1 q_2 q_3} \geq 0. \tag{C.27}$$

The boundary of the black hole sheet is characterized by: 1) One of the $\omega_a$ values being negative or zero,[137] or 2) Two of the $\mu_i$ values are negative or zero (The proof is shown below). In the charge space, this can be described as follows:

1. $\omega_a < 0$: For $q_1, q_2, q_3$ all positive, $j_1 < 0$ and $j_2 > 0$, we find that $\omega_2 < 0$ (or vice versa). In this case, gas configuration involving $j_2$ defines the boundary of the EER.

2. $\mu_i < 0, \mu_j < 0$: This case arises for $j_1, j_2$ both positive, one of the $q_i$ values is negative. Let us assume $q_1 < 0 < q_2 < q_3$, which corresponds to a $\mu_2 < 0, \mu_3 < 0$ black hole on the boundary. The dual giant configuration involves contributions along the $q_3$ direction. (Only when $q_2 = q_3$ rank 4 dual giants is allowed.)

**Proof** According to (41),

$$\omega_2 = -\frac{2\pi i}{\frac{S+2i\pi j_2}{S+2i\pi j_1} - \frac{S+2i\pi j_2}{S-2i\pi q_1} - \frac{S+2i\pi j_2}{S-2i\pi q_2} - \frac{S+2i\pi j_2}{S-2i\pi q_3} + 1}. \tag{C.28}$$

The real part of it is (positively) proportional to

$$
\begin{aligned}
4\pi^2 S^5 & (j_1 + 2j_2 + q_1 + q_2 + q_3) \\
& + 16\pi^4 S^3 \big(3j_1^2 j_2 + j_1^2 q_1 + j_1^2 q_2 + j_1^2 q_3 + j_1 q_1^2 + j_1 q_2^2 + j_1 q_3^2 + j_2 q_1^2 \\
& \qquad + j_2 q_2^2 + j_2 q_3^2 + q_1^2 q_2 + q_1^2 q_3 + q_1 q_2^2 + q_1 q_3^2 + q_2^2 q_3 + q_2 q_3^2\big) \\
& + S \big(128\pi^6 j_1^2 j_2 q_1^2 + 128\pi^6 j_1^2 j_2 q_2^2 + 128\pi^6 j_1^2 j_2 q_3^2 + 64\pi^6 j_1^2 q_1^2 q_2 + 64\pi^6 j_1^2 q_1^2 q_3 \\
& \qquad + 64\pi^6 j_1^2 q_1 q_2^2 + 64\pi^6 j_1^2 q_1 q_3^2 + 64\pi^6 j_1^2 q_2^2 q_3 + 64\pi^6 j_1^2 q_2 q_3^2 + 64\pi^6 j_1 q_1^2 q_2^2 \\
& \qquad + 64\pi^6 j_1 q_1^2 q_3^2 + 64\pi^6 j_1 q_2^2 q_3^2 + 64\pi^6 q_1^2 q_2^2 q_3 + 64\pi^6 q_1^2 q_2 q_3^2 + 64\pi^6 q_1 q_2^2 q_3^2\big) \\
& + \frac{256\pi^8}{S} \big(j_1^2 j_2 q_1^2 q_2^2 + j_1^2 j_2 q_1^2 q_3^2 + j_1^2 j_2 q_2^2 q_3^2 + j_1^2 q_1^2 q_2^2 q_3 + j_1^2 q_1^2 q_2 q_3^2 \\
& \qquad + j_1^2 q_1 q_2^2 q_3^2 + j_1 q_1^2 q_2^2 q_3^2 - j_2 q_1^2 q_2^2 q_3^2\big).
\end{aligned}
\tag{C.29}
$$

Since we are considering the boundary of the black hole sheet where the entropy goes to zero, we retain only the leading-order term of the above expression in $S$, as shown in the last line. Substituting $j_1$ and $j_2$ in terms of $q_1, q_2, q_3$, we obtain

$$\mathrm{Re}[\omega_2] \propto \left(q_1 q_2 + q_1 q_3 + q_2 q_3 - \sqrt{2q_1 q_2 q_3 + (q_2 q_3 + q_1 q_2 + q_3 q_1)^2}\right), \tag{C.30}$$

which is negative when $q_1, q_2, q_3$ are positive. (Here, we assume $j_1 < j_2$.)

On the other hand, if one of $q$s become negative, $\mathrm{Re}[\omega_2] > 0$ (and of course $\mathrm{Re}[\omega_1] > 0$). Therefore, at least one of $\mathrm{Re}[\mu_i]$ should become negative. Indeed, under the similar procedure, we obtain

$$\mathrm{Re}[\mu_1] \propto q_1^2 q_2^3 q_3^3 (1 + 2q_1 + 2q_2 + 2q_3) \propto q_2 q_3, \tag{C.31}$$

which is negative when one of $q_2$ and $q_3$ are negative. So we conclude that at the boundary of the black hole sheet, at least one of the following conditions is satisfied: $\omega_a \le 0$ or $\mu_i \le 0$.

## C.12 Impossibility of simultaneous negative $\mathrm{Re}(\omega_1)$ and $\mathrm{Re}(\omega_2)$

As explained in the main text, the complex chemical potentials can be expressed as follows:

$$
\begin{aligned}
\mu_I &= \frac{2\pi i z_I}{1 + z_1 + z_2 + z_3 + z_4}, & I &= 1, 2, 3, \\
\omega_1 &= -\frac{2\pi i z_4}{1 + z_1 + z_2 + z_3 + z_4}, & \omega_2 &= -\frac{2\pi i}{1 + z_1 + z_2 + z_3 + z_4},
\end{aligned}
\tag{C.32}
$$

---

[137] $\omega_a = 0$ or $\mu_i = 0$ occurs on a co-dimension one region on the boundary of black hole sheet.

where $\mu_1 + \mu_2 + \mu_3 = \omega_1 + \omega_2 + 2\pi i$ and the $z_I$'s are auxiliary parameters that conveniently express chemical potentials that are determined in terms of the on-shell black hole charges:

$$z_I = -\frac{S + 2\pi i J_2}{S - 2\pi i Q_I}, \qquad z_4 = \frac{S + 2\pi i J_2}{S + 2\pi i J_1}. \tag{C.33}$$

Now we investigate whether an on-shell BPS black hole solution can simultaneously satisfy $\mathrm{Re}(\omega_1) < 0$ and $\mathrm{Re}(\omega_2) < 0$. A necessary condition for achieving both $\mathrm{Re}(\omega_1) < 0$ and $\mathrm{Re}(\omega_2) < 0$ is the existence of a black hole configuration where $\mathrm{Re}(\omega_1) = 0$ and $\mathrm{Re}(\omega_2) = 0$, which simplifies our analysis. We will show that such a configuration is not possible.

For $\mathrm{Re}(\omega_1) = \mathrm{Re}(\omega_2) = 0$, the charge relations are given by:

$$J_L = \frac{2(Q_1 Q_2 + Q_1 Q_3 + Q_2 Q_3)}{2(Q_1 + Q_2 + Q_3) + 3}, \qquad J_R = 0, \tag{C.34}$$

and the BPS black hole charge relation can be expressed as:

$$Q_1 Q_2 Q_3 + \frac{J_L^2}{2} = \left(\frac{1}{2} + Q_1 + Q_2 + Q_3\right)(Q_1 Q_2 + Q_2 Q_3 + Q_3 Q_1 - J_L). \tag{C.35}$$

Solving these equations for $J_L$ yields:

$$\begin{aligned} 2(Q_1 + Q_2 + Q_3) + 1 + \frac{4(Q_1 Q_2 + Q_2 Q_3 + Q_3 Q_1)}{2(Q_1 + Q_2 + Q_3) + 3} \\ = \sqrt{(2Q_1 + 2Q_2 + 1)(2Q_1 + 2Q_3 + 1)(2Q_2 + 2Q_3 + 1)}. \end{aligned} \tag{C.36}$$

By multiplying both sides by $2(Q_1 + Q_2 + Q_3 + 3)$ and squaring the expression, we arrive at:

$$(1 + 2a)^2(3 + 2a)b - 4b^2 - 2(3 + 2a)^2 c = 0, \tag{C.37}$$

where $a = Q_1 + Q_2 + Q_3$, $b = Q_1 Q_2 + Q_2 Q_3 + Q_3 Q_1$, and $c = Q_1 Q_2 Q_3$. Note that while $a$ and $b$ are always positive, $c$ may be negative if one of the $Q$ values is negative.

We first argue that (C.37) cannot be satisfied if $c$ is negative. Since $a^2 = Q_1^2 + Q_2^2 + Q_3^2 + 2b$, it is straightforward to see that equation (C.37) remains positive in this case. Furthermore, when $c$ is positive (i.e., all $Q$ values are positive), equation (C.37) is always positive as well. Substituting $a$, $b$, and $c$ in terms of the $Q$ values confirms that all coefficients of monomials are positive. Therefore, it is impossible to satisfy both $\mathrm{Re}(\omega_1) = 0$ and $\mathrm{Re}(\omega_2) = 0$ simultaneously.

## C.13 Impossibility of simultaneous negative $\mathrm{Re}(\omega_1)$ and $\mathrm{Re}(\mu_1)$

We are interested in finding an expression for $\mathrm{Re}[\mu_1] = 0$, which is equivalent to requiring the following expression to be purely imaginary.

$$(S - 2\pi i Q_1)\left(\frac{1}{S + 2\pi i J_1} + \frac{1}{S + 2\pi i J_2} - \frac{1}{S - 2\pi i Q_1} - \frac{1}{S - 2\pi i Q_2} - \frac{1}{S - 2\pi i Q_3}\right) = \text{pure imaginary}. \tag{C.38}$$

Using the following non-linear constraint of BPS black holes

$$\pi i (S + 2\pi i J_1)(S + 2\pi i J_2) = (S - 2\pi i Q_1)(S - 2\pi i Q_2)(S - 2\pi i Q_3), \tag{C.39}$$

we get

$$\frac{J_1 + J_2}{2} = \frac{(Q_2 + Q_3)(Q_1 + 2Q_1^2 - 2Q_2 Q_3)}{1 + 2Q_1}. \tag{C.40}$$

Under a similar procedure, we also get the expression for $\text{Re}[\omega_1] = 0$, which is written as

$$J_1 = -2(1 + Q_1 + Q_2 + Q_3)J_2 + 2(Q_1Q_2 + Q_2Q_3 + Q_3Q_1). \tag{C.41}$$

Now we ask whether an on-shell BPS black hole solution can satisfy $\text{Re}(\mu_1) < 0$ and $\text{Re}(\omega_1) < 0$. The necessary condition for achieving $\text{Re}(\mu_1) < 0$ and $\text{Re}(\omega_1) < 0$ is that a black hole with $\text{Re}(\mu_1) = 0$ and $\text{Re}(\omega_1) = 0$ must exist, which simplifies the analysis. We will demonstrate that it is impossible for both $\text{Re}(\mu_1) = 0$ and $\text{Re}(\omega_1) = 0$ to hold. With $\text{Re}(\mu_1) = \text{Re}(\omega_1) = 0$, we have

$$\frac{\mu_1}{\omega_1} = -\frac{z_1}{z_4} = \frac{S + 2\pi i J_1}{S - 2\pi i Q_1} = \text{Real}.$$

Since $S$, $Q_1$, and $J_1$ are all real, the only way $\frac{\mu_1}{\omega_1}$ can be real is if $Q_1 + J_1 = 0$, which is only satisfied at a special co-dimension one surface on the boundary of the black hole sheet where the black hole entropy vanishes. Therefore in the bulk of the black hole sheet, $\mu_1$ and $\omega_1$ can never simultaneously be negative.

## C.14 Monotonicity of the entropy at the boundary between rank 2 (rank 4) and rank 4 (rank 6) phases along the index line

In this subsection, we show that the entropy at the boundary between rank 2 (rank 4) and rank 4 (rank 6) phases along the index line changes monotonically. First, we consider the index line passing through the boundary between rank 2 DDBH and rank 4 DDBH. We show that the change in entropy at the boundary between the two phases is monotonic.

Suppose the charges at the boundary are given by $(q_1, q_2, q_3, j_1, j_2)$ with $q_1 < q_2 < q_3$. The core black hole is described by the following charges:

$$(q_1, q_2, q_2, j_1, j_2), \tag{C.42}$$

so that the hair is a dual giant with charge $q_3 - q_2$.

Let us consider a rank 2 DDBH near the boundary, where the charges are given by

$$(q_1 - \epsilon, q_2 - \epsilon, q_3 - \epsilon, j_1 + \epsilon, j_2 + \epsilon), \tag{C.43}$$

with $\epsilon > 0$. The core black hole is described by the following charges:

$$(q_1 - \epsilon, q_2 - \epsilon, q_2 + \delta, j_1 + \epsilon, j_2 + \epsilon). \tag{C.44}$$

The change in entropy is

$$(\omega_1 + \omega_2 - \mu_1 - \mu_2)\epsilon + \mu_3\delta = \mu_3(\epsilon + \delta). \tag{C.45}$$

Since $\epsilon + \delta$ is positive, the change in entropy has the same sign as $\mu_3 = \mu_2$ of the core black hole.

On the other hand, let us consider a rank 4 DDBH near the boundary, where the charges are given by

$$(q_1 + \epsilon, q_2 + \epsilon, q_3 + \epsilon, j_1 - \epsilon, j_2 - \epsilon), \tag{C.46}$$

with $\epsilon > 0$. The core black hole is described by the following charges:

$$(q_1 + \epsilon, q_2 - \delta, q_2 - \delta, j_1 - \epsilon, j_2 - \epsilon). \tag{C.47}$$

The change in entropy is

$$(-\omega_1 - \omega_2 + \mu_1)\epsilon - 2\mu_2\delta = -2\mu_2(\epsilon + \delta). \tag{C.48}$$

Since $\epsilon + \delta$ is positive, the change in entropy has the opposite sign as $\mu_2 = \mu_3$.

Therefore, whenever the rank 2 DDBH has higher entropy compared to the one at the boundary, the rank 4 DDBH has lower entropy, and vice versa. This proves that the entropy is monotonic at the boundary.

This result can be readily generalized to boundaries between rank 4 and rank 6 DDBH phases or between rank 2 and rank 4 grey galaxy phases. In both cases, the entropy of the rank 6 DDBH and rank 4 grey galaxy phases is consistently lower than that of their counterparts.

For instance, in the case of the boundary between rank 2 and rank 4 grey galaxy phases, the change in entropy in the direction of the rank 2 phase is positively proportional to $\omega_1 = \omega_2$, which is always positive.[138] Conversely, the change in entropy in the direction of the rank 4 phase is negatively proportional to $\omega_1 = \omega_2$, which is always negative.

### C.15 Local maxima of hairy black holes along the index line

In this subsection, we show that along the index line, the charges corresponding to a hairy black hole with a core black hole characterized by $\nu_i = 0$ represent a local maximum. This result includes:

- Rank 2 hairy black holes (grey galaxies or rank 2 DDBH) with core black hole $\nu_i = 0$ and hair satisfying $\zeta_i \neq 0$

- Rank 4 DDBH with core black hole $\mu_i = \mu_j = 0$ with hair $\zeta_i \neq 0, \zeta_i \neq 0$.

First we consider a rank 2 DDBH, where we assume $\zeta_1 \leq \zeta_2 \leq \zeta_3$. The hair in this case is associated with $q_3$. As we move along the shadow on the black hole sheet, the variations in the charges and angular momenta are given by:

$$\delta j_1 = \delta j_2 = -\delta q_1 = -\delta q_2 = \epsilon, \tag{C.49}$$

where $\delta q_3$ is automatically determined by the relations.

We aim to show that $\mathrm{Re}[\mu_3] = 0$ corresponds to a local maximum along the shadow. If $\mathrm{Re}[\mu_3] = 0$, the change in the entropy is expressed as:

$$\delta S = \mathrm{Re}[\delta \zeta_i \nu_i] = \mathrm{Re}[-\mu_1 - \mu_2 + \omega_1 + \omega_2]\epsilon = 0, \tag{C.50}$$

where we used the relation between the chemical potentials. To determine if $\mathrm{Re}[\mu_3] = 0$ is a local maximum, we analyze the second derivative of the entropy: $\frac{\partial^2 S}{\partial \epsilon^2} < 0$. Using the non-linear charge relation on the black hole sheet, we express $q_3$ in terms of the other charges. Then, we compute the entropy of the black hole with the modified charges: $(q_1 - \epsilon, q_2 - \epsilon, q_3(\epsilon), j_1 + \epsilon, j_2 + \epsilon)$, and subtract the entropy of the black hole at $(q_1, q_2, q_3, j_1, j_2)$ with $\nu = 3$. The result shows that:

$$\delta S = \left( -\frac{8(j_1 + q_1)(j_1 + q_2)(j_2 + q_1)(j_2 + q_2)}{\mathcal{P}(q_1, q_2, j_1, j_2)} - 1 \right)\epsilon^2, \tag{C.51}$$

where

$$
\begin{aligned}
f = \Big( & j_1^2 + 2j_1\big(4j_2q_1 + 4j_2q_2 + j_2 + 2q_1^2 + 4q_1q_2 + q_1 + 2q_2^2 + q_2\big) + j_2^2 \\
& + 2j_2\big(2q_1^2 + 4q_1q_2 + q_1 + 2q_2^2 + q_2\big) + 4q_1^4 + 4q_1^3 - 8q_1^2q_2^2 + 4q_1^2q_2 \\
& + q_1^2 + 4q_1q_2^2 + 2q_1q_2 + 4q_2^4 + 4q_2^3 + q_2^2 \Big)^{3/2},
\end{aligned}
$$

---

[138]Note that a core black hole cannot have both $\omega_1$ and $\omega_2$ negative simultaneously.

which is positive.[139] This confirms that $\mu_3 = 0$ is indeed a local maximum along the shadow.

Similarly, one can show that the entropy of the black hole along the shadow of the rank 2 grey galaxy becomes locally maximal at $\omega_2 = 0$, assuming $\zeta_4 \leq \zeta_5$. The entropy variation is given by:

$$\delta S = \left(-1 + \frac{32(j_1 + q_1)(j_1 + q_2)(j_1 + q_3)}{(2(j_1 + q_1 + q_2 + q_3) + 1)^3}\right)\epsilon^2, \tag{C.52}$$

where, on $\omega_2 = 0$, $j_1$ is expressed in terms of $q_1, q_2, q_3$ as:

$$j_1 = \frac{\sqrt{\left(4q_1^2 + 4q_1(q_2 + q_3 + 1) + 4q_2^2 + 4q_2(q_3 + 1) + (2q_3 + 1)^2\right)^2 + 64q_1 q_2 q_3(q_1 + q_2 + q_3 + 1)} - g(q_i)}{8(q_1 + q_2 + q_3 + 1)}, \tag{C.53}$$

where $g(q_i) = 4q_1^2 + 4q_1 q_2 + 4q_1 q_3 + 4q_1 + 4q_2^2 + 4q_2 q_3 + 4q_2 + 4q_3^2 + 4q_3 + 1$. From this, we find that $\delta S < 0$, indicating that $\omega_2 = 0$ is a local maximum along the shadow. This conclusion can also be supported by the following reasoning: as argued in Appendix C.14 and Subsection E.4, the entropy along the index line decreases when it leaves the rank 2 phase and transitions either into the rank 4 phase or reaches the endpoint of the index line. Hence, since the only extremum along the segment of the rank 2 phase occurs at $\omega_2 = 0$, this extremum must necessarily be a local maximum.

By applying the same reasoning as for the rank 2 grey galaxy, one can similarly conclude that a rank 4 DDBH, characterized by a core black hole with $\mu_i = \mu_j = 0$ and hair $\zeta_i \neq 0, \zeta_j \neq 0$, has locally maximal entropy along the index line. The entropy along the index line decreases when it leaves the rank 4 phase and transitions either into the rank 6 phase or reaches the endpoint of the index line. Since the only extremum along the segment of the rank 4 phase occurs at $\mu_i = \mu_j = 0$, this extremum must necessarily be a local maximum.

By applying the same reasoning as for the rank 2 grey galaxy, one can similarly conclude that a rank 4 DDBH, characterized by a core black hole with $\mu_i = \mu_j = 0$ and hair $\zeta_i \neq 0, \zeta_j \neq 0$, possesses locally maximal entropy along the index line. The entropy along the index line decreases as it exits the rank 4 phase and transitions either into the rank 6 phase or reaches the endpoint of the index line. Thus, given that the only extremum along the segment of the rank 4 phase occurs at $\mu_i = \mu_j = 0$, this extremum must necessarily be a local maximum.

## D  Entropy maximization and allowed phases

In this Appendix we outline a strategy that can be used to algorithmically implement the maximization procedure (described at the end of §3.1) in order to evaluate the supersymmetric entropy at any charge that lies in the EER. The discussion of this subsection will lead to a classification of allowed phases of the supersymmetric partition function.

Let us first recall that the thermodynamical relation

$$d\left(N^2 S_{BH}\right) = \nu_i \, dZ_i \quad \implies \quad dS_{BH} = \nu_i d\zeta_i, \tag{D.1}$$

where the renormalized chemical potentials $\nu_i$, (of the zero temperature limit of Lorentzian black holes) are simply the real parts of the complex chemical potentials listed on the RHS of (41).

Although $\nu_i$ have definite known values for all $i = 1 \ldots 5$, $S_{BH}$ is only defined only on the black hole manifold (43). Consequently, the one-forms in (D.1) should be understood as pulled back onto the manifold (43). More explicitly, let $\alpha_j$, $j = 1 \ldots 4$ be a set of coordinates

---

[139]If this were not the case, it would result in an imaginary value, which is unphysical.

on the manifold (43). The pullback of (D.1) onto the manifold (43) takes the explicit form

$$V \equiv dS_{BH} = v^i \frac{\partial Z_i}{\partial \alpha^m} d\alpha^m \,. \tag{D.2}$$

We now define a vector field by raising the indices of the one form (D.2)

$$V^m = g^{mn} \partial_n S_{BH} = v^i \frac{\partial Z_i}{\partial \alpha^n} g^{nm} \,, \tag{D.3}$$

where $g^{mn}$ is the inverse metric in charge space. It is possible to verify that the vector field $V^m$ does not vanish anywhere on the black hole manifold.[140]

Imagine we are sitting at the point $\{\zeta_{BH}^m\}$ on the black hole manifold. The motion

$$\delta \zeta_{BH}^m = \epsilon V^m \,, \quad \epsilon > 0 \,, \tag{D.4}$$

changes the black hole entropy by

$$\delta S_{BH} = \epsilon V^m \partial_m S_{BH} = \epsilon (V^m g_{mn} V^n) > 0 \,. \tag{D.5}$$

In other words, motion in the direction $V^m$ always increases black hole entropy.

## D.1 Unconstrained flow

As we have explained above, in order to evaluate $N^2 S_{BPS}(\zeta_i)$, we are instructed to maximize over all black holes that contain the charge $\{\zeta_i\}$ within their bosonic cone. We can search for (at least) local maxima by employing the method of "gradient flow", in a manner we now describe.

We first pick a point $\zeta_{BH}^i$ on the black hole sheet, chosen so that $\zeta^i$ lies with its bosonic cone i.e.[141]

$$\zeta^i - \zeta_{BH}^i \geq 0 \,, \quad \forall \, i \,. \tag{D.6}$$

We flow $\zeta_{BH}^i$ along the black hole sheet in the direction of the vector $V^m$ according to

$$d\zeta_{BH}^i = V^i \,. \tag{D.7}$$

It follows from (D.5) that this motion always increases the black hole entropy. We keep flowing (and so constantly changing $\zeta_{BH}^i$) till we reach the point $\zeta_{BH}^i = \zeta_{BHa}^i$ at which one of the inequalities - lets say (D.6) at $i = i_1$ is saturated, i.e. when[142]

$$\zeta_{BHa}^{i_1} = \zeta^i \,. \tag{D.8}$$

Any further motion along $V^m$ (according to (D.7)) would violate the conditions (D.6).

## D.2 First constrained flow

While we cannot continue to flow in the direction of the vector $V^m$, we have not yet achieved our goal of maximizing entropy subject to (D.6), because we can still flow along the 3 dimensional intersection of the black hole sheet and the plane[143]

$$\zeta_{BH}^{i_1} = \zeta_{BHa}^{i_1} \,. \tag{D.9}$$

---

[140]We choose the obvious flat metric $\delta_{ij}$ in the 5 dimensional charge space. $g_{mn}$ is then the restriction of this metric to (43).

[141]Since we have assumed that $\{\zeta^i\}$ lies within the EER, such a choice of $\zeta_{BH}^i$ always exists.

[142]As the vector $V^i$ vanishes nowhere on the black hole sheet, the flow continues until we hit such a boundary.

[143]This is a flow on the space of black holes that saturate the inequality (D.6) at $i = i_1$.

It is entropically advantageous to flow along the vector $V_{i_1}^m$, where $V_{i_1}^m$ is the projection of $V^m$ tangent to the plane (D.9). The equation (D.5) still applies to this motion, except that the vector $V^m$ is replaced by $V_{i_1}^m$, and the black hole entropy continues to increase along the flow. In Appendix D.8 we demonstrate that the vector $V_{i_1}^m$ never vanishes (at any point on the black hole manifold, and for any value of $i_1$). Consequently, our flow continues until we reach the point $\zeta_{BH}^i = \zeta_{BHb}^i$ at which

$$\zeta_{BHb}^{i_2} = \zeta^{i_2}, \tag{D.10}$$

at which point second of the inequalities (D.6) is saturated.

## D.3  Second constrained flow and the rank 6 DDBH phase

Further motion can only take place on the two dimensional intersection of the black hole sheet and the two planes

$$\zeta_{BH}^{i_1} = \zeta_{BHa}^{i_1} = \zeta_{BHb}^{i_1}, \qquad \zeta_{BH}^{i_2} = \zeta_{BHb}^{i_2}. \tag{D.11}$$

It is advantageous to flow along $V_{i_1 i_2}^m$, the projection of $V^m$ onto the tangent space of the two planes (D.9).[144] As long as $V_{i_1 i_2}^m \neq 0$, the flow proceeds, and the entropy continues to increases.

It turns out that the vector field $V_{i_1 i_2}^m$ vanishes on the black hole sheet if and only if $(i_1, i_2) = (4, 5)$ and the black hole carries charges

$$(q, q, q, \zeta_4, \zeta_5), \quad \text{with} \quad \zeta_i \geq q, \quad i = 1, 2, 3. \tag{D.12}$$

Here $q = q(\zeta_4, \zeta_5)$ is determined from the requirement that the black hole charges $(q, q, q, \zeta_4, \zeta_5)$ obey (43). Such a point is a local maximum of the entropy[145] and so represents a (locally stable) rank 6 DDBH, in which the dual giants carry charges $(\zeta_1 - q, \zeta_2 - q, \zeta_3 - q)$. The inequality (D.12) arises from the requirement that the dual giants carry positive charges.

## D.4  The rank 4 DDBH phase

If $(i_1, i_2) = (4, 5)$, or if $(i_1, i_2) = (4, 5)$ but the inequality at the end of the previous paragraph is not obeyed, the flow of the previous subsubsection proceeds until we reach the point $\zeta_{BHc}^i$ at which a third - lets say $i_3^{th}$ - of (D.6) is saturated. At this point

$$\zeta_{BH}^{i_1} = \zeta_{BHa}^{i_1} = \zeta_{BHb}^{i_1} = \zeta_{BHc}^{i_1}, \qquad \zeta_{BH}^{i_2} = \zeta_{BHb}^{i_2} = \zeta_{BHc}^{i_2}, \qquad \zeta_{BH}^{i_3} = \zeta_{BHc}^{i_3}. \tag{D.13}$$

In this situation, we continue to flow along the 1-dimensional curve given by the intersection of the three planes (D.9) and the black hole sheet. As above, we choose to flow in the direction of the vector $V_{i_1 i_2 i_3}^m$, defined as the orthogonal projection of $V^m$ tangent to our one dimensional curve. The entropy always increases along such a flow.

This flow terminates before we hit yet another boundary, if and only if $V_{i_1 i_2 i_3}^m$ vanishes. This happens under one of two conditions. We study the first of these in this subsubsection, and the second in the next subsubsection.

The first case in which $V_{i_1 i_2 i_3}^m$ vanishes is when $(i_1, i_2, i_3) = (3, 4, 5)$ (of course 3 above can everywhere be interchanged with by 1 or 2) and the black hole carries charges

$$(q, q, \zeta_3, \zeta_4, \zeta_5), \quad \zeta_{1,2} > q, \tag{D.14}$$

where $q(\zeta_i)$ is determined by the requirement that these charges obey (43). This point is a maximum of the entropy along the curve of interest, and corresponds to a rank 4 DDBH phase. The duals in this phase carry charges $(\zeta_1 - q, \zeta_2 - q, 0, 0, 0)$. The inequality in (D.14) arises from the requirement duals in the DDBH all carry positive charge.

---

[144]The flow now proceeds among black holes that have the property that the charge $\{\zeta_i\}$ lies on both the $i_1^{th}$ and $i_2^{th}$ boundaries of their bosonic cone.

[145]Along the (two dimensional) intersection of $\zeta_{i_1} =$ constant, $\zeta_{i_2} =$ constant, and the black hole sheet (43).

## D.5 The rank 4 grey galaxy phase

$V^m_{i_1 i_2 i_3}$ also vanishes when $(i_1, i_2, i_3) = (1, 2, 3)$ and the black hole carries charges

$$(\zeta_1, \zeta_2, \zeta_3, j, j), \quad \zeta_{4,5} > j, \tag{D.15}$$

where $j = j(\zeta_i)$ is determined by the requirement that (43) be obeyed. This point is a maximum of the entropy along the curve of interest, and corresponds to a rank 4 grey galaxy whose gas carries charges $(0, 0, 0, \zeta_4 - j, \zeta_5 - j)$. The inequality in (D.15) arises from the requirement that the gas carries positive angular momentum.

## D.6 The rank 2 DDBH phase

If our flow does not meet the condition of either of the last two subsubsections, it continues until the charge of interest saturate the $i_4^{th}$ inequality (D.6). At this point the flow terminates, because the intersection of the four planes (representing constant values of $\zeta_{1_1}$, $\zeta_{i_2}$, $\zeta_{i_3}$ and $\zeta_{i_4}$) and the black hole sheet (43) is just a point.

There are two qualitatively different cases; $(i_1, i_2, i_3, i_4) = (2, 3, 4, 5)$ (or the interchange of 1 with either 2 or 3) and the case $(i_1, i_2, i_3, i_4) = (1, 2, 3, 4)$ (or the interchange of 4 with 5). We study the first of these in this subsubsection, and the second in the next subsubsection.

When $(i_1, i_2, i_3, i_4) = (2, 3, 4, 5)$ the black hole carries charges

$$(q, \zeta_2, \zeta_3, \zeta_4, \zeta_5), \quad \zeta_1 - q > 0, \tag{D.16}$$

with $q = q_{\zeta_i}$ determined from (43) (we can also interchange $1 \leftrightarrow 2$ or $1 \leftrightarrow 3$). The resultant phase is a rank 2 DDBH whose duals carry charges $(\zeta_1 - q, 0, 0, 0, 0)$. The inequality in (D.16) arises from the requirement that DDBH carries a positive charge.

## D.7 The rank 2 grey galaxy phase

When $(i_1, i_2, i_3, i_4) = (1, 2, 3, 4)$, the black carries charges $(q, \zeta_2, \zeta_3, \zeta_4, \zeta_5)$, with $q$ determined from (43) (we can also interchange $1 \leftrightarrow 2$ or $1 \leftrightarrow 3$). The resultant phase is a rank 2 DDBH whose duals carry charges

$$(\zeta_1 - q, 0, 0, 0, 0), \quad \zeta_1 > q. \tag{D.17}$$

The graviton gas carries angular momentum $\zeta_1 - q$, explaining the inequality in (D.17)

## D.8 Proof of §D.2

In this subsection, we prove the assertion stated in §D.2, that the vector $V^m_{i_1}$ never vanishes which is equivalent to the claims demonstrated below.

**Claim 1: The mixture of grey galaxy and DDBH is impossible.**

In other words, the entropically dominant phases are either a non-hairy black hole, a grey galaxy, or a DDBH. Let us prove this by contradiction. Consider a hairy black hole configuration consisting of a core black hole with charges $q_1, q_2, q_3$ and angular momenta $j_1, j_2$, along with hair characterized by $\Delta q_1 > 0$ and $\Delta j_1 > 0$. (There may be additional hair components, but we focus only on these two, without losing generality.) This configuration represents a mixture of a grey galaxy and a DDBH, as the hair contributes to both $q_1$ and $j_1$.

The change in the entropy of the core black hole by varying $q_1$ and $j_1$ is given as

$$\delta S^2 \propto \delta(q_1 q_2 + q_2 q_3 + q_3 q_2 - \frac{1}{2}(j_1 + j_2)) = \left(q_2 + q_3 - \frac{1}{2} \frac{\partial j_1}{\partial q_1}\right) \delta q_1$$

$$= \frac{2(j_2 + q_2)(j_2 + q_3)(2q_2 + 2q_3 + 1)}{(2j_2 + 2q_1 + 2q_2 + 2q_3 + 1)^2} \delta q_1, \tag{D.18}$$

which is always positive as long as $\delta q_1$ (and $j_1$) is positive. Note that we have used the charge relation (43) to express $j_1$ in terms of other charges,

$$j_1 = \frac{-j_2(2q_1 + 2q_2 + 2q_3 + 1) + 4q_1^2(q_2 + q_3) + 2q_1\left(2q_2^2 + 4q_2q_3 + q_2 + 2q_3^2 + q_3\right) + 2q_2q_3(2q_2 + 2q_3 + 1)}{2j_2 + 2q_1 + 2q_2 + 2q_3 + 1}.$$
(D.19)

Therefore, entropy increases until one of hairs is depleted. This process is repeated until either the hair with angular momentum or the hair with charge is fully depleted.

To summarize, a higher entropy configuration can always be found when a hairy black hole simultaneously carries both charge and angular momentum.

**Claim 2: For fixed charges, the configuration with equal angular momenta is entropically dominant on the black hole sheet**

For fixed charges $q_1, q_2, q_3$, the angular momentum $j_1$ on the black hole sheet can be expressed in terms of the other charges. The squared entropy $S^2$ is proportional to

$$S^2 \propto \frac{-j_2^2 + 2j_2\left(q_1(q_2 + q_3) + q_2q_3\right) + 2q_1q_2q_3}{2j_2 + 2q_1 + 2q_2 + 2q_3 + 1}.$$
(D.20)

To identify the angular momentum configuration that maximizes the entropy for fixed charges, we vary $j_2$ and solve the extremization condition:

$$\frac{\partial S}{\partial j_2} = 0.$$
(D.21)

A detailed analysis shows that the entropy is maximized if and only if $j_1 = j_2 = j_L$ given as

$$j_L = \frac{1}{2}\left(\sqrt{(2q_1 + 2q_2 + 1)(2q_1 + 2q_3 + 1)(2q_2 + 2q_3 + 1)} - 2q_1 - 2q_2 - 2q_3 - 1\right).$$
(D.22)

For fixed charges $q_1, q_2, q_3$, the configuration with equal angular momenta is entropically dominant on the black hole sheet.

**Claim 3: The black hole with equal charges is the maxima with angular momenta fixed**

The change in entropy along the black hole sheet is given by

$$\delta S^2 \propto (q_2 + q_3)\delta q_1 + (q_1 + q_3)\delta q_2 + (q_1 + q_2)\delta q_3 + \frac{1}{2}\delta j_1 + \frac{1}{2}\delta j_2.$$
(D.23)

In this case we consider $\delta j_1 = \delta j_2 = 0$. So the variation in entropy is given by

$$\delta S^2 \propto \left(q_2 + q_3 + (q_1 + q_2)\frac{\partial q_3}{\partial q_1}\right)\delta q_1 + \left(q_1 + q_3 + (q_1 + q_2)\frac{\partial q_3}{\partial q_2}\right)\delta q_2.$$
(D.24)

Now $\delta(S^2) = 0$ requires

$$\left(q_2 + q_3 + (q_1 + q_2)\frac{\partial q_3}{\partial q_1}\right) = 0, \quad \text{and} \quad \left(q_1 + q_3 + (q_1 + q_2)\frac{\partial q_3}{\partial q_2}\right) = 0.$$
(D.25)

In order for both terms to be zero, their subtraction must vanish. This requires $q_1 = q_2$. Due to the permutation symmetry among $q_1, q_2$, and $q_3$, it follows that $q_1 = q_2 = q_3$. It can be shown in a straightforward manner that the configuration with equal charges is entropically dominant on the black hole sheet.

Similarly, if one of the charges is fixed in addition to the two angular momenta, it can be shown easily that a black hole with two charges being equal is the entropically dominant on the black hole sheet.

## D.9 Proof of the algorithm to compute supersymmetric phase and entropy

Proof 1: Assume there exists a rank 2 DDBH with its core black hole carrying charges $(q_1, q_2, q_3 - a, j_1, j_2)$ with $a > 0$. The black hole charge relation and the inequality above implies:

$$\frac{a}{2}(j_1 + j_2 - (q_1 + q_2)(1 + 2a + 2q_1 + 2q_2 + 4q_3')) > 0,$$

where $q_3' = q_3 - a$. On the other hand, the entropy formula of the core black hole requires $j_1 + j_2 < 2(q_1 q_2 + q_2 q_3' + q_3' q_1)$. Substituting this, we obtain:

$$2(q_1 q_2 + q_2 q_3' + q_3' q_1) - (q_1 + q_2)(1 + 2a + 2q_1 + 2q_2 + 4q_3') > 0.$$

However, expanding and simplifying the terms shows this is negative:

$$-(1 + 2a)(q_1 + q_2) - 2(q_1^2 + q_2^2) - 2(q_1 q_2 + q_2 q_3 + q_3 q_1) < 0,$$

which is a contradiction. Now we assume a rank 4 DDBH, with its core black hole carrying charges $(q_1, q, q, j_1, j_2)$ with $q_1 < q < q_2$. However, this leads to a contradiction following a reasoning analogous to the rank 2 case. Similarly, for a rank 6 DDBH, such a configuration is not allowed, as it leads to contradictions in the same manner as above.

Proof 2: Assume a rank 2 grey galaxy, whose core black hole carries charges $(q_1, q_2, q_3, j_1, j_2 - a)$ with $a > 0$. However, this means that $\frac{a}{4}(1 + 2j_1 + 2q_1 + 2q_2 + 2q_3) < 0$ which is a contradiction. Similarly, let us assume a rank 4 grey galaxy, there must exist a black hole with $(q_1, q_2, q_3, j, j)$ with $j < j_1 < j_2$. Therefore, $\frac{1}{4}(-2j^2 + 2j_1 j_2 + (j_1 + j_2 - 2j)(1 + 2q_1 + 2q_2 + 2q_3)) < 0$, which is a contradiction.

# E Phases along an indicial line

In this Appendix, we track the entropy of shadow black holes along any given index line.

## E.1 Rank 2 segments

Consider the segment of an index line that happens to lie in the rank 2 phase in which the gas/duals carries only one $\zeta_i$ charge ($i$ could be any of the integers $1 \ldots 5$). The shadow of this segment is formed by connecting each point (on this segment of the index line to the black hole sheet) via rays that start on the index line, and proceed in the direction $-\hat{\zeta}_i$.

Another way of saying this is the following. Consider the plane whose tangent vector space is spanned by $t_I$ (see (8)) and $\hat{\zeta}_i$, and includes the index line of interest. Consider the half strip of this plane that is bounded by the relevant segment of the index line and half lines in the $-\hat{\zeta}_i$. The shadow of this segment of the index line is given by the intersection of this half strip with the black hole sheet.

Where along this shadow segment is the black hole entropy is maximized? An infinitesimal translation $\delta a$ along the index line changes the location of the shadow (i.e. charges of the shadow black hole) by

$$\delta a \, t_I + \delta b \, \hat{\zeta}_i, \tag{E.1}$$

where the the precise value of the infinitesimal $\delta b$ (which is proportional to $\delta a$) depends on the local geometry of the black hole sheet around the intersection point.[146] The resultant

---

[146]Note that $t_I$ is never parallel to a tangent to the black hole sheet. Consequently therefore, an infinitesimal motion along the index line always results in a nonzero $\delta b$.

change in entropy is given by

$$\delta S = \delta a \left( \sum_{i=1}^{5} t_I^i \nu_i \right) + \delta b \ \nu_i. \tag{E.2}$$

Recall, however, that $\sum_{i=1}^{5} t_I^i \nu_i$ vanishes at every point on the black hole sheet, so

$$\delta S = \delta b \nu_i. \tag{E.3}$$

Thus the point $\nu_i = 0$ is always an extremum of the entropy[147] In §C.15 we demonstrate that any such point is, In fact, always a local maximum.[148]

If $\nu_i$ nowhere vanishes then the entropy is monotonic on the interval, and maximized on an end point. The list of possibilities for the two end points of this interval are

- The end point of the index line itself. This always lies on a $(1/8)^{th}$ BPS plane, so has zero entropy and never represents the maximum of the entropy.

- The boundary of the black hole sheet. By definition this boundary also always has zero entropy, and so never captures the maximum entropy.

- The index line itself. At this point the index line pierces the black hole sheet and the system makes a phase transition from a DDBH to a grey galaxy phase. Such an end point represents a possible maximum of the entropy on the interval.

- The phase boundary of the rank 2 black hole phase and rank 4 black hole phase.[149] Such an end point also represents a possible maximum of the entropy on the interval.

## E.2 Rank 4 segments

The shadow of the segment of an index line that lies in a rank 4 phase always has $\zeta_i = \zeta_j$ (where $i$ and $j$ are two of either $1 \ldots 3$ or are 4 and 5), and so also has $\nu_i = \nu_j = \nu$. One finds this shadow as follows. Consider the 3-plane whose tangent space is spanned by $t_I$, $\hat{\zeta}_i$ and $\hat{\zeta}_j$ and contains the index line of interest. Consider a slab of a quadrant of this three plane bounded by the following four (regions of) two planes formed from

- Rays shot out in the $-\hat{\zeta}_i$ from every point on the interval.

- Rays shot out in the $-\hat{\zeta}_j$ from every point on the interval.

- Rays shot out in the directions $-\cos\theta\hat{\zeta}_i - \sin\theta\hat{\zeta}_j$, $\theta \in [0, \frac{\pi}{2}]$, from the topmost point of the interval.

- Rays shot out in the directions $-\cos\theta\hat{\zeta}_i - \sin\theta\hat{\zeta}_j$, $\theta \in [0, \frac{\pi}{2}]$, from the bottommost point of the interval.

This slab intersects the black hole sheet on a two dimensional strip, but intersects the $q_i = q_j$ submanifold of the black hole sheet on an interval. This interval is the shadow of the rank 4 interval of the index line.

---

[147]We emphasize that $\nu_i = 0$ is an extremum - In fact a maximum - of the entropy only in segment of the index line that is in the rank 2 phase equilibrated with a "gas" that carries charge $\zeta_i$, rather than some other charge. There is no special significance to $\mu_i = 0$ in a rank 2 phase with charge $\zeta_j$ when $i \neq j$.

[148]In fact we will later see that if such a point exists on the shadow of the relevant interval of the index line, it always represents the global maximum of the entropy - not just on this segment, but the full index line.

[149]Or a rank 6 black hole when $i$ is one of $1, 2, 3$ (say $i = 1$) and the other two charges are equal (say $q_2 - q_3 = 0$).

The analogue of (E.3) is

$$\delta S = \delta a \left( \sum_{i=1}^{5} t_I^i \, v_i \right) + \delta b_i \, v_i + \delta b_j \, v_j = \left( \delta b_i + \delta b_j \right) v. \tag{E.4}$$

Once again, the entropy has a local extremum on such an interval if and only if $v_i = v_j = v$ vanishes at some point on the shadow. Since $\omega_1$ and $\omega_2$ never simultaneously vanish at an interior point on the black hole sheet (see Appendix C.12 for a proof of this statement), such a rank 4 extremum only exists when $i, j$ are two of $(1, 2, 3)$, and so only for DDBH (and never grey galaxy) phases. Once again such a point - if it exists - is always a local maximum along the index line (see Appendix C.15) and In fact will always turns out to be the global maximum along the entire index line.

When the shadow of interval does not pass through such an extremum, the entropy on the interval is maximum at an end point. The first two possible end points listed in the previous sub subsection have zero entropy. A necessary condition for the third listed end point (intersection with the black hole sheet) to occur is that $\zeta_i - \zeta_j$ vanishes. Consequently such end points are not generic. In every other case (always assuming that $v$ nowhere vanishes on the interval) the entropy along the interval is maximized on the phase boundary between a rank 4 and a 2 rank solution.[150]

## E.3 Rank 6 segments

In this case, the shadow black hole has $q_1 = q_2 = q_3 = q$ and $v_1 = v_2 = v_3 = \mu$. This case is similar to that of the previous subsubsection, and we leave it to the reader to fill out the details. The analogue of (E.4) is

$$\delta S = (\delta b_1 + \delta b_2 + \delta b_3) \mu. \tag{E.5}$$

The entropy is extremized locally only if the shadow passes through the surface $\mu = 0$. As there is, however, no point on the black hole sheet on which $\mu_1$, $\mu_2$ and $\mu_3$ simultaneously vanish, this condition is never met. As a consequence, the maximum along such a segment of the index line lies at the intersection with the black hole sheet if the indicial charges $\zeta_1 - \zeta_2$ and $\zeta_2 - \zeta_3$ both vanish, or at the phase boundary with a rank 4[151] (when this is not the case).

## E.4 Local extrema of entropy

We have explained above that, as we move along any given interval of the index, the entropy of the shadow in the bulk of the interval is locally extremized if and only if the relevant chemical potential vanishes. Such a situation always represents a local maximum of the shadow entropy.

In Appendix C.14 we study the transition points between different intervals (e.g. the transition between a rank 2 and a rank 4 interval, or the transition (through the black hole sheet) between a DDBH and a grey galaxy interval. We demonstrate in that Appendix that

- The transition between two different DDBH phases or two different grey galaxy phases never represent a local extremum of the shadow entropy.[152] While the derivative of the entropy can (and generically does) jump across these transition points, its sign never changes.

---

[150]Potential phase boundaries between rank 4 and rank 6 never represent maximum entropy.

[151]When $q_i - q_j = 0$, the phase boundary can be directly with a rank 2 rather than a rank 4 phase.

[152]There is one fine tuned exception to this rule. When the phase transition between a rank 2 DDBH (with, e.g. $q_1$ gas charge) and a rank 4 DDBH (with $q_1$ and $q_2$ gas charges) happens at a point at which the shadow black holes have $\mu_1 = \mu_2 = 0$, then the maximum of the entropy - which can be seen as occurring either in the rank 2 phase or in the rank 4 phase, depending on which direction we approach from - happens at the phase transition point. See e.g (F.15) and (F.16).

- The transition between a DDBH and grey galaxy phase - which always happens through the black hole sheet - is a local maximum of the shadow entropy if all chemical potentials (both $\mu_i$ and $\omega_j$) are positive. On the other hand it is not an extremum if the intersection occurs where any chemical potential is negative.

In summary, the only extremal points (for the shadow entropy) on the index line are either internal points on an interval at which the relevant chemical potential vanishes, or (potentially) the transition between a DDBH and grey galaxy phase through the black hole sheet. Each such extremum - when it occurs - is a local maximum. The shadow entropy never has a local minimum along the index line. It follows, therefore, that any local maximum is also a global maximum on the index line. Every index line has exactly one such maximum.

## F   Index charges near the surface $q_1 + j_1 = 0$

We wish to look at the index line parametrized by

$$(q_1' = q_1 + j_1, q_2' = q_2 + j_1, q_3' = q_3 + j_1, j_2' = j_2 - j_1).$$

### F.1   $q_1 + j_1 = 0$

The charges on the index line are given by

$$(-\epsilon_1, q_2', q_3', \epsilon_1, j_2'), \tag{F.1}$$

and for $\epsilon_1 \in [-j_2', q_2']$ assuming $q_3' > q_2'$, the index line passes via the Bose-Fermi cone.

In this case, only one point on the index line lies in the EER and this point has the following charges

$$(0, q_2', q_3', 0, j_2').$$

- When $j_2' > 2q_2'q_3'$, the point this index line passes via is the rank 2 grey galaxy phase with a core black hole that lies at the boundary of the black hole sheet.

- When $j_2' = 2q_2'q_3'$, the index line intersects the black hole sheet at the boundary.

- When $q_2' < \sqrt{j_2'/2} < q_3'$, the index line passes via rank 2 DDBH phase with the core black hole again lying at the boundary of the black hole sheet and the charge $q_3$ is carried via the dual giant.

- When $\sqrt{j_2'/2} < q_2' < q_3'$, the index line passes via rank 4 DDBH phase with the core black hole again lying at the boundary of the black hole sheet.

### F.2   $q_1 + j_1 = \epsilon$ and $j_2' \gg 2q_2'q_3'$

We are interested in the index line with $q_1 + j_1 = \epsilon$ and $j_2' > 2q_2'q_3'$. Let us say $j_2' = 2q_2'q_3' + x$ with $x > 0$. In the charge space labeled by $(q_1, q_2, q_3, j_1, j_2)$, we are interested in the following index line

$$\left(\epsilon - \epsilon_1, q_2' - \epsilon_1, q_3' - \epsilon_1, \epsilon_1, 2q_2'q_3' + \epsilon_1 + x\right),$$

where $\epsilon_1$ is a coordinate along the index line. The relevant part of the index line (the part inside the Bose-Fermi cone) is given by the following range of $\epsilon_1$

$$-q_2'q_3' - \frac{x}{2} \le \epsilon_1 \le \frac{q_2' + \epsilon}{2}$$

(where we have assumed $q_2' < q_3'$).

We find that when

$$-\sqrt{\frac{2q_2'q_3'}{1+2q_2'q_3'}}\sqrt{\epsilon} - \frac{4q_2'q_3'}{(1+2q_2'q_3')^2}\epsilon \leq \epsilon_1 \leq \sqrt{\frac{2q_2'q_3'}{1+2q_2'q_3'}}\sqrt{\epsilon} + \frac{4q_2'q_3'}{(1+2q_2'q_3')^2}\epsilon,\qquad \text{(F.2)}$$

the index line passes via the rank 2 grey galaxy phase and is outside the EER for the values of $\epsilon_1$ outside these ranges.

In the special case, when $q_2' \leq \frac{\epsilon}{2q_3'}$, the index line enters rank 4 grey galaxy phase.

## F.3 $\quad q_1 + j_1 = \epsilon$ and $j_2' = 2q_2'q_3' + \epsilon_2$

In this subsection, we consider the index charges such that the index line intersects the black hole sheet. Therefore $j_2'$ is fixed such that it is a small number ($\epsilon_2$) away from $2q_2'q_3'$. The index line is parameterized by following charges

$$(\epsilon - \epsilon_1, q_2' - \epsilon_1, q_3' - \epsilon_1, \epsilon_1, 2q_2'q_3' + \epsilon_2).\qquad \text{(F.3)}$$

The intersection occurs only when $\epsilon_2$ is chosen in the following range

$$-\sqrt{\epsilon(1+2q_2'+2q_3')(8q_2'q_3')} + \delta\epsilon \leq \epsilon_2 \leq \sqrt{\epsilon(1+2q_2'+2q_3')(8q_2'q_3')} - \delta\epsilon,\qquad \text{(F.4)}$$

where $\delta = 2\left(\frac{q_2'(2q_2'+4q_3'+1)}{2q_2'+2q_3'+1} + q_3'\right)$. For these values of $\epsilon_2$, the intersection point lies at the following values of charges

$$(\epsilon - \epsilon_1, q_2' - \epsilon_1, q_3' - \epsilon_1, \epsilon_1, 2q_2'q_3' + \epsilon_2 + \epsilon_1).\qquad \text{(F.5)}$$

For $\epsilon_2 = \epsilon_2^{max} = \sqrt{\epsilon(1+2q_2'+2q_3')(8q_2'q_3')} - \delta\epsilon$, the intersection occurs at

$$\epsilon_1 = -\sqrt{\epsilon}\sqrt{\frac{2q_2'q_3'}{2q_2'+2q_3'+1} - \frac{\epsilon(4q_2'q_3')}{(2q_2'+2q_3'+1)^2}} := \epsilon_1^{min},$$

and for $\epsilon_2 = \epsilon_2^{min} = -\sqrt{\epsilon(1+2q_2'+2q_3')(8q_2'q_3')} + \delta\epsilon$, the intersection occurs at

$$\epsilon_1 = \sqrt{\epsilon}\sqrt{\frac{2q_2'q_3'}{2q_2'+2q_3'+1} + \frac{\epsilon(4q_2'q_3')}{(2q_2'+2q_3'+1)^2}} := \epsilon_1^{max}.$$

Let us analyze the location of intersection point in detail. Depending on the value of $\epsilon_2$, the index line intersects the black hole sheet either in the stable black hole regime ($\omega_2 > 0, \mu_i > 0$) or the unstable black hole regime ($\omega_2 < 0$ or $\mu_i < 0$).

The intersection point lies in the $\omega_2 < 0$ when

$$\frac{2\epsilon\left(2q_2'^2 + q_2' + 2q_3'^2 + q_3' - 2q_2'q_3'\right)}{2q_2'+2q_3'+1} < \epsilon_2 < \epsilon_2^{max}.\qquad \text{(F.6)}$$

Hence for this range of $\epsilon_2$, the dominant solution on the index line is a grey galaxy with the core black hole lying on $\omega_2 = 0$ sheet. The intersection point lies on the $\omega_2 = 0$ sheet when the above lower bound is saturated.

Furthermore, using (49), we find that at

$$\epsilon_2 = -\frac{2\left(6q_2'q_3' + q_2'(2q_2'+3) - 2q_3'^2 + q_3' + 1\right)}{2q_2'+2q_3'+1}\epsilon,\qquad \text{(F.7)}$$

the intersection point lies on $\mu_3 = 0$ surface. So for

$$\epsilon_2^{min} < \epsilon_2 < -\frac{2\left(6q_2'q_3' + q_2'(2q_2' + 3) - 2q_3'^2 + q_3' + 1\right)}{2q_2' + 2q_3' + 1}\epsilon\,, \tag{F.8}$$

$\mu_3 < 0$. Note that in this range $\omega_2 > 0$, hence this agrees with the expectation that both $\mu_i$ and $\omega_i$ cannot be negative in the same region.

Interchanging $q_3'$ and $q_2'$ in (F.7), we find the value of $\epsilon_2$ for which the intersection occurs on $\mu_2 = 0$ sheet. Therefore when

$$\epsilon_2^{min} < \epsilon_2 < -\frac{2\left(6q_2'q_3' + q_3'(2q_3' + 3) - 2q_2'^2 + q_2' + 1\right)}{2q_2' + 2q_3' + 1}\epsilon\,, \tag{F.9}$$

$\mu_2 < 0$. Comparing (F.8), (F.9) and assuming $q_3' > q_2'$, we find that for

$$-\frac{2\left(6q_2'q_3' + q_3'(2q_3' + 3) - 2q_2'^2 + q_2' + 1\right)}{2q_2' + 2q_3' + 1}\epsilon < \epsilon_2 < -\frac{2\left(6q_2'q_3' + q_2'(2q_2' + 3) - 2q_3'^2 + q_3' + 1\right)}{2q_2' + 2q_3' + 1}\epsilon\,, \tag{F.10}$$

$\mu_3 < 0$ and $\mu_2 > 0$, therefore the index will be dominated by a rank 2 DDBH with $q_3$ charge in the dual giant. Whereas for

$$\epsilon^{min} < \epsilon_2 < -\frac{2\left(6q_2'q_3' + q_3'(2q_3' + 3) - 2q_2'^2 + q_2' + 1\right)}{2q_2' + 2q_3' + 1}\epsilon\,, \tag{F.11}$$

both $\mu_3$ and $\mu_2$ are negative at the intersection point. Here either a rank 4 DDBH can dominate the index or a rank 2 DDBH can dominate. We will analyze this case later.

Comparing the ranges given above we find that the intersection point lies in the stable black hole region when

$$-\frac{2\left(6q_2'q_3' + q_2'(2q_2' + 3) - 2q_3'^2 + q_3' + 1\right)}{2q_2' + 2q_3' + 1}\epsilon < \epsilon_2 < \frac{2\epsilon\left(2q_2'^2 + q_2' + 2q_3'^2 + q_3' - 2q_2'q_3'\right)}{2q_2' + 2q_3' + 1}\,. \tag{F.12}$$

For this range of $\epsilon_2$, index is dominated by the black hole phase.

**Intersection point in both $\mu_3 < 0$ and $\mu_2 < 0$ region**

Let us fix value of $\epsilon_2$ lies in the range given in (F.11) and try to see if there is $\mu_2 = \mu_3 = 0$ black hole in the shadow of this index line. If we find such a black hole, then the dominant solution on the index line is given by a rank 4 DDBH, else it is given by rank 2 DDBH.

The core black hole for a rank 4 DDBH solution has the following charges

$$\left(\epsilon - \epsilon_1, q^{BH}, q^{BH}, \epsilon_1, 2q_2'q_3' + \epsilon_2 + \epsilon_1\right)\,, \tag{F.13}$$

where $q^{BH}$ is determined by the black hole charge relation and $q^{BH} < q_2' - \epsilon_1$. We find that the charge constraint is solved when

$$q^{BH} = \sqrt{q_2'q_3'} + q_e\epsilon\,. \tag{F.14}$$

For $q^{BH} < q_2' - \epsilon_1$, we see that the shadow lies rank 4 DDBH only when $q_3' - q_2' = O(\epsilon)$. If $q_3'$ is much larger than $q_2'$, the index will be dominated by rank 2 DDBH even when the intersection point lies in the region where both $\mu_2$ and $\mu_3$ are negative. In this case, the core black hole will have $\mu_3 = 0$ and $\mu_2 > 0$.

Let us consider a special case of index charges

$$q_3' - q_2' = \gamma\epsilon\,, \quad \text{and} \quad \epsilon_2 = -\frac{2\left(6q_2'q_3' + q_3'(2q_3' + 3) - 2q_2'^2 + q_2' + 1\right)}{2q_2' + 2q_3' + 1}\epsilon - \kappa\epsilon\,. \tag{F.15}$$

As $\epsilon_2$ lies in the range (F.11), this index line intersects the black hole sheet in the regime where both $\mu_2$ and $\mu_3$ are negative. We find a $\mu_2 = \mu_3 = 0$ black hole in the shadow of the above index line when

$$\kappa < \frac{2q_2'(\gamma + 2q_2'(8\gamma + 8q_2'(5\gamma + 4(2\gamma - 3)q_2' - 9) - 23) - 7) - 1}{(4q_2' + 1)^3}. \tag{F.16}$$

Therefore for this range of $\epsilon_2$, the dominant saddle along the index line will be a rank 4 DDBH with duals carrying both $q_2$ and $q_3$. For $\kappa$ greater than LHS of (F.16), the dominant solution will be a rank 2 DDBH with the dual giant carrying charge $q_3$.

**Various segments of index line**

With $\epsilon_2$ satisfying (F.4), let us consider various segments of the index line

- Grey galaxy segment: For

$$\epsilon_0 < \epsilon_1 \leq \sqrt{\frac{2q_2'q_3'}{1 + 2q_2'q_3'}}\sqrt{\epsilon},$$

  where $\epsilon_0$ is the value of $\epsilon_1$ at the intersection point. The index line lies in the rank 2 grey galaxy phase with $j_2$ condensing.

- Intersection point i.e. $\epsilon_1 = \epsilon_0$: Depending on the value of $\epsilon_2$, the index line intersects the black hole sheet either in the stable black hole region or in the unstable black hole region. If the index line intersects in the unstable region, the dominant solution along the index line is either a rank 2 grey galaxy, a rank 2 DDBH or a rank 4 DDBH, depending on the values of $\epsilon_2$, $q_2'$ and $q_3'$.

- $-\sqrt{\frac{2q_2'q_3'\epsilon}{1 + 2q_2'q_3'}} \leq \epsilon_1 < \epsilon_0$

  For generic values of index charges, this segment of the index line lies in the rank 2 DDBH phase where the dual giant carries charge $q_3$. But when the index charges satisfy (F.15), the index line lies in rank 4 DDBH phase when

$$\epsilon\left(\frac{2q_2'(\gamma - 2(\kappa + 2) + 4q_2'(3\gamma - 2(\kappa + 3) + 4(2\gamma - 3)q_2'))}{(4q_2' + 1)(8q_2' + 1)}\right) \leq \epsilon_1 < \epsilon_0,$$

  and enters rank 2 DDBH phase for

$$\epsilon_1 > \epsilon\left(\frac{2q_2'(\gamma - 2(\kappa + 2) + 4q_2'(3\gamma - 2(\kappa + 3) + 4(2\gamma - 3)q_2'))}{(4q_2' + 1)(8q_2' + 1)}\right),$$

  and then exists EER at $\epsilon_1 = -\sqrt{\frac{2q_2'q_3'\epsilon}{1 + 2q_2'q_3'}}$.

## F.4 $\quad q_1 + j_1 = \epsilon$ and $j_2' \ll 2q_2'q_3'$

Let us consider index charge $j_2' = 2q_2'q_3' - x$ and $x > \sqrt{\epsilon(1 + 2q_2' + 2q_3')(8q_2'q_3')}$ so that the index does not intersect the black hole sheet. The charges along the index line are given by

$$\left(\epsilon - \epsilon_1, q_2' - \epsilon_1, q_3' - \epsilon_1, \epsilon_1, 2q_2'q_3' + \epsilon_1 - x\right).$$

Assuming $q_3' > q_2'$, we find that the index line passes via rank 4 DDBH phase only for

$$2q_2'(q_3' - q_2') < x \leq 2q_2'q_3',$$

and for

$$-\frac{\sqrt{\left(1-2\sqrt{4q_2'q_3'-2x}\right)\left(2q_2'q_3'-x\right)\epsilon}}{\sqrt{-16q_2'q_3'+8x+1}} \le \epsilon_1 \le \frac{\sqrt{\left(1-2\sqrt{4q_2'q_3'-2x}\right)\left(2q_2'q_3'-x\right)\epsilon}}{\sqrt{-16q_2'q_3'+8x+1}}. \quad \text{(F.17)}$$

The charges of the core black hole in this phase are given by

$$\left(\epsilon-\epsilon_1, q^{BH}, q^{BH}, \epsilon_1, 2q_2'q_3'+\epsilon_1-x\right),$$

where $q^{BH}$ is found by solving the charge constraint.

For

$$\sqrt{\epsilon(1+2q_2'+2q_3')(8q_2'q_3')} < x \le 2q_2'(q_3'-q_2'),$$

the index line passes via a rank 2 DDBH phase with the charge $q_3'$ in the dual giant and for the following range of $\epsilon_1$

$$-\sqrt{\frac{q_2'(2q_2'q_3'-x)}{2q_2'^2+2q_2'q_3'+q_2'-x}}\sqrt{\epsilon} \le \epsilon_1 \le \sqrt{\frac{q_2'(2q_2'q_3'-x)}{2q_2'^2+2q_2'q_3'+q_2'-x}}\sqrt{\epsilon}. \quad \text{(F.18)}$$

The charges of the core black hole in this phase are given by

$$\left(\epsilon-\epsilon_1, q_2'-\epsilon_1, q^{BH}, \epsilon_1, 2q_2'q_3'+\epsilon_1-x\right),$$

where $q^{BH}$ is found by solving the charge constraint.

For a given value of $x$, for the values of $\epsilon_1$ outside the ranges given in (F.17) or (F.18), the index line is outside EER. To summarize, the index lines with $j_2' \gg 2q_2'q_3'$ pass either via a rank 4 DDBH phase or via a rank 2 DDBH phase.

# G The special case $q_1 = q_2 = q_3 = q$

In the next two Appendices we illustrate the discussion, and explicitly implement algorithms of §3 and §4 in two special cuts of charge space. In this Appendix we focus on the three dimensional cut of charge of charges $(q_1, q_2, q_3, j_1, j_2) = (q, q, q, j_1, j_2)$. In the next Appendix we will focus on the cut of charges $(q, q, q', j, j)$.

## G.1 Thermodynamics of vacuum black holes

The most general SUSY black holes with $q_1 = q_2 = q_3 = q$ were presented in [5].[153] In this subsection, we present a more detailed, ab initio discussion of existence region and thermodynamics of these black holes, and verify that our final results agree with the specialization of (43), (44), (41), (42), and (45) to these special charges.

In [5], the two parameter set of SUSY black holes with $q_1 = q_2 = q_3 = q$ were parameterized by two real numbers, $a$ and $b$,[154] subject to the inequalities[155]

$$a+b+ab \ge 0, \quad a < 1, \quad b < 1. \quad \text{(G.1)}$$

---

[153]All the charges in [5] are rescaled by a factor of $\frac{2}{\pi}$ in our normalization. The R-charge $Q$ is further divided by $\sqrt{3}$ to give the BPS condition (G.2).

[154]The BPS black hole solutions of Gutowski-Reall [1] are a special case of the above solutions where the parameters satisfy $a = b$.

[155]These inequalities arise from the requirement that the black hole does not have any naked singularity or closed timelike curves.

The charges and of these black holes, are given, as a function of $a$ and $b$, by[156]

$$e = \frac{E}{N^2} = \frac{(a+b)}{2(1-a)^2(1-b)^2}\left((1-a)(1-b)+(1+a)(1+b)(2-a-b)\right), \tag{G.3}$$

$$j_1 = \frac{J_1}{N^2} = \frac{(a+b)(2a+b+ab)}{2(1-a)^2(1-b)}, \tag{G.4}$$

$$j_2 = \frac{J_2}{N^2} = \frac{(a+b)(2b+a+ab)}{2(1-b)^2(1-a)}, \tag{G.5}$$

$$q = \frac{Q}{N^2} = \frac{(a+b)}{2(1-a)(1-b)}, \tag{G.6}$$

so that

$$j_L = \frac{j_1+j_2}{2} = -\frac{(a+b)^2(ab+a+b-3)}{4(a-1)^2(b-1)^2}, \tag{G.7}$$

$$j_R = \frac{j_1-j_2}{2} = \frac{(a+1)(b+1)(a-b)(a+b)}{4(a-1)^2(b-1)^2}. \tag{G.8}$$

The entropy of these black holes is given by,

$$S_{BH} = N^2 \frac{\pi(a+b)\sqrt{a+b+ab}}{(1-a)(1-b)}. \tag{G.9}$$

Notice that the entropy (G.9) vanishes when the first inequality of (G.1) is saturated.

The renormalized chemical potentials of these black holes are given, as a function of $a$ and $b$, by

$$3\mu_1 = 3\mu_2 = 3\mu_3 \equiv \mu = \omega_L,$$

$$\omega_L = \frac{3(a+b)(1-ab)}{2\sqrt{ab+a+b}\,(a^2+3a(b+1)+b(b+3)+1)},$$

$$\omega_R = \frac{(b-a)(a(b+2)+2b+1)}{2\sqrt{ab+a+b}\,(a^2+3a(b+1)+b(b+3)+1)}, \tag{G.10}$$

where $\omega_{L/R} = \omega_1 \pm \omega_2$.[157]

Upon solving for $a$ and $b$ in terms of $q$ and $j_L$ (using (G.6) and (G.7)) we find

$$a, b = \pm \frac{4q^2 - j_L + \sqrt{6qj_L + j_L^2 + j_L - 16q^3 - 3q^2}}{4q^2 + q}. \tag{G.11}$$

Plugging (G.11) into (G.1) turns this constraint into

$$j_L \le 3q^2 \equiv j_L^{\text{Max}}(q), \tag{G.12}$$

in agreement with the third of (44). The condition that $a$ and $b$ be real also gives

$$j_L \ge j_L^{GR}(q) \equiv \frac{1}{2}\left(-(1+6q)+(1+4q)^{\frac{3}{2}}\right) \tag{G.13}$$

---

[156]It is easily checked that $E$ in (G.3) is given in term of $Q$, $J_1$ and $J_2$ by the the specialization of the BPS bound (14)

$$E = J_1 + J_2 + 3Q. \tag{G.2}$$

[157]Consequently, $\mu_1 + \mu_2 + \mu_3 - \omega_1 - \omega_2 = \mu - \omega_L = 0$.

(this condition can be reobtained from (43)). In summary at any fixed value of $q$, $j_L$ ranges over the interval

$$\frac{1}{2}\left(-(1+6q)+(1+4q)^{\frac{3}{2}}\right) \leq j_L \leq 3q^2 \,. \tag{G.14}$$

Plugging (G.11) into (G.8) gives the following expression for $j_R$ in terms of of $(q, j_L)$[158]

$$j_R = \pm\frac{1}{2}\sqrt{(2j_L+1+6q)^2-(1+4q)^3} \,. \tag{G.15}$$

(G.15) can also be obtained from (43). This equation, together with the inequality (G.12), define the black hole sheet in $(q, j_L, j_R)$ space. Taking the partial derivative of $j_R$, w.t.t. $j_L$ (at fixed $q$) gives

$$\left.\frac{\partial j_R}{\partial j_L}\right|_q = \pm\frac{2j_L+6q+1}{\sqrt{(2j_L+6q+1)^2-(1+4q)^3}} \,. \tag{G.16}$$

Note that the modulus of the RHS of (G.16) - i.e. the modulus of the slope of a constant $q$ section of the black hole sheet - always greater than unity (see fig. 23), a point that will be useful below.

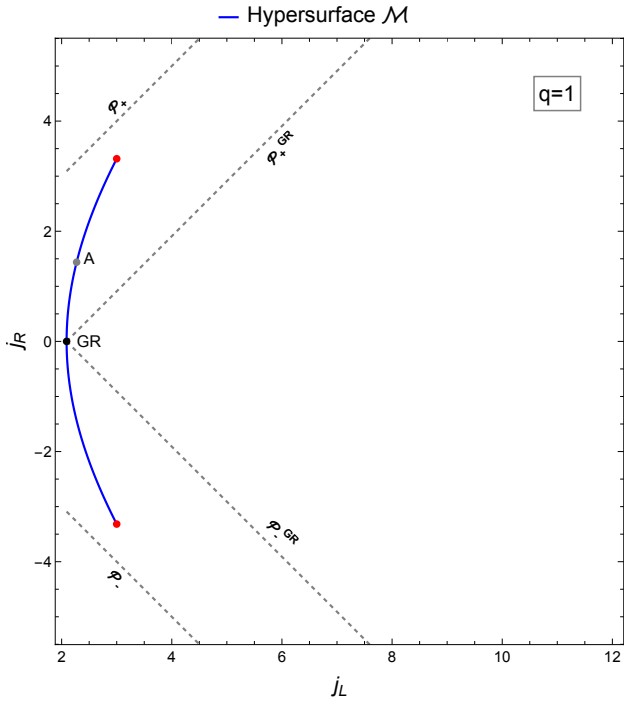

Figure 20: Constant $q(=1)$ cross section of the three dimensional space $(q, j_L, j_R)$. The blue arc is the curve on the surface $\mathcal{M}$ at $q=1$. The black hole solutions of [5] exist at all points on the blue arc. These black holes exist only up to $j_L = 3$ where they become zero size. The $45^o$ dashed lines are the cross sections of two pyramids, where the outer pyramid ($\mathcal{P}$) has its tip at origin and the inner pyramid ($\mathcal{P}^{GR}$) has its tip at the Gutowski-Reall black hole of charge $q=1$ (black dot labelled "GR").

---

[158]It follows immediately from this equation that, at any given $q$, the value of $|j_R|$ increases monotonically with $j_L$, from $j_R = 0$ when $j_L = j_L^{GR}(q)$ to $|j_R| = q^{\frac{3}{2}}\sqrt{2+9q}$ when $j_L = 3q^2$. The value of $|j_R|$ increases monotonically as $j_L$ increases at fixed $Q$. It is also easy to check that $|j_R|$ decreases as $Q$ increases at fixed $j_L$. Finally, substituting $(j_R = 0)$ yields the equation of the curve $j_L^{GR}(q)$ in the $j_R = 0$ plane on which the Gutowski-Reall black holes exist.

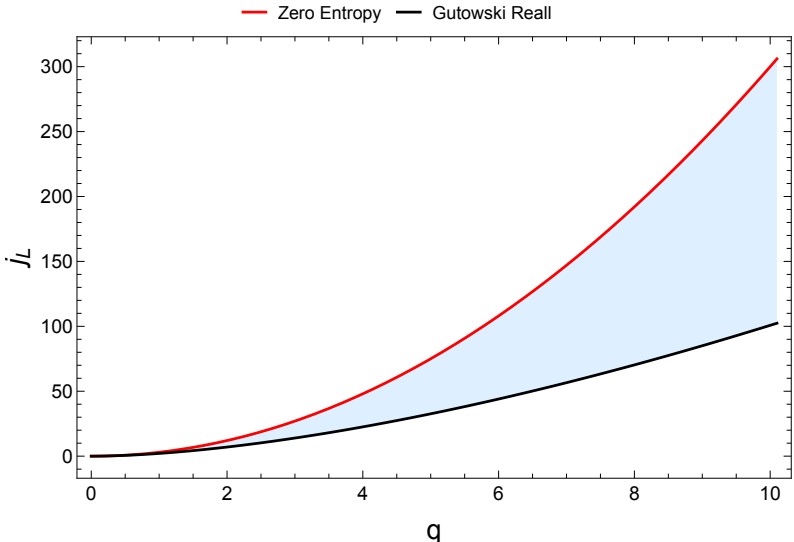

Figure 21: This is a cross section of the 2d black hole sheet with $q_1 = q_2 = q_3 = q$ such that $j_R$ is out of the plane of paper. The black curve represents black holes with $j_R = 0$(Gutowski-Reall black holes). The red black holes have maximum $j_R$ for a given $q$. The entropy of black holes on red curve vanishes.

We now have a complete picture of the embedding of the black hole sheet into the three dimensional space parameterized by $(q, j_L, j_R)$. At any fixed value of $q$, this sheet is the curve depicted in Fig. 20. This curve (which is topologically an interval or a $B_1$) - is simply the $q_1 = q_2 = q_3$ "core of the apple" in Fig 2. The full black hole sheet is a fibration of this "base" over $q$. The $C$ shape in Fig. 20 is of zero size at $q = 0$. Its size increases without bound as $q$ increases. The apex of the $C$ shape also shifts to higher values of $j_L$ as $q$ increases. Topologically, the black hole sheet is an infinite triangle, whose two finite edges occur at $j_L = \pm 3q^2$. Momentarily regarding $j_R$ as the $z$ axis, In Fig. 21 we depict the "top view" of the supersymmetric black hole sheet in the $j_L - q$ plane. The modulus of right angular momentum $|j_R|$ increases as one moves along the y-axis of figure.

Finally, plugging (G.11) into (G.9) yields a simple expression for the entropy as a function of $j_L$ and $q$

$$S_{BH}(q, j_L) = 2\pi \sqrt{3q^2 - j_L}, \tag{G.17}$$

in agreement with (45).

Inserting (G.11) into (G.10) yields an expression for the renormalized chemical potentials as functions of $q$ and $j_L$: we find

$$\mu = \omega_L = \frac{(6q + 3)j_L - 6q^2}{(-4j_L + 12q(4q + 1) + 1)\sqrt{3q^2 - j_L}},$$

$$\omega_R = -\frac{(6q + 1)\sqrt{j_L(j_L + 6q + 1) - q^2(16q + 3)}}{(-4j_L + 12q(4q + 1) + 1)\sqrt{3q^2 - j_L}}. \tag{G.18}$$

These expressions are consistent with (41) and (42) in the sense that the real part of the chemical potentials in (41) and (42) agrees with the chemical potential defined above [8].

It is not difficult to check that $\omega_1$ or $\omega_2$ are negative in a strip of the black hole sheet near the boundary. The region of the black hole sheet with $\omega_1 < 0$ is separated from the region of the black hole sheet with $\omega_1 > 0, \omega_2 > 0$ by the critical curve on the black hole sheet satisfying

$\omega_1 = 0$ and is given by

$$j_L^c(q) = \frac{1 + 18q(1+4q)^2 - (1+4q)^{\frac{3}{2}}(1+6q)\sqrt{1+12q}}{16(1+3q)} \tag{G.19}$$

($j_R$ along this curve is also determined, as a function of $q$, by plugging (G.19) into (G.15)).

## G.2   The special case $j_R = 0$

As a warm up, let us first analyse the extremely special case $j_R = 0$. In this case the only nonzero charges are $q$ and $j_L$. The black hole sheet is a curve - the curve of Gutowski-Reall (GR) black holes [1]. This curve is depicted in Fig 22 and takes the form $j_L = j_L^{GR}(q)$ (see (G.13)).

In this case the EER is simply the region $j_L \geq 0, q \geq 0$. The EER has two components. The grey galaxy component - the blue region in Fig 22 - is the region $j_L \geq j_L^{GR}(q)$. The DDBH component of the EER (the white region in Fig. 22 - is the region that obeys $j_L \leq j_L^{GR}(q)$.

The supersymmetric entropy of a point $A$ (with charges $(q, j_L)$), in the grey galaxy component of the EER, is simply $S_{BH}(q, j_L^{GR}(q))$ where $j_L^{GR}(q)$ is given in (G.13). Geometrically, the SUSY entropy of the point $A$ is the entropy of the GR black hole vertically below it (see Fig. 22).

The supersymmetric entropy of a point $B$ (with charges $(q, j_L)$), in the DDBH component of the EER, is simply $S_{BH}(q', j_L)$, where $q'$ is defined so that $j_L^{GR}(q') = j_L$. Geometrically, the SUSY entropy of the point $B$ is the entropy of the GR black hole, horizontally to its left (see Fig. 22).

An index line runs across Fig. 22 at $-45$ degrees. It follows immediately from the fact that the entropy of GR black holes is a monotonically increasing function of $q$ - and the constructions described in the previous two paragraphs - that the entropy along any such index line is maximum at the intersection with the black hole curve. This completes the analysis of the case $j_R = 0$.

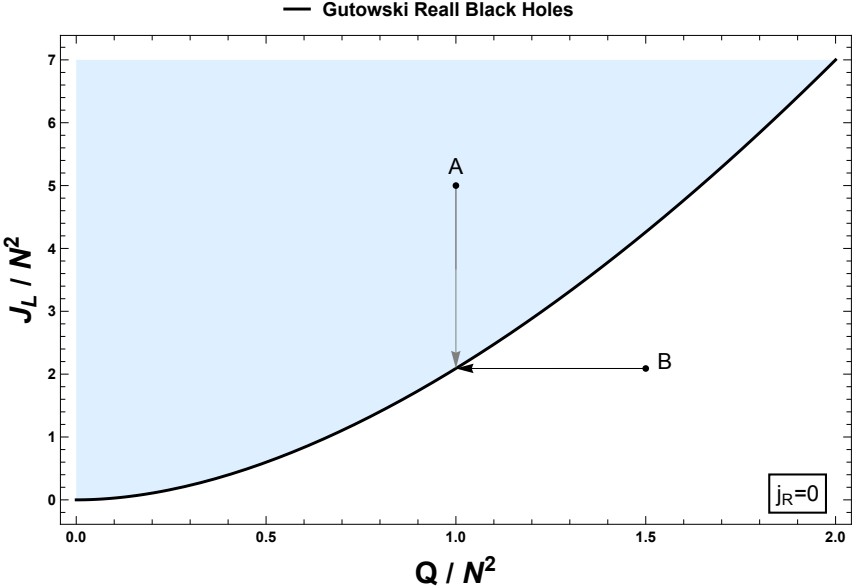

Figure 22: Phase diagram for $j_R = 0$ black holes; The black curve represents the one parameter Gutowski-Reall black holes. There exists a RBH at every point in the blue region above the Gutowski-Reall curve. The entropy of these RBHs is given by the entropy of the corresponding Gutowski-Reall black hole with the same charge ($Q$).

In the rest of this section we turn to the analysis of the general case (when $j_R$ is an arbitrary nonzero number).

## G.3 Equation for the EER

As we have explained in previous sections, the EER consists of a DDBH component and a grey galaxy component. In section 3 we have explained that the boundary of the EER is generated by rays emitted (in appropriate directions) from the end of the black hole sheet, which, in this case is the curve

$$(q, j_L, j_R) = \left( q, 3q^2, \pm q^{\frac{3}{2}} \sqrt{2 + 9q} \right). \tag{G.20}$$

The EER has two components, which we study in turn.

### G.3.1 The DDBH component of the EER

The DDBH component of the boundary of the EER is generated by rays emitted in the positive $\hat{q}$ direction from each point on the boundary of the black hole curve. The DDBH part of the EER is the region that obeys

$$\frac{1}{4} \left( (2j_L + 1 + 6q)^2 - (1 + 4q)^3 \right) \leq j_R^2,$$

$$j_R^2 \leq \left( \frac{j_L}{3} \right)^{\frac{3}{2}} \left( 2 + 9 \left( \frac{j_L}{3} \right)^{\frac{1}{2}} \right). \tag{G.21}$$

The first of (G.21) asserts that the DDBH component of the EER has a larger value of $q$ than the black hole sheet at the same values of $j_L$ and $j_R$.[159] The second of (G.21) asserts that $j_R$ has to lie within the surface formed from the union of rays emitted - in the direction of increasing $q$ - from the end points of the black hole sheet.

### G.3.2 The grey galaxy component of the EER

The grey galaxy component of the boundary of the EER is generated by rays emitted in the $\hat{j}_1$ direction from the $j_R > 0$ part of the black hole boundary curve, together with rays emitted in the $\hat{j}_2$ direction from the $j_R < 0$ segment of the black hole boundary curve. In equations, this part of the EER is given by the region

$$\frac{1}{4} \left( (2j_L + 1 + 6q)^2 - (1 + 4q)^3 \right) \geq j_R^2,$$

$$j_R^2 \leq \left( j_L - 3q^2 + q^{\frac{3}{2}} \sqrt{2 + 9q} \right)^2. \tag{G.22}$$

The first of (G.22) asserts that the grey galaxy component of the EER has a larger value of $j_L$ than the black hole sheet at the same values of $q$ and $j_R$.[160] The second of (G.22) asserts that $j_R$ has to lie within the surface formed from the union of rays emitted - in the direction of increasing $j_1$ (when $j_1 > j_2$) or the direction of increasing $j_2$ (when $j_2 > j_1$) - from the end points of the black hole sheet.

## G.4 Prediction for $S_{BPS}(q, j_L, j_R)$ in the grey galaxy component of the EER

Consider a point with charges $(q, j_L, j_R)$ that lies in the grey galaxy part of the EER. Consider the constant $q$ slice (at the value of $q$ carried by the charge of interest). Such a slice takes

---

[159]Recall that the quantity on the LHS of the first of (G.21) is a decreasing function of $q$ at fixed $j_L$.

[160]Recall that the quantity on the LHS of the first of (G.22) is an increasing function of $j_L$ at fixed $q$.

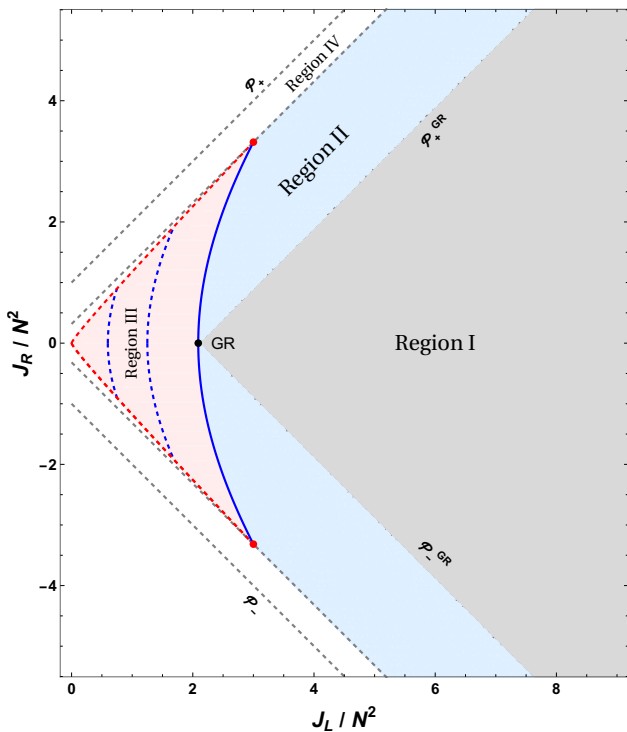

Figure 23: We depict a constant $q$ slice of charge space. The dark blue line depicts the intersection of the black hole sheet $\mathcal{M}$ with our constant $q$ slice. The dashed lines $\mathcal{P}_{\pm}$ and $\mathcal{P}_{\pm}^{GR}$ are the intersections of the planes $\mathcal{P}_{\pm} : |j_R| = j_L + q$ and $\mathcal{P}_{\pm}^{GR} : |j_R| = j_L - j_L^{GR}(q)$ with the constant $q$ plane.

the form depicted in Fig. 23. By assumption, our point lies in either the grey or blue regions (Region I or Region II) of Fig. 23.[161]

If our point lies in the Region I, the dominant solution is a rank 4 grey galaxy. The black hole at the centre of this solution is the Gutowski-Reall (GR) black hole[162] marked by the black dot in Fig 23 This black hole carries charges $(q, j_L, j_R) = (q, j_L^{GR}(q), 0)$. The gas in the solution carries charges $(q, j_L, j_R) = (0, j_L - j_L^{GR}(q), j_R)$. Because our point has been assumed to lie in the Region I, the charges of the gas are in the allowed range $(j_1^{gas} > 0, j_2^{gas} > 0)$. The supersymmetric entropy at the given charges is thus given by $S_{BH}(q, j_L^{GR}(q), 0)$.

If our point lies in the blue region (Region II) of Fig. 23, then the dominant phase is a rank 2 grey galaxy whose gas carries $j_1/j_2$ charge, depending on whether our point lies in the upper/lower part of Region II. The black hole at the centre of the grey galaxy solution is obtained by tracing back from the charge of interest to the black hole sheet, along a 45 degree line (if our charge lies in the upper part of Region II) or a $-45$ degree line (if our charge lies in the lower part or region 2). The entropy at these charges is, then, simply the entropy of the black hole determined by this construction.

The discussion of these paragraphs can be summarized in equations as follows. Within the

---

[161]This is the case because the the union of the grey and blue regions make up the constant $q$ slice of the grey galaxy part of the EER. In contrast, the pink part of Fig 23 is the constant $q$ slice of the the DDBH part of the of the EER.

[162]That this is the dominant solution at these charges, follows from the observation that no black hole at charge $\leq q$ carries a larger entropy than the Gutowski-Reall black hole at charge $q$.

grey galaxy component of the EER , the cohomological entropy is given by

$$
S(q, j_L, j_R) = \left\{ \begin{array}{ll} S_{BH}(q, j_L^{GR}(q)), & |j_R| \leq j_L - j_L^{GR}(q), \\ S_{BH}(q, j_L^B), & j_R \geq j_L - j_L^{GR}(q), \\ S_{BH}(q, j_L^B), & j_R \leq -j_L + j_L^{GR}(q), \end{array} \right.
\tag{G.23}
$$

where $S_{BH}$ is given in (G.17) and $j_L^B$ appearing in the second line is the real and positive solution to

$$
j_L - j_R = j_L^B - j_R^{BH}(q, j_L),
\tag{G.24}
$$

and in the third line, $j_L^B$ is the real and positive solution to

$$
j_L + j_R = j_L^B + j_R^{BH}(q, j_L).
\tag{G.25}
$$

## G.5 Prediction for $S_{BPS}(q, j_L, j_R)$ in the DDBH component of the EER

In the DDBH component of the EER, the dominant solution is a rank 6 DDBH solution. Any point in this component of the EER lies in the pink region of Fig 23. The dominant solution is the black hole that lies directly to its left. The entropy of this solution is given by

$$
S(q, j_L, j_R) = S_{BH}(q^{BH}(j_L, j_R), j_L),
\tag{G.26}
$$

where $q^{BH}(j_L, j_R)$ is the unique real solution of $q$ to (G.15).

## G.6 Prediction for the superconformal index

### G.6.1 The index line

In the context of the special cut of charges of interest to this section, distinct index lines are conveniently labeled

$$
\alpha = q + j_L, \qquad j_R^0 = j_R.
\tag{G.27}
$$

The index is only nontrivial if $\alpha > |j_R|$ (which, in particular means that $\alpha > 0$). Below we assume that this inequality is obeyed.

It is easily verified that the segment within the Bose-Fermi cone - of such an index line - extends between the points with charges $(q, j_L, j_R)$ given by

$$
(0, \alpha, j_R^0), \quad \text{and} \quad (\alpha, 0, j_R^0).
\tag{G.28}
$$

More generally, the segment of the index line within the Bose-Fermi cone is given by

$$
(x, \alpha - x, j_R^0), \quad \text{with} \quad 0 \leq x \leq \alpha.
\tag{G.29}
$$

In the rest of this subsection, we will trace the shadow of the part of the index line (on the black hole sheet), for various different ranges of the indicial parameters $\alpha$, $j_R^0$.

### G.6.2 $j_R^0 = 0$

In this case the shadow of all points in (G.29) with

$$
j_L^{GR}(x) \leq \alpha - x,
\tag{G.30}
$$

lie in the rank 4 grey galaxy phase. This inequality is met in the "large angular momentum" part of the index line. As shown in Fig 24, the charge $q$ and the value of $j_L$ increase monotonically (in this segment) as $x$ increases from zero to the value at which the inequality in (G.30) is

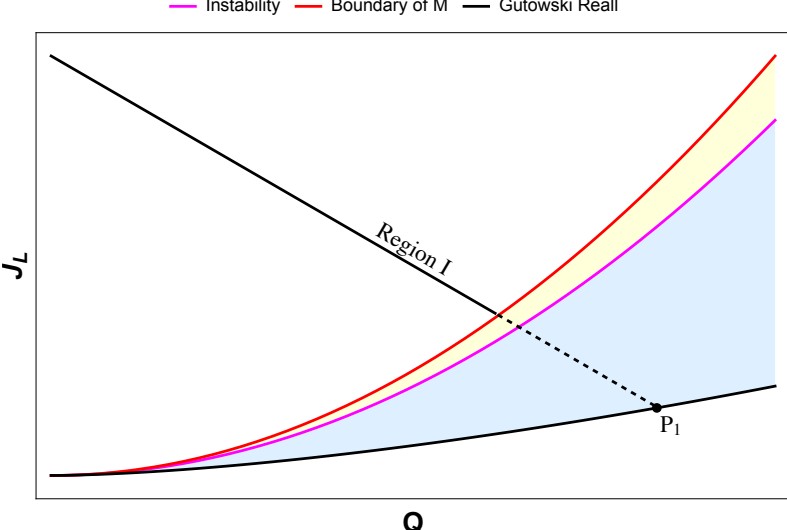

Figure 24: In this case we consider an index line which has $j_R = 0$ and hence intersects the Gutowski-Reall black holes on the black hole sheet.

saturated. As all chemical potentials are positive all through this segment, the supersymmetric entropy of the shadow also increases monotonically as $x$ increases.

On the other hand, the shadow of all points with

$$j_L^{GR}(x) \geq \alpha - x\,, \tag{G.31}$$

are in the rank 6 DDBH phase. This inequality is met in the "large charge" part of the index line. The charge $q$ and the value of $j_L$ decrease monotonically (in this segment) as $x$ increases from the value at which the inequality in (G.31) is saturated - up to $\alpha$. As all chemical potentials are positive all through this segment as well, the supersymmetric entropy of the shadow decreases monotonically as $x$ increases through this segment.

According to conjecture 2, in this case the large $N$ indicial entropy equals the entropy of the black hole at the intersection of the index line and the black hole sheet.

### G.6.3  $|j_R^0| \leq j_R^c(\alpha)$

As we have explained around (G.19), the black hole sheet hosts a distinguished curve, which separates the regions with $\omega_1 < 0$ or $\omega_2 < 0$ from regions in which $\omega_1, \omega_2 > 0$. The shadow of the index line behaves differently, depending on which of these regions the index line intersects the black hole sheet. In this subsubsection we study index lines that have the property that they intersect the black hole sheet in the region where $\omega_1, \omega_2 > 0$. This is the case when the indicial charge $|j_R^0|$ is not too large; more precisely when $|j_R^0| \leq j_R^c(\alpha)$

$$\begin{aligned} \alpha &= q + j_L^c(q)\,, \\ |j_R^c| &= \frac{1}{2}\sqrt{(2j_L^c(q) + 1 + 6q)^2 - (1 + 4q)^3}\,, \end{aligned} \tag{G.32}$$

where $j_L^c(q)$ is given in (G.19).

In this case the shadow of all points in (G.29) with

$$j_L - j_L^{GR}(x) > |j_R^0|\,, \tag{G.33}$$

lie in the rank 4 grey galaxy phase. The gas in this phase carries charges $(q^{gas}, j_L^{gas}, j_R^{gas}) = (0, \alpha - x - j_L^{GR}(x), j_R^0)$. At the value of $x$ at which the inequality (G.33)

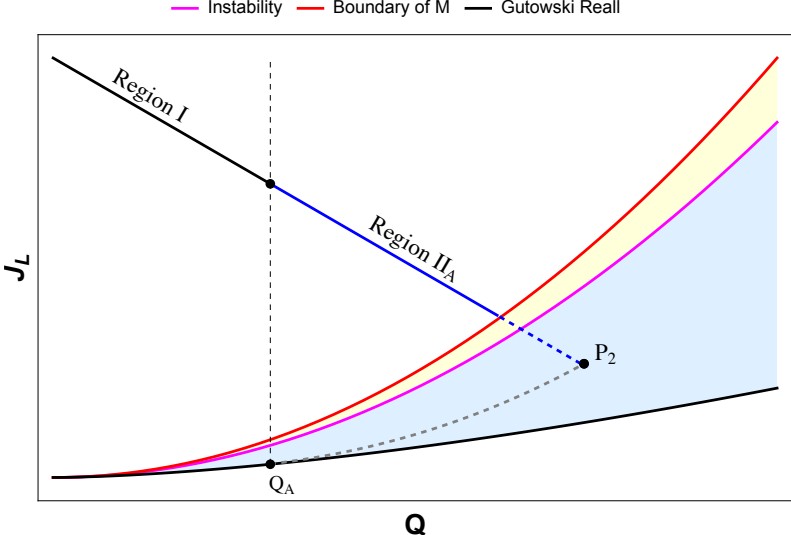

Figure 25: The case where the index line intersects the black hole sheet below $\omega_1 = 0$ curve (which is depicted as a pink curve).

is saturated, $|j_L|^{gas} = |j_R|^{gas}$, i.e. either $j_1^{gas}$ or $j_2^{gas}$ goes to zero (depending on whether $j_R > 0$ or $j_R < 0$. At larger values of $x$, the shadow of the index line moves into the rank 2 grey galaxy phase (in which the gas carries only $j_1$ or only $j_2$ charge, depending on whether $j_R > 0$ or $j_R < 0$). The shadow remains in the rank 2 grey galaxy phase while

$$|j_R^0| > |j_R^{BH}(x, \alpha - x)|. \tag{G.34}$$

The index line hits the black hole sheet at the value $x$ at which the inequality (G.34) is saturated. At all larger values of $x$, the shadow of the index line is in the DDBH phase, until it (the shadow) hits the edge of the black hole sheet when $x$ satisfies the following equations:

$$3(q^{BH}(\alpha - x, j_R^0))^2 = \alpha - x, \tag{G.35}$$

where $q^{BH}(j_l, j_R)$ is the value of $q$ on black hole sheet as function of $j_L$ and $j_R$. At larger values of $x$ the index line leaves the EER.

Since we have assumed $|j_R|^0 \leq j_R^c(\alpha)$, all chemical potentials are positive in all three segments described above. Consequently, the variation of the supersymmetric entropy as a function of $x$ is very similar to the previous subsubsection. The entropy increases monotonically as $x$ increases from 0 to the saturation of the inequality (G.34), and the decreases monotonically as $x$ decreases from this value to zero (when $x$ saturates (G.35)). The entropy stays zero at large values of $x$.

According to conjecture 2, in this case the large $N$ indicial entropy equals the entropy of the black hole at the intersection of the index line and the black hole sheet.

### G.6.4 $\quad |j_R^0| \geq j_R^c(\alpha)$

Once again, in this case the shadow of all points in (G.29) with (G.33) lie in the grey galaxy phase. As $x$ is further increased, however, the shadow now passes through the critical curve (G.19) at a value of $x$ that obeys

$$\alpha - x = j_L^c(x). \tag{G.36}$$

As $x$ is further increased we have two possibilities. If

$$|j_R^0| < \left| j_R^{BH} \left( \frac{1}{6} \left( \sqrt{12\alpha + 1} - 1 \right), \frac{1}{12} \left( \sqrt{12\alpha + 1} - 1 \right)^2 \right) \right|, \tag{G.37}$$

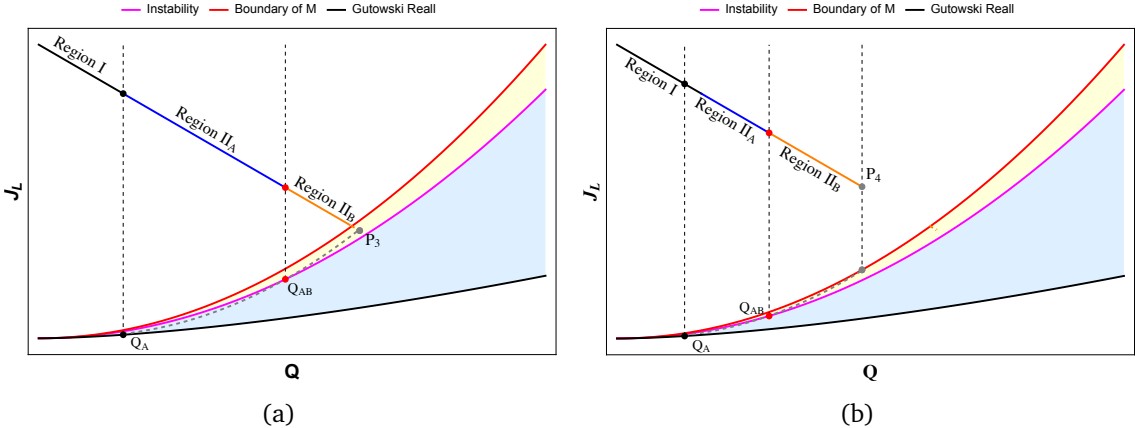

Figure 26: Fig (a) shows the index line in the case where it intersects the black hole sheet above $\omega_1 = 0$ curve (which is depicted as a pink curve). Fig (b) shows the index line in the case where it does not intersect the black hole sheet.

the subsequent evolution of the shadow is similar to that in the previous subsubsection. The shadow of all points with $x$ between the saturation values of (G.33) and (G.34) lie in the rank 2 grey galaxy phase. Once again, the shadows of all $x$ that lie between the saturation values of (G.34) and (G.35) lie in the rank 6 DDBH phase. All points with still larger values of $x$ lie outside the EER.

On the other hand if (G.37) is not obeyed, the index line nowhere pierces the black hole sheet. In this case, at values of $x$ that are larger than the solution of (G.36), the shadow of the index line remains in the rank 2 grey galaxy phase until it hits the boundary of the black hole sheet when

$$3x^2 + x - j_R^{BH}(x, 3x^2) = \alpha - j_R^0. \tag{G.38}$$

Points with still large values of $x$ exit the EER.

In both cases described above, the behaviour of the supersymmetric entropy at the shadow point is similar. The entropy increases monotonically until we reach the solution of (G.36). At this point, $\omega_1$ (if $j_R > 0$) or $\omega_2$ (if $j_R < 0$) becomes zero. The entropy then decreases as $x$ is further increased, continuing to decrease until the point along the index line exits the EER.

According to conjecture 2, in this case the large $N$ indicial entropy equals the entropy of the black hole whose charges are determined by the solution of (G.36).

## G.7 Summary

In this subsection, we will summarize our predictions for the index and the cohomology of $N = 4$ SYM in the $q_1 = q_2 = q_3 = q$ sector.

### G.7.1 Predictions for cohomology

The entropy at charges $(q, j_L, j_R)$ is given by

$$S(q, j_L, j_R) = \begin{cases} S_{BH}(q, j_L^{GR}(q)), & j_L^{GR}(q) \le j_L - |j_R| \implies \text{rank 4 grey galaxy,} \\ S_{BH}(q, j_L), & -j_L^{GR}(q) < |j_R| - j_L \le |j_R^{\max}| - 3q^2 \\ & \implies \text{rank 2 grey galaxy,} \\ S_{BH}(q^{BH}(j_L, j_R), j_L), & j_L < j_L^{GR}(q, j_R) \implies \text{rank 6 DDBH,} \end{cases} \tag{G.39}$$

where $j_R^{\max} = \pm\frac{1}{2}\sqrt{(6q^2+6q+1)^2-(4q+1)^3}$ and $S_{BH}$ is the entropy of the supersymmetric black holes given in (G.17) and $q^{BH}(j_L, j_R)$ can be found by solving (G.15). The entropy is non-zero only if the entropies of the core black holes in all three lines are real and positive.

### G.7.2  Prediction for the index

The supersymmetric index is labeled by two charges, namely $\alpha = q + j_L$ and $j_R^0$.

The expression for the index entropy as a function of $\alpha$ and $j_R^0$ is given by

$$
I(\alpha, j_R^0) = 
\begin{cases}
S_{BH}(x(\alpha, j_R^0), \alpha - x(\alpha, j_R^0)), & |j_R| \leq |j_R^c(\alpha)| \implies \text{pure black hole}, \\
S_{BH}(q, j_L^c(q)), & |j_R^c(\alpha)| < |j_R| \leq \alpha \\
& \qquad\qquad \implies \text{rank 2 grey galaxy}, \\
0, & |j_R| > \alpha.
\end{cases}
\tag{G.40}
$$

$j_R^c(\alpha)$ is defined in (G.32) and $j_L^c(q)$ is given in (G.19). In the first line, $x(\alpha, j_R^0)$ is the real positive solution to

$$
j_R^{BH}(x, \alpha - x) = j_R^0.
\tag{G.41}
$$

In the second line, $q$ is the real positive solution to

$$
q + j_L^c(q) - j_R^{BH}(q, j_L^c(q)) = \alpha - j_R^0.
\tag{G.42}
$$

## H  Equal angular momenta and two equal $SO(6)$ charges

### H.1  Vacuum black holes

In this Appendix we study supersymmetric states with two of the R-symmetry charges equal (say $q_1 = q_2 = q$ and $q_3 = q'$) and the two angular momentum equal to each other ($j_1 = j_2 = j$). The states we study carry the charges $\{\zeta_i\} = (q, q, q', j, j)$.[163]

BPS black holes with such charges lie on the curve $q_1 = q_2$ on the red equatorial plane of Fig 2. This curve may be seen in Fig. 5 (at small values of charge) and Fig. 6 (at larger values of charge). Quantitatively, these black holes can be parameterized by two charges $(q, q')$. Their angular momentum $j$ is determined as a function of $q$ and $q'$ by the black hole sheet equation (43)[164]

$$
j(q, q') = \left(q + q' + \frac{1}{2}\right)\sqrt{4q + 1} - \left(2q + q' + \frac{1}{2}\right).
\tag{H.2}
$$

Their entropy is given (in units of $N^2$) by

$$
s(q, q') = 2\pi\sqrt{q^2 + 2qq' - j(q, q')}.
\tag{H.3}
$$

The black hole solutions are nonsingular when the argument of the square root in (H.3) has to be positive. This condition yields a lower bound on $q'$,

$$
q' \geq \frac{1}{4}\left(-1 - 2q + \sqrt{1 + 4q}\right).
\tag{H.4}
$$

---

[163]Of course, the energy of these states is determined by the BPS bound (13) to be $E/N^2 = 2q + q' + 2j$.

[164]On the charge cut of interest to this Appendix, (43) specializes to

$$
\mathcal{P}(q, q', j) \equiv q^2 q' + \frac{1}{2}j^2 - \left(\frac{1}{2} + 2q + q'\right)\left(q^2 + 2qq' - j\right) = 0.
\tag{H.1}
$$

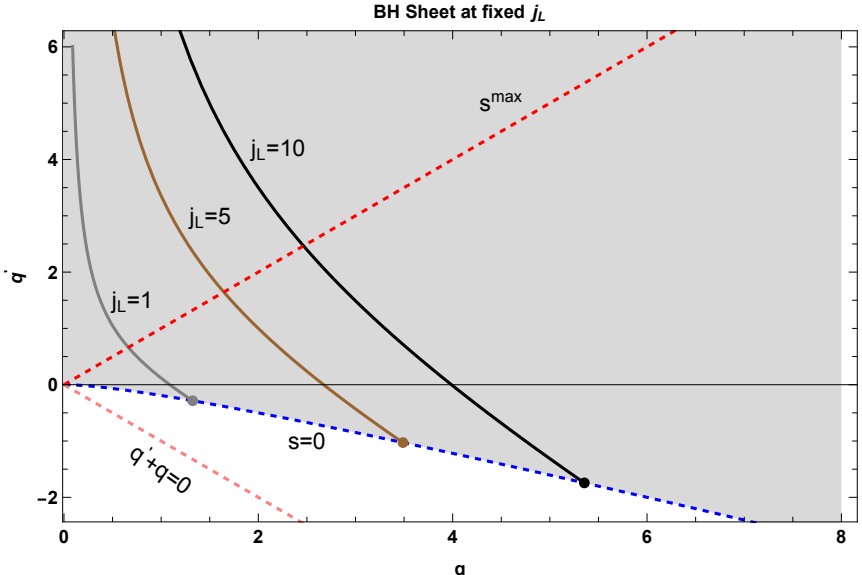

Figure 27: We display a "top view" of the black hole sheet (grey color), i.e. a projection of the black hole sheet to the $q, q'$ plane. The solid curves represent the intersection of the black hole sheet with planes of constant $j$ ("level surfaces"). Note that the black hole sheet rises in the direction of larger $q$ and larger $q'$. The black hole sheet ends on the blue dashed curve, where the black hole entropy vanishes. The entropy at any given $j$ is largest on the red dashed curve (this curve represents Gutowski-Reall black holes with equal charges and angular momenta).

The above inequality is saturated on a one-dimensional curve (blue dashed curve in Figure 27) in the three dimensional charge space $(q, q', j)$. The entropy of the black holes on this curve vanishes and the black hole sheet ends on this curve.[165] It follows that the black hole sheet is defined by the equations (H.2) together with the inequalities

$$q > 0, \qquad q' > \frac{1}{4}\left(-1 - 2q + \sqrt{1+4q}\right).$$  (H.5)

The allowed region (H.5) is the region shaded in Figure 27.

In order to visualize the black hole sheet, it is useful to study its "level surfaces", i.e. intersections with planes of constant $j$. They are given by

$$q' = \frac{j}{4q}\left(1 + \sqrt{1+4q}\right) + \frac{\sqrt{1+4q}}{4}\left(1 - \sqrt{1+4q}\right)$$  (H.6)

(see Figure 27 for a sketch of these curves at various values of $j$). The entropy along constant $j$ curves starts at zero when $q = 0$ (at this point $q' \to \infty$), and increases and attains a maximum at a value $q = q_0$ and then starts to decrease to zero as it approaches the zero entropy curve (H.4) (see Figure 28).

The curve on the black hole sheet where the black holes have maximum value of entropy at a given value of $j$ is obtained by maximizing (H.3) with respect to $q$ at fixed $j$ after substituting (H.6). This gives the equation,

$$q' = q.$$  (H.7)

---

[165]From (H.4), we can further infer that for all $q > 0$, $q' < 0$ on the zero entropy curve. This tells us that this curve lies outside the bosonic cone.

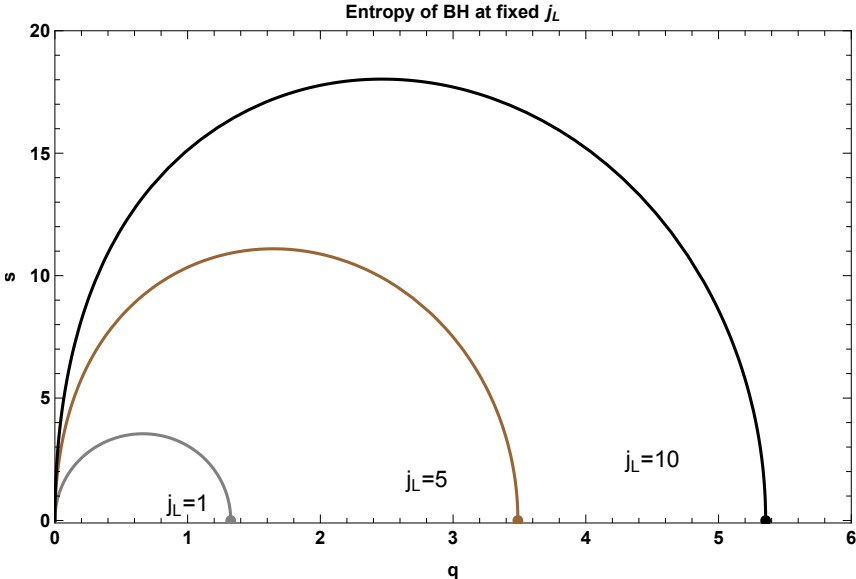

Figure 28: Entropy on the cross sections of the BH sheet at various values of constant $j_L$.

Therefore, the entropy at a given $j$ is maximized when all the charges are equal (red dashed line in Figure 27), i.e. on the one parameter set of Gutowski-Reall black holes.[166]

## H.2 Stability of vacuum black holes

In this section, we will check if any vacuum supersymmetric black hole solution on the black hole sheet itself is unstable to emission of gas/dual giants and form a more entropic configuration. An easy way is to first construct constant entropy contours on the BH sheet. To obtain them, we eliminate $j$ form (H.3) and (H.1), and write the resulting constraint in terms of $q'(q)$ at fixed $s$,

$$q' = \frac{1}{4}\left(\frac{\left(1 + 2q + \sqrt{4q+1}\right)\left(\frac{s}{2\pi}\right)^2}{q^2} + \left(-1 - 2q + \sqrt{4q+1}\right)\right). \tag{H.8}$$

The contours of constant entropy are shown in Figure 29.

The allowed charges for the non-black hole components - dual giants and graviton gas - are,

$$q \geq 0, \qquad q' \geq 0, \qquad j \geq 0. \tag{H.9}$$

Let us start with a BH solution whose charges are given by some point on the $S = const$ (say $S = 10$) curve. The potential black holes that it can decay to would lie in a cone defined by the inequalities (H.9), and whose tip is at the point of interest. From the Figure 29, it is clear that all the black holes on the black hole sheet that lie inside this cone have less entropy than the original black hole at the tip of the cone (this follows because the cones described above extend down and two the left of any point, and so move towards lower entropy). Therefore, no vacuum supersymmetric black hole is unstable in the microcanonical ensemble towards the formation of either DDBH or grey galaxies (or their combination).

More analytically, the stability of the vacuum supersymmetric black hole can be demonstrated using the thermodynamics of the vacuum black holes. Consider a vacuum black hole

---

[166]This result will later (in §H.3) help us in identifying the dominant phase of BPS configurations away from the black hole sheet.

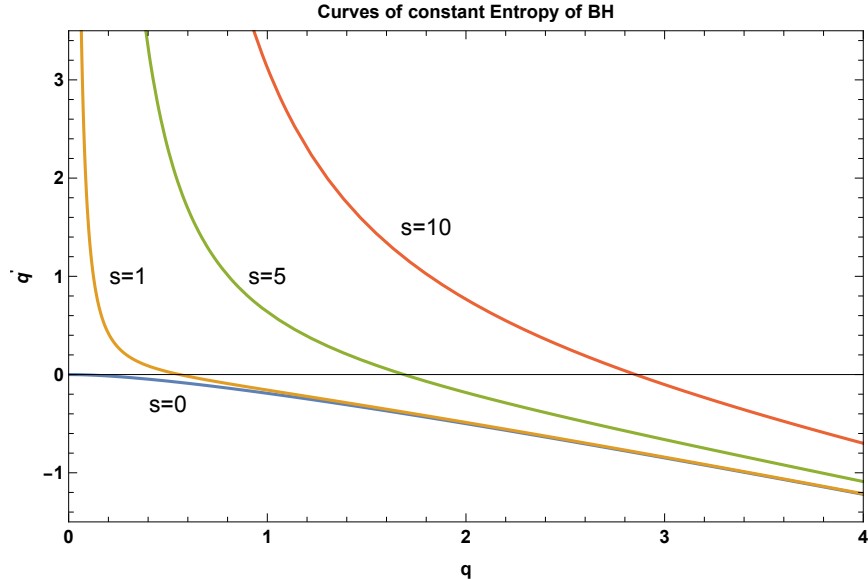

Figure 29: Contours of constant entropy on the BH sheet. Note that $j$ varies as we move along the contours. The above plot is a projection of the contours on the $q'-q$ plane.

with the charges $(q, q', j)$. Let $(\delta q, \delta q', \delta j)$ be the amount of charges that the black hole loses to the other component. For the resulting configuration to again lie on the black hole sheet, the three charges satisfy the condition that arises from varying the constraint (H.1),

$$\delta j = \frac{\partial j}{\partial q}\delta q + \frac{\partial j}{\partial q'}\delta q'. \tag{H.10}$$

Note that the variation $\delta j > 0$, if $\delta q > 0$ and $\delta q' > 0$, since the coefficients are always positive,

$$\frac{\partial j}{\partial q} = \sqrt{4q+1}-1, \tag{H.11}$$

$$\frac{\partial j}{\partial q'} = \frac{2\left(3q - \sqrt{4q+1} + q' + 1\right)}{\sqrt{4q+1}}, \tag{H.12}$$

for the charges on BH sheet (given in (H.5)). The variation of the black hole entropy (given by (H.3)) is given by,

$$-(\delta s)_{BH} = \frac{\partial s}{\partial q'}\delta q' + \frac{\partial s}{\partial q}\delta q + \frac{\partial s}{\partial j}\delta j \tag{H.13}$$

$$= \left(\frac{\partial s}{\partial j}\frac{\partial j}{\partial q} + \frac{\partial s}{\partial q}\right)\delta q + \left(\frac{\partial s}{\partial j}\frac{\partial j}{\partial q'} + \frac{\partial s}{\partial q'}\right)\delta q'. \tag{H.14}$$

The two quantities in the brackets evaluate to,

$$\frac{\partial s}{\partial j}\frac{\partial j}{\partial q} + \frac{\partial s}{\partial q} = \frac{2\pi\left(\left(\sqrt{4q+1}-1\right)(q'+1) + q\left(\sqrt{4q+1}-3\right)\right)}{\sqrt{2q+\frac{1}{2}}\sqrt{2q^2 + q\left(-2\sqrt{4q+1}+4q'+4\right) - \left(\sqrt{4q+1}-1\right)(2q'+1)}}, \tag{H.15}$$

$$\frac{\partial s}{\partial j}\frac{\partial j}{\partial q'} + \frac{\partial s}{\partial q'} = -\frac{\pi\left(-2q+\sqrt{4q+1}-1\right)}{\sqrt{q^2 + q\left(-\sqrt{4q+1}+2q'+2\right) - \frac{1}{2}\left(\sqrt{4q+1}-1\right)(2q'+1)}}. \tag{H.16}$$

For black hole on the BH sheet, i.e. those that lie above the zero entropy curve (H.4), the above two quantities are always positive and therefore establishing the stability of vacuum black holes in the micro-canonical ensemble.

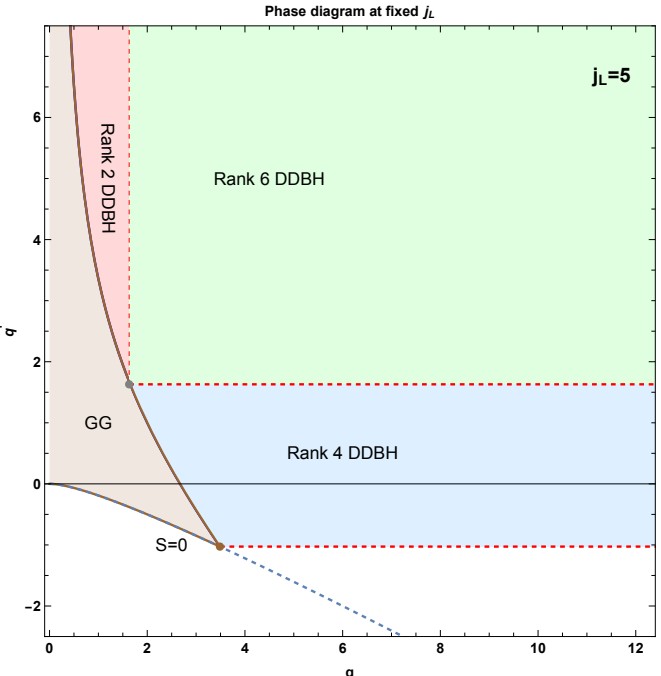

Figure 30: The constant $j$ cross section of the conjectured phase diagram of $\mathcal{N} = 4$ SYM in the subsector $q_1 = q_2 = q, q_3 = q'$ and $j_1 = j_2 = j$. The (brown shaded) region to the left of and below the black hole sheet (Brown curve) is dominated by a rank 4 grey galaxy phase. To the right of the black hole sheet, the charge space is divided into regions of rank 2, 4, and 6 DDBHs. All these three regions lie below the black hole sheet. The bottom red dashed line is the boundary of the EER at the given value of $j$.

## H.3 Phase diagram away from black hole sheet

In this section we will describe the phase diagram of $\mathcal{N} = 4$ SYM in the micro-canonical ensemble in the subsector of interest where we take two R-Symmetry charges and the two angular momenta to be equal. We will use the algorithm from §3.3 to identify the dominant supersymmetric phase at any given values of the charges $(q, q, q', j)$.

The black hole sheet ($\mathcal{P}(q, q')$) in this case is a 2-dimensional surface in the three dimensional charge space $(q, q', j)$. The first step of the algorithm is to identify the sign of $\mathcal{P}(q, q', j)$ - defined in (H.1). The function $\mathcal{P}$ splits the Extensive Entropy Region (EER) into two chambers depending on the sign of $\mathcal{P}$.

- **Case 1 $\mathcal{P}(q, q', j) = 0$:** If the charges satisfy the inequalities (H.5) then the dominant phase is the vacuum black hole phase. If the charges are outside the region specified by (H.5), then there is no supersymmetric black hole phase at those charges, i.e. the point of interest is outside the EER.

- **Case 2 $\mathcal{P}(q, q', j) > 0$:** If there exists a positive solution for $j_{gas}$ satisfying the following equation,

$$\mathcal{P}(q, q', j - j_{gas}) = 0, \tag{H.17}$$

and if the charges $q$ and $q'$ satisfy (H.5), then the rank 4 grey galaxy phase is dominant at those charges. The core black hole in such a phase will carry the charges $(q, q, q', j - j_{gas}, j - j_{gas})$ and the gas will carry the charges $(0, 0, 0, j_{gas}, j_{gas})$. This phase fills out the Brown region in Figure 30 (this is the region that lies to the "left" and also

"above" the black hole sheet). The core black hole that lies at the centre of any given grey galaxy solutions is obtained by starting at any given point in the brown region and moving directly downwards (going towards the negative $j$ axis, into the plane) until one hits the black hole sheet. The point of impact will give the charges of the seed black hole and the distance to impact will give the angular momentum in the gas. It is easy to verify that for all charges that satisfy (H.5), there will always be a solution for (H.17). Geometrically this is clear from Figure 30 since every point "above" (with a higher value of $j$) the black hole sheet can be obtained by forming rank 4 grey galaxies.

- **Case 3** $\mathcal{P}(q, q', j) < 0$: In this case the dominant phase will be a DDBH configuration. To identify which DDBH phase is dominant we will have to check the following sequentially,

   (i) If there exists a positive solution for $q_D$ and $q'_D$ satisfying,

$$\mathcal{P}(q - q_D, q' - q'_D, j) = 0, \tag{H.18}$$
$$q - q_D = q' - q'_D, \tag{H.19}$$

   then the dominant phase will be a rank 6 supersymmetric DDBH whose core black hole has charges $(q - q_D, q - q_D, q - q_D, j)$[167] and the three dual giants carry the charges $q_D, q_D$ and $q'_D$ respectively. In Figure 30, the green region contains the rank 6 DDBH solutions. The core black hole of the rank 6 DDBH is the Gutowski-Reall black hole carrying the same $j$ (grey dot in Figure 30). The charges carried by the dual giants are obtained by the distances along the $x$ and $y$ axis from the point of interest to the Gutowski-Reall black hole.

  (ii) If (H.18) and (H.19) have no allowed solutions, and if $q > q'$ and there exists a positive solution for $q_D$ satisfying,

$$\mathcal{P}(q - q_D, q', j) = 0. \tag{H.20}$$

   If $(q - q_D, q')$ do satisfy (H.5) then our charges lie outside the EER, and we predict that the supersymmetric entropy vanishes at these charges, at leading order in $N^2$. If these charges do obey (H.5), on the other hand, our charges lie in rank 4 super-symmetric DDBH phase. The core black hole carries charges $(q - q_D, q - q_D, q', j)$ and two dual giants carry $q_D$ charge each. The blue region in Figure 30 contains the rank 4 DDBH configurations.[168] The core black hole in this case is obtained by moving horizontally from the point of interest towards negative $q$ axis until we hit the black hole sheet (brown curve). The horizontal distance then gives the charge carried by each of the two dual giants.

 (iii) If $q' > q$ and there exists a positive solution for $q'_D$ satisfying,

$$\mathcal{P}(q, q' - q'_D, j) = 0, \tag{H.21}$$

   then the dominant phase at those charges will be a rank 2 DDBH with the core black hole carrying the charges $(q, q, q' - q'_D, j)$ and one dual giant in the $q_3$ plane carrying the charge $q'_D$. In Figure 30 this region is colored pink. The seed black hole is obtained by moving vertically along the negative $q'$ axis until we hit the black hole sheet (brown curve).

Note that it is convenient to identify various phases of DDBH in constant $j$ cross sections, since the dual giants do not carry any angular momentum.

---

[167]Or $(q' - q'_D, q' - q'_D, q' - q'_D, j)$.

[168]This conclusion from our algorithm can be independently verified as follows. It is clear from our analysis of entropy on constant $j$ slices in §H.2 (see Figure 28) that the maximum entropy configuration is obtained when dual giants carry only $q$ charge.

### H.4 Prediction for the superconformal index

#### H.4.1 The space of index lines

On this special cut of charges, index lines may be labeled by the following two indicial charges:

$$\alpha_1 = \frac{2q + q'}{3} + j\,, \tag{H.22}$$

$$\alpha_2 = q' - q\,. \tag{H.23}$$

Our indices lie within the indicial diamond (i.e. within the Bose Fermi cone and so are non-trivial) provided

$$-\frac{3}{2}\alpha_1 \le \alpha_2 \le 3\alpha_1\,. \tag{H.24}$$

When (H.24) are obeyed, our index lines lie on one of the red lines in Fig. 15[169] The first (left) inequality in (H.24) is saturated when $q + j = 0$ (this happens at the half BPS point $H'_3$ in Figs 15 (see also Figs 12 and 14). The second (right) inequality in (H.24) is saturated at the quarter BPS point at the other end of the red line in Fig. 15.

The charges of points on an index line labeled by $\alpha_1$ and $\alpha_2$ are given by

$$(q, q', j) = \left(\alpha_1 - \frac{\alpha_2}{3} + x, \alpha_1 + \frac{2\alpha_2}{3} + x, -x\right)\,, \tag{H.25}$$

where the parameter $x$ ranges over[170]

$$-\alpha_1 - \frac{\alpha_2}{6} \le x \le 0\,. \tag{H.26}$$

We see from (H.25) that as $x$ increases, the index line proceeds toward larger $q$, larger $q'$ but smaller $j$. In other words the index line proceeds "Northeast" and downwards in Fig. 27.

#### H.4.2 Phases along a shadow

As the index line ranges over the charges (H.25), (H.26) (at any given values of $\alpha_1$ and $\alpha_2$), its shadow traces out a curve on the black hole sheet. Every point on the shadow lies in one of four phases, namely

- A grey galaxy phase (for points on the index line that lie above the black hole sheet). In this case the shadow point is obtained by a downward vertical projection.

- The rank 6 DDBH phase (for points on the index line that lie below the black hole sheet and within the 90 degree wedge (parallel to the $q$ and $q'$ axes) of the Gutowski Reall black hole at the corresponding value of $j$). The shadow of such points are given by Gutowski Reall black holes at the corresponding value of $j$. The collection of all such shadows track the dotted red Gutowski Reall curve in Fig. 27)

- The rank 4 DDBH phase. This phase dominates when the index line lies below the black hole sheet, outside the wedge of the Gutowski Reall black hole (see previous point) and with $q > q'$. In this case the projection of the corresponding point leftwards along the $q$ axis (i.e. at constant $q'$ and constant $j$) onto the black hole sheet yields its shadow. If such a projection fails to intersect the black hole sheet, the corresponding indicial point lies outside the EER.

---

[169]They lie in a diamond of the form depicted in Fig. 15(a) when $\alpha_1 < \frac{1}{6}$ (small indicial charges) but on a diamond of the form depicted in Fig. 15(b) when $\alpha_1 > \frac{1}{6}$ (larger indicial charges).

[170]As explained around (53), the index line exits the Bose-Fermi cone beyond this range of $x$.

- The rank 2 DDBH phase. This phase dominates when the index line lies below the black hole sheet, outside the wedge of the Gutowski Reall black hole (see above) and when $q' > q$. In this case the projection of the corresponding point downwards along the $q'$ axis (i.e. at constant $q$ and constant $j$) onto the black hole sheet yields its shadow. If such a projection fails to intersect the black hole sheet, the corresponding indicial point lies outside the EER.

As we move along a given index line (allowing $x$ in (H.25) to increase) we sample several of the phases described above. Exactly which phases we sample depends on the precise values of $\alpha_1$ and $\alpha_2$. When $\alpha_2 > 0$ (so that $q' > q$) we always start out above the black hole sheet (see Fig. 31) and so in the grey galaxy phase, continue in this phase until we meet the black hole sheet, then emerge out into a rank 2, and then eventually a rank 6 DDBH phase. When $\alpha_2 < 0$, on the other hand, we have two possibilities. When $|\alpha_2|$ is not too large (more precisely, when the values of $\alpha_i$ are such that the index line intersects the black hole sheet), we once again start out above the black hole sheet (see Fig. 31) and so in the grey galaxy phase, continue in this phase until we meet the black hole sheet, then emerge out into a rank 4, and then eventually a rank 6 DDBH phase. When (recall negative) $\alpha_2$ has a modulus that is large enough so that the corresponding index line misses the black hole sheet all together, on the other hand, the smallest value of $x$ that lies in the EER already occurs in the rank 4 DDBH phase. As $x$ is increased, we move into the rank 6 DDBH phase, and then once again exit the EER. The shadow of such index lines never enters the grey galaxy phase.

We are chiefly interested in the maximum entropy point along the index line. This point turns out to be easy to determine, as we describe in the next subsection.

### H.4.3 Three distinct indicial phases

All index lines either intersect the black hole sheet (i.e. belong to the colored region of the indicial diamond) or do not. In the current example, those index lines that fail to intersect the black hole sheet do so because they pass by "below" the solid blue line in Fig. 31.

In Fig. 31 we have presented a top view of the black hole sheet. The brown and orange curves, respectively, represent the curves $\mu_3 = 0$ and $\mu_1 = \mu_2 = 0$ on the black hole sheet. Index lines are of three qualitative varieties

- An index line that intersects the black hole sheet in between the brown and the orange curve (i.e. at a "stable" black hole point) has maximum entropy (along the index line, as a function of $x$) at this intersection point.

- An index line that intersects the black hole sheet above the brown curve in Fig. 31 has a shadow that passes through a point on the curve $\mu_3 = 0$. In this case, the maximum indicial entropy (along the index line, as a function of $x$) occurs at this point. The index is then in the rank 2 DDBH phase.

- Finally, an index line that either intersects the black hole sheet below the orange curve in Fig. 31 - or fails to intersect the black hole sheet completely- always has a shadow that passes through a point on the curve $\mu_3 = 0$. Such an index line attains its maximum entropy (along the index line, as a function of $x$) at this point. The index is then in the rank 4 DDBH phase.

We now describe how the indicial entropy may be computed, in equations, in each of the three cases listed above. All through the rest of this subsection we assume that (H.24) holds, so that the indicial charges lie within the Bose Fermi cone.

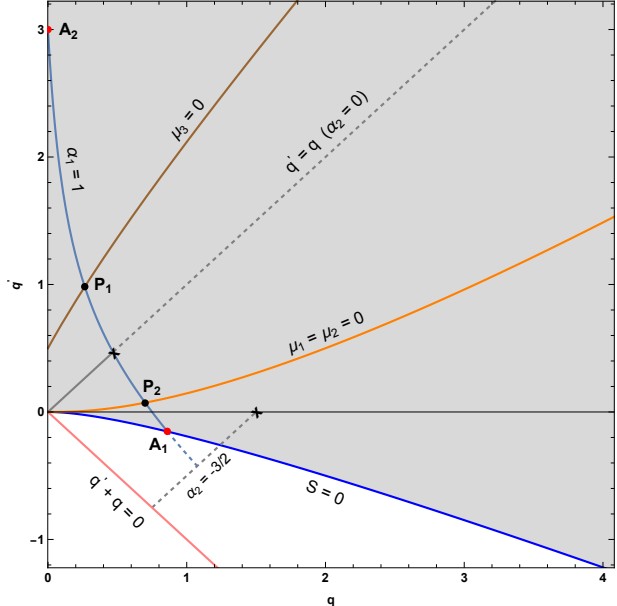

Figure 31: In this figure we study a set of index lines with fixed $\alpha_1 = 1$ and varying $\alpha_2$. The blue curve depicts the point of intersection of the corresponding index lines with the black hole sheet. At the red points $A_1$ and $A_2$, the index lines intersect the black hole sheet at points of zero entropy. The brown and orange curves depict the locations on the black hole sheet where $\mu_3 = 0$ and $\mu_1 = \mu_2 = 0$. When the index line intersects the black hole sheet at a point where all renormalized chemical potentials are positive (i.e. between the orange and brown curves) this point dominates the index line (the index is then in the "back hole phase". However index lines which intersect the black hole sheet either above $P_1$ or below $P_2$ (or fail to intersect the black hole sheet) are not dominated by the black hole phase. Index lines that intersect below $P_2$ are dominated by shadow points that lie along the orange curve (in the direction of smaller values of smaller values of $q$), i.e. by rank 4 DDBH phases. Index lines that intersect the black hole sheet above $P_1$ are dominated by shadow points along the brown curve, again towards smaller values of $q$, and lie in rank 2 DDBH phases.

*Black hole phase*

In the first case (the black hole phase) we solve the following equations for $q, q', j$ as functions of $\alpha_1$ and $\alpha_2$.

$$\frac{2q + q'}{3} + j = \alpha_1,$$
$$q' - q = \alpha_2, \tag{H.27}$$
$$j = \left(q + q' + \frac{1}{2}\right)\sqrt{4q + 1} - \left(2q + q' + \frac{1}{2}\right).$$

The solution to these equations yields the value of $(q, q', j)$ at the intersection of index line in question and black hole sheet. A solution to these equations is considered legal if the black hole charges lie on a legal point (positive entropy point) of the black hole sheet, that, moreover, lies between brown and orange curves in fig. 31. In equations one needs to check if $(\mu_1 = \mu_2)$ and $\mu_3$ are positive at the corresponding values of $(q, q', j)$, i.e. if

$$\frac{2q^2 + 4q - (2q + 1)\sqrt{4q + 1} + 1}{2q} < q' < \frac{1}{2}\left(2q + \sqrt{4q + 1}\right). \tag{H.28}$$

*Rank 2 DDBH phase*

When solution to (H.27) (intersection of the index line with the black hole curve) satisfies

$$q' > \frac{1}{2}\left(2q + \sqrt{4q+1}\right), \tag{H.29}$$

the index line intersects the black hole sheet above the brown curve in Fig. 31 above, and the index lies in the rank 2 DDBH phase (recall we assume that the indicial charges obey (H.24)).

In this case, the dominant phase along the index line is a rank 2 DDBH. The equations that determine the shadow black hole which sits at the core of the rank 2 DDBH are as follows:

$$q + j = \alpha_1 - \frac{1}{3}\alpha_2, \tag{H.30}$$

$$q' = \frac{1}{2}\left(2q + \sqrt{4q+1}\right), \tag{H.31}$$

$$j = q\left(2\sqrt{4q+1} - 1\right). \tag{H.32}$$

The first equation asserts that the indicial charge $q + j$ of the shadow black hole in a rank 2 phase equals indicial charge $q + j \equiv \alpha_1 - \frac{\alpha_2}{3}$ on the indicial line (as the gas in this phase carries only $q'$ charge). The second and third equations assert that the charges $(q, q', j)$ lie on the black hole sheet, and also have $\mu_3 = 0$. A solution to these equations is legal if the charges lie on the black hole sheet (i.e. if the black hole entropy is positive) and if $q'$ in the black hole (which we get by solving (H.30)) is smaller than the value of $q'$ - at the same values of $q$ and $j$ - in the indicial line. This is the case when

$$\alpha_2 + q - q' > 0. \tag{H.33}$$

This condition guarantees that gas in the rank 2 DDBH carries positive charge.

When (H.24) are obeyed, the equations (H.30) always have a unique legal solution. The indicial entropy is then obtained by plugging the charges of this solution into (H.3).

*Rank 4 DDBH phase*

If the index line either intersects the black hole sheet below the orange curve, or if the index line does not intersect the black hole sheet at all (and $\alpha_2 < 0$), we are in the rank 4 DDBH phase. The first case occurs when (H.27) has a legal (positive entropy) solution that satisfies the inequality:

$$q' < \frac{2q^2 + 4q - (2q+1)\sqrt{4q+1} + 1}{2q}. \tag{H.34}$$

The second case occurs when (H.27) does not have a positive entropy solution (but the indicial charges lie within the Bose Fermi cone, i.e. obey (H.24), as we assume throughout this subsection).

In either case the dominant phase along the index line is a rank 4 DDBH. The equations that determine the shadow black hole which sits at the core of the rank 4 DDBH are as follows:

$$q' + j = \alpha_1 + \frac{2}{3}\alpha_2, \tag{H.35}$$

$$q' = \frac{2q^2 + 4q - (2q+1)\sqrt{4q+1} + 1}{2q}, \tag{H.36}$$

$$j = \frac{\sqrt{4q+1} - 1}{q} + q\left(2\sqrt{4q+1} - 7\right) + \frac{1}{2}\left(7\sqrt{4q+1} - 11\right). \tag{H.37}$$

The first equation asserts that the indicial charge $q' + j$ of the shadow black hole in a rank 2 phase equals indicial charge $q' + j \equiv \alpha_1 + \frac{2\alpha_2}{3}$ on the indicial line (as the gas in this phase

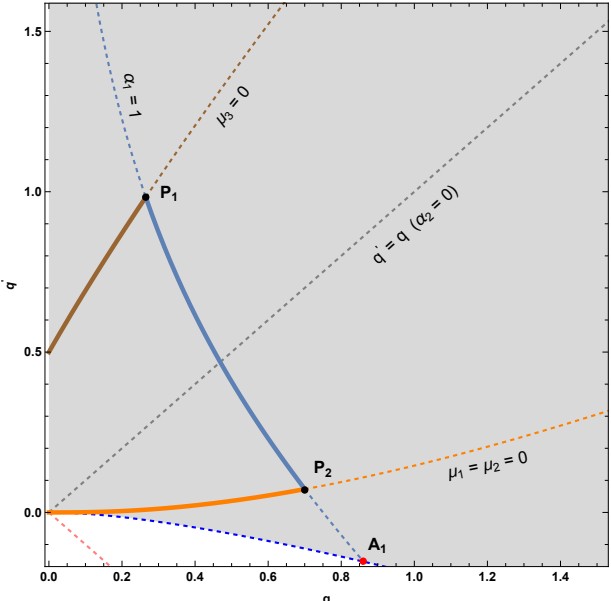

Figure 32: The path followed by the charges of the core black hole of the dominant configuration on the index lines as we vary $\alpha_2$.

carries only $q$ charge). The second and third equations assert that the charges $(q, q', j)$ lie on the black hole sheet, and also have $\mu_1 = \mu_2 = 0$. A solution to these equations is legal if the charges lie on the black hole sheet (i.e. if the black hole entropy is positive) and if $q$ in the black hole (which we get by solving (H.30)) is smaller than the the value of $q$ (at the black hole values of $q'$ and $j$, i.e. at the values of $q'$ and $j$ on the solution fo (H.30). This condition is met when

$$q' - \alpha_2 - q > 0 \,. \tag{H.38}$$

The indicial entropy is then given by plugging the legal solution $(q, q', j)$ of (H.35) (such a solution always exists when (H.24) is obeyed) into (H.3).

### H.4.4 Indicial phase diagram as a function of $\alpha_2$ at fixed $\alpha_1$

Very roughly, the indicial charge $\alpha_1$ can be thought of as a measure of the overall scale of indicial charges; the base $R^+$ coordinate, in a presentation in which the space if indices is viewed as a fibration of the indicial diamond over $R^+$. On the other hand, the coordinate $\alpha_2$ measures how skew the charge distribution is between $q$ and $q'$. It may be thought of as a coordinate along the red lines in given indicial diamonds Fig. 15. In other words, $\alpha_1$ may be thought of as a "which diamond" coordinate, while $\alpha_2$ may be thought of as a "where on red line of given diamond" coordinate. In this section we will study the phase diagram as a function of $\alpha_2$ at fixed $\alpha_1$.

*The case $\alpha_1 > \frac{1}{6}$.*

If we fix $\alpha_1 = 1$ and vary $\alpha_2$, the various resultant index lines intersect the black hole sheet along the light blue curve depicted in Fig 31. As we have explained in the previous section, this intersection point maximizes entropy when it lies between the brown and orange curves. When, on the other hand, the intersection happens to lie above the brown curve, the saddle point changes to a rank 2 DDBH phase, and the indicial entropy equals the entropy of a black hole at some point on the brown curve (see Fig. 32). When the intersection happens below the orange curve - or when the indicial line fails to intersect the black hole sheet altogether- the saddle point switches to a rank 4 DDBH, and the indicial entropy equals the entropy of

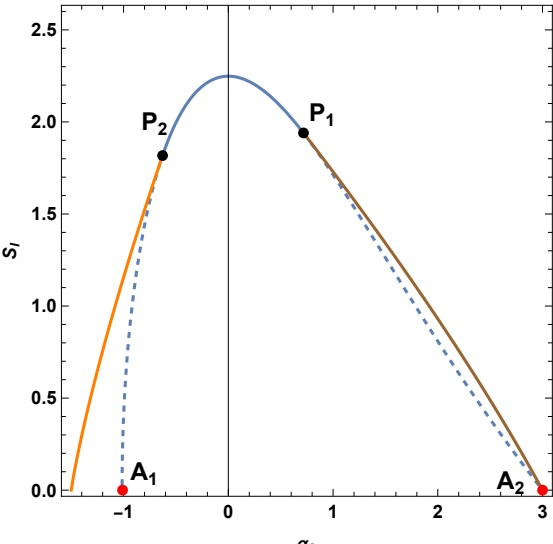

Figure 33: The indicial entropy is computed with $\alpha_1 = 1$ while varying $\alpha_2$. The blue and orange solid lines represent the indicial entropy predicted by H.4.3. The blue line (solid and dashed) corresponds to the black hole entropy, determined at the intersection of the black hole sheet and an index line. The orange line depicts the entropy of rank 4 DDBH solutions with $\mu_1 = \mu_2 = 0$, and the brown line shows the entropy of rank 2 DDBH solutions with $\mu_3 = 0$.

a black holes at some point on the orange curve (see Fig. 32). This general structure is also visible in Fig 15(b). The point $H_3'$ in that figure represents the largest value of $\alpha_2$. Moving up the red curve corresponds to decreasing $\alpha_2$. It is apparent from Fig. 15(b) that, as $\alpha_2$ is decreased from its maximum value, we move from a rank 2 DDBH to a black hole to a rank 4 DDBH phase.

In Fig 33 below we present a quantitative plot of the Indicial phase diagram as a function of $\alpha_2$ at $\alpha_1 = 1$. See also the third and fourth images in Fig. 34 for quantitative plots of the phase diagram at larger values of $\alpha_1$.

*The case $\alpha_1 < \frac{1}{6}$.*

The indicial phase diagrams at fixed $\alpha_1$ change qualitatively when $\alpha_1 < \frac{1}{6}$. This point may be understood as follows. As $\alpha_1$ is decreased, the light blue curve in Fig. 31 shifts to the left, and the point $A_2$ comes lower and lower. At the critical value $\alpha_1 = \frac{1}{6}$,[171] $A_2$ meets the intersection of brown curve and the $q'$ axis. At all lower values of $\alpha_1$, the blue curve lies everywhere "below" (i.e at smaller $q'$ values than) the brown curve. At such values of $\alpha_1$, the rank 2 DDBH phase (the solid brown curve in Fig. 32) is simply absent. At these values of $\alpha_1$, the indicial phase diagram starts out in the black hole phase (at the largest allowed values of $\alpha_2$) and then makes a single phase transition to the rank 4 DDBH phase as $\alpha_2$ is lowered. This is also clearly visible in Fig. 15(a).

In the first of Fig 34 we have presented a quantitative plot of the indicial entropy as a function of $\alpha_2$ at fixed $\alpha_1 = 0.15 < \frac{1}{6}$. Notice that this phase diagram has only two phases.

---

[171]The critical value is obtained by substituting the charges $q' = 1/2, q = j = 0$ in (H.27). This critical point was first observed in [74] in the case of non-supersymmetric black holes.

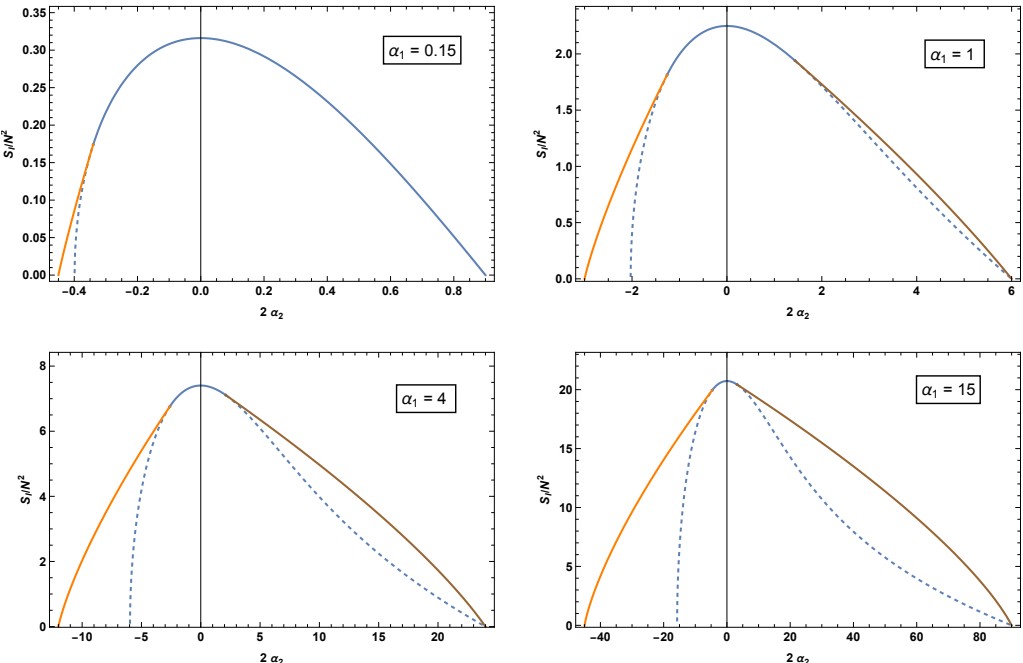

Figure 34: In each plot, the indicial entropy $S_I$ is calculated with $\alpha_1 = \frac{2q+q'}{3} + j_L$ fixed at $\alpha_1 = 0.15, 1, 4, 15$, respectively, while varying $2(q'-q) = 2\alpha_2$. The deviation between the black hole entropy at the intersection (blue solid and dotted curves) and the indicial entropy from DDBH (orange and brown curves) increases as $\alpha_1$ and $\alpha_2$ become large.

# I Legendre transforming the index

In this paper, we have (in particular) analyzed the superconformal index in the microcanonical ensemble. In principle, the constructions of this paper also predict new phases of the canonical superconformal index, i.e. the superconformal index defined by the formulae (1), (2). In this appendix we present a first analysis (making some assumptions) for the predictions made for the canonical index (1) from the new (microcanonical) phases constructed in this paper.

This Appendix is organized as follows. We first review the general formalism for going between the canonical and microcanonical indices. We then first apply this general formalism to the standard "black hole phase" (this exercise was first performed in [7–9,23]) and recover standard known results. Finally, we apply the same formalism to the new phases predicted in this paper, and find a prediction for the indicial chemical potentials of the relevant phases.

## I.1 Transforming between ensembles

In (6) in the introduction, we have presented an equation for the canonical index in terms of the microcanonical quantity $n_I(Z_i')$. Recall $Z_i'$ is some (any) representative charge along a given index line. The quantity $n_I(Z_i')$ is class valued, which means that it depends on which index line the charge $Z_i'$ lies in, but not on which representative is chosen to label this line.[172]

---

[172]Indeed, the factor $(-1)^{2(Z_1'+Z_2'+Z_3'-Z_4'-Z_5')}$ in (7) was inserted to ensure that $n_I(Z_i')$ (defined in (7)) is "class valued". Infact $n_I(Z_i')$ it is defined to ensure that all bosons contribute as $+1$ and all fermions contribute as $-1$, to the sum in (6). The factor that multiplies $n_I(Z_i')$ in (6) is also class valued: if the representative charge $Z_i'$ changes to the next value along the same indicial line, both factors, i.e. the $(-1)^{\cdots}$ and the $e^{\cdots}$ pick up a minus sign, so their product is left invariant.

We define the indicial entropy via the formula

$$n_I(Z_i') = e^{S_I(Z_i')}. \tag{I.1}$$

Note that $n_I(Z_i')$ (and so $e^{S_I(Z_i')}$) can be either positive or negative.[173] With this definition, it follows from (6) that

$$\begin{aligned} \mathcal{I}_W(v_i) &= \sum_{Z_i' \in \Lambda_I} (-1)^{F(Z_i')} e^{S_I(Z_i')} e^{-\sum_{i=1}^5 v_i Z_i'} \\ &= \sum_{Z_i' \in \Lambda_I} e^{S_I(Z_i') - \sum_{i=1}^5 (v_i Z_i') + \pi i F(Z_i')}, \end{aligned} \tag{I.2}$$

where we have defined

$$(-1)^{F(Z_i')} = (-1)^{2(Z_1' + Z_2' + Z_3' - Z_4' - Z_5')} \tag{I.3}$$

(here $F$ stands for Fermion number) and the summation is performed over the set of index lines (i.e. over one representative charge on each index line; the indicial charge lattice $\Lambda_I$ is defined in §2.4).

The formula (I.2) determines canonical index in terms of the microcanonical indicial data (i.e. the indicial entropy). In the saddle point approximation the sum on the RHS of (I.2) can, presumably be replaced by a maxmimzation, yielding,

$$\mathcal{I}_W(v_i) = \text{Ext}_{Z_i} \left( e^{S_I(Z_i') - \sum_{i=1}^5 (v_i Z_i') + \pi i F(Z_i')} \right). \tag{I.4}$$

We now turn to the question of inverting these formulae, i.e. of evauating the indicial entropy given the superconformal index. As we have explained in §2.6, the superconformal index (the quantity on the LHS of (I.2)) is left invariant if the chemical potential vectors $v_i$ are shifted by $2\pi i$ times an element of $\Lambda^*$ (the dual charge lattice). In explicit gravitational computations of the superconformal index at large $N$, this periodicity is realized as follows. An infinite number of gravitational saddles [33] - one for each vector in $\Lambda^*$ - contribute to the gravitational path integral that computes $\mathcal{I}_W(v_i)$. The action for each of these saddles is a smooth function of $v_i$ that is not left invariant by shifts by $2\pi i v$ (with $v \in \Lambda^*$). The full gravitational result is given by summing over each of these saddle point contributions, and takes the form

$$\mathcal{I}_W(v_i) = \sum_{\lambda \in \Lambda^*} e^{F_W(v_i + \lambda_i)}. \tag{I.5}$$

Consequently, (I.2) can be rewritten as[174]

$$\mathcal{I}_W(v_i) = \sum_{\lambda \in \Lambda^*} e^{F_W(v_i + \lambda_i)} = \sum_{Z_i' \in \Lambda_I} e^{S_I(Z_i') - \sum_{i=1}^5 (v_i Z_i') + \pi i F(Z_i')}. \tag{I.6}$$

In order to invert (I.6), we multiply both sides by $e^{v_i \tilde{Z}_i'}$ (here $\tilde{Z}_i$ is a charge vector in the charge lattice $\Lambda$) and integrate $v_i$ over a contour $C$ such that

- $v.t_I = \pi i n$ for some choice of an odd integer $n$ and, moreover, each of the 5 vector components, $v_i$ is imaginary.

- The integration contour $C$ is obtained by setting $v = v_0 + \mu$ and integrating $\mu$ over a unit cell of dual lattice $\Lambda_I^*$.[175]

---

[173]Note that this equation unambiguously defines the exponential of the entropy function, but leaves the entropy function itself ambiguous upto a shift of integer multiples of $2\pi i$.

[174]Note, of course, that (I.6) applies only when the chemical potentials obey (2) for some choice of the odd integer $n$.

[175]Recall that two values of $\mu$ that differ by a shift in the dual indicial lattice represent the same physical index.

On the RHS, the integral yields a $\delta$ function that sets $\tilde{Z}'_i \equiv Z'_i$ (i.e. $Z'_i$ and $\tilde{Z}'_i$ are equal as vectors in the indicial charge lattice $\Lambda_I$), and so we conclude that

$$e^{S_I(Z'_i)} = (-1)^{F(Z'_i)} \int_C d\nu \, e^{\nu_i Z'_i} \left( \sum_{\lambda \in \Lambda^*} e^{F_W(\nu_i + \lambda_i)} \right). \tag{I.7}$$

The contour integral $C$ plus the sum over $\Lambda^*$ in (I.7) combine together into the sum over integrals on the contours $\tilde{C}_n$ (for all odd integers $n$). The contour $\tilde{C}_n$ sets $\nu = \nu_0(n) + \mu$ where $\nu_0(n)$ is any vector that obeys the condition $\nu_0(n).t_I = \pi i n$ (where $n$ is any odd integer) and $\mu$ is integrated over all imaginary vectors (vectors with all components imaginary) that obey $\mu.t_I = 0$. We conclude

$$e^{S_I(Z'_i)} = (-1)^{F(Z'_i)} \left( \sum_{n=\text{odd integers}} \int_{\tilde{C}_n} d\nu \, e^{F_W(\nu_i) + \sum_i \nu_i Z'_i} \right). \tag{I.8}$$

In the saddle point approximation, (I.8) simplifies to

$$
\begin{aligned}
e^{S(Z'_i)} &\approx (-1)^{F(Z'_i)} \sum_{n=-\infty}^{\infty} \text{Ext}_{\nu.t_I = \pi(2n+1)i} \left( e^{\left( F_W(\nu_i) + \nu_i Z'_i \right)} \right) \\
&\approx (-1)^{F(Z'_i)} \text{Max}_{n \in \mathbb{Z}} \left[ \text{Ext}_{\nu.t_I = \pi(2n+1)i} \left( e^{\left( F_W(\nu_i) + \nu_i Z'_i \right)} \right) \right].
\end{aligned}
\tag{I.9}
$$

Note that the shift $Z' \to Z' + t_I$ leaves the RHS of (I.9) unchanged, in agreement with class valued nature of the entropy on the LHS.

## I.2 The black hole phase

As explained in the main text, the analysis of this paper predicts that the superconformal index lies in one of 9 distinct phases. One of these - the black hole phase - has been extensively analyzed over the last six years or so. As reviewed around (40), in this case the function $F_W(\nu_i)$ is given by

$$F_W(\nu_i) = \frac{N^2 \mu_1 \mu_2 \mu_3}{2\omega_1 \omega_2}. \tag{I.10}$$

When $F_W$ takes the form in (I.10), the maximization over $n$ (in (I.9)) always occurs at either $n = \pm 1$ (these two values yield complex conjugates of the entropy). Performing the extremization (I.9) at $n = \pm 1$ yields an explicit but ugly formula for $\nu_i$ as a function of $Z'_i$, and also of $S_{BH}(Z'_i)$.[176] At real (and integer) values of $Z'_i$ of physical interest, $\nu_i$ and $S_{BH}(Z'_i)$ are both complex numbers.[177]

We emphasize that the Legendre transformation procedure, described above, determines the indicial entropy, but gives absolutely no information about which charge (on a given indicial line) dominates the summation along the given indicial line.

---

[176]Explicitly, the entropy $S_{BH}$ as a function of charges is computed in [8]; $S_{BH}$ in this paper equals $-2\pi i f$ in [8], and $f$ is defined in Eq. 2.76 of [8].

[177]The physical meaning of the imaginary part of $S_{BH}(Z'_i)$ was explained in [36]; the number indicial states equals $e^{\text{Re}(S_{BH})} \cos(\text{Im}(S_{BH} + \alpha))$ where $\alpha$ is a function of $\zeta_i = \frac{Z_i}{N^2}$ and so is effectively constant when charges vary by order unity. Note that averaging this number of states over any window of charges (with an interval large compared to unity but small compared to $N^2$) and then taking the log, yields entropy $\text{Re}(S_{BH})$ up to subleading corrections in $\frac{1}{N^2}$.

### I.3 Grey galaxy or DDBH phases

Consider, to start with, a rank 2 grey galaxy phase, whose central black hole has $Re(\omega_1) = 0$. All rank two grey galaxies, with the same central black hole - but different amounts of $J_1$ from the gas - carry the same complex entropy. Two different index lines, whose maxima occur on the grey galaxy phase with the same central black hole (but different $J_1$ in gas) therefore carry the same entropy. As the charges of these two solutions differ only by $J_1$ (and as the chemical potential is the derivative of the entropy w.r.t. charge, as can be seen from (I.4)), we conclude that a rank 2 grey galaxy indicial phase carries indicial chemical potentials with $\omega_1 = 0$.

As we have already mentioned, the black hole at the centre of a rank 2 grey galaxy phase has $Re(\omega_1) = 0$ but does not, in general, have $Im(\omega_1) = 0$. It follows that the pure black hole at the centre of a grey galaxy can be assigned two different chemical potentials; one when approached as the limit of the pure black hole phase (seen this way $\omega_1$ is purely imaginary), and the second, when approached as the limit of a grey galaxy (seen this way $\omega_1$ is simply zero). The fact that the chemical potential is discontinuous along the phase transition boundary is a simple consequence of the fact (argued for in the main text in this paper), that the microcanonical indicial entropy, as a function of charges, has phase transitions in the large $N$ limit. While entropies are continuous across a (microcanonical) phase transition, their normal derivatives (across a phase transition wall) are not. As a consequence, chemical potentials jump across phase transitions in such situations.

The fact that $\omega_1$ vanishes in the appropriate rank 2 grey galaxy phase also explains why $J_1$ condenses in this phase, in a manner similar to the analysis presented in [25]. This discussion suggests that the canonical grey galaxy phase represents a completely distinct saddle point - both in the unitary matrix integral as well as from a Euclidean gravitational viewpoint- from the much studied black hole saddle point. The independent determination of these saddle points (both in the matrix model as well as in gravity) is a key outstanding question; one that we leave to future work.

In the discussion above, we have focussed on a particular rank 2 grey galaxy phase. The generalization to other rank two phases (grey galaxy as well as DDBH) is straightforward: for instance the rank 2 DDBH phase that has a central black hole with $Re(\mu_1) = 0$ simply has $\mu_1 = 0$. The generalization to rank 4 phases is also straightforward. For instance, the rank 4 DDBH phase whose central black hole has $Re(\mu_1) = Re(\mu_2) = 0$, itself turns out to have $\mu_1 = \mu_2 = 0$.

In addition to the general points mentioned above, one could go ahead and compute the explicit expressions for the free energy $F_W(\nu_i)$ for each of these phases. This exercise is straightforward in principle. In any of these phases we have a formula that determines the charges of the central black hole, $Z^{BH}$ in terms of the indicial charges $Z'$ by some function $Z^{BH}(Z')$. The canonical index is then given by the formula

$$\text{Ext}_{Z_i'}\left((-1)^{F(Z_i')}e^{-\nu_i Z_i'}e^{S_{BH}((Z^{BH}(Z_i')))}\right) = e^{F_W^{BH}(\nu_i)}. \tag{I.11}$$

The complicated nature of the function $Z^{BH}(Z')$ appears to make actually carrying out the extremization in (I.11) a messy proposition, one that we leave to future work.

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
