# Peer review of "Supersymmetric Grey Galaxies, Dual Dressed Black Holes and the Superconformal Index"

_SciPost Physics, doi:SciPost Phys. 19, 072 (2025)_

## Round 1 · Referee Report · Anonymous (Referee 1) · 2025-5-11

Report

In this manuscript, the authors performed a systematic analysis of the microcanonical phase diagram of the 1/16-th BPS states in the 4d N=4 Super Yang Mills theory. Motivated by recent developments in constructing black hole solutions dressed by rotating objects such as gravitons (“grey galaxies”) and charged objects such as D-branes (“dual dressed black holes”), as well as the explicit construction of their field theory counterparts, the authors conjectured the existence of similarly dressed supersymmetric black holes solutions. Their result presents a dramatic update to the phase diagram of the 1/16-th BPS sector – instead of having only extensive number of states concentrated in a 4-dimensional sheet of 5 possible charges, the dressed black holes cover a 5-dimensional “extensive entropy region”. After carefully analyzing how different regions of the vacuum black hole sheet should be dressed differently, the authors present a method to determine the dominant entropy solution for arbitrary given charges.

Importantly, the outcome of their analysis, under an assumption of “unobstructed saddle conjecture”, suggests new predictions that can be tested against the field theory index. The authors performed such tests numerically in some cases and found moderate support.

The analysis in this manuscript is a remarkable tour de force and represents the state-of-the-art understanding of black hole states in holographic field theories. It opens many possible new directions and questions for future exploration.

Nonetheless, there are some questions/suggestions that I hope the authors can clarify or consider before I recommend this manuscript for publication.

1) In analyzing what are the possible dressings one can add to the black hole, the authors discussed the distinction of the “allowed region” (the possible charges of all operators built out of BPS letters) and the “Bosonic cone” (the possible charges one can arrive at by only including bosonic letters). The authors argued in Appendix B.5 that for charges much greater than one, the supersymmetric gas lie in a region that asymptotes to the bosonic cone. If I understand correctly, the argument uses two facts: (1) fermionic single trace operators can be excited at most once; (2) single trace graviton gas operators contain at most three fermionic letters. However, in Table 2, one sees that there exist bosonic single trace operators that nonetheless contain two fermionic letters. It seems such operators can in principle be excited arbitrary many times (at least when it is somewhat small compared to N^2), which seems to lead to states which lie outside the bosonic cone. One might argue that these will not be the typical gas states, but it is unclear one can invoke entropic arguments since the entropy of the gas modes is negligible in the final solution anyway. Could the authors clarify whether this would affect the analysis?
As an aside, the second paragraph of Appendix B.5 seems to contain several typos, such as “n = 1, 2, …, n”, and the two summations over m are written the same but gives different answers.

2) It seems to me that the “unobstructed saddle conjecture” has relatively less support. In principle, given the entropy S(x) of the solutions the authors found along the direction summed over by the index (parametrized by x), one could perform a simple-minded saddle point analysis of \int e^{S(x)} e^{\pi i n x} with n being an odd integer and check explicitly whether the saddle point contribution agrees with S_{max} (x). It could be that the difference is small enough such that it has not been picked up in the numerical tests. Are there some simple cases where one could work out the expression of S(x) such that this test could in principle be performed?

3) In Figure 17, one observes several bumps in the log plot of the index. Since this is a log plot, such bumps might be viewed as signaling first order competitions between several different saddles, reminiscent of the discussion in https://arxiv.org/abs/2206.15357. Do the authors have any speculations on this feature?

4) Several periods are missing, such as in footnotes 11 and 13 and immediately before footnote 27.

Recommendation

Ask for minor revision

  • validity: -
  • significance: -
  • originality: -
  • clarity: -
  • formatting: -
  • grammar: -

Author:  Diksha Jain  on 2025-07-11  [id 5631]

(in reply to Report 1 on 2025-05-11)

We thank the referee for their insightful questions that have helped us improve the paper. Below, we address each of their questions:

1) The reason why the charges of gas modes lie almost within the Bosonic Cone at high energies is twofold a) All bosonic single trace gas operators (including the operators with two fermionic letters flagged by the referee) turn out to have charges that lie strictly within the Bosonic cone. b) While fermionic single trace operators carry charges that violate the Bosonic cone condition, the fractional violation of this condition - due to such operators - goes to zero in the large charge limit, for the reasons summarized by the referee.

In our original writing of section 2.8 we had omitted to emphasize point a) above. We have modified section 2.8 and Appendix B.5 to emphasize this point. We have also improved Table 2 (the improved table now carries more fine grained information about the charges of all single trace operators; the final column in the table can be used to verify our claim that the charges of all bosonic single trace operators (including the operators that are bilinear in fermions) lie in the bosonic cone, in the manner explained in the new Footnote 43). We have also taken the opportunity to correct an error in the precise estimate of Appendix B.6: the violation of the Bosonic cone condition scales like $1/E^\frac{1}{4}$, not $1/E^\frac{1}{5}$ as we had incorrectly originally claimed.

We once again thank the referee for their question. We hope that our explanation - and the revisions described above - satisfactorily address the referee's question.

2) We thank the referee for flagging this point, and for their suggestions. In response to the referees suggestions, we have added a new Appendix - Appendix A - to the paper. In this Appendix we first study the analogue of the unobstructed saddle conjecture in two simple toy models and find that it fails in the first but holds in the second one. The key qualitative feature present in the second toy model (where the conjecture holds), but absent in the first toy model (where it fails) is a degree of randomness in the precise number of super symmetric states as a function of location on the index line. In this Appendix, we explain that we expect our physical situation to have this feature and hence would obey the unobstructed saddle conjecture. We hope the referee finds this discussion satisfactory.

Even with the new discussion of Appendix A (which, in our opinion, makes a plausible case for the unobstructed saddle conjecture), we agree that the evidence in favour of this conjecture is less persuasive than the evidence for the dressed concentration conjecture. We have emphasized this point, first by adding the words (and more tentative) in the description of this conjecture in the abstract. We have also emphasized the limited a priori evidence for the unobstructed saddle conjectue in the newly written text under the statement of the conjecture (in subsection 1.3 of the introduction).

We thank the referee again for their prodding on this point, and hope that they find our modifications satisfactory.

3) We thank the referee for this suggestion. We have added a new subsection 5.3.1, which gives a quantitative explanation for the bumps that the referee has flagged. In brief they are a smoking gun of the oscillations that follow from the imaginary part of the entropy of the black hole. The explanation is essentially identical to that provided - for a similar phenomenon - in paper [36]. We hope this addresses the referees question on this point.

4) We thank the referee for pointing this out, and have accounted for these points.

We hope that the above points address the referees concerns and once again thank the referee for helping us improve our draft.

---

## Round 1 · Referee Report · Anonymous (Referee 2) · 2025-5-14

Report

Building on recent developments by some of the authors and their collaborators—concerning the existence of novel gravitational configurations obtained by dressing known black hole solutions (the core) with either gravitons or dual giants—this paper presents two intriguing conjectures related to the analysis of the superconformal index of 4d $\mathcal{N}=4$ SYM:

  • The dressed concentration conjecture.
  • The unobstructed saddle conjecture.

In my opinion, and given their potential relevance as well, more refined tests of these conjectures are necessary and of interest. In particular, it is unclear to me how backreaction from the dressing - if present, may affect the final form of the entropy, or of the free energy of the newly conjectured solutions, and consequently it is also unclear how this may affect the final form of the conjectures and of some of the results presented in this paper e.g. the phase diagrams. Leaving that seemingly weak point aside, the paper has a lot of value.

Before recommending it for publication, however, there is one issue I would like the authors to address.

In the discussion section it is suggested that the supersymmetric free energy (not just the entropy) of the dressed black hole configuration is equal to the one of its BPS black hole core \begin{equation} \mathcal{F}=\mathcal{F}[\omega] \end{equation} i.e. the well-known expression quoted in their equation (38).

Precisely, in their words:

a) "The resultant phases all include central black holes whose chemical potentials (approximately) saturate one of the inequalities listed above. These facts suggest that the grand canonical index is indeed given by the usual formula (38) at chemical potentials that obey (68),..."

It is important to note that this suggestion is in clash with the dressed concentration conjecture. Let me explain.

The dressed concentration conjecture states that:

In the microcanonical ensemble the entropy (not the free energy) computed by the superconformal index equals the entropy of the core BPS black holes, i.e. the known entropy formula for the known BPS black holes in $AdS_5\times S_5\,$, \begin{equation} S_{BPS}=S_{BPS}[J_{BH}], \end{equation} evaluated at the core indicial charges $J_{BH}=J_{core}\equiv J_{c}\,$, not at the total indicial charges of the system, $J$. By indicial charges -- borrowing the author's notation, I mean charges that commute with the two supercharges that organize the cohomologies that the superconformal index counts.

The important relation \begin{equation} J_{c}=J_{c}[J] \end{equation} is fixed by the procedure outlined in the dressed concentration conjecture.

As will be explained next, the fact that $J_c[J] \neq J$ implies that the onshell action of the newly proposed configurations should not match the well-known BPS free-energy $\mathcal{F}$ reported in (38). Let me explain this, schematically.

Allow me to start from the following well-known extremization principle (being careless about numerical factors) \begin{equation} \underset{\omega}{ext}\Bigg(\mathcal{F}[\omega]+ \omega J\Bigg)\, =\, S_{BPS}[J] + \text{i} Q[J] \end{equation} and its inverse transform \begin{equation} \underset{J}{ext}\Bigg(S_{BPS}[J] + \text{i} Q[J] -\omega J\Bigg)\, =\, \mathcal{F}[\omega ] \end{equation} where $\mathcal{F}\,$, again, stands for the expression reported in equation (38).

The function of indicial charges $Q=Q[J]$ is defined by the non-linear constraint \begin{equation} \mathcal{P}(J,Q[J])=0 \end{equation} (which it is the same polynomial relation $\mathcal{P}$ or $P$ proposed by the authors as a test function in their diagram in page 43) and $Q$ is the average of the R-charge operator $\widehat{Q}\,$. This is the operator used to represent $(-1)^F$ as $e^{\pi \text{i} \widehat{Q}}\,$. This operator is not an indicial charge (i.e., it does not commute with the supercharge that organizes the cohomology of 1/16 BPS states of interest).

Now the dressed concentration conjecture seems to indicate the substitution of the right-hand side of the first extremization above as follows \begin{equation} S_{BPS}[J] + \text{i} Q[J] \rightarrow S_{BPS}[J_c] + \text{i} Q[J_c] \end{equation} were again $J_{c}= J_{c}[J]$ and \begin{equation} \mathcal{P}(J_c,Q[J_c])=0\,. \end{equation}

At last, going back to grandcanonical ensemble, one obtains

$\underset{J}{ext}\Bigg(S_{BPS}[J_c] + \text{i} Q[J_c] -\omega J\Bigg)=\mathcal{F}_{new} [\omega] = $ $\frac{dJ_c}{dJ} \times \mathcal{F}[\omega_{dressed}]\neq\mathcal{F}[\omega]$

where the dressed chemical potential $\omega_{dressed}$ is defined as \begin{equation} \omega_{dressed} = \omega \frac{dJ}{dJ_c} \end{equation} Note that the extremization procedure fixes $J$ and $\frac{dJ_c}{dJ}$ as functions of the meaningful $\omega\,$. These latter relations can be found, for instance, in perturbations of $\omega$, say about 0 or 1, assuming that one knows $\frac{dJ_c}{dJ}$ for all $J$, or at least in some asymptotic region.

Of course, I do not think it is necessary to perform the precise version of this analysis in this paper - although that would give more value to the paper; my point here is to illustrate as clearly as possible that the observation a) - is in conflict with the conjectured approach.

This is important to highlight because, should a) be correct, then none of the newly conjectured black hole solutions would correspond to new saddles of the index in grandcanonical ensemble -- well within the region (68), and that would be very strange. All known saddle point approaches to this problem have convincingly shown that every known large-$N$ saddle points of the index (in grandcanonical ensemble) has a dual complex gravitational configuration -- with differing on-shell actions, well within the region (68). I expect these solutions -- if they exist, to correspond to new large-$N$ saddles of the superconformal index in grandcanonical ensemble.

Recommendation

Ask for major revision

  • validity: -
  • significance: -
  • originality: -
  • clarity: -
  • formatting: -
  • grammar: -

Author:  Diksha Jain  on 2025-07-11  [id 5634]

(in reply to Report 2 on 2025-05-14)

We thank the referee for their thoughtful and perceptive comments, questions and suggestions.

1) "In my opinion, and given their potential relevance as well, more refined tests of these conjectures are necessary and of interest. In particular, it is unclear to me how backreaction from the dressing - if present, may affect the final form of the entropy, or of the free energy of the newly conjectured solutions, and consequently it is also unclear how this may affect the final form of the conjectures and of some of the results presented in this paper e.g. the phase diagrams. "

Our Response: The backreaction of the matter (bulk gas in the case of Grey Galaxy solutions, and the dual giant in the case of DDBH solutions) was investigated in detail in references [25] [26] [27] of our draft. There it was demonstrated that effect of this backreaction on the entropy was subleading in $N$, and so irrelevant for a large $N$ analysis. This point is also referred to in the third item (in the itemized list of points to note about Grey Galaxies and DDBHs) in section 1.2 in the introduction.

We hope this clarification satisfies the referee on this point.

2) "In the discussion section it is suggested that the supersymmetric free energy (not just the entropy) of the dressed black hole configuration is equal to the one of its BPS black hole core

$$F=F[\omega]$$
i.e. the well-known expression quoted in their equation (38). Precisely, in their words: a) "The resultant phases all include central black holes whose chemical potentials (approximately) saturate one of the inequalities listed above. These facts suggest that the grand canonical index is indeed given by the usual formula (38) at chemical potentials that obey (68),..."

Our Response: The referee appears to have misunderstood us. We did not intend to predict that the new phases, described in this paper, have the same grand canonical free energy as a function of chemical potentials as usual vacuum black holes. Infact the line from our discussion section quoted by the referee continued to say "but suggests the existence of new saddle points (or substantially loop corrected saddle points) at values of chemical potentials that approach parametrically near to values of chemical potentials that approach (68).'' The statement "is indeed given by (38) at chemcial potentials that obey (68)" was simply meant to reiterate our expectation that the usual black holes would dominate the grand canonical ensemble when all $Re(\nu_i)$ are positive .

We apologize for the poor wording that seems to have led to this confusion.

3) "Of course, I do not think it is necessary to perform the precise version of this analysis in this paper - although that would give more value to the paper..."

In response to the referees prodding (and related insightful and very useful comments and suggestions) we have added a new Appendix - Appendix I (and subsections 2.4 and 2.6) - to our draft. In this Appendix we review the relationship between the grand canonical and microcanonical description of the black hole phase, and describe how we expect this relationship to be modified in the new Grey Galaxy and DDBH phases. In particular, we explain that we expect $\omega_1$ to simply vanish (in terms of both real as well as imaginary part) in a Grey Galaxy phase that carries $J_1$ gas charge (similar expectations apply to other Grey Galaxy and DDBH phases).

In view of the newly added Appendix, we have, correspondingly, also modified the comments in the last paragraph of section 1.3 of the introduction, and completely rewritten the paragraphs in the discussion section (those that the referee found confusing, see the previous points). We hope this extra material, the additions and clarrifications now address the referees points.

We would also like, once again, to thank the referee for their insightful comments.

---

## Round 1 · Referee Report · Anonymous (Referee 3) · 2025-5-15

Report

The manuscript proposes two conjectures regarding the supersymmetric phases of large N $\mathcal{N}=4$ super Yang Mills theory, both of which are rather surprising and interesting.

The first, the dressed concentration conjecture, says that the leading SUSY entropy for a set of charges is not necessarily given by that of the the SUSY black hole with those charges. It is instead determined via a maximization procedure that may return the entropy of supersymmetric grey galaxy or DDBH solutions over that of the pure SUSY black hole. The second, the unobstructed saddle conjecture, says that the leading large N behavior of the oscillating sum on the index line is captured in a single maximal degeneracy $n(Z_i)$, for the interacting theory, in that sum.

The two conjectures yield supersymmetric and indicial phase diagrams characterized by different types of supersymmetric grey galaxy, DDBH, and pure black hole solutions in AdS. The result is an impressive update for the supersymmetric phases of large N $\mathcal{N}=4$ super Yang-Mills.

1) In Section 1.2, the authors motivate the 'dressing' of the concentration conjecture by stating (in paragraph starting at line 154) that field theoretic enumerations of supersymmetric states at finite N does not fit well with the concentration explanation. Apart from an analogy between the states found in refs. [17,20,22] and grey galaxy/DDBH solutions, does the dressed concentration conjecture extrapolated to low N lead to a better explanation for the distribution of charges of known fortuitous cohomologies at low values of N? If the situation is yet unclear, the motivating paragraph could be potentially misleading for the reader.

2) From the supersymmetric and indicial phase diagrams (e.g. slices shown in Figures 6 or 14), it appears that a slight perturbation away from the 1/2-BPS limits $H_i$ while $j_1 \neq j_2$ would put one in a rank 2 grey galaxy phase. However one may have expected instead to end up in a rank 2 DDBH phase given that we've perturbed slightly away from a limit where one of the charges $Q_i$ is very large. Is there a physical explanation for why the 1/2-BPS limit (associated with one large charge) is continuously connected to the rank 2 grey galaxy phase (associated with one large angular momentum)?

3) I am curious whether the location of the maximal degeneracy along an index line in the interacting theory could be also inferred by examining the location of the maximal degeneracy of the free BPS partition function refined by five charges at sufficiently large N $\sim O(10)$. This comment is motivated by the findings in supersymmetric SYK of ref. [24], where the Q-cohomology is localized at or near the maximal dimension vector space inside a cochain complex prior to the cohomology. This could be a consistency check for the corresponding locations determined for the interacting theory in the paper using the dressed concentration conjecture, and it would be interesting if there is agreement.

4) As mentioned in Section 6, it would be of interest to know how the indicial phases proposed in the paper are related to those suggested by other approaches to the index. For example, the giant graviton expansion for the N=4 index is known to undergo wall-crossing due to the presence of poles in the space of chemical potentials, see, e.g., discussion around (1.12) of https://arxiv.org/abs/2109.02545. The giant graviton bound-states that contribute to the finite N index can therefore change discontinously depending on the intensive charge space region. Could the boundaries between rank 2 and 4 DDBH and pure black holes be related to walls in the giant graviton expansion? Though this is not necessary for the current submission, a canonical understanding of the indical phase diagram could be helpful for determining potential connections to other approaches to the index.

I would be happy to recommend the paper for publication once the authors get a chance to respond to the above points.

Recommendation

Ask for minor revision

  • validity: -
  • significance: -
  • originality: -
  • clarity: -
  • formatting: -
  • grammar: -

Author:  Diksha Jain  on 2025-07-11  [id 5635]

(in reply to Report 3 on 2025-05-15)

We thank the referee for their thoughtful and perceptive comments, questions and suggestions. Below we address their questions:

1) We understand the referee's concern. To address this point we have reworded the relevant paragraph to "The concentration explanation feels satisfyingly economical. However, the results of recent field theoretic enumerations of supersymmetric states [11-22] have not found any clear evidence for the concentration conjecture, at least at the (usually small) values of $N$ and charges at which these studies have been performed."

We hope this modification addresses the referees concerns.

2) This is indeed a curious point. It results from the half BPS black hole being a very singular point (cuspy) point in the space of black holes. As a result, the $\omega_1=0$ sheet, the $\omega_2=0$ sheet, the $\mu_1=0$ sheet and the $\mu_2=0$ sheet all meet at the half BPS black hole that carries $Q_3$ charge. As a consequence, a small perturbation away from the half BPS point can trigger different instabilities, depending on the direction of the perturbation. (Some of this is made quantitative in Appendix C. , where the neighbourhood of $1/8$ BPS black holes is studied in detail).

We do not have a definite physical understanding for why a small angular momentum on top of a large charge can trigger an instability to condensation of angular momentum. Perhaps it is a consequence of the fact that the black hole becomes very small- and so has a small moment of inertia - near the half BPS point, so that a small change in angular momentum leads to a large change in angular velocity. Therefore even though the angular momentum is small, the black holes can still have large angular velocity, triggering an instability.

3) Unfortunately this is not the case. This point was already investigated in the original paper (ref [11] in our draft). In Free Yang Mills theory, the maximum entropy along an indicial line occurs at the wrong location to reproduce the black hole charge relation, and produces too large an entropy. We have described the situation in Footnote 120 (in the new Appendix A.3), where we also explain why - in our opinion - this disagreement is not surprising (the basic point is that the entropy of states along an index line is too regular a function of location, because the computation is done in a free theory, whose spectrum shows no chaotic irregularity).

We hope this addition addresses the referees questions.

4) In response to the referees suggestions above we have added a new Appendix - Appendix I - which discusses the new phases of this paper in the Grand Canonical Ensemble. We hope that the referee finds this new appendix satisfactory.

We find the referee's suggestions linking the phases transitions described in our paper to wall crossing in the space of chemical potentials a very interesting suggestion for future work, but, unfortunately, have nothing concrete to add to their suggestion at this point.

We once again thank the referee for their comments that have helped improve our draft.

---

## Round 2 · Referee Report · Anonymous (Referee 1) · 2025-7-24

Report

I would like to thank the authors for carefully considering my comments and implementing changes in the new version of the manuscript. I am generally satisfied with their changes.

However, I have some comments on the newly added Appendix. A.3. The authors state there: "The number of black hole states as a function of charge is expected to be determined by diagonalizing a ‘one loop’ Hamiltonian that is expected to display features of chaos". They interpret this as saying that the number of BPS states has some random fluctuations as a function of charges.

I believe that the Hamiltonian exhibiting chaos does not imply that the number of black hole states display any randomness. One way to probe this question is to ask whether the number of BPS states, for a fixed charge, would have small random fluctuations if one consider an ensemble of Hamiltonians within the same symmetry class. If the starting Hamiltonian is sufficiently generic such that all the BPS states that could be lifted are already lifted, the BPS spectrum should not fluctuate at all.

Within a fixed theory, we can ask what we expect from the gravity side. At the perturbative level, one should in principle be able to estimate the number of states by studying loop corrections of quantum fields on the BPS black hole background, and it is hard to see how this would produce genuine randomness. One might also wonder about non-perturbative corrections, but there is an independent gravity argument (see around (2.3) in https://arxiv.org/pdf/2307.13051) that the degeneracy of BPS states does not receive wormhole corrections.

The authors refer to [45] in the quote. As far as I understand, the chaos they were studying is the chaos of BPS wavefunctions rather than the number of states.

Recommendation

Publish (surpasses expectations and criteria for this Journal; among top 10%)

---

## Round 2 · Referee Report · Anonymous (Referee 2) · 2025-8-4

Report

Thanks to the authors for the relevant clarifications and the updates to the draft. As I said, the current version of the draft includes some really interesting results and merits publication in this journal.

Recommendation

Publish (meets expectations and criteria for this Journal)

---

## Round 2 · Referee Report · Anonymous (Referee 3) · 2025-8-11

Report

I thank the authors for their explanation of points raised in the comments. The authors have sufficiently addressed my comments through the minor revision and cover letter. I recommend the manuscript for publication.

Recommendation

Publish (meets expectations and criteria for this Journal)

---

## Round 2 · Author Response

In this revised version, we addressed the comments/questions raised by referees.

---

## Round 2 · List of Changes

1. In response to referee 1, we added Appendix A on the unobstructed saddle conjecture, giving an example where it fails and another example where it works. We explained why we expect it to work in the physical situation.

  2. In response to referee 1, we have improved the explanation of the fact that the gas lies within the bosonic cone in section 2.8 and Appendix B.5. We have also improved Table 2 (the improved table now carries more fine grained information about the charges of all single trace operators; the final column in the table can be used to verify our claim that the charges of all bosonic single trace operators lie in the bosonic cone, in the manner explained in the new Footnote 43). We have also corrected some minor errors in Appendix B.6.

  3. In response to Referees 2 and 3, we added an Appendix I (and subsections 2.4 and 2.6) on the manifestation of our new phases in the Canonical Ensemble. As the material of this new Appendix contains the material of Appendix B.4 in our previous version, we have removed Appendix B.4 in this version.

  4. In response to referee 1, we added section 5.3.1 to explain oscillations in Figure 17. The oscillations are coming from the imaginary part of the entropy.

  5. In response to referee 1, we added the words "(and more tentative)" in the abstract. We have also emphasized the limited a priori evidence for the unobstructed saddle conjecture in the newly written text under the statement of the conjecture (in subsection 1.3 of the introduction).

  6. In the process of addressing the referees' questions and comments, we have also taken the opportunity to fix additional typos and grammatical mistakes, etc, that caught our eye.

  7. The new additions to the draft have been flagged at appropriate places in the updated introduction and discussion sections.

---

## Editorial Decision

published